# The $\varphi$ Curve: The Shape of Generalization through the Lens of Norm-based Capacity Control

**Yichen Wang** [*]
Department of Computer Sciences
University of Wisconsin-Madison, US
`yichen.wang@wisc.edu`

**Yudong Chen**
Department of Computer Sciences
University of Wisconsin-Madison, US
`yudong.chen@wisc.edu`

**Lorenzo Rosasco**
Malga - DIBRIS, University of Genova, IT
Istituto Italiano di Tecnologia, IT
`lorenzo.rosasco@unige.it`

**Fanghui Liu**
School of Mathematical Sciences
Institute of Nature Sciences
Shanghai Jiao Tong University
DCS, University of Warwick
`fanghui.liu@sjtu.edu.cn`

## Abstract

Understanding how the test risk scales with model complexity is a central question in machine learning. Classical theory is challenged by the learning curves observed for large over-parametrized deep networks. Capacity measures based on parameter count typically fail to account for these empirical observations. To tackle this challenge, we consider norm-based capacity measures and develop our study for random features based estimators, widely used as simplified theoretical models for more complex networks. In this context, we provide a precise characterization of how the estimator's norm concentrates and how it governs the associated test error. Our results show that the predicted learning curve admits a phase transition from under- to over-parameterization, but no double descent behavior. This confirms that more classical U-shaped behavior is recovered considering appropriate capacity measures based on models norms rather than size. From a technical point of view, we leverage deterministic equivalence as the key tool and further develop new deterministic quantities which are of independent interest.

## 1 Introduction

How the test risk scales with the data size and model size is always a central question in machine learning, both empirically and theoretically. This is characterized as the shape of *generalization*, i.e., learning curves, that can be formulated as classical U-shaped curves [54], double descent [5], and scaling laws [27, 59].

In these learning curves, the model size, i.e., the number of parameters, provides a basic measure of the capacity of a machine learning (ML) model. However it is well known that model size cannot describe the "true" model capacity [2, 63], especially for over-parameterized neural networks [4, 62] and large language models (LLMs) [8]. The focus on the number of parameters results in an inaccurate characterization of the learning curve, and consequently, an improper data-parameter configuration in practice. For instance, even for the same architecture (model size), the learning curve

---

[*]Most of this work was done when Yichen was a visiting student at University of Warwick. Correspondence to Fanghui Liu (fanghui.liu@{sjtu.edu.cn, warwick.ac.uk}).

39th Conference on Neural Information Processing Systems (NeurIPS 2025).

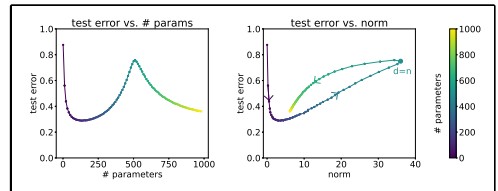
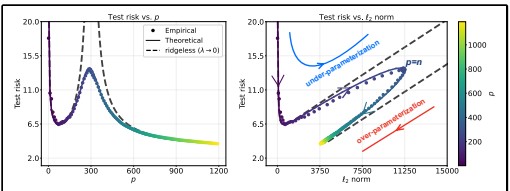

(a) Empirical observations from Figure 8.12 of [43]  (b) Our theory

Figure 1: Fig. 1(a) presents previous empirical observations from [43, Fig. 8.12] in the random feature model. Fig. 1(b) precisely characterize the learning curve from our theory and perfectly matches our experiments (shown by points) with training data $\{(\boldsymbol{x}_i, y_i)\}_{i=1}^n$, with $n = 300$, sub-sampled from the MNIST [30] with $d = 748$. The feature map is defined as $\varphi(\boldsymbol{x}, \boldsymbol{w}) = \mathrm{erf}(\langle \boldsymbol{x}, \boldsymbol{w} \rangle)$ with random initialization $\boldsymbol{w} \sim \mathcal{N}(0, \boldsymbol{I})$. Note that whether the curve is finally lower than before is different between Fig. 1(a) and Fig. 1(b), mainly because of data, see more discussion in Appendix G.3.

can be totally different, e.g., double descent may disappear [39, 40]. A natural question raises that: *What is the shape of generalization under the lens of a suitable model capacity than model size?*

In a ML model, its parameters can be represented as vectors, matrices, or tensors, and hence the model size is characterized by their dimensions. However, to evaluate the "size" of parameters, a more suitable metric is their norm. This is termed as *normed based capacity*, a perspective pioneered in the classical results indicated by [2]. Indeed, norm based capacity/complexity are widely considered to be more effective in characterizing generalization behavior; see e.g. [42, 52, 16, 33] and references therein. For instance, path-norm based model capacity empirically demonstrates a quite strong correlation to generalization while other metrics of model capacity may not [26]. Additionally, minimum norm-based solution received much attention as a possible way to understand the learning performance of over-parameterized neural networks in the interpolation regime; see e.g. [31, 55, 4, 62, 40].

Empirical observations on the learning curve under norm-based capacity have been discussed in the lecture notes [43, Fig. 8.12], as shown in Fig. 1(a): when changing the model capacity from model size to parameters' norm, the learning curve is changed from double descent to a "$\varphi$"-shaped curve. However, a precise mathematical framework on obtaining/understanding this curve is still lacking. The goal of this paper is to investigate this curve by addressing the following fundamental question:

*What is the relationship between test risk and norm-based model capacity, and how can it be precisely characterized?*

In this work, we take the first step toward answering this question, as illustrated in Fig. 1(b). Compared to the classical double descent curve w.r.t. model size $p$, we quantitatively characterize the relationship: test risk *vs.* norm-based capacity. Our theoretical predictions (shown as curves) precisely predict the empirical results (shown as points), and the curve is more close to the "$\varphi$"-shaped curve. More broadly, our results address how the learning curve behaves under more suitable model capacities—specifically, whether classical phenomena such as the U-shaped curve, double descent, or scaling laws persist or are fundamentally altered. We believe this opens the door to rethinking the role of model capacity and the nature of learning curves (e.g., scaling laws) in the era of LLMs.

## 1.1 Contributions and findings

We consider linear and random features models (RFMs) regression to precisely characterize the relationship between the test risk and the capacity measured by the estimator's norm. The key technical tool we leverage is the *deterministic equivalence* technique from random matrix theory [10, 14], where the test risk $\mathcal{R}$ (depending on data $\boldsymbol{X}$, target function $f^*$, and the regularization parameter $\lambda$) can be well approximated by a deterministic quantity R (with data size $n$ and model size $p$), i.e.,

$$\mathcal{R}(\boldsymbol{X}, f_*, \lambda) = (1 + \mathcal{O}(n^{-1/2}) + \mathcal{O}(p^{-1/2})) \cdot \mathsf{R}(\boldsymbol{\Sigma}, f_*, \lambda_*), \quad \textit{asymptotically or non-asymptotically}$$

where $\mathsf{R}(\boldsymbol{\Sigma}, f_*, \lambda_*)$ is the exact deterministic characterization only depends on $f^\star$, expected data covariance $\boldsymbol{\Sigma}$, "re-scaled" regularization parameter $\lambda$, or other deterministic quantities. In our work,

Table 1: Summary of our main results for RFMs on deterministic equivalents and their relationship.

| Type | Results | Regularization | Deterministic equivalents N | Relationship between R and N |
|---|---|---|---|---|
| Deterministic equivalence | Theorem 3.1 | $\lambda > 0$ | Asymptotic | - |
| | Corollary 3.2 | $\lambda \to 0$ | Asymptotic | - |
| | Theorem E.2 | $\lambda > 0$ | Non-asymptotic | - |
| Relationship | Proposition 4.1 | $\lambda \to 0$ | - | Over-parameterized regime |
| | Corollary E.3 | $\lambda \to 0$ | - | Under $\boldsymbol{\Lambda} = \boldsymbol{I}_m$ $(n < m < \infty)$ |
| | Corollary 4.2 | $\lambda \to 0$ | - | Under Assumption 2 (power-law) |
| | Proposition 4.3 | $\lambda > 0$ | - | Under Assumption 2 (power-law) |

we aim to build the deterministic equivalents N of the estimator's $\ell_2$ norm $\mathcal{N}$, both *asymptotically* and *non-asymptotically*, and derive a corresponding relationship between R and N, allowing a precise characterization, i.e.

> **Our target**
>
> $$\mathcal{N}(\boldsymbol{X}, f_*, \lambda) = (1 + \mathcal{O}(n^{-1/2}) + \mathcal{O}(p^{-1/2})) \cdot \mathsf{N}(\boldsymbol{\Sigma}, f_*, \lambda_*) \implies \mathsf{R} = g(\mathsf{N}) \text{ for some function } g.$$

The main results are given by Table 1 for RFMs, which covers random features ridge regression as well as min-norm estimator ($\lambda = 0$). Results for linear regression are deferred to Appendix D due to page limit. Deriving results N on norm-based capacity is more chandelling than for test risk. This is because, we need to explore **new deterministic quantities**, which are of independent interest and more broadly useful. Specifically, we derive the deterministic equivalents w.r.t. $\text{Tr}(\boldsymbol{A} \boldsymbol{X}^\top \boldsymbol{X} (\boldsymbol{X}^\top \boldsymbol{X} + \lambda)^{-1})$ for any positive semi-definite (PSD) matrix $\boldsymbol{A}$ while previous work only handled $\boldsymbol{A} := \boldsymbol{I}$ [1, 37, 14]. Moreover, non-asymptotic results, those valid for finite $n, p = \Omega(1)$ rather than in the asymptotic regime $n, p \to \infty$, on norm-based capacity require more technical conditions. In particular, they involve non-asymptotic bounds on *deterministic equivalents of differences between random quantities*. Due to the complexity of the formulations, we present these results in the appendix.

After that, we establish the characterization of $\mathsf{R} = g(\mathsf{N})$ under isotropic features and further illustrate the scaling law under classical power law scaling assumptions. The derivation requires non-trivial calculation and integral approximation by eliminating the model size $p$. We have the following findings from this characterization.

- **Norm-based capacity suffices to characterize generalization, whereas effective dimension and smoother do not:** Our results on deterministic equivalence demonstrate that the estimator's norm includes the information of the test risk's bias and variance[2], respectively. In contrast, typical model capacity, e.g., effective dimension [63] and smoother [12] can only characterize the test risk's variance and thus are insufficient to characterize generalization.

- **Phase transition exists but double descent does not exist:** There exists a phase transition from under- to over-parameterized regime, as shown in Fig. 1(b). In the under-parameterized regime, we still observe the same U-shaped curve, whether we consider the norm N or model size $p$ as the model capacity. This curve can be precisely described as a hyperbola for the min-norm interpolator (linear regression) under isotropic features.

  But in the over-parameterized regime, when the norm N increases, the test risk R also increases (almost linearly if the regularization is small). This differs from double descent: when the model size $p$ increases, the test risk decreases. Our empirical results on Fig. 1 verify this theoretical prediction. More importantly, this curve aligns more with classical statistical intuition—a U-shaped curve—rather than the double descent phenomenon. We conclude that *with suitably chosen model capacity, the learning curve more closely follows a U-shape curve than a double descent*, potentially observable in more complex models and real-world datasets, see Appendices H.2 and H.3, respectively.

---

[2]Strictly speaking, it also requires knowing whether the model is under-parameterized or over-parameterized, as the self-consistent equations differ between these two regimes.

- **Scaling law is not monotone in norm-based capacity:** We study the scaling law of RFMs under norm-based capacity in a multiplication style by taking model size $p := n^q$ ($q \geq 0$), leading to $\mathsf{R} = Cn^{-a}\mathsf{N}^b$ with $a \geq 0$, $b \in \mathbb{R}$, and $C > 0$. Note that $b \in \mathbb{R}$ can be positive or negative, resulting in different behaviors of $\mathsf{R}$. This differs from the classical scaling law that is monotonically decreasing in the model size.
- **Controlling norm-based capacity can be achieved by the tuned regularization parameter $\lambda$:** Norm-based capacity appears less intuitive used in practice when compared to model size. Our results demonstrate that the norm decreases monotonically with increasing $\lambda$, and in both under- and over-parameterized regimes. Accordingly, such one-to-one correspondence allows for controlling norm via $\lambda$, related to the known L-curve [21].

We remark that, our theory cannot fully recover the "$\varphi$"-curve shown in Fig. 1(a), where the curve in some over-parameterized regimes is above that in the under-parameterized regime. This is because, some real-world datasets may not satisfy the well-behaved data assumption in Assumption 1. We also emphasize that we do *not* claim that $\ell_2$ norm-based capacity (or other norm-based capacity) is the best metric of model capacity. Rather, this work aims to show how the test risk behaves when a more suitable model capacity than model size is used to measure capacity. For completeness, we discuss the "$\varphi$"-curve under real-world dataset as well as other metrics of model capacity evaluated in Appendix H. All code and replication materials (including our reproduction of OpenAI's deep double-descent results [40]) are available at `github.com/yichenblue/norm-capacity`.

**Notations:** In this paper we generally adopt the following convention. Caligraphic letters (e.g., $\mathcal{N}_\lambda, \mathcal{R}_\lambda, \mathcal{B}_{\mathcal{N},\lambda}, \mathcal{V}_{\mathcal{R},\lambda},$) denote random quantities, and upright letters (e.g., $\mathsf{N}_\lambda, \mathsf{R}_\lambda, \mathsf{B}_{\mathsf{N},\lambda}, \mathsf{V}_{\mathsf{R},\lambda}$) denote their deterministic equivalents. The letters N, R, B, V above (in any font) signify quantities related to the solution norm, test risk, bias, and variance, respectively. With $\lambda$ denoting the $\ell_2$-regularization parameter, setting $\lambda = 0$ corresponds to the min-norm interpolator. The superscripts $^{\mathsf{LS}}$ and $^{\mathsf{RFM}}$ denote quantities defined for linear regression and random feature regression, respectively.

We denote by $\gamma$ the ratio between the parameter size and the data size, i.e., $\gamma := d/n$ in ridge regression and $\gamma := p/n$ in RFMs. For asymptotic results, we adopt the notation $u \sim v$, meaning that the ratio $u/v$ tends to one as the dimensions $n, d$ ($p$ for RFMs) tend to infinity. A complete list notations can be found in Appendix A.

## 1.2 Related work

The relationship between the test risk, the data size, and the model size is classically characterized by the U-shaped curve [54]: larger models tend to overfit. This can not explain the success of deep learning (with even more parameters than data), leading to a new concept: double descent [5], where the test risk has a second descent when transitioning from under- to over-parameterized regimes. Moreover recent scaling law [27] shows that the test risk is monotonically decreasing with model size, typically in the under-parameterized regime for LLMs.

**Model capacity metrics:** Beyond model size as a capacity measure, there is considerable effort to define alternative capacity measures, e.g, degrees of freedom from statistics [17, 18, 47], effective dimension/rank [63, 3], smoother [12], flatness [48], as well as norm-based capacity [42, 33]. The norm's asymptotic characterization is given in specific settings [25] but the risk-norm relationship is not directly studied. Besides, training strategies can be also explained as implicit regularization [61, 41], affecting the model capacity as well. We refer to the survey [26] for details.

**Deterministic equivalents:** Random matrix theory (RMT) provides powerful mathematical tools to precisely characterize the relationship between the test risk $\mathcal{R}$ and $n, p, d$ via deterministic equivalence, in an asymptotic regime ($n, p, d \to \infty$, [35, 20, 57, 60, 1]), or non-asymptotic regime [22, 10, 37]. We refer the reader to [11] for further details. Complementary to RMT approaches, techniques from statistical physics are also possible to derive the deterministic equivalence, e.g., replica methods [6, 19, 34] and dynamical mean field theory [28, 36, 38].

## 2 Preliminaries

We overview RFMs via deterministic equivalents here; see more details in Appendix B with additional preliminaries on linear regression.

Random features models (RFMs) [49, 32] can be regarded as two-layer neural networks with $f(\boldsymbol{x}; \boldsymbol{a}) = \frac{1}{\sqrt{p}} \sum_{j=1}^{p} a_j \varphi(\boldsymbol{x}, \boldsymbol{w}_j)$, where $\varphi : \mathbb{R}^d \times \mathbb{R}^d \to \mathbb{R}$ is a nonlinear activation function. The

first-layer parameters $\{\boldsymbol{w}_i\}_{i=1}^p$ are sampled i.i.d. from a probability measure $\mu_{\boldsymbol{w}}$ and kept unchanged during training. We only train $\boldsymbol{a}$ by solving the following random features ridge regression

$$\hat{\boldsymbol{a}} := \underset{\boldsymbol{a} \in \mathbb{R}^p}{\arg\min} \left\{ \sum_{i=1}^n (y_i - f(\boldsymbol{x}_i; \boldsymbol{a}))^2 + \lambda \|\boldsymbol{a}\|_2^2 \right\} = \left( \boldsymbol{Z}^\top \boldsymbol{Z} + \lambda \mathbf{I}_p \right)^{-1} \boldsymbol{Z}^\top \boldsymbol{y}, \quad \boldsymbol{Z} \in \mathbb{R}^{n \times p}, \quad (1)$$

where the feature matrix is $[\boldsymbol{Z}]_{ij} = p^{-1/2}\varphi(\boldsymbol{x}_i; \boldsymbol{w}_j)$ and $\lambda \geq 0$ is the regularization parameter. We also consider min-$\ell_2$-norm solution ($\lambda = 0$), i.e., $\hat{\boldsymbol{a}}_{\min} = \arg\min_{\boldsymbol{a}} \|\boldsymbol{a}\|_2, s.t. \boldsymbol{Z}\boldsymbol{a} = \boldsymbol{y}$.

Following [14], under proper assumptions on $\varphi$ (e.g., bounded, squared-integrable), we can define a compact integral operator $\mathbb{T}: L^2(\mu_{\boldsymbol{x}}) \to \mathcal{V} \subseteq L^2(\mu_{\boldsymbol{w}})$ for any $f \in L_2(\mu_{\boldsymbol{x}})$ such that

$$(\mathbb{T}f)(\boldsymbol{w}) := \int_{\mathbb{R}^d} \varphi(\boldsymbol{x}; \boldsymbol{w}) f(\boldsymbol{x}) \mathrm{d}\mu_{\boldsymbol{x}}, \quad \mathbb{T} = \sum_{k=1}^\infty \xi_k \psi_k \phi_k^*,$$

where $(\xi_k)_{k \geq 1} \subseteq \mathbb{R}$ are the eigenvalues and $(\psi_k)_{k \geq 1}$ and $(\phi_k)_{k \geq 1}$ are orthonormal bases of $L^2(\mu_{\boldsymbol{x}})$ and $\mathcal{V}$ for spectral decomposition respectively. We denote $\boldsymbol{\Lambda} := \mathrm{diag}(\xi_1^2, \xi_2^2, \ldots) \in \mathbb{R}^{\infty \times \infty}$ and assume all eigenvalues are non-zero and arranged in non-increasing order.

Accordingly, the covariate feature matrix can be represented as $\boldsymbol{G} := [\boldsymbol{g}_1, \ldots, \boldsymbol{g}_n]^\top \in \mathbb{R}^{n \times \infty}$ with $\boldsymbol{g}_i := (\psi_k(\boldsymbol{x}_i))_{k \geq 1}$ and the weight feature matrix is $\boldsymbol{F} := [\boldsymbol{f}_1, \ldots, \boldsymbol{f}_p]^\top \in \mathbb{R}^{p \times \infty}$ with $\boldsymbol{f}_j := (\xi_k \phi_k(\boldsymbol{w}_j))_{k \geq 1}$. Then the feature matrix can be denoted by $\boldsymbol{Z} = \frac{1}{\sqrt{p}} \boldsymbol{G} \boldsymbol{F}^\top \in \mathbb{R}^{n \times p}$. Note that $\boldsymbol{f}$ has covariance matrix $\mathbb{E}[\boldsymbol{f}\boldsymbol{f}^\top] = \boldsymbol{\Lambda}$, and we further introduce $\widehat{\boldsymbol{\Lambda}}_{\boldsymbol{F}} := \mathbb{E}_{\boldsymbol{z}}[\boldsymbol{z}\boldsymbol{z}^\top | \boldsymbol{F}] = \frac{1}{p} \boldsymbol{F}\boldsymbol{F}^\top \in \mathbb{R}^{p \times p}$.

Assuming that $f_* \in L^2(\mu_{\boldsymbol{x}})$ admits $f_*(\boldsymbol{x}) = \sum_{k \geq 1} \boldsymbol{\theta}_{*,k} \psi_k(\boldsymbol{x})$, we have a bias-variance decomposition of the excess risk

$$\mathcal{R}^{\mathrm{RFM}} := \mathbb{E}_\varepsilon \left\| \boldsymbol{\theta}_* - \frac{1}{\sqrt{p}} \boldsymbol{F}^\top \hat{\boldsymbol{a}} \right\|_2^2 = \left\| \boldsymbol{\theta}_* - \frac{1}{\sqrt{p}} \boldsymbol{F}^\top \mathbb{E}_\varepsilon[\hat{\boldsymbol{a}}] \right\|_2^2 + \mathrm{Tr}\left( \widehat{\boldsymbol{\Lambda}}_{\boldsymbol{F}} \mathrm{Cov}_\varepsilon(\hat{\boldsymbol{a}}) \right),$$

where the first RHS term is the *bias*, denoted by $\mathcal{B}_{\mathcal{R},\lambda}^{\mathrm{RFM}}$, and the second term is the *variance*, denoted by $\mathcal{V}_{\mathcal{R},\lambda}^{\mathrm{RFM}}$. Similarly, under proper assumptions (to be detailed later), they admit the following deterministic equivalents, asymptotically [53] and non-asymptotically [14]

$$\mathcal{B}_{\mathcal{R},\lambda}^{\mathrm{RFM}} \sim \mathsf{B}_{\mathrm{R},\lambda}^{\mathrm{RFM}} := \frac{\nu_2^2}{1 - \Upsilon(\nu_1, \nu_2)} \left[ \langle \boldsymbol{\theta}_*, (\boldsymbol{\Lambda} + \nu_2 \mathbf{I})^{-2} \boldsymbol{\theta}_* \rangle + \chi(\nu_2) \langle \boldsymbol{\theta}_*, \boldsymbol{\Lambda}(\boldsymbol{\Lambda} + \nu_2 \mathbf{I})^{-2} \boldsymbol{\theta}_* \rangle \right],$$

$$\mathcal{V}_{\mathcal{R},\lambda}^{\mathrm{RFM}} \sim \mathsf{V}_{\mathrm{R},\lambda}^{\mathrm{RFM}} := \frac{\sigma^2 \Upsilon(\nu_1, \nu_2)}{1 - \Upsilon(\nu_1, \nu_2)}, \quad (2)$$

where $(\nu_1, \nu_2)$ satisfy the self-consistent equations

$$n - \frac{\lambda}{\nu_1} = \mathrm{Tr}(\boldsymbol{\Lambda}(\boldsymbol{\Lambda} + \nu_2 \mathbf{I})^{-1}), \quad p - \frac{p\nu_1}{\nu_2} = \mathrm{Tr}(\boldsymbol{\Lambda}(\boldsymbol{\Lambda} + \nu_2 \mathbf{I})^{-1}), \quad (3)$$

and $\Upsilon(\nu_1, \nu_2)$ and $\chi(\nu_2)$ are defined as

$$\Upsilon(\nu_1, \nu_2) := \frac{p}{n} \left[ \left( 1 - \frac{\nu_1}{\nu_2} \right)^2 + \left( \frac{\nu_1}{\nu_2} \right)^2 \frac{\mathrm{Tr}\left( \boldsymbol{\Lambda}^2(\boldsymbol{\Lambda} + \nu_2)^{-2} \right)}{p - \mathrm{Tr}\left( \boldsymbol{\Lambda}^2(\boldsymbol{\Lambda} + \nu_2)^{-2} \right)} \right], \quad \chi(\nu_2) := \frac{\mathrm{Tr}\left( \boldsymbol{\Lambda}(\boldsymbol{\Lambda} + \nu_2)^{-2} \right)}{p - \mathrm{Tr}\left( \boldsymbol{\Lambda}^2(\boldsymbol{\Lambda} + \nu_2)^{-2} \right)}.$$

## 3 Deterministic equivalents under norm-based capacity

To mathematically characterize the phenomena in Fig. 1 under norm-based capacity, in this section, we firstly derive the bias-variance decomposition for the norm $\mathbb{E}_\varepsilon \|\hat{\boldsymbol{a}}\|_2^2 =: \mathcal{N}_\lambda^{\mathrm{RFM}} = \mathcal{B}_{\mathcal{N},\lambda}^{\mathrm{RFM}} + \mathcal{V}_{\mathcal{N},\lambda}^{\mathrm{RFM}}$ (with definition later), then relate $\mathcal{B}_{\mathcal{N},\lambda}^{\mathrm{RFM}}$ and $\mathcal{V}_{\mathcal{N},\lambda}^{\mathrm{RFM}}$ to their respective deterministic equivalents $\mathsf{B}_{\mathrm{N},\lambda}^{\mathrm{RFM}}$ and $\mathsf{V}_{\mathrm{N},\lambda}^{\mathrm{RFM}}$. In the next section, we aim to precisely characterize the learning curves under norm-based capacities via deterministic equivalence.

To derive the deterministic equivalence, we need the following assumption on well-behaved data and random features.

**Assumption 1** (Concentration of the eigenfunctions [14])**.** Recall the random vectors $\boldsymbol{\psi} := (\xi_k \psi_k(\boldsymbol{x}))_{k\geq 1}$ and $\boldsymbol{\phi} := (\xi_k \phi_k(\boldsymbol{w}))_{k\geq 1}$. There exists $C_* > 0$ such that for any PSD matrix $\boldsymbol{A} \in \mathbb{R}^{\infty \times \infty}$ with $\mathrm{Tr}(\boldsymbol{\Lambda A}) < \infty$ and any $t \geq 0$, we have

$$\mathbb{P}\left(\left|\boldsymbol{\psi}^\top \boldsymbol{A}\boldsymbol{\psi} - \mathrm{Tr}(\boldsymbol{\Lambda A})\right| \geq t\|\boldsymbol{\Lambda}^{1/2}\boldsymbol{A}\boldsymbol{\Lambda}^{1/2}\|_{\mathbf{F}}\right) \leq C_* e^{-\frac{t}{C_*}},$$

$$\mathbb{P}\left(\left|\boldsymbol{\phi}^\top \boldsymbol{A}\boldsymbol{\phi} - \mathrm{Tr}(\boldsymbol{\Lambda A})\right| \geq t\|\boldsymbol{\Lambda}^{1/2}\boldsymbol{A}\boldsymbol{\Lambda}^{1/2}\|_{\mathbf{F}}\right) \leq C_* e^{-\frac{t}{C_*}}.$$

This assumptions holds for sub-Gaussian distributions and more generally, distributions that satisfy a log-Sobolev or convex Lipschitz concentration inequality [10]. Next we present the deterministic equivalence results of $\mathcal{N}_\lambda^{\mathrm{RFM}}$, deferring the proof to Appendix E.1.

**Theorem 3.1** (Deterministic equivalence of $\mathcal{N}_\lambda^{\mathrm{RFM}}$)**.** *Given RFMs in Section 2, the bias-variance decomposition of its norm $\mathbb{E}_\varepsilon\|\hat{\boldsymbol{a}}\|_2^2$ is given by $\mathbb{E}_\varepsilon\|\hat{\boldsymbol{a}}\|_2^2 =: \mathcal{N}_\lambda^{\mathrm{RFM}} = \mathcal{B}_{\mathcal{N},\lambda}^{\mathrm{RFM}} + \mathcal{V}_{\mathcal{N},\lambda}^{\mathrm{RFM}}$, where $\mathcal{B}_{\mathcal{N},\lambda}^{\mathrm{RFM}}$ and $\mathcal{V}_{\mathcal{N},\lambda}^{\mathrm{RFM}}$ are defined as*

$$\mathcal{B}_{\mathcal{N},\lambda}^{\mathrm{RFM}} := \langle \boldsymbol{\theta}_*, \boldsymbol{G}^\top \boldsymbol{Z}(\boldsymbol{Z}^\top \boldsymbol{Z} + \lambda \boldsymbol{I})^{-2}\boldsymbol{Z}^\top \boldsymbol{G}\boldsymbol{\theta}_* \rangle, \quad \mathcal{V}_{\mathcal{N},\lambda}^{\mathrm{RFM}} := \sigma^2 \mathrm{Tr}\left(\boldsymbol{Z}^\top \boldsymbol{Z}(\boldsymbol{Z}^\top \boldsymbol{Z} + \lambda \boldsymbol{I})^{-2}\right).$$

*Under Assumption 1, we have the following asymptotic deterministic equivalents $\mathcal{B}_{\mathcal{N},\lambda}^{\mathrm{RFM}} \sim \mathsf{B}_{\mathsf{N},\lambda}^{\mathrm{RFM}}$, $\mathcal{V}_{\mathcal{N},\lambda}^{\mathrm{RFM}} \sim \mathsf{V}_{\mathsf{N},\lambda}^{\mathrm{RFM}}$ and thus $\mathcal{N}_\lambda^{\mathrm{RFM}} \sim \mathsf{N}_\lambda^{\mathrm{RFM}} := \mathsf{B}_{\mathsf{N},\lambda}^{\mathrm{RFM}} + \mathsf{V}_{\mathsf{N},\lambda}^{\mathrm{RFM}}$*

$$\mathsf{B}_{\mathsf{N},\lambda}^{\mathrm{RFM}} := \frac{p\langle \boldsymbol{\theta}_*, \boldsymbol{\Lambda}(\boldsymbol{\Lambda}+\nu_2\boldsymbol{I})^{-2}\boldsymbol{\theta}_*\rangle}{p - \mathrm{Tr}\left(\boldsymbol{\Lambda}^2(\boldsymbol{\Lambda}+\nu_2\boldsymbol{I})^{-2}\right)} + \frac{p\chi(\nu_2)}{n} \cdot \underbrace{\boxed{\frac{\nu_2^2\left[\langle \boldsymbol{\theta}_*, (\boldsymbol{\Lambda}+\nu_2\boldsymbol{I})^{-2}\boldsymbol{\theta}_*\rangle + \chi(\nu_2)\langle \boldsymbol{\theta}_*, \boldsymbol{\Lambda}(\boldsymbol{\Lambda}+\nu_2\boldsymbol{I})^{-2}\boldsymbol{\theta}_*\rangle\right]}{1 - \Upsilon(\nu_1,\nu_2)}}}_{\mathsf{B}_{\mathsf{R},\lambda}^{\mathrm{RFM}}},$$

$$\mathsf{V}_{\mathsf{N},\lambda}^{\mathrm{RFM}} := \frac{p\chi(\nu_2)}{n\Upsilon(\nu_1,\nu_2)} \cdot \underbrace{\boxed{\frac{\sigma^2\Upsilon(\nu_1,\nu_2)}{1 - \Upsilon(\nu_1,\nu_2)}}}_{\mathsf{V}_{\mathsf{R},\lambda}^{\mathrm{RFM}}}. \tag{4}$$

**Remark:** This theorem establishes asymptotic equivalence; a more complex non-asymptotic analysis is developed in Appendix E.2. Numerical validation is provided through experiments on synthetic and real-world datasets in Appendix H.1 and Appendix H.2, respectively.

By comparing Eq. (2) (test risk) and Eq. (4) (norm) via deterministic equivalence, we conclude that

- Bias: the test risk's bias in Eq. (2) has been included in the the second term of $\mathsf{B}_{\mathsf{N},\lambda}^{\mathrm{RFM}}$ (see the red area in Eq. (4)) with a rescaled factor $\frac{p\chi(\nu_2)}{n}$.

- Variance: we find that the variance term of the norm $\mathsf{V}_{\mathsf{N},\lambda}^{\mathrm{RFM}}$ equals the variance term of the test risk $\mathsf{V}_{\mathsf{R},\lambda}^{\mathrm{RFM}}$ (see the blue area in Eq. (4)) in Eq. (2) multiplied by a factor $\frac{p\chi(\nu_2)}{n\Upsilon(\nu_1,\nu_2)}$.

Hence norm-based capacity (on the second layer) suffices to characterize the test risk in RFMs. Here we discuss whether **other classical metrics** of model capacity can characterize the generalization.

- Effective dimension [63]: It is defined as $\mathrm{Tr}(\boldsymbol{\Lambda}(\boldsymbol{\Lambda}+\nu_{1(2)}\boldsymbol{I})^{-1})$ or similar formulation, e.g., $\mathrm{Tr}(\boldsymbol{\Lambda}^2(\boldsymbol{\Lambda}+\nu_{1(2)}\boldsymbol{I})^{-2})$. These effective dimensions increase monotonically with $p$, thus exhibit double descent.

- Smoother [12]: It is defined as $n\mathrm{Tr}(\widehat{\boldsymbol{\Lambda}}_{\boldsymbol{F}}\boldsymbol{Z}^\top \boldsymbol{Z}(\boldsymbol{Z}^\top \boldsymbol{Z} + \lambda)^{-2})$, which corresponds to the variance of the test risk $\mathcal{V}_{\mathcal{R}}^{\mathrm{RFM}}$ scaled by the factor $\frac{n}{\sigma^2}$. Therefore, it first increases and then decreases with $p$, reaching a peak near at the interpolation threshold ($p = n$).

The above two metrics offer a variance-based measure of model capacity: they capture the variance component of test risk but contain no information about the target function $\boldsymbol{\theta}^*$, and thus cannot fully characterize generalization. In summary, *norm-based capacity suffices to characterize generalization, whereas effective dimension and smoother do not.*

**Norm-based capacity over different layers:** In RFMs, if we use the norm of the first layer, i.e., $\|\boldsymbol{W}\|_{\mathrm{F}}$ as model capacity, we will obtain a reshaped double descent curve as Fig. 15. This is because, the first layer's parameters are with random Gaussian initialization and then untrained, we directly

have $\mathbb{E}[\|\boldsymbol{W}\|_{\mathrm{F}}] = \sqrt{2} \cdot \Gamma(\frac{dp+1}{2})/\Gamma(\frac{dp}{2}) \approx \sqrt{dp - \frac{1}{2}}$, increasing with $p$. For two-layer neural networks with both trained layers, path norm is empirically verified as the most suitable (data-independent) model capacity for neural networks. We find that the curve aligns more closely with the norm-based capacity in RFMs of the second-layer parameters, rather than that of the first layer, see more discussion in Appendix H.3.

For better illustration, we consider a special case of Theorem 3.1, the min-norm estimator ($\lambda = 0$), which will be used later, and derive its deterministic equivalence; see the proof in Appendix E.1.

**Corollary 3.2** (Asymptotic deterministic equivalence of $\mathsf{N}_0^{\mathrm{RFM}}$). *Under Assumption 1, for the min-$\ell_2$-norm estimator $\hat{\boldsymbol{a}}_{\min}$, in the under-parameterized regime ($p < n$), we have*

$$\mathcal{B}_{\mathcal{N},0}^{\mathrm{RFM}} \sim \frac{p\langle\boldsymbol{\theta}_*, \boldsymbol{\Lambda}(\boldsymbol{\Lambda} + \lambda_p \boldsymbol{I})^{-2}\boldsymbol{\theta}_*\rangle}{n - \mathrm{Tr}(\boldsymbol{\Lambda}^2(\boldsymbol{\Lambda} + \lambda_p \boldsymbol{I})^{-2})} + \frac{p\langle\boldsymbol{\theta}_*, (\boldsymbol{\Lambda} + \lambda_p \boldsymbol{I})^{-1}\boldsymbol{\theta}_*\rangle}{n - p}, \quad \mathcal{V}_{\mathcal{N},0}^{\mathrm{RFM}} \sim \frac{\sigma^2 p}{\lambda_p(n - p)},$$

*where $\lambda_p$ is from $\mathrm{Tr}(\boldsymbol{\Lambda}(\boldsymbol{\Lambda} + \lambda_p \boldsymbol{I})^{-1}) \sim p$. In the over-parameterized regime ($p > n$), we have*

$$\mathcal{B}_{\mathcal{N},0}^{\mathrm{RFM}} \sim \frac{p\langle\boldsymbol{\theta}_*, (\boldsymbol{\Lambda} + \lambda_n \boldsymbol{I})^{-1}\boldsymbol{\theta}_*\rangle}{p - n}, \quad \mathcal{V}_{\mathcal{N},0}^{\mathrm{RFM}} \sim \frac{\sigma^2 p}{\lambda_n(p - n)},$$

*where $\lambda_n$ is defined by $\mathrm{Tr}(\boldsymbol{\Lambda}(\boldsymbol{\Lambda} + \lambda_n \boldsymbol{I})^{-1}) \sim n$.*

**Remark:** $\mathcal{V}_{\mathcal{N},0}^{\mathrm{RFM}}$ admits the similar formulation in under-/over-parameterized regimes but differs in $\lambda_n$ and $\lambda_p$. An interesting point to note is that, in the over-parameterized regime, $\lambda_n$ is a constant when $n$ constant. Therefore, $\mathcal{B}_{\mathcal{N},0}^{\mathrm{RFM}}$ and $\mathcal{V}_{\mathcal{N},0}^{\mathrm{RFM}}$ are proportional to each other.

We need to analyze RFMs separately in the under-/over-parameterized regimes when $\lambda \to 0$, leading to different self-consistent equations in these two settings.

- In the under-parameterized regime, $\nu_1$ converges to $0$, and $\nu_2$ converges to a value $\lambda_p$ satisfying $\mathrm{Tr}(\boldsymbol{\Lambda}(\boldsymbol{\Lambda} + \lambda_p \boldsymbol{I})^{-1}) = p$.
- In the over-parameterized regime, $\nu_2$ converges to a constant $\lambda_n$ satisfying $\mathrm{Tr}(\boldsymbol{\Lambda}(\boldsymbol{\Lambda} + \lambda_n \boldsymbol{I})^{-1}) = n$, and $\nu_1$ converges to $\nu_2(1 - {}^n/{}_p)$.

These differing asymptotic behaviors of $\nu_1$ and $\nu_2$ between the two regimes enable a more precise characterization of the risk–norm relationship, which will be described in the next section.

## 4 Characterization of learning curves

By giving the deterministic equivalents of the norm, we are ready to plot the learning curve under norm-based capacity, see Fig. 1(b) for illustration. In some special cases, the mathematical formulation of learning curves can be given. Accordingly, in this section, we firstly discuss the shape of learning curves from the lens of norm-based capacity in Section 4.1. Then we take the example of min-$\ell_2$-norm interpolator, and precisely characterize the learning curve by reshaping scaling laws in Section 4.2.

### 4.1 The shape description of learning curves

Here we conduct the bias-variance decomposition, and track how bias and variance behave w.r.t. model size, norm, and the regularization parameter $\lambda$, as shown in Fig. 2, which will provide a more detailed description and understanding on learning curves.

**Reshape bias-variance trade-offs and double descent:** We plot the bias and variance components of the test risk over model size $p$ and norm, see Fig. 2(a) and Fig. 2(b), respectively. Note that, our theory (shown in curve) can precisely predict experimental results (shown by points). Fig. 2(a) aligns closely with [35, Figure 6] on the double descent when increasing the model size $p$ from the under- to over-parameterized regimes. However, even in the classical under-parameterized setting, the conventional bias-variance trade-off no longer holds: the bias follows a U-shaped curve, whereas the variance grows monotonically. This was discussed recently by [56, 50] on "whether we should remove bias-variance trade-offs from ML textbooks".

When examining bias-variance vs. norm (see Fig. 2(b)), we observe that: *i)* in the under-parameterized regime, bias exhibits a U-shaped dependence on norm, while variance increases monotonically. This

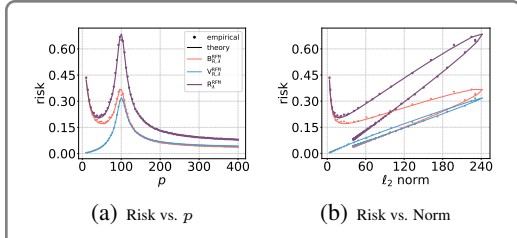
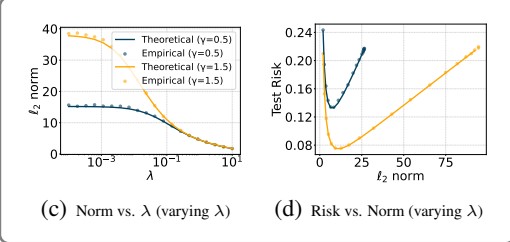

|   |   |
|:-:|:-:|
| (a) Risk vs. $p$ | (b) Risk vs. Norm |

|   |   |
|:-:|:-:|
| (c) Norm vs. $\lambda$ (varying $\lambda$) | (d) Risk vs. Norm (varying $\lambda$) |

Figure 2: The curves of bias and variance in RFMs are over model size $p$ in Fig. 2(a) and over norm $\mathbb{E}_{\varepsilon}\|\hat{\boldsymbol{a}}\|_2^2$ in Fig. 2(b), respectively. Fig. 2(c) establishes a one-to-one correspondence between the norm and $\lambda$ for a fixed $p$ across varying $\lambda$ values. Fig. 2(d) examines the relationship between risk and norm under the same conditions. Training data $\{(\boldsymbol{x}_i, y_i)\}_{i\in[n]}$, $n = 100$, sampled from the model $y_i = \boldsymbol{g}_i^{\top}\boldsymbol{\theta}_* + \varepsilon_i$, $\sigma^2 = 0.04$, $\boldsymbol{g}_i \sim \mathcal{N}(0, \boldsymbol{I})$, $\boldsymbol{f}_i \sim \mathcal{N}(0, \boldsymbol{\Lambda})$, with $\xi_k^2(\boldsymbol{\Lambda}) = k^{-3/2}$ and $\boldsymbol{\theta}_{*,k} = k^{-1}$.

result matches with that for model size in Fig. 2(a); *ii)* in the over-parameterized regime, both bias and variance increase monotonically with norm. These findings reshape the traditional understanding of bias-variance trade-offs and double descent.

Since the self-consistent equation differs from under-parameterized to over-parameterized regimes, the learning curve plotted against the norm (see Fig. 1(b) and Fig. 2(d)) is not **single-valued** because of such phase transition: a single norm value may correspond to two distinct error levels in the under- and over-parameterized regimes. However, when analyzed separately, each regime exhibits a one-to-one relationship between test risk and norm. Notably, our analytical and empirical findings suggest that i) sufficient over-parameterization is always better than under-parameterization in terms of lower test risk, which also coincides with [53]. ii) More importantly, this curve aligns more with classical statistical intuition—a U-shaped curve—rather than the double descent phenomenon. We conclude that ***with suitably chosen model capacity, the learning curve more closely follows a U-shape than a double descent***. We conjecture that this behavior is universal in more complex models and real-world datasets; see Appendices H.2 and H.3 for details.

**Control the norm via regularization.** Norm-based capacity appears less intuitive used in practice when compared to model size. To control model norm, one can either fix the regularization parameter and vary the model size $p$ or fix $p$ and constrain the weight norm, The latter approach is mathematically equivalent to tuning the regularization parameter $\lambda$ in random feature ridge regression, as evidenced by the equivalence to the constrained optimization problem: $\min_{\boldsymbol{\beta}} \|\boldsymbol{y} - \boldsymbol{Z}\boldsymbol{a}\|^2$ s.t. $\|\boldsymbol{a}\|_2 = B$. This yields a ridge-type solution: $\hat{\boldsymbol{a}} = (\boldsymbol{Z}^{\top}\boldsymbol{Z} + \lambda\boldsymbol{I})^{-1}\boldsymbol{Z}^{\top}\boldsymbol{y}$ subject to $\|\hat{\boldsymbol{a}}\|_2 = B$, where $\lambda$ is uniquely determined by the norm constraint $B$ (with $\partial\|\hat{\boldsymbol{a}}\|_2^2/\partial\lambda < 0$ guaranteeing a one-to-one mapping). We empirically verified this in the random feature model by fixing the training sample size $n$ and ratio $\gamma$, and varying $\lambda$ to control the estimator norm. As shown in Fig. 2(c) (under-parameterized with $\gamma = 0.5$) and Fig. 2(d) (over-parameterized with $\gamma = 1.5$), the norm decreases monotonically with increasing $\lambda$, and in both under- and over-parameterized regimes, the test risk exhibits a U-shaped dependence on norm capacity, consistent with the known L-curve behavior [21]. Further discussion can be found in Appendix G.1.

### 4.2 Mathematical formulation of learning curves

Firstly, we show that the risk-norm relationship is **linear** in over-parameterized regime, see the proof in Appendix E.3.

**Proposition 4.1** (Linear learning curve). *The deterministic equivalents* $\mathsf{R}_0^{\texttt{RFM}}$ *and* $\mathsf{N}_0^{\texttt{RFM}}$, *in over-parameterized regimes ($p > n$) admit the linear relationship with the constant slope* $\lambda_n$

$$\mathsf{R}_0^{\texttt{RFM}} = \lambda_n \mathsf{N}_0^{\texttt{RFM}} + C_{\boldsymbol{\theta}_*, \boldsymbol{\Lambda}, n, \sigma}, \tag{5}$$

*where* $\lambda_n$ *satisfying* $\mathrm{Tr}(\boldsymbol{\Lambda}(\boldsymbol{\Lambda} + \lambda_n\boldsymbol{I})^{-1}) \sim n$ *and* $C_{\boldsymbol{\theta}_*, \boldsymbol{\Lambda}, n, \sigma}$ *are two constants independent of $p$ but dependent on* $\boldsymbol{\theta}_*, \boldsymbol{\Lambda}, n$, *and* $\sigma$, *as defined in Appendix E.3.*

**Remark:** Characterizing the relationship between risk and norm for ridge estimators ($\lambda > 0$) becomes particularly challenging. As shown in Eq. (3), the parameters $p$, $\lambda$, $\nu_1$, and $\nu_2$ are intricately coupled, making it extremely difficult to solve for $\nu_1$ and $\nu_2$—let alone derive an explicit (even approximate)

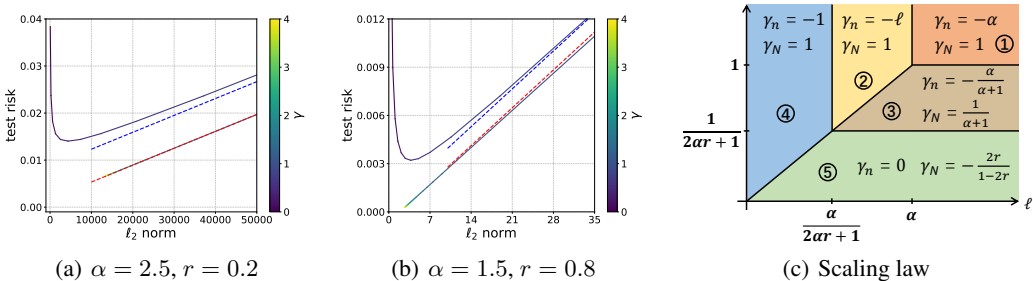

(a) $\alpha = 2.5, r = 0.2$      (b) $\alpha = 1.5, r = 0.8$      (c) Scaling law

Figure 3: **Fig. 3(a) and Fig. 3(b):** Validation of Corollary 4.2. The solid line represents the result of the deterministic equivalents, well approximated by the red dashed line of Eq. (6) in the over-parameterized regime, and the blue dashed line of Eq. (6) when $p \to n$ in the under-parameterized regime. **Fig. 3(c):** The value of exponents $\gamma_n$ and $\gamma_N$ in different regions (divided by $q$ and $\ell$) for $r \in (0, \frac{1}{2})$. Variance dominated region is colored by orange, yellow and brown, bias dominated region is colored by blue and green.

relationship between risk and norm. In the case of linear regression, a complete description of the risk-norm relationship under ridge regularization can be established, as presented in Appendix D.

The relationship in the under-parameterized regime is also complicated as well. We consider the special case of isotropic features in Corollary E.3 and give an approximation in Corollary 4.2 under the power-law assumption, given as below.

**Assumption 2** (Power-law, [14]). We assume that $\{\xi_k^2\}_{k=1}^{\infty}$ in $\mathbf{\Lambda}$ and $\boldsymbol{\theta}_*$ satisfy

$$\xi_k^2 = k^{-\alpha}, \quad \theta_{*,k} = k^{-\frac{1+2\alpha\tau}{2}}, \text{ with } \alpha > 1, \ r > 0.$$

The assumption coincides with the source condition $\|\mathbf{\Lambda}^{-r}\boldsymbol{\theta}_*\|_2 < \infty$ $(r > 0)$ and capacity condition $\mathrm{Tr}(\mathbf{\Lambda}^{1/\alpha}) < \infty$ $(\alpha > 1)$ [9]. Under power-law, we need to handle the self-consistent equations to approximate the infinite summation. We have the following approximation.

**Corollary 4.2** (Relationship for min-$\ell_2$ norm interpolator under power law). *Under Assumption 2, the deterministic equivalents $\mathsf{R}_0^{\mathrm{RFM}}$ and $\mathsf{N}_0^{\mathrm{RFM}}$ admit [3] the following relationship with $C_{n,\alpha,r,1} < C_{n,\alpha,r,2}$*

$$\mathsf{R}_0^{\mathrm{RFM}} \approx \left(n/C_\alpha\right)^{-\alpha} + \begin{cases} C_{n,\alpha,r,1} & \text{if } p > n, \\ C_{n,\alpha,r,2} & \text{if } p \to n^-. \end{cases} \tag{6}$$

*where $C_{n,\alpha,r,1(2)}$ are constants (see Appendix E.3 for details) that only depend on $n$, $\alpha$ and $r$. The notation $p \to n^-$ means that $p$ approaches to $n$ in the under-parameterized regime $(p < n)$.*

**Remark:** In the over-parameterized regime, the relationship between $\mathsf{R}_0^{\mathrm{RFM}}$ and $\mathsf{N}_0^{\mathrm{RFM}}$ is a monotonically increasing linear function, with a growth rate controlled by the factor decaying with $n$. In the under-parameterized regime, as $p \to n$ (which also leads to $\mathsf{R}_0^{\mathrm{RFM}}$ and $\mathsf{N}_0^{\mathrm{RFM}} \to \infty$), $\mathsf{R}_0^{\mathrm{RFM}}$ still grows linearly w.r.t $\mathsf{N}_0^{\mathrm{RFM}}$, with the same growth rate factor decaying with $n$. Furthermore, since $C_{n,\alpha,r,1} < C_{n,\alpha,r,2}$, the test risk curve shows that over-parameterization is better than under-parameterization. This approximation is also empirically verified to be precise in Fig. 3.

To study scaling law, we follow the same setting of [14] by choosing $p = n^q$ and $\lambda = n^{-(\ell-1)}$ with $q, l \geq 0$. We have the scaling law as below; see the proof in Appendix F.

**Proposition 4.3.** *Under Assumption 2, for $r \in (0, \frac{1}{2})$, taking $p = n^q$ and $\lambda = n^{-(\ell-1)}$ with $q, l \geq 0$, we formulate the scaling law under norm-based capacity in different areas as*

$$\mathsf{R}_\lambda^{\mathrm{RFM}} = \Theta\left(n^{\gamma_n} \cdot \left(\mathsf{N}_\lambda^{\mathrm{RFM}}\right)^{\gamma_N}\right), \quad \gamma_n \leq 0, \ \gamma_N \in \mathbb{R},$$

*where the rate $\{\gamma_n, \gamma_N\}$ in different areas is given in Fig. 3(c).*

---

[3]The symbol $\approx$ here denotes using an integral to approximate an infinite sum when calculating $\mathrm{Tr}(\cdot)$.

**Remark:** In all regions of Fig. 3(c), $\gamma_n \leq 0$, which aligns with the classical scaling law—that increasing the number of training samples leads to a reduction in test risk. As for $\gamma_N$, in regions ①, ②, ③, and ④, $\gamma_N > 0$, indicating that when $q$ is large (i.e., $p$ is large), the test risk increases monotonically with the norm. In contrast, in region ⑤, $\gamma_N < 0$, meaning that when $q$ is small (i.e., $p$ is small), the risk decreases monotonically with the norm. This again resembles the traditional U-shaped curve. These findings highlight the dual role of model norm in generalization: while a larger norm can be beneficial in low-complexity regimes, it becomes detrimental when the model is already sufficiently complex.

## 5    Conclusion and future work

This paper derives a precise characterization of the learning curve under the $\ell_2$-norm based capacity for both linear models and RFMs. It implies that, with suitably chosen model capacity, the learning curve more closely follows a U-shape than a double descent, and accordingly reshapes scaling laws. One limitation may be that the studied model is relatively simple, however, deterministic equivalence on complex models requires more exploration [13].

In future work, we will investigate the relationship between test risk and model complexity under (stochastic) gradient descent training. Leveraging recent advances in characterizing learning dynamics [45, 44, 7], we aim to precisely analyze the evolution of model norms and establish rigorous theoretical connections between norm dynamics and generalization behavior. Besides, our new deterministic quantities provide a possible way to study distribution shift and out-of-distribution (OOD) [46] with a precise estimation, which requires the deterministic equivalence of $\mathrm{Tr}(\boldsymbol{A}(\boldsymbol{X}^\top \boldsymbol{X} + \lambda \boldsymbol{I})^{-1} \boldsymbol{B}(\boldsymbol{X}^\top \boldsymbol{X} + \lambda \boldsymbol{I})^{-1})$ for two matrices $\boldsymbol{A}$ and $\boldsymbol{B}$.

## Acknowledgment

Y. C. was supported in part by National Science Foundation grants CCF-2233152. F. L. was supported by Royal Society KTP R1 241011 Kan Tong Po Visiting Fellowships and Warwick-SJTU seed fund. L. R. acknowledges the financial support of: the European Commission (Horizon Europe grant ELIAS 101120237), the Ministry of Education, University and Research (FARE grant ML4IP R205T7J2KP) the European Research Council (grant SLING 819789), the US Air Force Office of Scientific Research (FA8655-22-1-7034), the Ministry of Education, the grant BAC FAIR PE00000013 funded by the EU - NGEU and the MIUR grant (PRIN 202244A7YL). This work represents only the view of the authors. The European Commission and the other organizations are not responsible for any use that may be made of the information it contains. We thank Zulip[4] for the project organization tool, and Sulis[5] for GPU computation resources.

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

# Contents

# A   Notations

Table 2 summarizes the notations used throughout the main text and appendices.

Table 2: Core notations used the main text and appendix.

| Notation | Dimension(s) | Definition |
|---|---|---|
| $\mathcal{N}_\lambda^{\mathrm{LS}}$ | - | The $\ell_2$ norm of the linear regression estimator under regularization $\lambda$ for linear regression |
| $\mathcal{B}_{\mathcal{N},\lambda}^{\mathrm{LS}}$ | - | The bias of $\mathcal{N}_\lambda^{\mathrm{LS}}$ |
| $\mathcal{V}_{\mathcal{N},\lambda}^{\mathrm{LS}}$ | - | The variance of $\mathcal{N}_\lambda^{\mathrm{LS}}$ |
| $\mathsf{N}_\lambda^{\mathrm{LS}}$ | - | The deterministic equivalent of $\mathcal{N}_\lambda^{\mathrm{LS}}$ |
| $\mathsf{B}_{\mathsf{N},\lambda}^{\mathrm{LS}}$ | - | The deterministic equivalent of $\mathcal{B}_{\mathcal{N},\lambda}^{\mathrm{LS}}$ |
| $\mathsf{V}_{\mathsf{N},\lambda}^{\mathrm{LS}}$ | - | The deterministic equivalent of $\mathcal{V}_{\mathcal{N},\lambda}^{\mathrm{LS}}$ |
| $\|\boldsymbol{v}\|_2$ | - | Euclidean norms of vectors $\boldsymbol{v}$ |
| $\|\boldsymbol{v}\|_{\boldsymbol{\Sigma}}$ | - | $\sqrt{\boldsymbol{v}^\top \boldsymbol{\Sigma} \boldsymbol{v}}$ |
| $n$ | - | Number of training samples |
| $d$ | - | Dimension of the data for linear regression |
| $p$ | - | Number of features for random feature model |
| $\lambda$ | - | Regularization parameter |
| $\lambda_*$ | - | Effective regularization parameter for linear ridge regression |
| $\nu_1 , \nu_2$ | - | Effective regularization parameters for random feature ridge regression |
| $\sigma_k(\boldsymbol{M})$ | - | The $k$-th eigenvalue of $\boldsymbol{M}$ |
| $\boldsymbol{x}$ | $\mathbb{R}^d$ | The data vector |
| $\boldsymbol{X}$ | $\mathbb{R}^{n\times d}$ | The data matrix |
| $\boldsymbol{\Sigma}$ | $\mathbb{R}^{d\times d}$ | The covariance matrix of $\boldsymbol{x}$ |
| $y$ | $\mathbb{R}$ | The label |
| $\boldsymbol{y}$ | $\mathbb{R}^n$ | The label vector |
| $\boldsymbol{\beta}_*$ | $\mathbb{R}^d$ | The target function for linear regression |
| $\hat{\boldsymbol{\beta}}$ | $\mathbb{R}^d$ | The estimator of ridge regression model |
| $\hat{\boldsymbol{\beta}}_{\min}$ | $\mathbb{R}^d$ | The min-$\ell_2$-norm estimator of ridge regression model |
| $\varepsilon$ | $\mathbb{R}$ | The noise |
| $\varepsilon_i$ | $\mathbb{R}$ | The $i$-th noise |
| $\boldsymbol{\varepsilon}$ | $\mathbb{R}^n$ | The noise vector |
| $\sigma^2$ | $\mathbb{R}$ | The variance of the noise |
| $\boldsymbol{w}_i$ | $\mathbb{R}^d$ | The $i$-th weight vector for random feature model |
| $\varphi(\cdot;\cdot)$ | - | Nonlinear activation function for random feature model |
| $\boldsymbol{z}_i$ | $\mathbb{R}^p$ | The $i$-th feature for random feature model |
| $\boldsymbol{Z}$ | $\mathbb{R}^{n\times p}$ | Feature matrix for random feature model |
| $\hat{\boldsymbol{a}}$ | $\mathbb{R}^p$ | The estimator of random feature ridge regression model |
| $\hat{\boldsymbol{a}}_{\min}$ | $\mathbb{R}^p$ | The min-$\ell_2$-norm estimator of random feature ridge regression model |
| $f_*(\cdot)$ | - | The target function |
| $\mu_{\boldsymbol{x}}$ | - | The distribution of $\boldsymbol{x}$ |
| $\mu_{\boldsymbol{w}}$ | - | The distribution of $\boldsymbol{w}$ |
| $\mathbb{T}$ | - | An integral operator defined by $(\mathbb{T}f)(\boldsymbol{w}) := \int_{\mathbb{R}^d} \varphi(\boldsymbol{x};\boldsymbol{w}) f(\boldsymbol{x}) \mathrm{d}\mu_{\boldsymbol{x}}, \quad \forall f \in L_2(\mu_{\boldsymbol{x}})$ |
| $\mathcal{V}$ | - | The image of $\mathbb{T}$ |
| $\xi_k$ | $\mathbb{R}$ | The $k$-th eigenvalue of $\mathbb{T}$, defined by $\mathbb{T} = \sum_{k=1}^\infty \xi_k \psi_k \phi_k^*$ |
| $\psi_k$ | - | The $k$-th eigenfunction of $\mathbb{T}$ in the space $L_2(\mu_{\boldsymbol{x}})$, defined by the decomposition $\mathbb{T} = \sum_{k=1}^\infty \xi_k \psi_k \phi_k^*$ |
| $\phi_k$ | - | The $k$-th eigenfunction of $\mathbb{T}$ in the space $\mathcal{V}$, defined by the decomposition $\mathbb{T} = \sum_{k=1}^\infty \xi_k \psi_k \phi_k^*$ |
| $\boldsymbol{\Lambda}$ | $\mathbb{R}^{\infty\times\infty}$ | The spectral matrix of $\mathbb{T}$, $\boldsymbol{\Lambda} = \mathrm{diag}(\xi_1^2, \xi_2^2, \ldots) \in \mathbb{R}^{\infty\times\infty}$ |
| $\boldsymbol{g}_i$ | $\mathbb{R}^\infty$ | $\boldsymbol{g}_i := (\psi_k(\boldsymbol{x}_i))_{k\geq 1}$ |
| $\boldsymbol{f}_i$ | $\mathbb{R}^\infty$ | $\boldsymbol{f}_i := (\xi_k \phi_k(\boldsymbol{w}_i))_{k\geq 1}$ |
| $\boldsymbol{G}$ | $\mathbb{R}^{n\times\infty}$ | $\boldsymbol{G} := [\boldsymbol{g}_1,\ldots,\boldsymbol{g}_n]^\top \in \mathbb{R}^{n\times\infty}$ with $\boldsymbol{g}_i := (\psi_k(\boldsymbol{x}_i))_{k\geq 1}$ |
| $\boldsymbol{F}$ | $\mathbb{R}^{p\times\infty}$ | $\boldsymbol{F} := [\boldsymbol{f}_1,\ldots,\boldsymbol{f}_p]^\top \in \mathbb{R}^{p\times\infty}$ |
| $\widehat{\boldsymbol{\Lambda}}_{\boldsymbol{F}}$ | $\mathbb{R}^{p\times p}$ | $\widehat{\boldsymbol{\Lambda}}_{\boldsymbol{F}} := \mathbb{E}_{\boldsymbol{z}}[\boldsymbol{z}\boldsymbol{z}^\top|\boldsymbol{F}] = \frac{1}{p}\boldsymbol{F}\boldsymbol{F}^\top \in \mathbb{R}^{p\times p}$ |
| $\boldsymbol{\theta}_{*,k}$ | $\mathbb{R}$ | The coefficients associated with the eigenfunction $\psi_k$ in the expansion of $f_*(\boldsymbol{x}) = \sum_{k\geq 1} \boldsymbol{\theta}_{*,k}\psi_k(\boldsymbol{x})$ |
| $\boldsymbol{\theta}_*$ | $\mathbb{R}^\infty$ | $\boldsymbol{\theta}_* = (\boldsymbol{\theta}_{*,k})_{k\geq 1}$ |

[1] Replacing $\mathcal{N}$ with $\mathcal{R}$ (N with R), we get the notations associated to the test risk.
[2] Replacing $\lambda$ with 0, we get the notations associated to the min-$\ell_2$-norm solution.
[3] Replacing LS with RFM, we get the notations associated to random feature regression.

# B Preliminary and background

We provide an overview of the preliminary results used in this work. For self-contained completeness, we include results on asymptotic deterministic equivalence in Appendix B.1, results on ridge regression in Appendix B.2, and results on random feature ridge regression in Appendix B.3. Additionally, Appendix B.4 presents results on non-asymptotic deterministic equivalence, along with definitions of quantities required for these results. Finally, Appendix B.5 introduces key results for deriving the scaling law.

## B.1 Asymptotic deterministic equivalence

For the ease of description, we include preliminary results on asymptotic deterministic equivalence here. In fact, these assumptions and results can be recovered from non-asymptotic results, e.g., [37].

For linear regression, the asymptotic deterministic equivalence aim to find $\mathcal{B}_{\mathcal{R},\lambda}^{\mathsf{LS}} \sim \mathsf{B}_{\mathsf{R},\lambda}^{\mathsf{LS}}$, $\mathcal{V}_{\mathcal{R},\lambda}^{\mathsf{LS}} \sim \mathsf{V}_{\mathsf{R},\lambda}^{\mathsf{LS}}$, where $\mathsf{B}_{\mathsf{R},\lambda}^{\mathsf{LS}}$ and $\mathsf{V}_{\mathsf{R},\lambda}^{\mathsf{LS}}$ are some deterministic quantities. For asymptotic results, a series of assumptions in high-dimensional statistics via random matrix theory are required, on well-behaved data, spectral properties of $\boldsymbol{\Sigma}$ under nonlinear transformation in high-dimensional regime. We put the assumption from [1] here that are also widely used in previous literature [15, 51].

**Assumption 3.** [1, Well-behaved data] We assume that:

(A1) The sample size $n$ and dimension $d$ grow to infinity with $\frac{d}{n} \to \gamma > 0$.

(A2) $\boldsymbol{X} = \boldsymbol{T}\boldsymbol{\Sigma}^{1/2}$, where $\boldsymbol{T} \in \mathbb{R}^{n \times d}$ has i.i.d. sub-Gaussian entries with zero mean and unit variance.

(A3) $\boldsymbol{\Sigma}$ is invertible with $\|\boldsymbol{\Sigma}\|_{\mathrm{op}} < \infty$ and its spectral measure $\frac{1}{d}\sum_{i=1}^{d} \delta_{\sigma_i}$ converges to a compactly supported probability distribution $\mu$ on $\mathbb{R}^+$.

(A4) $\|\boldsymbol{\beta}_*\|_2 < \infty$ and the measure $\sum_{i=1}^{d}(\boldsymbol{v}_i^\top \boldsymbol{\beta}_*)^2 \delta_{\sigma_i}$ converges to a measure $\nu$ with bounded mass, where $\boldsymbol{v}_i$ is the unit-norm eigenvector of $\boldsymbol{\Sigma}$ related to its respective eigenvalue $\sigma_i$.

**Definition B.1** (Effective regularization)**.** For $n$, $\boldsymbol{\Sigma}$, and $\lambda \geq 0$, we define the *effective regularization* $\lambda_*$ to be the unique non-negative solution to the self-consistent equation

$$n - \frac{\lambda}{\lambda_*} \sim \mathrm{Tr}(\boldsymbol{\Sigma}(\boldsymbol{\Sigma} + \lambda_*)^{-1}). \tag{7}$$

**Definition B.2** (Degrees of freedom)**.**

$$\mathrm{df}_1(\lambda_*) := \mathrm{Tr}(\boldsymbol{\Sigma}(\boldsymbol{\Sigma} + \lambda_*)^{-1}), \quad \mathrm{df}_2(\lambda_*) := \mathrm{Tr}(\boldsymbol{\Sigma}^2(\boldsymbol{\Sigma} + \lambda_*)^{-2}).$$

**Proposition B.3.** *[1, Restatement of Proposition 1] Assume (A1), (A2), (A3), we consider $\boldsymbol{A}$ and $\boldsymbol{B}$ with bounded operator norm, admitting the convergence of the empirical measures, i.e., $\sum_{i=1}^{d} \boldsymbol{v}_i^\top \boldsymbol{A} \boldsymbol{v}_i \cdot \delta_{\sigma_i} \to \nu_A$ and $\sum_{i=1}^{d} \boldsymbol{v}_i^\top \boldsymbol{B} \boldsymbol{v}_i \cdot \delta_{\sigma_i} \to \nu_B$ with bounded total variation, respectively. Then, for $\lambda \geq 0$, with $\lambda_*$ satisfying Eq. (7), we have the following* **asymptotic deterministic equivalence**

$$\mathrm{Tr}(\boldsymbol{A}\boldsymbol{X}^\top \boldsymbol{X}(\boldsymbol{X}^\top \boldsymbol{X} + \lambda)^{-1}) \sim \mathrm{Tr}(\boldsymbol{A}\boldsymbol{\Sigma}(\boldsymbol{\Sigma} + \lambda_*)^{-1}), \tag{8}$$

$$\mathrm{Tr}(\boldsymbol{A}\boldsymbol{X}^\top \boldsymbol{X}(\boldsymbol{X}^\top \boldsymbol{X} + \lambda)^{-1}\boldsymbol{B}\boldsymbol{X}^\top \boldsymbol{X}(\boldsymbol{X}^\top \boldsymbol{X} + \lambda)^{-1}) \sim \mathrm{Tr}(\boldsymbol{A}\boldsymbol{\Sigma}(\boldsymbol{\Sigma} + \lambda_*)^{-1}\boldsymbol{B}\boldsymbol{\Sigma}(\boldsymbol{\Sigma} + \lambda_*)^{-1})$$
$$+ \lambda_*^2 \mathrm{Tr}(\boldsymbol{A}(\boldsymbol{\Sigma} + \lambda_*)^{-2}\boldsymbol{\Sigma}) \cdot \mathrm{Tr}(\boldsymbol{B}(\boldsymbol{\Sigma} + \lambda_*)^{-2}\boldsymbol{\Sigma}) \cdot \frac{1}{n - \mathrm{df}_2(\lambda_*)}, \tag{9}$$

$$\mathrm{Tr}(\boldsymbol{A}(\boldsymbol{X}^\top \boldsymbol{X} + \lambda)^{-1}) \sim \frac{\lambda_*}{\lambda} \mathrm{Tr}(\boldsymbol{A}(\boldsymbol{\Sigma} + \lambda_*)^{-1}), \tag{10}$$

$$\mathrm{Tr}(\boldsymbol{A}(\boldsymbol{X}^\top \boldsymbol{X} + \lambda)^{-1}\boldsymbol{B}(\boldsymbol{X}^\top \boldsymbol{X} + \lambda)^{-1}) \sim \frac{\lambda_*^2}{\lambda^2} \mathrm{Tr}(\boldsymbol{A}(\boldsymbol{\Sigma} + \lambda_*)^{-1}\boldsymbol{B}(\boldsymbol{\Sigma} + \lambda_*)^{-1})$$
$$+ \frac{\lambda_*^2}{\lambda^2} \mathrm{Tr}(\boldsymbol{A}(\boldsymbol{\Sigma} + \lambda_*)^{-2}\boldsymbol{\Sigma}) \cdot \mathrm{Tr}(\boldsymbol{B}(\boldsymbol{\Sigma} + \lambda_*)^{-2}\boldsymbol{\Sigma}) \cdot \frac{1}{n - \mathrm{df}_2(\lambda_*)}. \tag{11}$$

**Proposition B.4.** *[1, Restatement of Proposition 2] Assume **(A1)**, **(A2)**, **(A3)**, we consider $\boldsymbol{A}$ and $\boldsymbol{B}$ with bounded operator norm, admitting the convergence of the empirical measures, i.e., $\sum_{i=1}^{d} \boldsymbol{v}_i^\top \boldsymbol{A} \boldsymbol{v}_i \cdot \delta_{\sigma_i} \to \nu_A$ and $\sum_{i=1}^{d} \boldsymbol{v}_i^\top \boldsymbol{B} \boldsymbol{v}_i \cdot \delta_{\sigma_i} \to \nu_B$ with bounded total variation, respectively. Then, for $\lambda \in \mathbb{C} \backslash \mathbb{R}_+$, with $\lambda_*$ satisfying Eq. (7), we have the following **asymptotic deterministic equivalence***

$$\mathrm{Tr}(\boldsymbol{A}\boldsymbol{T}^\top(\boldsymbol{T}\boldsymbol{\Sigma}\boldsymbol{T}^\top + \lambda)^{-1}\boldsymbol{T}) \sim \mathrm{Tr}(\boldsymbol{A}(\boldsymbol{\Sigma} + \lambda_*)^{-1}), \tag{12}$$

$$\mathrm{Tr}(\boldsymbol{A}\boldsymbol{T}^\top(\boldsymbol{T}\boldsymbol{\Sigma}\boldsymbol{T}^\top + \lambda)^{-1}\boldsymbol{T}\boldsymbol{B}\boldsymbol{T}^\top(\boldsymbol{T}\boldsymbol{\Sigma}\boldsymbol{T}^\top + \lambda)^{-1}\boldsymbol{T}) \sim \mathrm{Tr}(\boldsymbol{A}(\boldsymbol{\Sigma} + \lambda_*)^{-1}\boldsymbol{B}(\boldsymbol{\Sigma} + \lambda_*)^{-1})$$

$$+ \lambda_*^2 \mathrm{Tr}(\boldsymbol{A}(\boldsymbol{\Sigma} + \lambda_*)^{-2}) \cdot \mathrm{Tr}(\boldsymbol{B}(\boldsymbol{\Sigma} + \lambda_*)^{-2}) \cdot \frac{1}{n - \mathrm{df}_2(\lambda_*)}. \tag{13}$$

Note that the results in Proposition B.3, B.4 still hold even for the random features model. We will explain this in details in Appendix E.

## B.2 Deterministic equivalence for ridge regression

We consider $n$ samples $\{\boldsymbol{x}_i\}_{i=1}^n$ sampled i.i.d. from a distribution $\mu_{\boldsymbol{x}}$ over $\mathbb{R}^d$ with covariance matrix $\boldsymbol{\Sigma} := \mathbb{E}[\boldsymbol{x}\boldsymbol{x}^\top] \in \mathbb{R}^{d \times d}$. The label $y_i$ is generated by a linear target function parameterized by $\boldsymbol{\beta}_* \in \mathbb{R}^d$, i.e., $y_i = \boldsymbol{x}_i^\top \boldsymbol{\beta}_* + \varepsilon_i$, where $\varepsilon_i$ is additive noise independent of $\boldsymbol{x}_i$ satisfying $\mathbb{E}[\varepsilon_i] = 0$ and $\mathrm{var}(\varepsilon_i) = \sigma^2$. We can write the model in a compact form as $\boldsymbol{y} = \boldsymbol{X}\boldsymbol{\beta}_* + \boldsymbol{\varepsilon}$, where the data matrix as $\boldsymbol{X} \in \mathbb{R}^{n \times d}$, the label vector $\boldsymbol{y} \in \mathbb{R}^n$, and the noise vector as $\boldsymbol{\varepsilon} \in \mathbb{R}^n$. The estimator of ridge regression is given by $\hat{\boldsymbol{\beta}} = (\boldsymbol{X}^\top \boldsymbol{X} + \lambda \boldsymbol{I})^{-1} \boldsymbol{X}^\top \boldsymbol{y}$. We also consider min-$\ell_2$-norm solution in the over-parameterized regime, i.e., $\hat{\boldsymbol{\beta}}_{\min} = \arg\min_{\boldsymbol{\beta}} \|\boldsymbol{\beta}\|_2$, s.t. $\boldsymbol{X}\boldsymbol{\beta} = \boldsymbol{y}$. The excess risk of $\hat{\boldsymbol{\beta}}$ admits a bias-variance decomposition

$$\mathcal{R}^{\mathrm{LS}} := \mathbb{E}_\varepsilon \|\boldsymbol{\beta}_* - \hat{\boldsymbol{\beta}}\|_{\boldsymbol{\Sigma}}^2 = \|\boldsymbol{\beta}_* - \mathbb{E}_\varepsilon[\hat{\boldsymbol{\beta}}]\|_{\boldsymbol{\Sigma}}^2 + \mathrm{Tr}(\boldsymbol{\Sigma}\mathrm{Cov}_\varepsilon(\hat{\boldsymbol{\beta}})),$$

where the first RHS term is the *bias*, denoted by $\mathcal{B}_{\mathcal{R},\lambda}^{\mathrm{LS}}$, and the second term is the *variance*, denoted by $\mathcal{V}_{\mathcal{R},\lambda}^{\mathrm{LS}}$. Accordingly, the bias-variance decomposition is given by

$$\mathcal{B}_{\mathcal{R},\lambda}^{\mathrm{LS}} := \|\boldsymbol{\beta}_* - \mathbb{E}_\varepsilon[\hat{\boldsymbol{\beta}}]\|_{\boldsymbol{\Sigma}}^2 = \lambda^2 \langle \boldsymbol{\beta}_*, (\boldsymbol{X}^\top \boldsymbol{X} + \lambda \boldsymbol{I})^{-1} \boldsymbol{\Sigma} (\boldsymbol{X}^\top \boldsymbol{X} + \lambda \boldsymbol{I})^{-1} \boldsymbol{\beta}_* \rangle, \tag{14}$$

$$\mathcal{V}_{\mathcal{R},\lambda}^{\mathrm{LS}} := \mathrm{Tr}\left(\boldsymbol{\Sigma}\mathrm{Cov}_\varepsilon(\hat{\boldsymbol{\beta}})\right) = \sigma^2 \mathrm{Tr}(\boldsymbol{\Sigma}\boldsymbol{X}^\top \boldsymbol{X} (\boldsymbol{X}^\top \boldsymbol{X} + \lambda \boldsymbol{I})^{-2}). \tag{15}$$

Under proper assumptions (to be detailed later), we have the following deterministic equivalents, asymptotically [1] and non-asymptotically [10]

$$\mathcal{B}_{\mathcal{R},\lambda}^{\mathrm{LS}} \sim \mathsf{B}_{\mathsf{R},\lambda}^{\mathrm{LS}} := \frac{\lambda_*^2 \langle \boldsymbol{\beta}_*, \boldsymbol{\Sigma}(\boldsymbol{\Sigma} + \lambda_* \boldsymbol{I})^{-2} \boldsymbol{\beta}_* \rangle}{1 - n^{-1} \mathrm{Tr}(\boldsymbol{\Sigma}^2(\boldsymbol{\Sigma} + \lambda_* \boldsymbol{I})^{-2})}, \quad \mathcal{V}_{\mathcal{R},\lambda}^{\mathrm{LS}} \sim \mathsf{V}_{\mathsf{R},\lambda}^{\mathrm{LS}} := \frac{\sigma^2 \mathrm{Tr}(\boldsymbol{\Sigma}^2(\boldsymbol{\Sigma} + \lambda_* \boldsymbol{I})^{-2})}{n - \mathrm{Tr}(\boldsymbol{\Sigma}^2(\boldsymbol{\Sigma} + \lambda_* \boldsymbol{I})^{-2})}, \tag{16}$$

where $\lambda_*$ is the non-negative solution to the self-consistent equation $n - \frac{\lambda}{\lambda_*} = \mathrm{Tr}(\boldsymbol{\Sigma}(\boldsymbol{\Sigma} + \lambda_* \boldsymbol{I})^{-1})$.

Accordingly, the risk admits the following deterministic equivalents via bias-variance decomposition.

**Proposition B.5.** *[1, Restatement of Proposition 3] Given the bias variance decomposition in Eq. (14) and Eq. (15), $\boldsymbol{X}$, $\boldsymbol{\Sigma}$ and $\boldsymbol{\beta}_*$ satisfy Assumption 3, we have the following asymptotic deterministic equivalents $\mathcal{R}_\lambda^{\mathrm{LS}} \sim \mathsf{R}_\lambda^{\mathrm{LS}} := \mathsf{B}_{\mathsf{R},\lambda}^{\mathrm{LS}} + \mathsf{V}_{\mathsf{R},\lambda}^{\mathrm{LS}}$ such that $\mathcal{B}_{\mathcal{R},\lambda}^{\mathrm{LS}} \sim \mathsf{B}_{\mathsf{R},\lambda}^{\mathrm{LS}}$, $\mathcal{V}_{\mathcal{R},\lambda}^{\mathrm{LS}} \sim \mathsf{V}_{\mathsf{R},\lambda}^{\mathrm{LS}}$, where $\mathsf{B}_{\mathsf{R},\lambda}^{\mathrm{LS}}$ and $\mathsf{V}_{\mathsf{R},\lambda}^{\mathrm{LS}}$ are defined by Eq. (16).*

**Proposition B.6.** *[1, Restatement of results in Sec 5] Under the same assumption as Proposition B.5, for the minimum $\ell_2$-norm estimator $\hat{\boldsymbol{\beta}}_{\min}$, we have for the under-parameterized regime ($d < n$):*

$$\mathcal{B}_{\mathcal{R},0}^{\mathrm{LS}} = 0, \quad \mathcal{V}_{\mathcal{R},0}^{\mathrm{LS}} \sim \sigma^2 \frac{d}{n - d}.$$

*In the over-parameterized regime ($d > n$), we have*

$$\mathcal{B}_{\mathcal{R},0}^{\mathrm{LS}} \sim \frac{\lambda_n^2 \langle \boldsymbol{\beta}_*, \boldsymbol{\Sigma}(\boldsymbol{\Sigma} + \lambda_n \boldsymbol{I})^{-2} \boldsymbol{\beta}_* \rangle}{1 - n^{-1} \mathrm{Tr}(\boldsymbol{\Sigma}^2(\boldsymbol{\Sigma} + \lambda_n \boldsymbol{I})^{-2})}, \quad \mathcal{V}_{\mathcal{R},0}^{\mathrm{LS}} \sim \frac{\sigma^2 \mathrm{Tr}(\boldsymbol{\Sigma}^2(\boldsymbol{\Sigma} + \lambda_n \boldsymbol{I})^{-2})}{n - \mathrm{Tr}(\boldsymbol{\Sigma}^2(\boldsymbol{\Sigma} + \lambda_n \boldsymbol{I})^{-2})},$$

*where $\lambda_n$ defined by $\mathrm{Tr}(\boldsymbol{\Sigma}(\boldsymbol{\Sigma} + \lambda_n \boldsymbol{I})^{-1}) \sim n$.*

## B.3 Deterministic equivalence for random feature ridge regression

Recall Eq. (1), the parameter $\boldsymbol{a}$ can be learned by the following empirical risk minimization with an $\ell_2$ regularization

$$\hat{\boldsymbol{a}} := \underset{\boldsymbol{a} \in \mathbb{R}^p}{\arg\min} \left\{ \sum_{i=1}^n \left( y_i - \frac{1}{\sqrt{p}} \sum_{j=1}^p \boldsymbol{a}_j \varphi(\boldsymbol{x}, \boldsymbol{w}_j) \right)^2 + \lambda \|\boldsymbol{a}\|_2^2 \right\} = (\boldsymbol{Z}^\top \boldsymbol{Z} + \lambda \boldsymbol{I})^{-1} \boldsymbol{Z}^\top \boldsymbol{y} \,.$$

Assuming that the target function $f_* \in L^2(\mu_{\boldsymbol{x}})$ admits $f_*(\boldsymbol{x}) = \sum_{k \geq 1} \boldsymbol{\theta}_{*,k} \psi_k(\boldsymbol{x})$, the excess risk $\mathcal{R}^{\mathrm{RFM}} := \mathbb{E}_\varepsilon \left\| \boldsymbol{\theta}_* - \frac{\boldsymbol{F}^\top \hat{\boldsymbol{a}}}{\sqrt{p}} \right\|_2^2$ admits the following bias-variance decomposition

$$\mathcal{B}_{\mathcal{R},\lambda}^{\mathrm{RFM}} := \left\| \boldsymbol{\theta}_* - \frac{\boldsymbol{F}^\top \mathbb{E}_\varepsilon[\hat{\boldsymbol{a}}]}{\sqrt{p}} \right\|_2^2 = \left\| \boldsymbol{\theta}_* - p^{-1/2} \boldsymbol{F}^\top (\boldsymbol{Z}^\top \boldsymbol{Z} + \lambda \boldsymbol{I})^{-1} \boldsymbol{Z}^\top \boldsymbol{G} \boldsymbol{\theta}_* \right\|_2^2 \,, \tag{17}$$

$$\mathcal{V}_{\mathcal{R},\lambda}^{\mathrm{RFM}} := \mathrm{Tr}\left( \widehat{\boldsymbol{\Lambda}}_{\boldsymbol{F}} \mathrm{Cov}_\varepsilon(\hat{\boldsymbol{a}}) \right) = \sigma^2 \mathrm{Tr}(\widehat{\boldsymbol{\Lambda}}_{\boldsymbol{F}} \boldsymbol{Z}^\top \boldsymbol{Z} (\boldsymbol{Z}^\top \boldsymbol{Z} + \lambda \boldsymbol{I})^{-2}) \,. \tag{18}$$

Accordingly, the risk admits the following deterministic equivalents via bias-variance decomposition.

**Proposition B.7.** *[14, Asymptotic version of Theorem 3.3] Given the bias variance decomposition in Eq. (17) and Eq. (18), under Assumption 1, we have the following asymptotic deterministic equivalents $\mathcal{R}_\lambda^{\mathrm{RFM}} \sim \mathsf{R}_\lambda^{\mathrm{RFM}} := \mathsf{B}_{\mathsf{R},\lambda}^{\mathrm{RFM}} + \mathsf{V}_{\mathsf{R},\lambda}^{\mathrm{RFM}}$ such that $\mathcal{B}_{\mathcal{R},\lambda}^{\mathrm{RFM}} \sim \mathsf{B}_{\mathsf{R},\lambda}^{\mathrm{RFM}}$, $\mathcal{V}_{\mathcal{R},\lambda}^{\mathrm{RFM}} \sim \mathsf{V}_{\mathsf{R},\lambda}^{\mathrm{RFM}}$, where $\mathsf{B}_{\mathsf{R},\lambda}^{\mathrm{RFM}}$ and $\mathsf{V}_{\mathsf{R},\lambda}^{\mathrm{RFM}}$ are defined by Eq. (2).*

Note that the above results are delivered in a non-asymptotic way [14], but more notations and technical assumptions are required. We give an overview of non-asymptotic deterministic equivalence as below.

## B.4 Non-asymptotic deterministic equivalence

Regarding non-asymptotic results, we require a series of notations and assumptions. We give a brief introduction here for self-completeness. More details can be found in [10, 37, 14].

Given $\boldsymbol{x} \in \mathbb{R}^d$ with $d \in \mathbb{N}$, the associated covariance matrix is given by $\boldsymbol{\Sigma} = \mathbb{E}[\boldsymbol{x}\boldsymbol{x}^\top]$. We denote the eigenvalue of $\boldsymbol{\Sigma}$ in non-increasing order as $\sigma_1 \geq \sigma_2 \geq \sigma_3 \geq \cdots \geq \sigma_d$.

We introduce the non-asymptotic version of Eq. (7) as below.

**Definition B.8** (Effective regularization). Given $n$, $\boldsymbol{\Sigma}$, and $\lambda \geq 0$, the *effective regularization* $\lambda_*$ is defined as the unique non-negative solution of the following self-consistent equation

$$n - \frac{\lambda}{\lambda_*} = \mathrm{Tr}\left( \boldsymbol{\Sigma} (\boldsymbol{\Sigma} + \lambda_*)^{-1} \right).$$

**Remark:** Existence and uniqueness of $\lambda_*$ are guaranteed since the left-hand side of the equation is monotonically increasing in $\lambda_*$, while the right-hand side is monotonically decreasing.

In the next, we introduce the following definitions on "effective dimension", a metric to describe the model capacity, widely used in statistical learning theory.

Define $r_{\boldsymbol{\Sigma}}(k) := \frac{\mathrm{Tr}(\boldsymbol{\Sigma}_{\geq k})}{\|\boldsymbol{\Sigma}_{\geq k}\|_{\mathrm{op}}} = \frac{\sum_{j=k}^d \sigma_j}{\sigma_k}$ as the intrinsic dimension, we require the following definition

$$\rho_\lambda(n) := 1 + \frac{n \sigma_{\lfloor \eta_* \cdot n \rfloor}}{\lambda} \left\{ 1 + \frac{r_{\boldsymbol{\Sigma}}(\lfloor \eta_* \cdot n \rfloor) \vee n}{n} \log \left( r_{\boldsymbol{\Sigma}}(\lfloor \eta_* \cdot n \rfloor) \vee n \right) \right\}, \tag{19}$$

where $\eta_* \in (0, 1/2)$ is a constant that will only depend on $C_*$ defined in Assumption 4. And we used the convention that $\sigma_{\lfloor \eta_* \cdot n \rfloor} = 0$ if $\lfloor \eta_* \cdot n \rfloor > d$.

In this section we consider functionals that depend on $\boldsymbol{X}$ and deterministic matrices. For a general PSD matrix $\boldsymbol{A} \in \mathbb{R}^{d \times d}$, define the functionals

$$\Phi_1(\boldsymbol{X}; \boldsymbol{A}, \lambda) := \text{Tr}\left(\boldsymbol{A}\boldsymbol{\Sigma}^{1/2}(\boldsymbol{X}^\top\boldsymbol{X} + \lambda)^{-1}\boldsymbol{\Sigma}^{1/2}\right), \tag{20}$$

$$\Phi_2(\boldsymbol{X}; \boldsymbol{A}, \lambda) := \text{Tr}\left(\boldsymbol{A}\boldsymbol{X}^\top\boldsymbol{X}(\boldsymbol{X}^\top\boldsymbol{X} + \lambda)^{-1}\right), \tag{21}$$

$$\Phi_3(\boldsymbol{X}; \boldsymbol{A}, \lambda) := \text{Tr}\left(\boldsymbol{A}\boldsymbol{\Sigma}^{1/2}(\boldsymbol{X}^\top\boldsymbol{X} + \lambda)^{-1}\boldsymbol{\Sigma}(\boldsymbol{X}^\top\boldsymbol{X} + \lambda)^{-1}\boldsymbol{\Sigma}^{1/2}\right), \tag{22}$$

$$\Phi_4(\boldsymbol{X}; \boldsymbol{A}, \lambda) := \text{Tr}\left(\boldsymbol{A}\boldsymbol{\Sigma}^{1/2}(\boldsymbol{X}^\top\boldsymbol{X} + \lambda)^{-1}\frac{\boldsymbol{X}^\top\boldsymbol{X}}{n}(\boldsymbol{X}^\top\boldsymbol{X} + \lambda)^{-1}\boldsymbol{\Sigma}^{1/2}\right). \tag{23}$$

These functionals can be approximated through quantities that scale proportionally to

$$\Psi_1(\lambda_*; \boldsymbol{A}) := \text{Tr}\left(\boldsymbol{A}\boldsymbol{\Sigma}(\boldsymbol{\Sigma} + \lambda_*\boldsymbol{I})^{-1}\right), \tag{24}$$

$$\Psi_2(\lambda_*; \boldsymbol{A}) := \frac{1}{n} \cdot \frac{\text{Tr}\left(\boldsymbol{A}\boldsymbol{\Sigma}^2(\boldsymbol{\Sigma} + \lambda_*\boldsymbol{I})^{-2}\right)}{n - \text{Tr}\left(\boldsymbol{\Sigma}^2(\boldsymbol{\Sigma} + \lambda_*\boldsymbol{I})^{-2}\right)}. \tag{25}$$

The following theorem gathers the approximation guarantees for the different functionals stated above, and is obtained by modifying [14, Theorem A.2]. We generalize Eq. (28) for any PSD matrix $\boldsymbol{A}$, which will be required for our results on the deterministic equivalence of $\ell_2$ norm. The proof can be found in Appendix C.

**Theorem B.9** (Dimension-free deterministic equivalents, Theorem A.2 of [14]). *Assume the features $\{\boldsymbol{x}_i\}_{i \in [n]}$ satisfy Assumption 4 with a constant $C_* > 0$. Then for any $D, K > 0$, there exist constants $\eta_* \in (0, 1/2)$, $C_{D,K} > 0$ and $C_{*,D,K} > 0$ ensuring the following property holds. For any $n \geq C_{D,K}$ and $\lambda > 0$, if the following condition is satisfied:*

$$\lambda \cdot \rho_\lambda(n) \geq \|\boldsymbol{\Sigma}\|_{\text{op}} \cdot n^{-K}, \quad \rho_\lambda(n)^{5/2}\log^{3/2}(n) \leq K\sqrt{n}, \tag{26}$$

*then for any PSD matrix $\boldsymbol{A}$, with probability at least $1 - n^{-D}$, we have that*

$$|\Phi_1(\boldsymbol{X}; \boldsymbol{A}, \lambda) - \frac{\lambda_*}{\lambda}\Psi_1(\lambda_*; \boldsymbol{A})| \leq C_{*,D,K}\frac{\rho_\lambda(n)^{5/2}\log^{3/2}(n)}{\sqrt{n}} \cdot \frac{\lambda_*}{\lambda}\Psi_1(\lambda_*; \boldsymbol{A}), \tag{27}$$

$$|\Phi_2(\boldsymbol{X}; \boldsymbol{I}, \lambda) - \Psi_1(\lambda_*; \boldsymbol{I})| \leq C_{*,D,K}\frac{\rho_\lambda(n)^4\log^{3/2}(n)}{\sqrt{n}}\Psi_1(\lambda_*; \boldsymbol{I}), \tag{28}$$

$$|\Phi_3(\boldsymbol{X}; \boldsymbol{A}, \lambda) - \left(\frac{n\lambda_*}{\lambda}\right)^2\Psi_2(\lambda_*; \boldsymbol{A})| \leq C_{*,D,K}\frac{\rho_\lambda(n)^6\log^{5/2}(n)}{\sqrt{n}} \cdot \left(\frac{n\lambda_*}{\lambda}\right)^2\Psi_2(\lambda_*; \boldsymbol{A}), \tag{29}$$

$$|\Phi_4(\boldsymbol{X}; \boldsymbol{A}, \lambda) - \Psi_2(\lambda_*; \boldsymbol{A})| \leq C_{*,D,K}\frac{\rho_\lambda(n)^6\log^{3/2}(n)}{\sqrt{n}}\Psi_2(\lambda_*; \boldsymbol{A}). \tag{30}$$

Next, we present some of the concepts to be used in deriving random feature ridge regression. Similar to how ridge regression depends on $\lambda_*$, as defined in Definition B.8, the deterministic equivalence of random feature ridge regression relies on $\nu_1$ and $\nu_2$, which are the solutions to the coupled equations

$$n - \frac{\lambda}{\nu_1} = \text{Tr}\left(\boldsymbol{\Lambda}(\boldsymbol{\Lambda} + \nu_2)^{-1}\right), \quad p - \frac{p\nu_1}{\nu_2} = \text{Tr}\left(\boldsymbol{\Lambda}(\boldsymbol{\Lambda} + \nu_2)^{-1}\right). \tag{31}$$

Writing $\nu_1$ as a function of $\nu_2$ produces the equations as below

$$1 + \frac{n}{p} - \sqrt{\left(1 - \frac{n}{p}\right)^2 + 4\frac{\lambda}{p\nu_2}} = \frac{2}{p}\text{Tr}\left(\boldsymbol{\Lambda}(\boldsymbol{\Lambda} + \nu_2)^{-1}\right), \nu_1 := \frac{\nu_2}{2}\left[1 - \frac{n}{p} + \sqrt{\left(1 - \frac{n}{p}\right)^2 + 4\frac{\lambda}{p\nu_2}}\right]. \tag{32}$$

For random features, our results also depend on the capacity of $\boldsymbol{\Lambda}$. Recall the definition of $r_{\boldsymbol{\Lambda}}(k) := \frac{\text{Tr}(\boldsymbol{\Lambda}_{\geq k})}{\|\boldsymbol{\Lambda}_{\geq k}\|_{\text{op}}}$ as the intrinsic dimension of $\boldsymbol{\Lambda}$ at level $k$, we sequentially define the following quantities that can be found in [37, 14].

$$M_{\boldsymbol{\Lambda}}(k) = 1 + \frac{r_{\boldsymbol{\Lambda}}(\lfloor \eta_* \cdot k \rfloor) \vee k}{k} \log\left( r_{\boldsymbol{\Lambda}}(\lfloor \eta_* \cdot k \rfloor) \vee k \right), \tag{33}$$

$$\rho_\kappa(p) = 1 + \frac{p \cdot \xi^2_{\lfloor \eta_* \cdot p \rfloor}}{\kappa} M_{\boldsymbol{\Lambda}}(p), \tag{34}$$

$$\widetilde{\rho}_\kappa(n, p) = 1 + \mathbb{1}\{n \leq p/\eta_*\} \cdot \left\{ \frac{n \xi^2_{\lfloor \eta_* \cdot n \rfloor}}{\kappa} + \frac{n}{p} \cdot \rho_\kappa(p) \right\} M_{\boldsymbol{\Lambda}}(n), \tag{35}$$

where the constant $\eta_* \in (0, 1/2)$ only depends on $C_*$ introduced in Assumption 1.

For an integer $\mathsf{m} \in \mathbb{N}$, we split the covariance matrix $\boldsymbol{\Lambda}$ into low degree part and high degree part as

$$\boldsymbol{\Lambda}_0 := \operatorname{diag}(\xi_1^2, \xi_2^2, \ldots, \xi_{\mathsf{m}}^2), \quad \boldsymbol{\Lambda}_+ := \operatorname{diag}(\xi_{\mathsf{m}+1}^2, \xi_{\mathsf{m}+2}^2, \ldots).$$

After we define the high degree feature covariance $\boldsymbol{\Lambda}_+$, we can define the function $\gamma(\kappa) := \kappa + \operatorname{Tr}(\boldsymbol{\Lambda}_+)$. To simplify the statement, we assume that we can choose $\mathsf{m}$ such that $p^2 \xi_{\mathsf{m}+1}^2 \leq \gamma(p\lambda/n)$, which is always satisfied under Assumption 1. For convenience, we will further denote

$$\gamma_+ := \gamma(p\nu_1), \qquad \gamma_\lambda := \gamma(p\lambda/n). \tag{36}$$

For random feature ridge regression, we will first demonstrate that the $\ell_2$ norm concentrates around a quantity that depends only on $\widehat{\boldsymbol{\Lambda}}_{\boldsymbol{F}}$. To this end, we define the following functionals with respect to $\boldsymbol{Z}$.

$$\Phi_3(\boldsymbol{Z}; \boldsymbol{A}, \kappa) := \operatorname{Tr}\left( \boldsymbol{A} \widehat{\boldsymbol{\Lambda}}_{\boldsymbol{F}}^{1/2} (\boldsymbol{Z}^\top \boldsymbol{Z} + \kappa)^{-1} \widehat{\boldsymbol{\Lambda}}_{\boldsymbol{F}} (\boldsymbol{Z}^\top \boldsymbol{Z} + \kappa)^{-1} \widehat{\boldsymbol{\Lambda}}_{\boldsymbol{F}}^{1/2} \right),$$

$$\Phi_4(\boldsymbol{Z}; \boldsymbol{A}, \kappa) := \operatorname{Tr}\left( \boldsymbol{A} \widehat{\boldsymbol{\Lambda}}_{\boldsymbol{F}}^{1/2} (\boldsymbol{Z}^\top \boldsymbol{Z} + \kappa)^{-1} \frac{\boldsymbol{Z}^\top \boldsymbol{Z}}{n} (\boldsymbol{Z}^\top \boldsymbol{Z} + \kappa)^{-1} \widehat{\boldsymbol{\Lambda}}_{\boldsymbol{F}}^{1/2} \right). \tag{37}$$

Given that $\boldsymbol{Z}$ consists of i.i.d. rows with covariance $\widehat{\boldsymbol{\Lambda}}_{\boldsymbol{F}} = \boldsymbol{F}\boldsymbol{F}^\top/p$, we will demonstrate that the aforementioned functionals can be approximated by those of $\boldsymbol{F}$, which, in turn, can be represented using the following functionals:

$$\widetilde{\Phi}_5(\boldsymbol{F}; \boldsymbol{A}, \kappa) := \frac{1}{n} \cdot \frac{\widetilde{\Phi}_6(\boldsymbol{F}; \boldsymbol{A}, \kappa)}{n - \widetilde{\Phi}_6(\boldsymbol{F}; \boldsymbol{I}, \kappa)},$$

$$\widetilde{\Phi}_6(\boldsymbol{F}; \boldsymbol{A}, \kappa) := \operatorname{Tr}\left( \boldsymbol{A}(\boldsymbol{F}\boldsymbol{F}^\top)^2 (\boldsymbol{F}\boldsymbol{F}^\top + \kappa)^{-2} \right). \tag{38}$$

**Proposition B.10** (Deterministic equivalents for $\Phi(\boldsymbol{Z})$ conditional on $\boldsymbol{F}$, Proposition B.6 of [14]). *Assume $\{\boldsymbol{z}_i\}_{i \in [n]}$ and $\{\boldsymbol{f}\}_{i \in [p]}$ satisfy Assumption 1 with a constant $C_* > 0$, and $\boldsymbol{F} \in \mathcal{A}_{\boldsymbol{F}}$ defined in [14, Eq. (79)]. Then for any $D, K > 0$, there exist constants $\eta_* \in (0, 1/2)$, $C_{D,K} > 0$ and $C_{*,D,K} > 0$ ensuring the following property holds. Let $\rho_\kappa(p)$ and $\tilde{\rho}_\kappa(n, p)$ be defined as per Eq. (34) and Eq. (35), $\gamma_+$ be defined as Eq. (36). For any $n \geq C_{D,K}$ and $\lambda > 0$, if the following condition is satisfied:*

$$\lambda \geq n^{-K}, \qquad \tilde{\rho}_\lambda(n, p)^{5/2} \log^{3/2}(n) \leq K\sqrt{n}, \qquad \tilde{\rho}_\lambda(n, p)^2 \cdot \rho_{\gamma_+}(p)^{5/2} \log^3(p) \leq K\sqrt{p},$$

*then for any PSD matrix $\boldsymbol{A} \in \mathbb{R}^{p \times p}$ (independent of $\boldsymbol{Z}$ conditional on $\boldsymbol{F}$), we have with probability at least $1 - n^{-D}$ that*

$$\left| \Phi_3(\boldsymbol{Z}; \boldsymbol{A}, \lambda) - \left(\frac{n\nu_1}{\lambda}\right)^2 \widetilde{\Phi}_5(\boldsymbol{F}; \boldsymbol{A}, p\nu_1) \right| \leq C_{*,D,K} \cdot \mathcal{E}_1(n, p) \cdot \left(\frac{n\nu_1}{\lambda}\right)^2 \widetilde{\Phi}_5(\boldsymbol{F}; \boldsymbol{A}, p\nu_1), \tag{39}$$

$$\left| \Phi_4(\boldsymbol{Z}; \boldsymbol{A}, \lambda) - \widetilde{\Phi}_5(\boldsymbol{F}; \boldsymbol{A}, p\nu_1) \right| \leq C_{*,D,K} \cdot \mathcal{E}_1(n, p) \cdot \widetilde{\Phi}_5(\boldsymbol{F}; \boldsymbol{A}, p\nu_1), \tag{40}$$

*where the rate $\mathcal{E}_1(n, p)$ is given by $\mathcal{E}_1(n, p) := \frac{\tilde{\rho}_\lambda(n,p)^6 \log^{5/2}(n)}{\sqrt{n}} + \frac{\tilde{\rho}_\lambda(n,p)^2 \cdot \rho_{\gamma_+}(p)^{5/2} \log^3(p)}{\sqrt{p}}.$*

## B.5 Scaling law

For the derivation of the scaling law, we use the results in [14, Appendix D]. We define $T^s_{\delta,\gamma}(\nu)$ as

$$T^s_{\delta,\gamma}(\nu) := \sum_{k=1}^{\infty} \frac{k^{-s-\delta\alpha}}{(k^{-\alpha} + \nu)^{\gamma}}, \quad s \in 0, 1, \ 0 \leq \delta \leq \gamma.$$

Under Assumption 2, according to [14, Appendix D], we have the following results

$$T^s_{\delta\gamma}(\nu) = O\left(\nu^{1/\alpha[s-1+\alpha(\delta-\gamma)]\wedge 0}\right). \tag{41}$$

Next, we present some rates of the quantities used in the deterministic equivalence of random feature ridge regression. The rate of $\nu_2$ is given by

$$\nu_2 \approx O\left(n^{-\alpha(1\wedge q\wedge \ell/\alpha)}\right), \tag{42}$$

and in particular, for $\Upsilon(\nu_1, \nu_2)$ and $\chi(\nu_2)$, we have

$$1 - \Upsilon(\nu_1, \nu_2) = O(1), \tag{43}$$

$$\chi(\nu_2) = n^{-q}O\left(\nu_2^{-1-1/\alpha}\right). \tag{44}$$

## C  Proofs on additional non-asymptotic deterministic equivalents

In this section, we aim to generalize Eq. (28) for any PSD matrix $\boldsymbol{A}$, i.e.

$$\left|\Phi_2(\boldsymbol{X}; \boldsymbol{A}) - \Psi_2(\mu_*; \boldsymbol{A})\right| \leq \widetilde{\mathcal{O}}(n^{-\frac{1}{2}}) \cdot \Psi_2(\mu_*; \boldsymbol{A}),$$

that is required to derive our non-asymptotic deterministic equivalence for the bias term of the $\ell_2$ norm.

By introducing a change of variable $\mu_* := \mu_*(\lambda) = \lambda/\lambda_*$, we find that $\mu_*$ satisfies the following fixed-point equation:

$$\mu_* = \frac{n}{1 + \mathrm{Tr}(\boldsymbol{\Sigma}(\mu_*\boldsymbol{\Sigma} + \lambda)^{-1})}. \tag{45}$$

We define $\boldsymbol{t}$ and $\boldsymbol{T}$ as follows

$$\boldsymbol{t} = \boldsymbol{\Sigma}^{-1/2}\boldsymbol{x}, \quad \boldsymbol{T} = \boldsymbol{X}\boldsymbol{\Sigma}^{-1/2}.$$

And the following resolvents are also defined

$$\boldsymbol{R} := (\boldsymbol{X}^{\top}\boldsymbol{X} + \lambda)^{-1}, \quad \overline{\boldsymbol{R}} := (\mu_*\boldsymbol{\Sigma} + \lambda)^{-1}, \quad \boldsymbol{M} := \boldsymbol{\Sigma}^{1/2}\boldsymbol{R}\boldsymbol{\Sigma}^{1/2}, \quad \overline{\boldsymbol{M}} := \boldsymbol{\Sigma}^{1/2}\overline{\boldsymbol{R}}\boldsymbol{\Sigma}^{1/2}.$$

Since the proof relies on a leave-one-out argument, we define $\boldsymbol{X}_- \in \mathbb{R}^{(n-1)\times d}$ as the data matrix obtained by removing one data. We also introduce the associated resolvent and rescaled resolvent:

$$\boldsymbol{R}_- := (\boldsymbol{X}_-^{\top}\boldsymbol{X}_- + \lambda)^{-1}, \overline{\boldsymbol{R}}_- := \left(\frac{n}{1+\kappa}\boldsymbol{\Sigma} + \lambda\right)^{-1}, \boldsymbol{M}_- := \boldsymbol{\Sigma}^{1/2}\boldsymbol{R}_-\boldsymbol{\Sigma}^{1/2}, \overline{\boldsymbol{M}}_- := \boldsymbol{\Sigma}^{1/2}\overline{\boldsymbol{R}}_-\boldsymbol{\Sigma}^{1/2},$$

where $\kappa = \mathbb{E}[\mathrm{Tr}(\boldsymbol{M}_-)]$.

For the sake of narrative convenience, we introduce a functional used in [37]

$$\Psi_1(\mu_*; \boldsymbol{A}) := \mathrm{Tr}(\boldsymbol{A}\boldsymbol{\Sigma}(\mu_*\boldsymbol{\Sigma} + \lambda)^{-1}).$$

Next, we give the proof of Eq. (28). We consider the functional

$$\Phi_2(\boldsymbol{X}; \boldsymbol{A}) = \mathrm{Tr}(\boldsymbol{A}\boldsymbol{\Sigma}^{-1/2}\boldsymbol{X}^{\top}\boldsymbol{X}(\boldsymbol{X}^{\top}\boldsymbol{X} + \lambda)^{-1}\boldsymbol{\Sigma}^{1/2}) = \mathrm{Tr}(\boldsymbol{A}\boldsymbol{T}^{\top}\boldsymbol{T}\boldsymbol{M}).$$

**Remark:** Note that, to align more closely with the proof in [37], the $\Phi_2(\boldsymbol{X}; \boldsymbol{A})$ defined here differs slightly from the $\Phi_2(\boldsymbol{X}; \boldsymbol{A}, \lambda)$ in Eq. (28). However, the two definitions are equivalent if we take $\boldsymbol{A}$ here as $\boldsymbol{A} = \boldsymbol{\Sigma}^{-1/2}\boldsymbol{B}\boldsymbol{\Sigma}^{1/2}$, which recovers the formulation in Eq. (28).

We show that $\Phi_2(\boldsymbol{X}; \boldsymbol{A})$ is well approximated by the following deterministic equivalent:

$$\Psi_2(\mu_*; \boldsymbol{A}) = \mathrm{Tr}(\boldsymbol{A}\mu_*\boldsymbol{\Sigma}(\mu_*\boldsymbol{\Sigma} + \lambda)^{-1}) = \mathrm{Tr}(\boldsymbol{A}\boldsymbol{\Sigma}(\boldsymbol{\Sigma} + \lambda_*)^{-1}).$$

**Theorem C.1** (Deterministic equivalent for $\mathrm{Tr}(\boldsymbol{A}\boldsymbol{T}^\top \boldsymbol{T}\boldsymbol{M})$). *Assume the features $\{\boldsymbol{x}_i\}_{i \in [n]}$ satisfy Assumption 4 with a constant $C_* > 0$. Then for any $D, K > 0$, there exist constants $\eta \in (0, 1/2)$, $C_{D,K} > 0$, and $C_{*,D,K} > 0$ ensuring the following property holds. For any $n \geq C_{D,K}$ and $\lambda > 0$, if the following condition is satisfied:*

$$\lambda \cdot \rho_\lambda(n) \geq n^{-K}, \qquad \rho_\lambda(n)^2 \log^{\frac{3}{2}}(n) \leq K\sqrt{n}, \tag{46}$$

*then for any PSD matrix $\boldsymbol{A}$, with probability at least $1 - n^{-D}$, we have that*

$$\left| \Phi_2(\boldsymbol{X}; \boldsymbol{A}) - \Psi_2(\mu_*; \boldsymbol{A}) \right| \leq C_{*,D,K} \frac{\rho_\lambda(n)^4 \log^{\frac{3}{2}}(n)}{\sqrt{n}} \Psi_2(\mu_*; \boldsymbol{A}). \tag{47}$$

**Remark:** Theorem C.1 generalizes Eq. (28). Note that there are some differences between $\rho_\lambda$ as defined in Eq. (19) and $\nu_\lambda$ as defined in [37]. However, based on the discussion in [14, Appendix A], $\nu_\lambda$ can be easily adjusted to match $\rho_\lambda$. Therefore, while we follow the argument in [37], we use $\rho_\lambda$ directly in this work to minimize additional notation.

Following the approach outlined in [37], our proof involves separately bounding the deterministic and martingale components. This is accomplished in the following two propositions.

**Proposition C.2** (Deterministic part of $\mathrm{Tr}(\boldsymbol{A}\boldsymbol{T}^\top \boldsymbol{T}\boldsymbol{M})$). *Under the same assumption as Theorem C.1, there exist constants $C_K$ and $C_{*,K}$, such that for all $n \geq C_K$ and $\lambda > 0$ satisfying Eq. (46), and for any PSD matrix $\boldsymbol{A}$, we have*

$$\left| \mathbb{E}[\Phi_2(\boldsymbol{X}; \boldsymbol{A})] - \Psi_2(\mu_*; \boldsymbol{A}) \right| \leq C_{*,K} \frac{\rho_\lambda(n)^4}{\sqrt{n}} \Psi_2(\mu_*; \boldsymbol{A}). \tag{48}$$

**Proposition C.3** (Martingale part of $\mathrm{Tr}(\boldsymbol{A}\boldsymbol{T}^\top \boldsymbol{T}\boldsymbol{M})$). *Under the same assumption as Theorem C.1, there exist constants $C_{K,D}$ and $C_{*,D,K}$, such that for all $n \geq C_{K,D}$ and $\lambda > 0$ satisfying Eq. (46), and for any PSD matrix $\boldsymbol{A}$, we have with probability at least $1 - n^{-D}$ that*

$$\left| \Phi_2(\boldsymbol{X}; \boldsymbol{A}) - \mathbb{E}[\Phi_2(\boldsymbol{X}; \boldsymbol{A})] \right| \leq C_{*,D,K} \frac{\rho_\lambda(n)^3 \log^{\frac{3}{2}}(n)}{\sqrt{n}} \Psi_2(\mu_*; \boldsymbol{A}). \tag{49}$$

Theorem C.1 is obtained by combining the bounds (48) and (49). Next, we prove the two propositions above separately.

*Proof of Proposition C.2.* First, by Sherman-Morrison identity

$$\boldsymbol{M} = \boldsymbol{M}_- - \frac{\boldsymbol{M}_- \boldsymbol{t} \boldsymbol{t}^\top \boldsymbol{M}_-}{1 + \boldsymbol{t}^\top \boldsymbol{M}_- \boldsymbol{t}}, \quad \text{and} \quad \boldsymbol{M}\boldsymbol{t} = \frac{\boldsymbol{M}_- \boldsymbol{t}}{1 + \boldsymbol{t}^\top \boldsymbol{M}_- \boldsymbol{t}},$$

we decompose $\mathbb{E}[\Phi_2(\boldsymbol{X}; \boldsymbol{A})]$ as

$$\mathbb{E}\left[\mathrm{Tr}(\boldsymbol{A}\boldsymbol{T}^\top \boldsymbol{T}\boldsymbol{M})\right] = n\mathbb{E}\left[\frac{\boldsymbol{t}^\top \boldsymbol{M}_- \boldsymbol{A}\boldsymbol{t}}{1 + S}\right]$$

$$= n\frac{\mathbb{E}[\mathrm{Tr}(\boldsymbol{A}\boldsymbol{M}_-)]}{1 + \kappa} + n\mathbb{E}\left[\frac{\kappa - S}{(1 + \kappa)(1 + S)} \boldsymbol{t}^\top \boldsymbol{M}_- \boldsymbol{A}\boldsymbol{t}\right],$$

where we denoted $S = \boldsymbol{t}^\top \boldsymbol{M}_- \boldsymbol{t}$. Therefore, bounding the following two terms is sufficient

$$
\begin{aligned}
&\left| \mathbb{E}[\Phi_2(\boldsymbol{X}; \boldsymbol{A})] - \Psi_2(\mu_*; \boldsymbol{A}) \right| \\
&\leq \left| \frac{n\mathbb{E}[\mathrm{Tr}(\boldsymbol{A}\boldsymbol{M}_-)]}{1 + \kappa} - \Psi_2(\mu_*; \boldsymbol{A}) \right| + \left| n\mathbb{E}\left[ \frac{\kappa - S}{(1 + \kappa)(1 + S)} \boldsymbol{t}^\top \boldsymbol{M}_- \boldsymbol{A}\boldsymbol{t} \right] \right|.
\end{aligned}
\tag{50}
$$

For the first term, recall that $\tilde{\mu}_*$ is the solution of the equation (45) where we replaced $n$ by $n - 1$, and $\tilde{\mu}_- := n/(1 + \kappa)$. By [37, Proposition 2], we have

$$\left| \mathbb{E}[\mathrm{Tr}(\boldsymbol{A}\boldsymbol{M}_-)] - \Psi_1(\tilde{\mu}_*; \boldsymbol{A}) \right| \leq \mathcal{E}_{1,n-1}^{(\mathsf{D})} \cdot \Psi_1(\tilde{\mu}_*; \boldsymbol{A}),$$

where $\mathcal{E}_{1,n-1}^{(\mathrm{D})} = C_{*,K}\frac{\rho_\lambda(n)^{5/2}}{\sqrt{n-1}}$. For $n \geq C$, we have $\mathcal{E}_{1,n-1}^{(\mathrm{D})} \leq C\mathcal{E}_{1,n}^{(\mathrm{D})}$ and by [37, Lemma 3], we have

$$|\Psi_1(\tilde{\mu}_*; \boldsymbol{A}) - \Psi_1(\mu_*; \boldsymbol{A})| \leq C\frac{\rho_\lambda(n)}{n}\Psi_1(\mu_*; \boldsymbol{A}).$$

Combining the above bounds, we obtain

$$|\mathbb{E}[\mathrm{Tr}(\boldsymbol{A}\boldsymbol{M}_-)] - \Psi_1(\mu_*; \boldsymbol{A})| \leq \mathcal{E}_{1,n}^{(\mathrm{D})} \cdot \Psi_1(\mu_*; \boldsymbol{A}).$$

Furthermore, from the proof of [37, Proposition 4, Claim 3], we have

$$\frac{|\mu_* - \tilde{\mu}_-|}{\tilde{\mu}_-} \leq C_{*,K}\frac{\rho_\lambda(n)^{5/2}}{\sqrt{n}}.$$

Then we conclude that

$$\begin{aligned}
|\mu_* - \tilde{\mu}_-| &\leq C_{*,K}\frac{\rho_\lambda(n)^{5/2}}{\sqrt{n}} \cdot \tilde{\mu}_- \\
&\leq C_{*,K}\frac{\rho_\lambda(n)^{5/2}}{\sqrt{n}} \cdot \left(1 + C_{*,K}\frac{\rho_\lambda(n)^{5/2}}{\sqrt{n}}\right)\mu_* \\
&\leq C_{*,K}\frac{\rho_\lambda(n)^{5/2}}{\sqrt{n}} \cdot \mu_*,
\end{aligned}$$

where we use condition (46) in the last inequality.

Combining this inequality with the previous bounds, we obtain

$$\begin{aligned}
\left|\frac{n\mathbb{E}[\mathrm{Tr}(\boldsymbol{A}\boldsymbol{M}_-)]}{1+\kappa} - \Psi_2(\mu_*; \boldsymbol{A})\right| &= |\tilde{\mu}_-\mathbb{E}[\mathrm{Tr}(\boldsymbol{A}\boldsymbol{M}_-)] - \mu_*\Psi_1(\mu_*; \boldsymbol{A})| \\
&\leq \tilde{\mu}_-\,|\mathbb{E}[\mathrm{Tr}(\boldsymbol{A}\boldsymbol{M}_-)] - \Psi_1(\mu_*; \boldsymbol{A})| + \frac{|\tilde{\mu}_- - \mu_*|}{\mu_*} \cdot \mu_*\Psi_1(\mu_*; \boldsymbol{A}) \\
&\leq C\mathcal{E}_{1,n}^{(\mathrm{D})} \cdot \mu_*\Psi_1(\mu_*; \boldsymbol{A}) \\
&= C\mathcal{E}_{1,n}^{(\mathrm{D})} \cdot \Psi_2(\mu_*; \boldsymbol{A}).
\end{aligned}$$

In the next, we aim to estimate the second term in Eq. (50). Here we can reduce $\boldsymbol{A}$ to be a rank-one matrix $\boldsymbol{A} := \boldsymbol{v}\boldsymbol{v}^\top$ following [37, Eq. (77)]. We simply apply Hölder's inequality and obtain

$$\begin{aligned}
&n\left|\mathbb{E}\left[\frac{\kappa - S}{(1+\kappa)(1+S)}\boldsymbol{t}^\top\boldsymbol{M}_-\boldsymbol{A}\boldsymbol{t}\right]\right| \\
&= n\mathbb{E}\left[\left|\frac{\kappa - S}{(1+\kappa)(1+S)}\boldsymbol{t}^\top\boldsymbol{M}_-\boldsymbol{v}\boldsymbol{v}^\top\boldsymbol{t}\right|\right] \\
&\leq n\mathbb{E}_{\boldsymbol{M}_-}\left[\mathbb{E}_{\boldsymbol{t}}\left[(\kappa - S)^2\right]^{1/2}\mathbb{E}_{\boldsymbol{t}}\left[(\boldsymbol{t}^\top\boldsymbol{M}_-\boldsymbol{v}\boldsymbol{v}^\top\boldsymbol{t})^2\right]^{1/2}\right] \\
&\leq n\mathbb{E}_{\boldsymbol{M}_-}\left[\mathbb{E}_{\boldsymbol{t}}\left[(\kappa - S)^2\right]\right]^{1/2}\mathbb{E}_{\boldsymbol{M}_-}\left[\mathbb{E}_{\boldsymbol{t}}\left[(\boldsymbol{t}^\top\boldsymbol{M}_-\boldsymbol{v}\boldsymbol{v}^\top\boldsymbol{t})^2\right]\right]^{1/2} \\
&\leq n\mathbb{E}_{\boldsymbol{M}_-}\left[\mathbb{E}_{\boldsymbol{t}}\left[(\kappa - S)^2\right]\right]^{1/2}\mathbb{E}_{\boldsymbol{M}_-}\left[\mathbb{E}_{\boldsymbol{t}}\left[(\boldsymbol{t}^\top\boldsymbol{M}_-\boldsymbol{v})^4\right]^{1/2}\mathbb{E}_{\boldsymbol{t}}\left[(\boldsymbol{v}^\top\boldsymbol{t})^4\right]^{1/2}\right]^{1/2}.
\end{aligned}$$

Each of these terms can be bounded, according to the proof of [37, Proposition 2], for the first term, we get

$$\mathbb{E}_{\boldsymbol{M}_-}\left[\mathbb{E}_{\boldsymbol{t}}\left[(\boldsymbol{t}^\top\boldsymbol{M}_-\boldsymbol{t} - \kappa)^2\right]\right]^{1/2} \leq C_{*,K}\frac{\rho_\lambda(n)}{\sqrt{n}}.$$

For the second term, first according to [37, Lemma 2], we have

$$\mathbb{E}_{\boldsymbol{t}}\left[(\boldsymbol{t}^\top\boldsymbol{M}_-\boldsymbol{v})^4\right]^{1/2} \leq C_{*,K}\boldsymbol{v}^\top\boldsymbol{M}_-^2\boldsymbol{v},$$

$$\mathbb{E}_{\boldsymbol{t}}\left[(\boldsymbol{v}^\top\boldsymbol{t})^4\right]^{1/2} \leq C_{*,K}\boldsymbol{v}^\top\boldsymbol{v}.$$

Thus we have

$$\mathbb{E}_{\boldsymbol{M}_-}\left[\mathbb{E}_{\boldsymbol{t}}\left[(\boldsymbol{t}^\top \boldsymbol{M}_-\boldsymbol{v})^4\right]^{1/2}\mathbb{E}_{\boldsymbol{t}}\left[(\boldsymbol{v}^\top \boldsymbol{t})^4\right]^{1/2}\right]^{1/2}\leq \mathbb{E}_{\boldsymbol{M}_-}\left[C_{*,K}\boldsymbol{v}^\top \boldsymbol{M}_-^2\boldsymbol{v}\boldsymbol{v}^\top \boldsymbol{v}\right]^{1/2}$$

$$= C_{*,K}\mathbb{E}_{\boldsymbol{M}_-}\left[\mathrm{Tr}(\boldsymbol{A}\boldsymbol{M}_-^2\boldsymbol{A})\right]^{1/2}.$$

Then according to [37, Lemma 4.(b)], we have

$$\mathbb{E}_{\boldsymbol{M}_-}\left[\mathrm{Tr}(\boldsymbol{A}\boldsymbol{M}_-^2\boldsymbol{A})\right]\leq C_{*,K}\rho_\lambda^2(n)\mathrm{Tr}(\boldsymbol{A}\overline{\boldsymbol{M}}_-^2\boldsymbol{A})=C_{*,K}\rho_\lambda^2(n)\mathrm{Tr}(\boldsymbol{A}\overline{\boldsymbol{M}}_-)^2,$$

where the last inequality holds due to $\boldsymbol{A}\overline{\boldsymbol{M}}_-$ being a rank-1 matrix. Combining the bounds for the second term, we have

$$\mathbb{E}_{\boldsymbol{M}_-}\left[\mathbb{E}_{\boldsymbol{t}}\left[(\boldsymbol{t}^\top \boldsymbol{M}_-\boldsymbol{v})^4\right]^{1/2}\mathbb{E}_{\boldsymbol{t}}\left[(\boldsymbol{v}^\top \boldsymbol{t})^4\right]^{1/2}\right]^{1/2}\leq C_{*,K}\rho_\lambda(n)\mathrm{Tr}(\boldsymbol{A}\overline{\boldsymbol{M}}_-)\leq C_{*,K}\rho_\lambda^2(n)\mathrm{Tr}(\boldsymbol{A}\overline{\boldsymbol{M}}).$$

By combining the above bounds for the first and second term, we have

$$n\left|\mathbb{E}\left[\frac{\kappa-S}{(1+\kappa)(1+S)}\boldsymbol{T}^\top \boldsymbol{M}_-\boldsymbol{A}\boldsymbol{T}\right]\right|\leq C_{*,K}\frac{\rho_\lambda^3(n)}{\sqrt{n}}n\mathrm{Tr}(\boldsymbol{A}\overline{\boldsymbol{M}})$$

$$\leq C_{*,K}\frac{\rho_\lambda^4(n)}{\sqrt{n}}\mu_*\mathrm{Tr}(\boldsymbol{A}\overline{\boldsymbol{M}}),$$

where we use $\mu_*=\frac{n}{1+\mathrm{Tr}(\overline{\boldsymbol{M}})}\geq \frac{n}{2\rho_\lambda(n)}$ according to [37, Lemma 3] in the last inequality.

Combining the above bounds concludes the proof. $\qquad\square$

*Proof of Proposition Proposition C.3.* The martingale argument follows a similar approach to the proofs of [37, Propositions 3 and 5]. The key remaining steps are to adjust Step 2 in [37, Proposition 3] and establish high-probability bounds for each term in the martingale difference sequence.

We rewrite this term as a martingale difference sequence

$$S_n := \mathrm{Tr}(\boldsymbol{A}\boldsymbol{T}^\top \boldsymbol{T}\boldsymbol{M})-\mathbb{E}[\mathrm{Tr}(\boldsymbol{A}\boldsymbol{T}^\top \boldsymbol{T}\boldsymbol{M})]=\sum_{i=1}^n(\mathbb{E}_i-\mathbb{E}_{i-1})\,\mathrm{Tr}(\boldsymbol{A}\boldsymbol{T}^\top \boldsymbol{T}\boldsymbol{M})=:\sum_{i=1}^n\Delta_i\,,$$

where $\mathbb{E}_i$ is denoted as the expectation over $\{\boldsymbol{x}_{i+1},\cdots,\boldsymbol{x}_n\}$.

We show below that $|\Delta_i|\leq R$ with probability at least $1-n^{-D}$ with

$$R=C_{*,D,K}\frac{\rho_\lambda(n)^2\log(n)}{n}\Psi_2(\mu_*;\boldsymbol{A}). \tag{51}$$

For Step 3 and bounding $\mathbb{E}_{i-1}[\Delta_i\mathbb{1}_{\Delta_i\not\in[-R,R]}]$, observe that with probability at least $1-n^{-D}$, by [37, Lemma 4.(b)]

$$\mathbb{E}_{i-1}[\Delta_i^2]^{1/2}\leq 2\mathbb{E}_{i-1}\left[\frac{(\boldsymbol{t}^\top \boldsymbol{M}_-\boldsymbol{A}\boldsymbol{t})^2}{(1+S)^2}\right]^{1/2}$$

$$\leq C_{*,D,K}\frac{\rho_\lambda(n)^3\log^{1/2}(n)}{n}\mu_*\mathrm{Tr}(\boldsymbol{A}\overline{\boldsymbol{M}})$$

$$\leq C_{*,D,K}\frac{\rho_\lambda(n)^3\log^{1/2}(n)}{n}\Psi_2(\mu_*;\boldsymbol{A}).$$

We establish a high-probability bound for $\Delta_i$ by first decomposing it and strategically adding and subtracting carefully chosen terms. Observing that

$$\Delta_i=(\mathbb{E}_i-\mathbb{E}_{i-1})\,\mathrm{Tr}(\boldsymbol{A}\boldsymbol{T}^\top \boldsymbol{T}\boldsymbol{M})=(\mathbb{E}_i-\mathbb{E}_{i-1})\left(\mathrm{Tr}(\boldsymbol{A}\boldsymbol{T}^\top \boldsymbol{T}\boldsymbol{M})-\mathrm{Tr}(\boldsymbol{A}\boldsymbol{T}_i^\top \boldsymbol{T}_i\boldsymbol{M}_i)\right),$$

where $\boldsymbol{M}_i$ is the rescaled resolvent removes $\boldsymbol{x}_i$, and we used that $\mathbb{E}_i\left[\boldsymbol{A}\boldsymbol{T}_i^\top \boldsymbol{T}_i\boldsymbol{M}_i\right]=\mathbb{E}_{i-1}\left[\boldsymbol{A}\boldsymbol{T}_i^\top \boldsymbol{T}_i\boldsymbol{M}_i\right]$, and we'll write (recall that $S_i=\boldsymbol{t}_i^\top \boldsymbol{M}_i\boldsymbol{t}_i$)

$$\mathrm{Tr}(\boldsymbol{A}\boldsymbol{T}^\top \boldsymbol{T}\boldsymbol{M})-\mathrm{Tr}(\boldsymbol{A}\boldsymbol{T}_i^\top \boldsymbol{T}_i\boldsymbol{M}_i)=\mathrm{Tr}(\boldsymbol{A}(\boldsymbol{t}_i\boldsymbol{t}_i^\top +\boldsymbol{T}_i^\top \boldsymbol{T}_i)\boldsymbol{M})-\mathrm{Tr}(\boldsymbol{A}\boldsymbol{T}_i^\top \boldsymbol{T}_i\boldsymbol{M}_i)$$

$$=\boldsymbol{t}_i^\top \boldsymbol{M}\boldsymbol{A}\boldsymbol{t}_i+\mathrm{Tr}(\boldsymbol{A}\boldsymbol{T}_i^\top \boldsymbol{T}_i\boldsymbol{M})-\mathrm{Tr}(\boldsymbol{A}\boldsymbol{T}_i^\top \boldsymbol{T}_i\boldsymbol{M}_i)$$

$$=\frac{1}{(1+S_i)}\left\{\boldsymbol{t}_i^\top \boldsymbol{M}_i\boldsymbol{A}\boldsymbol{t}_i-\mathrm{Tr}(\boldsymbol{A}\boldsymbol{T}_i^\top \boldsymbol{T}_i\boldsymbol{M}_i\boldsymbol{t}_i\boldsymbol{t}_i^\top \boldsymbol{M}_i)\right\}$$

$$=\frac{1}{(1+S_i)}\mathrm{Tr}(\boldsymbol{t}_i\boldsymbol{t}_i^\top \boldsymbol{M}_i\boldsymbol{A}(\boldsymbol{I}-\boldsymbol{T}_i^\top \boldsymbol{T}_i\boldsymbol{M}_i)).$$

Observing that
$$\boldsymbol{I} - \boldsymbol{T}_i^\top \boldsymbol{T}_i \boldsymbol{M}_i = \lambda \boldsymbol{\Sigma}^{-1} \boldsymbol{M}_i,$$
we can write for $j \in \{i-1, i\}$, with probability at least $1 - n^{-D}$,

$$
\left| \mathbb{E}_j \left[ \frac{1}{(1+S_i)} \mathrm{Tr}(\boldsymbol{t}_i \boldsymbol{t}_i^\top \boldsymbol{M}_i \boldsymbol{A} (\boldsymbol{I} - \boldsymbol{T}_i^\top \boldsymbol{T}_i \boldsymbol{M}_i)) \right] \right|
\begin{aligned}
&\leq \lambda \mathbb{E}_j \left[ |\boldsymbol{t}_i^\top \boldsymbol{M}_i \boldsymbol{A} \boldsymbol{\Sigma}^{-1} \boldsymbol{M}_i \boldsymbol{t}_i| \right] \\
&\leq \mathbb{E}_j \left[ |\boldsymbol{t}_i^\top \boldsymbol{M}_i \boldsymbol{A} \boldsymbol{t}_i| \right] \\
&\leq C_{*,D} \log(n) \mathbb{E}_j \left[ \mathrm{Tr}(\boldsymbol{A} \boldsymbol{M}_i) \right] \\
&\leq C_{*,D} \rho_\lambda(n) \log(n) \mathrm{Tr}(\boldsymbol{A}\overline{\boldsymbol{M}}) \\
&\leq C_{*,D} \frac{\rho_\lambda(n)^2 \log(n)}{n} \mu_* \mathrm{Tr}(\boldsymbol{A}\overline{\boldsymbol{M}}) \\
&= C_{*,D} \frac{\rho_\lambda(n)^2 \log(n)}{n} \Psi_2(\mu_*; \boldsymbol{A}),
\end{aligned}
$$

where we used that $\boldsymbol{M}_i \preceq \boldsymbol{\Sigma}/\lambda$ by definition in the second inequality, [37, Lemma 4.(b)] in the fourth inequality, and $\mu_* = \frac{n}{1+\mathrm{Tr}(\overline{\boldsymbol{M}})} \geq \frac{n}{2\rho_\lambda(n)}$ in the last inequality.

Applying a union bound and adjusting the choice of $D$, we conclude that with probability at least $1 - n^{-D}$, the following holds for all $i \in [n]$:

$$|\Delta_i| \leq C_{*,D,K} \frac{\rho_\lambda(n)^2 \log(n)}{n} \Psi_2(\mu_*; \boldsymbol{A}).$$

$\square$

# D Main results and proofs for linear regression

In this section, we study the asymptotic and non-asymptotic deterministic equivalent of the (ridge/ridgeless) estimator norm for linear regression. Based on these results, we are able to mathematically characterize the test risk under norm-based capacity. Table 3 presents our main results for linear regression.

Table 3: Summary of our main results for **Linear Regression**.

| Type | Results | Regularization | Deterministic equivalents N | Relationship between R and N |
|---|---|---|---|---|
| Deterministic equivalence | Proposition D.4 | $\lambda > 0$ | Asymptotic | - |
| | Corollary D.5 | $\lambda \to 0$ | Asymptotic | - |
| | Theorem D.6 | $\lambda > 0$ | Non-asymptotic | - |
| Relationship | Proposition D.7 | $\lambda > 0$ | - | Under $\boldsymbol{\Sigma} = \boldsymbol{I}_d$ |
| | Proposition D.9 | $\lambda \to 0$ | - | Under-parameterized regime |
| | Corollary D.8 | $\lambda \to 0$ | - | Under $\boldsymbol{\Sigma} = \boldsymbol{I}_d$ |
| | Proposition D.10 | $\lambda \to 0$ | - | Under Assumption 6 (power-law) |

To deliver our results, we need the following lemma for the bias-variance decomposition of the estimator's norm.

**Lemma D.1** (Bias-variance decomposition of $\mathcal{N}_\lambda^{\mathrm{LS}}$). *We have the bias-variance decomposition* $\mathbb{E}_\varepsilon \|\hat{\boldsymbol{\beta}}\|_2^2 =: \mathcal{N}_\lambda^{\mathrm{LS}} = \mathcal{B}_{\mathcal{N},\lambda}^{\mathrm{LS}} + \mathcal{V}_{\mathcal{N},\lambda}^{\mathrm{LS}}$, *where* $\mathcal{B}_{\mathcal{N},\lambda}^{\mathrm{LS}}$ *and* $\mathcal{V}_{\mathcal{N},\lambda}^{\mathrm{LS}}$ *are defined as*

$$\mathcal{B}_{\mathcal{N},\lambda}^{\mathrm{LS}} := \langle \boldsymbol{\beta}_*, (\boldsymbol{X}^\top \boldsymbol{X})^2 (\boldsymbol{X}^\top \boldsymbol{X} + \lambda \boldsymbol{I})^{-2} \boldsymbol{\beta}_* \rangle, \quad \mathcal{V}_{\mathcal{N},\lambda}^{\mathrm{LS}} := \sigma^2 \mathrm{Tr}(\boldsymbol{X}^\top \boldsymbol{X} (\boldsymbol{X}^\top \boldsymbol{X} + \lambda \boldsymbol{I})^{-2}).$$

And we present the proof of Lemma D.1 as below.

*Proof of Lemma D.1.* Here we give the bias-variance decomposition of $\mathbb{E}_\varepsilon \|\hat{\boldsymbol{\beta}}\|_2^2$. The formulation of $\mathbb{E}_\varepsilon \|\hat{\boldsymbol{\beta}}\|_2^2$ is given by

$$\mathbb{E}_\varepsilon \|\hat{\boldsymbol{\beta}}\|_2^2 = \| \left( \boldsymbol{X}^\top \boldsymbol{X} + \lambda \boldsymbol{I} \right)^{-1} \boldsymbol{X}^\top \boldsymbol{y} \|_2^2,$$

which can be decomposed as

$$
\begin{aligned}
\mathbb{E}_\varepsilon \|\hat{\boldsymbol{\beta}}\|_2^2 &= \mathbb{E}_\varepsilon \| \left( \boldsymbol{X}^\top \boldsymbol{X} + \lambda \boldsymbol{I} \right)^{-1} \boldsymbol{X}^\top (\boldsymbol{X}\boldsymbol{\beta}_* + \varepsilon) \|_2^2 \\
&= \| \left( \boldsymbol{X}^\top \boldsymbol{X} + \lambda \boldsymbol{I} \right)^{-1} \boldsymbol{X}^\top \boldsymbol{X}\boldsymbol{\beta}_* \|_2^2 + \mathbb{E}_\varepsilon \| \left( \boldsymbol{X}^\top \boldsymbol{X} + \lambda \boldsymbol{I} \right)^{-1} \boldsymbol{X}^\top \varepsilon \|_2^2 \\
&= \langle \boldsymbol{\beta}_*, (\boldsymbol{X}^\top \boldsymbol{X})^2 (\boldsymbol{X}^\top \boldsymbol{X} + \lambda \boldsymbol{I})^{-2} \boldsymbol{\beta}_* \rangle + \sigma^2 \mathrm{Tr}(\boldsymbol{X}^\top \boldsymbol{X}(\boldsymbol{X}^\top \boldsymbol{X} + \lambda \boldsymbol{I})^{-2}) \\
&=: \mathcal{B}_{\mathcal{N},\lambda}^{\mathsf{LS}} + \mathcal{V}_{\mathcal{N},\lambda}^{\mathsf{LS}} \, .
\end{aligned}
$$

Accordingly, we can see that it shares the similar spirit with the bias-variance decomposition. $\qquad\square$

Our first goal is to relate $\mathcal{B}_{\mathcal{N},\lambda}^{\mathsf{LS}}$ and $\mathcal{V}_{\mathcal{N},\lambda}^{\mathsf{LS}}$ to their respective deterministic equivalents. And next, we will present the results for both asymptotic and non-asymptotic regime separately.

## D.1 Asymptotic deterministic equivalence for ridge regression

In this section, we establish the asymptotic approximation guarantees for linear regression, focusing on the relationships between the $\ell_2$ norm of the estimator and its deterministic equivalent. These results can be recovered by our non-asymptotic results, but we put them here just for completeness.

Before presenting the results on deterministic equivalence for ridge regression and their proofs, we begin by introducing a couple of useful corollaries from Propositions B.3 and B.4.

**Corollary D.2.** *Under the same condition of Proposition B.3, we have*

$$
\mathrm{Tr}(\boldsymbol{A}\boldsymbol{X}^\top \boldsymbol{X}(\boldsymbol{X}^\top \boldsymbol{X} + \lambda)^{-2}) \sim \frac{\mathrm{Tr}(\boldsymbol{A}\boldsymbol{\Sigma}(\boldsymbol{\Sigma} + \lambda_* \boldsymbol{I})^{-2})}{n - \mathrm{df}_2(\lambda_*)} \, . \tag{52}
$$

*Specifically, if $\boldsymbol{A} = \boldsymbol{\Sigma}$, we have*

$$
\mathrm{Tr}(\boldsymbol{\Sigma}\boldsymbol{X}^\top \boldsymbol{X}(\boldsymbol{X}^\top \boldsymbol{X} + \lambda)^{-2}) \sim \frac{\mathrm{df}_2(\lambda_*)}{n - \mathrm{df}_2(\lambda_*)} \, . \tag{53}
$$

**Corollary D.3.** *Under the same condition of Proposition B.4, we have*

$$
\mathrm{Tr}(\boldsymbol{A}\boldsymbol{T}^\top (\boldsymbol{T}\boldsymbol{\Sigma}\boldsymbol{T}^\top + \lambda)^{-2}\boldsymbol{T}) \sim \frac{\mathrm{Tr}(\boldsymbol{A}(\boldsymbol{\Sigma} + \lambda_* \boldsymbol{I})^{-2})}{n - \mathrm{df}_2(\lambda_*)} \, . \tag{54}
$$

Using the equation

$$
\mathrm{Tr}(\boldsymbol{A}\boldsymbol{X}^\top \boldsymbol{X}(\boldsymbol{X}^\top \boldsymbol{X}+\lambda)^{-2}) = \frac{1}{\lambda} \left( \mathrm{Tr}(\boldsymbol{A}\boldsymbol{X}^\top \boldsymbol{X}(\boldsymbol{X}^\top \boldsymbol{X} + \lambda)^{-1}) - \mathrm{Tr}(\boldsymbol{A}(\boldsymbol{X}^\top \boldsymbol{X})^2 (\boldsymbol{X}^\top \boldsymbol{X} + \lambda)^{-2}) \right) \, ,
$$

we can directly obtain Corollaries D.2 and D.3 from Propositions B.3 and B.4.

Now we are ready to derive the deterministic equivalence, i.e., $\mathbb{E}_\varepsilon \|\hat{\boldsymbol{\beta}}\|_2^2$, under the bias-variance decomposition. Our results can handle ridge estimator $\hat{\boldsymbol{\beta}}$ in Proposition D.4 and interpolator $\hat{\boldsymbol{\beta}}_{\min}$ in Corollary D.5, respectively.

**Proposition D.4** (Asymptotic deterministic equivalence of $\mathcal{N}_\lambda^{\mathsf{LS}}$.)**.** *Given the bias variance decomposition of $\mathbb{E}_\varepsilon \|\hat{\boldsymbol{\beta}}\|_2^2$ in Lemma D.1, under Assumption 3, we have the following asymptotic deterministic equivalents $\mathcal{N}_\lambda^{\mathsf{LS}} \sim \mathsf{N}_\lambda^{\mathsf{LS}} := \mathsf{B}_{\mathsf{N},\lambda}^{\mathsf{LS}} + \mathsf{V}_{\mathsf{N},\lambda}^{\mathsf{LS}}$ such that $\mathcal{B}_{\mathcal{N},\lambda}^{\mathsf{LS}} \sim \mathsf{B}_{\mathsf{N},\lambda}^{\mathsf{LS}}$, $\mathcal{V}_{\mathcal{N},\lambda}^{\mathsf{LS}} \sim \mathsf{V}_{\mathsf{N},\lambda}^{\mathsf{LS}}$, where these quantities are from Lemma D.1 and Eq. (55).*

$$
\mathsf{B}_{\mathsf{N},\lambda}^{\mathsf{LS}} := \langle \boldsymbol{\beta}_*, \boldsymbol{\Sigma}^2 (\boldsymbol{\Sigma} + \lambda_* \boldsymbol{I})^{-2} \boldsymbol{\beta}_* \rangle + \frac{\mathrm{Tr}(\boldsymbol{\Sigma}(\boldsymbol{\Sigma} + \lambda_* \boldsymbol{I})^{-2})}{n} \cdot \underbrace{\frac{\lambda_*^2 \langle \boldsymbol{\beta}_*, \boldsymbol{\Sigma}(\boldsymbol{\Sigma} + \lambda_* \boldsymbol{I})^{-2} \boldsymbol{\beta}_* \rangle}{1 - n^{-1}\mathrm{Tr}(\boldsymbol{\Sigma}^2 (\boldsymbol{\Sigma} + \lambda_* \boldsymbol{I})^{-2})}}_{\mathsf{B}_{\mathsf{R},\lambda}^{\mathsf{LS}}} \, ,
$$

$$
\mathsf{V}_{\mathsf{N},\lambda}^{\mathsf{LS}} := \frac{\mathrm{Tr}(\boldsymbol{\Sigma}(\boldsymbol{\Sigma} + \lambda_* \boldsymbol{I})^{-2})}{\mathrm{Tr}(\boldsymbol{\Sigma}^2 (\boldsymbol{\Sigma} + \lambda_* \boldsymbol{I})^{-2})} \cdot \underbrace{\frac{\sigma^2 \mathrm{Tr}(\boldsymbol{\Sigma}^2 (\boldsymbol{\Sigma} + \lambda_* \boldsymbol{I})^{-2})}{n - \mathrm{Tr}(\boldsymbol{\Sigma}^2 (\boldsymbol{\Sigma} + \lambda_* \boldsymbol{I})^{-2})}}_{\mathsf{V}_{\mathsf{R},\lambda}^{\mathsf{LS}}} \, . \tag{55}
$$

We remark that, by checking Eq. (16) and Eq. (55), ***norm-based capacity suffices to characterize generalization while effective dimension can not***, where effective dimension is defined as $\mathrm{Tr}(\mathbf{\Sigma}(\mathbf{\Sigma} + \lambda_* \boldsymbol{I})^{-1})$ [63] or similar formulation, e.g., $\mathrm{Tr}(\mathbf{\Sigma}^2(\mathbf{\Sigma} + \lambda_* \boldsymbol{I})^{-2})$.

- Bias: for the bias term, we find that the second term of $\mathsf{B}^{\mathsf{LS}}_{\mathsf{N},\lambda}$ rescales $\mathsf{B}^{\mathsf{LS}}_{\mathsf{R},\lambda}$ in Eq. (16) by a factor ${\color{red}\frac{\mathrm{Tr}(\mathbf{\Sigma}(\mathbf{\Sigma}+\lambda_*\boldsymbol{I})^{-2})}{n}}$.

- Variance: we find that the variance term of the norm $\mathsf{V}^{\mathsf{LS}}_{\mathsf{N},\lambda}$ equals the variance term of the test risk $\mathsf{V}^{\mathsf{LS}}_{\mathsf{R},\lambda}$ in Eq. (16) multiplied by a factor ${\color{blue}\frac{\mathrm{Tr}(\mathbf{\Sigma}(\mathbf{\Sigma}+\lambda_*\boldsymbol{I})^{-2})}{\mathrm{Tr}(\mathbf{\Sigma}^2(\mathbf{\Sigma}+\lambda_*\boldsymbol{I})^{-2})}}$. That means, under isotropic features $\mathbf{\Sigma} = \boldsymbol{I}_d$, they are the same.

Accordingly, the norm-based capacity is able to characterize the bias and variance of the excess risk. We provide a quantitative analysis of this relationship in Appendix D.3.2. Below, we present the proof of Proposition D.4.

*Proof of Proposition D.4.* We give the asymptotic deterministic equivalents for $\mathcal{B}^{\mathsf{LS}}_{\mathcal{N},\lambda}$ and $\mathcal{V}^{\mathsf{LS}}_{\mathcal{N},\lambda}$, respectively. For the bias term $\mathcal{B}^{\mathsf{LS}}_{\mathcal{N},\lambda}$, we use Eq. (9) by taking $\boldsymbol{A} = \boldsymbol{\beta}_*\boldsymbol{\beta}_*^\top$ and $\boldsymbol{B} = \boldsymbol{I}$ and thus obtain

$$
\begin{aligned}
\mathcal{B}^{\mathsf{LS}}_{\mathcal{N},\lambda} &= \langle \boldsymbol{\beta}_*, (\boldsymbol{X}^\top\boldsymbol{X})^2(\boldsymbol{X}^\top\boldsymbol{X} + \lambda\boldsymbol{I})^{-2}\boldsymbol{\beta}_* \rangle \\
&= \mathrm{Tr}(\boldsymbol{\beta}_*\boldsymbol{\beta}_*^\top(\boldsymbol{X}^\top\boldsymbol{X})^2(\boldsymbol{X}^\top\boldsymbol{X} + \lambda\boldsymbol{I})^{-2}) \\
&\sim \mathrm{Tr}(\boldsymbol{\beta}_*\boldsymbol{\beta}_*^\top\mathbf{\Sigma}^2(\mathbf{\Sigma} + \lambda_*\boldsymbol{I})^{-2}) \\
&\quad + \lambda_*^2\mathrm{Tr}(\boldsymbol{\beta}_*\boldsymbol{\beta}_*^\top\mathbf{\Sigma}(\mathbf{\Sigma} + \lambda_*\boldsymbol{I})^{-2}) \cdot \mathrm{Tr}(\mathbf{\Sigma}(\mathbf{\Sigma} + \lambda_*\boldsymbol{I})^{-2}) \cdot \frac{1}{n - \mathrm{Tr}(\mathbf{\Sigma}^2(\mathbf{\Sigma} + \lambda_*\boldsymbol{I})^{-2})} \\
&= \langle \boldsymbol{\beta}_*, \mathbf{\Sigma}^2(\mathbf{\Sigma} + \lambda_*\boldsymbol{I})^{-2}\boldsymbol{\beta}_* \rangle + \frac{\mathrm{Tr}(\mathbf{\Sigma}(\mathbf{\Sigma} + \lambda_*\boldsymbol{I})^{-2})}{n} \cdot \frac{\lambda_*^2\langle \boldsymbol{\beta}_*, \mathbf{\Sigma}(\mathbf{\Sigma} + \lambda_*\boldsymbol{I})^{-2}\boldsymbol{\beta}_* \rangle}{1 - n^{-1}\mathrm{Tr}(\mathbf{\Sigma}^2(\mathbf{\Sigma} + \lambda_*\boldsymbol{I})^{-2})} \\
&=: \mathsf{B}^{\mathsf{LS}}_{\mathsf{N},\lambda}\,.
\end{aligned}
$$

For the variance term $\mathcal{V}^{\mathsf{LS}}_{\mathcal{N}}$, we use Eq. (52) by taking $\boldsymbol{A} = \boldsymbol{I}$ and obtain

$$
\mathcal{V}^{\mathsf{LS}}_{\mathcal{N}} = \sigma^2\mathrm{Tr}(\boldsymbol{X}^\top\boldsymbol{X}(\boldsymbol{X}^\top\boldsymbol{X} + \lambda\boldsymbol{I})^{-2}) \sim \frac{\sigma^2\mathrm{Tr}(\mathbf{\Sigma}(\mathbf{\Sigma} + \lambda_*\boldsymbol{I})^{-2})}{n - \mathrm{Tr}(\mathbf{\Sigma}^2(\mathbf{\Sigma} + \lambda_*\boldsymbol{I})^{-2})} =: \mathsf{V}^{\mathsf{LS}}_{\mathsf{N},\lambda}\,.
$$

$\square$

**The deterministic equivalent of the norm for the min-$\ell_2$-norm estimator.** We have the following results on the characterization of the deterministic equivalence of $\|\hat{\boldsymbol{\beta}}_{\min}\|_2$.

**Corollary D.5** (Asymptotic deterministic equivalence of the norm of interpolator). *Under Assumption 3, for the minimum $\ell_2$-norm estimator $\hat{\boldsymbol{\beta}}_{\min}$, we have the following deterministic equivalence: for the under-parameterized regime ($d < n$), we have*

$$
\mathcal{B}^{\mathsf{LS}}_{\mathcal{N},0} = \|\boldsymbol{\beta}_*\|_2^2\,, \quad \mathcal{V}^{\mathsf{LS}}_{\mathcal{N},0} \sim \frac{\sigma^2}{n - d}\mathrm{Tr}(\mathbf{\Sigma}^{-1})\,.
$$

*In the over-parameterized regime ($d > n$), we have*

$$
\mathcal{B}^{\mathsf{LS}}_{\mathcal{N},0} \sim \langle \boldsymbol{\beta}_*, \mathbf{\Sigma}(\mathbf{\Sigma} + \lambda_n\boldsymbol{I})^{-1}\boldsymbol{\beta}_* \rangle\,,
$$
$$
\mathcal{V}^{\mathsf{LS}}_{\mathcal{N},0} \sim \frac{\sigma^2\mathrm{Tr}(\mathbf{\Sigma}(\mathbf{\Sigma} + \lambda_n\boldsymbol{I})^{-2})}{n - \mathrm{Tr}(\mathbf{\Sigma}^2(\mathbf{\Sigma} + \lambda_n\boldsymbol{I})^{-2})} = \frac{\sigma^2}{\lambda_n}\,,
$$

*where $\lambda_n$ is defined by $\mathrm{Tr}(\mathbf{\Sigma}(\mathbf{\Sigma} + \lambda_n\boldsymbol{I})^{-1}) \sim n$.*

**Remark:** The asymptotic behavior of $\lambda_*$ differs between the under-parameterized and over-parameterized regimes as $\lambda \to 0$, though the ridge regression estimator $\hat{\boldsymbol{\beta}}$ converges to the min-$\ell_2$-norm estimator $\hat{\boldsymbol{\beta}}_{\min}$. To be specific, in the under-parameterized regime, $\lambda_*$ converges to 0 as $\lambda \to 0$; while in the over-parameterized regime, $\lambda_*$ converges to a constant that admits $\mathrm{Tr}(\mathbf{\Sigma}(\mathbf{\Sigma} + \lambda_n\boldsymbol{I})^{-1}) \sim n$ when $\lambda \to 0$. Accordingly, for the minimum $\ell_2$-norm estimator, it is necessary to analyze the two regimes separately. And we show that the solution $\lambda_n$ to the self-consistent equation $\mathrm{Tr}(\mathbf{\Sigma}(\mathbf{\Sigma} + \lambda_n\boldsymbol{I})^{-1}) \sim n$ can be obtained from the variance $\mathsf{V}^{\mathsf{LS}}_{\mathsf{N},0} = \sigma^2/\lambda_n$.

*Proof of Corollary D.5.* We separate the results in the under-parameterized and over-parameterized regimes.

In the under-parameterized regime ($d < n$), for minimum norm estimator $\hat{\beta}_{\min}$, we have (for $\boldsymbol{X}^\top \boldsymbol{X}$ is invertible)

$$\hat{\beta}_{\min} = \left(\boldsymbol{X}^\top \boldsymbol{X}\right)^{-1} \boldsymbol{X}^\top \boldsymbol{y} = \left(\boldsymbol{X}^\top \boldsymbol{X}\right)^{-1} \boldsymbol{X}^\top (\boldsymbol{X}\beta_* + \varepsilon) = \beta_* + \left(\boldsymbol{X}^\top \boldsymbol{X}\right)^{-1} \boldsymbol{X}^\top \varepsilon\,.$$

Accordingly, we can directly obtain the bias-variance decomposition as well as their deterministic equivalents

$$\mathcal{B}_{\mathcal{N},0}^{\mathsf{LS}} = \|\beta_*\|_2^2\,, \quad \mathcal{V}_{\mathcal{N},0}^{\mathsf{LS}} = \sigma^2 \mathrm{Tr}(\boldsymbol{X}^\top \boldsymbol{X}(\boldsymbol{X}^\top \boldsymbol{X})^{-2}) \sim \sigma^2 \frac{\mathrm{Tr}(\boldsymbol{\Sigma}^{-1})}{n-d}\,,$$

where we use Eq. (52) and take $\lambda \to 0$ for the variance term.

In the over-parameterized regime ($d > n$), we take the limit $\lambda \to 0$ within ridge regression and use Proposition D.4. Define $\lambda_n$ as $\mathrm{Tr}(\boldsymbol{\Sigma}(\boldsymbol{\Sigma} + \lambda_n \boldsymbol{I})^{-1}) \sim n$, we have for the bias term

$$\mathcal{B}_{\mathcal{N},0}^{\mathsf{LS}} \sim \langle \beta_*, \boldsymbol{\Sigma}^2 (\boldsymbol{\Sigma} + \lambda_n \boldsymbol{I})^{-2} \beta_* \rangle + \frac{\mathrm{Tr}(\boldsymbol{\Sigma}(\boldsymbol{\Sigma} + \lambda_n \boldsymbol{I})^{-2})}{n} \cdot \frac{\lambda_n^2 \langle \beta_*, \boldsymbol{\Sigma}(\boldsymbol{\Sigma} + \lambda_n \boldsymbol{I})^{-2} \beta_* \rangle}{1 - n^{-1}\mathrm{Tr}(\boldsymbol{\Sigma}^2 (\boldsymbol{\Sigma} + \lambda_n \boldsymbol{I})^{-2})}$$

$$= \langle \beta_*, \boldsymbol{\Sigma}(\boldsymbol{\Sigma} + \lambda_n \boldsymbol{I})^{-1} \beta_* \rangle - \lambda_n \langle \beta_*, \boldsymbol{\Sigma}(\boldsymbol{\Sigma} + \lambda_n \boldsymbol{I})^{-2} \beta_* \rangle$$

$$\quad + \frac{\mathrm{Tr}(\boldsymbol{\Sigma}(\boldsymbol{\Sigma} + \lambda_n \boldsymbol{I})^{-2})}{n} \cdot \frac{\lambda_n^2 \langle \beta_*, \boldsymbol{\Sigma}(\boldsymbol{\Sigma} + \lambda_n \boldsymbol{I})^{-2} \beta_* \rangle}{1 - n^{-1}\mathrm{Tr}(\boldsymbol{\Sigma}^2 (\boldsymbol{\Sigma} + \lambda_n \boldsymbol{I})^{-2})}$$

$$= \langle \beta_*, \boldsymbol{\Sigma}(\boldsymbol{\Sigma} + \lambda_n \boldsymbol{I})^{-1} \beta_* \rangle\,.$$

For the variance term, we have

$$\mathcal{V}_{\mathcal{N},0}^{\mathsf{LS}} \sim \frac{\sigma^2 \mathrm{Tr}(\boldsymbol{\Sigma}(\boldsymbol{\Sigma} + \lambda_n \boldsymbol{I})^{-2})}{n - \mathrm{Tr}(\boldsymbol{\Sigma}^2 (\boldsymbol{\Sigma} + \lambda_n \boldsymbol{I})^{-2})}\,.$$

Finally we conclude the proof. $\qquad\square$

## D.2 Non-asymptotic analysis on the deterministic equivalents of estimator's norm

To derive the non-asymptotic results, we make the following assumption on well-behaved data.

**Assumption 4** (Data concentration [37])**.** There exist $C_* > 0$ such that for any PSD matrix $\boldsymbol{A} \in \mathbb{R}^{d \times d}$ with $\mathrm{Tr}(\boldsymbol{\Sigma A}) < \infty$ and $t \geq 0$, we have

$$\mathbb{P}\left(\left|\boldsymbol{X}^\top \boldsymbol{A} \boldsymbol{X} - \mathrm{Tr}(\boldsymbol{\Sigma A})\right| \geq t \|\boldsymbol{\Sigma}^{1/2} \boldsymbol{A} \boldsymbol{\Sigma}^{1/2}\|_{\mathrm{F}}\right) \leq C_* e^{-\frac{t}{C_*}}\,.$$

**Assumption 5** ([14])**.** There exists $C > 1$

$$\frac{\langle \beta_*, \boldsymbol{\Sigma}(\boldsymbol{\Sigma} + \lambda_*)^{-1} \beta_* \rangle}{\langle \beta_*, \boldsymbol{\Sigma}^2 (\boldsymbol{\Sigma} + \lambda_*)^{-2} \beta_* \rangle} \leq C\,.$$

**Remark:** This assumption holds in many settings of interest, such as power law assumptions like those in Assumption 6, since under this assumption the numerator and denominator are bounded sums of finite terms. It is a technical assumption used to address the difference between two deterministic equivalents that are needed in our work for norm-based capacity. In fact, this assumption is used for RFMs in [14] as the authors also face with the issue on the difference between two deterministic equivalents.

Based on the above two assumptions, we are ready to deliver the following result, our results can also numerically validated by Fig. 11 in Appendix H.2.

**Theorem D.6** (Deterministic equivalents of the $\ell_2$-norm of the estimator.)**.** *Assume well-behaved data $\{x_i\}_{i=1}^n$ satisfy Assumption 4 and Assumption 5. Then for any $D, K > 0$, there exist constants $\eta_* \in (0, 1/2)$ and $C_{*,D,K} > 0$ ensuring the following property holds. For any $n \geq C_{*,D,K}$, $\lambda > 0$, if the following condition is satisfied:*

$$\lambda \geq n^{-K}\,, \quad \rho_\lambda(n)^{5/2} \log^{3/2}(n) \leq K\sqrt{n}\,,$$

*then with probability at least $1 - n^{-D}$, we have that*

$$\left|\mathcal{B}_{\mathcal{N},\lambda}^{\mathsf{LS}} - \mathsf{B}_{\mathsf{N},\lambda}^{\mathsf{LS}}\right| \leq C_{x,D,K} \frac{\rho_\lambda(n)^6 \log^{3/2}(n)}{\sqrt{n}} \mathsf{B}_{\mathsf{N},\lambda}^{\mathsf{LS}},$$

$$\left|\mathcal{V}_{\mathcal{N},\lambda}^{\mathsf{LS}} - \mathsf{V}_{\mathsf{N},\lambda}^{\mathsf{LS}}\right| \leq C_{x,D,K} \frac{\rho_\lambda(n)^6 \log^{3/2}(n)}{\sqrt{n}} \mathsf{V}_{\mathsf{N},\lambda}^{\mathsf{LS}}.$$

Next, we give the proof of Proposition D.4 below.

*Proof of Theorem D.6.* **Part 1: Deterministic equivalents for the bias term.**

Here we prove the deterministic equivalents of $\mathcal{B}_{\mathcal{N},\lambda}^{\mathsf{LS}}$ and $\mathcal{V}_{\mathcal{N},\lambda}^{\mathsf{LS}}$. First, we decompose $\mathcal{B}_{\mathcal{N},\lambda}^{\mathsf{LS}}$ into

$$\mathcal{B}_{\mathcal{N},\lambda}^{\mathsf{LS}} = \mathrm{Tr}\left(\boldsymbol{\beta}_*\boldsymbol{\beta}_*^\top \boldsymbol{X}^\top \boldsymbol{X}(\boldsymbol{X}^\top\boldsymbol{X} + \lambda)^{-1}\right) - \lambda\mathrm{Tr}\left(\boldsymbol{\beta}_*\boldsymbol{\beta}_*^\top \boldsymbol{X}^\top \boldsymbol{X}(\boldsymbol{X}^\top\boldsymbol{X} + \lambda)^{-2}\right),$$

$$= \Phi_2(\boldsymbol{X}; \tilde{\boldsymbol{A}}_1, \lambda) - n\lambda\Phi_4(\boldsymbol{X}; \tilde{\boldsymbol{A}}_2, \lambda),$$

where $\tilde{\boldsymbol{A}}_1 := \boldsymbol{\beta}_*\boldsymbol{\beta}_*^\top$, $\tilde{\boldsymbol{A}}_2 := \boldsymbol{\Sigma}^{-1/2}\boldsymbol{\beta}_*\boldsymbol{\beta}_*^\top\boldsymbol{\Sigma}^{-1/2}$. Therefore, using Theorem B.9, with probability at least $1 - n^{-D}$, we have

$$\left|\Phi_2(\boldsymbol{X}; \tilde{\boldsymbol{A}}_1, \lambda) - \Psi_1(\lambda_*; \tilde{\boldsymbol{A}}_1)\right| \leq C_{x,D,K} \frac{\rho_\lambda(n)^{5/2} \log^{3/2}(n)}{\sqrt{n}} \Psi_1(\lambda_*; \tilde{\boldsymbol{A}}_1),$$

$$\left|n\lambda\Phi_4(\boldsymbol{X}; \tilde{\boldsymbol{A}}_2, \lambda) - n\lambda\Psi_2(\lambda_*; \tilde{\boldsymbol{A}}_2)\right| \leq C_{x,D,K} \frac{\rho_\lambda(n)^6 \log^{3/2}(n)}{\sqrt{n}} n\lambda\Psi_2(\lambda_*; \tilde{\boldsymbol{A}}_2).$$

Combining the above bounds, we deduce that

$$\left|\mathcal{B}_{\mathcal{N},\lambda}^{\mathsf{LS}} - \left(\Psi_1(\lambda_*; \tilde{\boldsymbol{A}}_1) - n\lambda\Psi_2(\lambda_*; \tilde{\boldsymbol{A}}_2)\right)\right|$$

$$\leq C_{x,D,K} \frac{\rho_\lambda(n)^6 \log^{3/2}(n)}{\sqrt{n}} \left(\Psi_1(\lambda_*; \tilde{\boldsymbol{A}}_1) + n\lambda\Psi_2(\lambda_*; \tilde{\boldsymbol{A}}_2)\right).$$

Note that

$$\Psi_1(\lambda_*; \tilde{\boldsymbol{A}}_1) - n\lambda\Psi_2(\lambda_*; \tilde{\boldsymbol{A}}_2) = \mathsf{B}_{\mathsf{N},\lambda}^{\mathsf{LS}}.$$

For $n\lambda\Psi_2(\lambda_*; \tilde{\boldsymbol{A}}_2)$, recall that $\Psi_2(\lambda_*; \boldsymbol{A}) := \frac{1}{n}\frac{\mathrm{Tr}(\boldsymbol{A}\boldsymbol{\Sigma}^2(\boldsymbol{\Sigma}+\lambda_*\boldsymbol{I})^{-2})}{n - \mathrm{Tr}(\boldsymbol{\Sigma}^2(\boldsymbol{\Sigma}+\lambda_*\boldsymbol{I})^{-2})}$, and according to Definition B.8 and Assumption 5, we have

$$n\lambda\Psi_2(\lambda_*; \tilde{\boldsymbol{A}}_2) = \lambda\frac{\mathrm{Tr}(\boldsymbol{\beta}_*\boldsymbol{\beta}_*^\top\boldsymbol{\Sigma}(\boldsymbol{\Sigma}+\lambda_*\boldsymbol{I})^{-2})}{n - \mathrm{Tr}(\boldsymbol{\Sigma}^2(\boldsymbol{\Sigma}+\lambda_*\boldsymbol{I})^{-2})}$$

$$\leq \lambda_*\mathrm{Tr}(\boldsymbol{\beta}_*\boldsymbol{\beta}_*^\top\boldsymbol{\Sigma}(\boldsymbol{\Sigma}+\lambda_*\boldsymbol{I})^{-2})$$

$$= \mathrm{Tr}(\boldsymbol{\beta}_*\boldsymbol{\beta}_*^\top\boldsymbol{\Sigma}(\boldsymbol{\Sigma}+\lambda_*\boldsymbol{I})^{-1}) - \mathrm{Tr}(\boldsymbol{\beta}_*\boldsymbol{\beta}_*^\top\boldsymbol{\Sigma}^2(\boldsymbol{\Sigma}+\lambda_*\boldsymbol{I})^{-2})$$

$$\leq \left(1 - \frac{1}{C}\right)\mathrm{Tr}(\boldsymbol{\beta}_*\boldsymbol{\beta}_*^\top\boldsymbol{\Sigma}(\boldsymbol{\Sigma}+\lambda_*)^{-1}),$$

and therefore

$$\Psi_1(\lambda_*; \tilde{\boldsymbol{A}}_1) + n\lambda\Psi_2(\lambda_*; \tilde{\boldsymbol{A}}_2) \leq \left(2 - \frac{1}{C}\right)\mathrm{Tr}(\boldsymbol{\beta}_*\boldsymbol{\beta}_*^\top\boldsymbol{\Sigma}(\boldsymbol{\Sigma}+\lambda_*)^{-1})$$

$$\leq (2C-1)\frac{1}{C}\mathrm{Tr}(\boldsymbol{\beta}_*\boldsymbol{\beta}_*^\top\boldsymbol{\Sigma}(\boldsymbol{\Sigma}+\lambda_*)^{-1})$$

$$\leq (2C-1)\left(\Psi_1(\lambda_*; \tilde{\boldsymbol{A}}_1) - n\lambda\Psi_2(\lambda_*; \tilde{\boldsymbol{A}}_2)\right).$$

Then we conclude that

$$\left|\mathcal{B}_{\mathcal{N},\lambda}^{\mathsf{LS}} - \mathsf{B}_{\mathsf{N},\lambda}^{\mathsf{LS}}\right| \leq C_{x,D,K} \frac{\rho_\lambda(n)^6 \log^{3/2}(n)}{\sqrt{n}} \mathsf{B}_{\mathsf{N},\lambda}^{\mathsf{LS}},$$

with probability at least $1 - n^{-D}$.

**Part 2: Deterministic equivalents for the variance term.** Next, we prove the deterministic equivalent of $\mathcal{V}_{\mathcal{N},\lambda}^{\mathsf{LS}}$. First, note that $\mathcal{V}_{\mathcal{N},\lambda}^{\mathsf{LS}}$ can be written in terms of the functional $\Phi_4(\boldsymbol{X}; \boldsymbol{A}, \lambda)$ defined in Eq. (23)

$$\mathcal{V}_{\mathcal{N},\lambda}^{\mathsf{LS}} = n\sigma_\varepsilon^2 \Phi_4(\boldsymbol{X}; \boldsymbol{\Sigma}^{-1}, \lambda)\,.$$

Thus, under the assumptions, we can apply Theorem B.9 to obtain that with probability at least $1 - n^{-D}$

$$\left| n\sigma_\varepsilon^2 \Phi_4(\boldsymbol{X}; \boldsymbol{\Sigma}^{-1}, \lambda) - n\sigma_\varepsilon^2 \Psi_2(\lambda_*; \boldsymbol{\Sigma}^{-1}) \right| \leq C_{x,D,K} \frac{\rho_\lambda(n)^6 \log^{3/2}(n)}{\sqrt{n}} n\sigma_\varepsilon^2 \Psi_2(\lambda_*; \boldsymbol{\Sigma}^{-1})\,.$$

Recall that $\Psi_2(\lambda_*; \boldsymbol{A}) := \frac{1}{n} \frac{\mathrm{Tr}(\boldsymbol{A}\boldsymbol{\Sigma}^2(\boldsymbol{\Sigma}+\lambda_*\boldsymbol{I})^{-2})}{n-\mathrm{Tr}(\boldsymbol{\Sigma}^2(\boldsymbol{\Sigma}+\lambda_*\boldsymbol{I})^{-2})}$, then we have

$$\left| \mathcal{V}_{\mathcal{N},\lambda}^{\mathsf{LS}} - \mathsf{V}_{\mathsf{N},\lambda}^{\mathsf{LS}} \right| \leq C_{x,D,K} \frac{\rho_\lambda(n)^6 \log^{3/2}(n)}{\sqrt{n}} \mathsf{V}_{\mathsf{N},\lambda}^{\mathsf{LS}}\,,$$

with probability at least $1 - n^{-D}$. $\qquad\square$

## D.3 Characterization of learning curves

By deriving deterministic equivalents for the norm in linear regression, we can now analyze learning curves through the lens of norm-based capacity. In certain cases, these learning curves can even be expressed in closed form.

In this section, we first examine the general characteristics of learning curves from a norm-based capacity perspective in Appendix D.3.1. We then provide a precise characterization of these curves in Appendix D.3.2.

### D.3.1 The shape description of learning curves

We plot the bias and variance components of the test risk over $\gamma := \frac{d}{n}$ and norm, see Fig. 4(a) and Fig. 4(b), respectively. Note that, our theory (shown in curve) can precisely predict experimental results (shown by points).

Fig. 4(a) reveals a clear bias-variance tradeoff in the over-parameterized regime (where $\gamma > 1$, as shown in the right portion of Fig. 4(a)). Specifically, we observe that: *i)* The bias exhibits a strictly increasing relationship with the parameter $\gamma$. *ii)* The variance demonstrates a corresponding strictly decreasing trend.

However, in the under-parameterized regime ($\gamma < 1$), both bias and variance increase monotonically with $\gamma$, therefore the bias-variance tradeoff does not exist. In particular, for the min-$\ell_2$-norm interpolator, since the bias equals 0, the risk is entirely composed of variance.

Because the self-consistent equation differs between the under- and over-parameterized regimes, the learning curve plotted against the norm (see Fig. 4(b)) is not single-valued—this is due to the phase transition between regimes. Specifically, a single norm value can correspond to two distinct error levels, depending on whether the model is under- or over-parameterized. However, when examined separately, each regime displays a one-to-one relationship between test risk and norm.

### D.3.2 Mathematical formulation of learning curves

In this section, we give the mathematical formulation of learning curves in several settings of interest. First we give some concrete examples on the relationship between $\mathsf{R}$ and $\mathsf{N}$ in terms of isotropic features.

**Proposition D.7** (Isotropic features for ridge regression, see Fig. 5)**.** *Consider covariance matrix* $\boldsymbol{\Sigma} = \boldsymbol{I}_d$, *the deterministic equivalents* $\mathsf{R}_\lambda^{\mathsf{LS}}$ *and* $\mathsf{N}_\lambda^{\overline{\mathsf{LS}}}$ *satisfy*

$$\left( \|\boldsymbol{\beta}_*\|_2^2 - \mathsf{R}_\lambda^{\mathsf{LS}} - \mathsf{N}_\lambda^{\mathsf{LS}} \right) \left( \|\boldsymbol{\beta}_*\|_2^2 + \mathsf{R}_\lambda^{\mathsf{LS}} - \mathsf{N}_\lambda^{\mathsf{LS}} \right)^2 d + 2\|\boldsymbol{\beta}_*\|_2^2 \left( \left( \|\boldsymbol{\beta}_*\|_2^2 + \mathsf{R}_\lambda^{\mathsf{LS}} - \mathsf{N}_\lambda^{\mathsf{LS}} \right)^2 - 4\|\boldsymbol{\beta}_*\|_2^2 \mathsf{R}_\lambda^{\mathsf{LS}} \right) \lambda = 2 \left( \left( \mathsf{R}_\lambda^{\mathsf{LS}} - \mathsf{N}_\lambda^{\mathsf{LS}} \right)^2 - \|\boldsymbol{\beta}_*\|_2^4 \right) d\sigma^2 \,.$$

**Remark:** $\mathsf{R}_\lambda^{\mathsf{LS}}$ and $\mathsf{N}_\lambda^{\mathsf{LS}}$ formulates a third-order polynomial. When $\lambda \to \infty$, it degenerates to $\mathsf{R}_\lambda^{\mathsf{LS}} = (\|\boldsymbol{\beta}_*\|_2 - \sqrt{\mathsf{N}_\lambda^{\mathsf{LS}}})^2$ when $\mathsf{N}_\lambda^{\mathsf{LS}} \leq \|\boldsymbol{\beta}_*\|_2^2$. Hence $\mathsf{R}_\lambda^{\mathsf{LS}}$ is monotonically decreasing with respect to $\mathsf{N}_\lambda^{\mathsf{LS}}$, empirically verified by Fig. 5. Besides, if we take $\lambda = \frac{d\sigma^2}{\|\boldsymbol{\beta}_*\|_2^2}$, which is the **optimal**

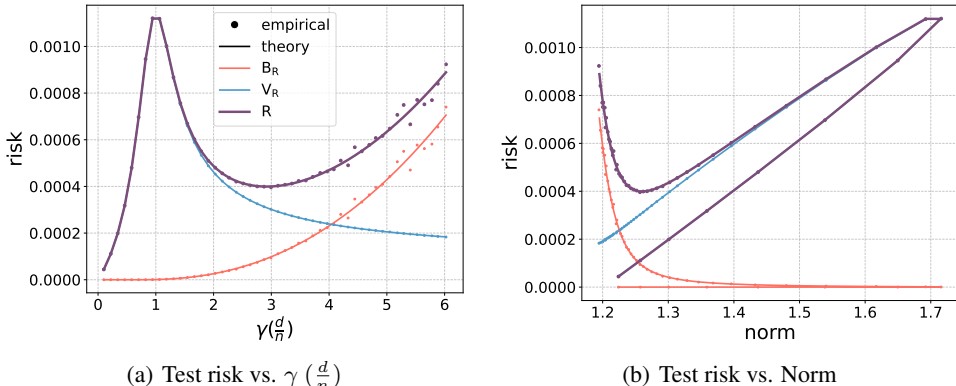

(a) Test risk vs. $\gamma\left(\frac{d}{n}\right)$          (b) Test risk vs. Norm

Figure 4: The relationship between the test risk R, norm N, their bias and variance ($B_R$, $V_R$, $B_N$, $V_N$), and the ratio $\gamma := \frac{d}{n}$ for linear regression model. Training data $\{(\boldsymbol{x}_i, y_i)\}_{i \in [n]}$, $d = 1000$, sampled from a linear model $y_i = \boldsymbol{x}_i^\top \boldsymbol{\beta}_* + \varepsilon_i$, $\sigma^2 = 0.0004$, $\boldsymbol{x}_i \sim \mathcal{N}(0, \boldsymbol{\Sigma})$, with $\sigma_k(\boldsymbol{\Sigma}) = k^{-1}$, $\boldsymbol{\beta}_{*,k} = k^{-3/2}$. The ridge $\lambda = 0.005$. Note that in the under-parameterized regime ($d < n$), the bias of the test risk is zero.

**regularization parameter** discussed in [57, 39], the relationship in Proposition D.7 will become $R_\lambda^{\mathsf{LS}} = \|\boldsymbol{\beta}_*\|_2^2 - N_\lambda^{\mathsf{LS}}$, which corresponds to a straight line. This is empirically shown in Fig. 5 with $\lambda = 50$. In addition to isotropic features, we further examine the relationship under the power-law assumption for the data.

Apart from sufficiently large $\lambda$ and optimal $\lambda$ mentioned before, below we consider min-$\ell_2$-norm estimator. Note that when $\lambda \to 0$, the ridge regression estimator $\hat{\boldsymbol{\beta}}$ converges to the min-$\ell_2$-norm estimator $\hat{\boldsymbol{\beta}}_{\min}$. However, the behavior of $\lambda_*$ differs between the under-parameterized and over-parameterized regimes as $\lambda \to 0$. Thus, the min-$\ell_2$-norm estimator requires **separate analysis of the two regimes**.

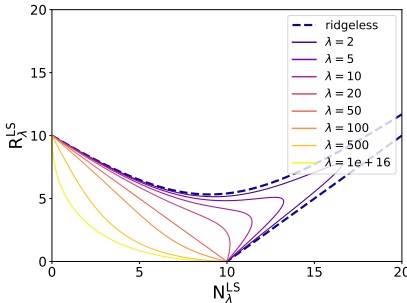

Figure 5: Relationship between $R_\lambda^{\mathsf{LS}}$ and $N_\lambda^{\mathsf{LS}}$ under the linear model $y_i = \boldsymbol{x}_i^\top \boldsymbol{\beta}_* + \varepsilon_i$, with $d = 500$, $\boldsymbol{\Sigma} = \boldsymbol{I}_d$, $\|\boldsymbol{\beta}_*\|_2^2 = 10$, and $\sigma^2 = 1$. The dashed line corresponds to the ridgeless regression curve.

*Proof of Proposition D.7.* According to the formulation of $B_{N,\lambda}^{\mathsf{LS}}$ and $V_{N,\lambda}^{\mathsf{LS}}$ in Eq. (55), for $\boldsymbol{\Sigma} = \boldsymbol{I}_d$, we have

$$B_{N,\lambda}^{\mathsf{LS}} = \frac{1}{(1+\lambda_*)^2}\|\boldsymbol{\beta}_*\|_2^2 + \frac{d}{n(1+\lambda_*)^2} \cdot \frac{\lambda_*^2 \frac{1}{(1+\lambda_*)^2}\|\boldsymbol{\beta}_*\|_2^2}{1 - \frac{d}{n(1+\lambda_*)^2}}, \quad V_{N,\lambda}^{\mathsf{LS}} = \frac{\sigma^2 \frac{d}{(1+\lambda_*)^2}}{n - \frac{d}{(1+\lambda_*)^2}},$$

$$N_\lambda^{\mathsf{LS}} = \frac{d}{(1+\lambda_*)^2}\|\boldsymbol{\beta}_*\|_2^2 + \frac{d}{n(1+\lambda_*)^2} \cdot \frac{\lambda_*^2 \frac{d}{(1+\lambda_*)^2}\|\boldsymbol{\beta}_*\|_2^2}{1 - \frac{d}{n(1+\lambda_*)^2}} + \frac{\sigma^2 \frac{d}{(1+\lambda_*)^2}}{n - \frac{d}{(1+\lambda_*)^2}},$$

where $\lambda_*$ admits a closed-form solution

$$\lambda_* = \frac{d + \lambda - n + \sqrt{4\lambda n + (n - d - \lambda)^2}}{2n} \, .$$

Recall the formulation $\mathsf{B}^{\mathsf{LS}}_{\mathsf{R},\lambda}$ and $\mathsf{V}^{\mathsf{LS}}_{\mathsf{R},\lambda}$ (for test risk) in Eq. (16), for $\boldsymbol{\Sigma} = \boldsymbol{I}_d$, we have

$$\mathsf{B}^{\mathsf{LS}}_{\mathsf{R},\lambda} = \frac{\lambda_*^2 \frac{1}{(1+\lambda_*)^2} \|\boldsymbol{\beta}_*\|_2^2}{1 - \frac{d}{n(1+\lambda_*)^2}} \, , \quad \mathsf{V}^{\mathsf{LS}}_{\mathsf{R},\lambda} = \frac{\sigma^2 \frac{d}{(1+\lambda_*)^2}}{n - \frac{d}{(1+\lambda_*)^2}} \, , \quad \mathsf{R}^{\mathsf{LS}}_{\lambda} = \frac{\lambda_*^2 \frac{d}{(1+\lambda_*)^2} \|\boldsymbol{\beta}_*\|_2^2}{1 - \frac{d}{n(1+\lambda_*)^2}} + \frac{\sigma^2 \frac{d}{(1+\lambda_*)^2}}{n - \frac{d}{(1+\lambda_*)^2}} \, .$$

Accordingly, to establish the relationship between $\mathsf{R}^{\mathsf{LS}}_{\lambda}$ and $\mathsf{N}^{\mathsf{LS}}_{\lambda}$, we combine their formulation and eliminate $n$ to obtain[6]

$$2((\mathsf{R}^{\mathsf{LS}}_{\lambda} - \mathsf{N}^{\mathsf{LS}}_{\lambda})^2 - \|\boldsymbol{\beta}_*\|_2^4)d\sigma^2 = (\|\boldsymbol{\beta}_*\|_2^2 - \mathsf{R}^{\mathsf{LS}}_{\lambda} - \mathsf{N}^{\mathsf{LS}}_{\lambda})(\|\boldsymbol{\beta}_*\|_2^2 + \mathsf{R}^{\mathsf{LS}}_{\lambda} - \mathsf{N}^{\mathsf{LS}}_{\lambda})^2 d$$
$$+ 2\|\boldsymbol{\beta}_*\|_2^2((\|\boldsymbol{\beta}_*\|_2^2 + \mathsf{R}^{\mathsf{LS}}_{\lambda} - \mathsf{N}^{\mathsf{LS}}_{\lambda})^2 - 4\|\boldsymbol{\beta}_*\|_2^2 \mathsf{R}^{\mathsf{LS}}_{\lambda})\lambda \, .$$

$\square$

**Corollary D.8** (Isotropic features for min-$\ell_2$-norm interpolator, see Fig. 5)**.** *Consider covariance matrix $\boldsymbol{\Sigma} = \boldsymbol{I}_d$, the relationship between $\mathsf{R}^{\mathsf{LS}}_0$ and $\mathsf{N}^{\mathsf{LS}}_0$ from under-parameterized to over-parameterized regimes admit*

$$\mathsf{R}^{\mathsf{LS}}_0 = \begin{cases} \mathsf{N}^{\mathsf{LS}}_0 - \|\boldsymbol{\beta}_*\|_2^2 \, , & \text{if } d < n \text{ (under-parameterized)}; \\ \sqrt{[\mathsf{N}^{\mathsf{LS}}_0 - (\|\boldsymbol{\beta}_*\|_2^2 - \sigma^2)]^2 + 4\|\boldsymbol{\beta}_*\|_2^2 \sigma^2} - \sigma^2 \, , & \text{o/w} \, . \end{cases}$$

*For the variance part of $\mathsf{R}^{\mathsf{LS}}_0$ and $\mathsf{N}^{\mathsf{LS}}_0$, we have $\mathsf{V}^{\mathsf{LS}}_{\mathsf{R},0} = \mathsf{V}^{\mathsf{LS}}_{\mathsf{N},0}$; For the respective bias part, we have $\mathsf{B}^{\mathsf{LS}}_{\mathsf{R},0} + \mathsf{B}^{\mathsf{LS}}_{\mathsf{N},0} = \|\boldsymbol{\beta}_*\|_2^2$.*

**Remark:** In the under-parameterized regime, the test error $\mathsf{R}^{\mathsf{LS}}_0$ is a linear function of the norm $\mathsf{N}^{\mathsf{LS}}_0$. In the over-parameterized regime, $\mathsf{R}^{\mathsf{LS}}_0$ and $\mathsf{N}^{\mathsf{LS}}_0$ formulates a rectangular hyperbola: $\mathsf{R}^{\mathsf{LS}}_0$ decreases with $\mathsf{N}^{\mathsf{LS}}_0$ if $\mathsf{N}^{\mathsf{LS}}_0 < \|\boldsymbol{\beta}_*\|_2^2 - \sigma^2$ while $\mathsf{R}^{\mathsf{LS}}_0$ increases with $\mathsf{N}^{\mathsf{LS}}_0$ if $\mathsf{N}^{\mathsf{LS}}_0 > \|\boldsymbol{\beta}_*\|_2^2 - \sigma^2$.

*Proof of Corollary D.8.* According to Proposition B.6 and Corollary D.5, for minimum $\ell_2$-norm estimator and $\boldsymbol{\Sigma} = \boldsymbol{I}_d$, for the under-parameterized regime ($d < n$), we have

$$\mathsf{B}^{\mathsf{LS}}_{\mathsf{R},0} = 0 \, , \quad \mathsf{V}^{\mathsf{LS}}_{\mathsf{R},0} = \frac{\sigma^2 d}{n - d} \, ; \qquad \mathsf{B}^{\mathsf{LS}}_{\mathsf{N},0} = \|\boldsymbol{\beta}_*\|_2^2 \, , \quad \mathsf{V}^{\mathsf{LS}}_{\mathsf{N},0} = \frac{\sigma^2 d}{n - d} \, .$$

From these expressions, we can conclude that

$$\mathsf{R}^{\mathsf{LS}}_0 = \mathsf{B}^{\mathsf{LS}}_{\mathsf{R},0} + \mathsf{V}^{\mathsf{LS}}_{\mathsf{R},0} = \frac{\sigma^2 d}{n - d} \, ; \qquad \mathsf{N}^{\mathsf{LS}}_0 = \mathsf{B}^{\mathsf{LS}}_{\mathsf{N},0} + \mathsf{V}^{\mathsf{LS}}_{\mathsf{N},0} = \|\boldsymbol{\beta}_*\|_2^2 + \frac{\sigma^2 d}{n - d} \, .$$

Finally, in the under-parameterized regime, it follows that

$$\mathsf{R}^{\mathsf{LS}}_0 = \mathsf{N}^{\mathsf{LS}}_0 - \|\boldsymbol{\beta}_*\|_2^2 \, .$$

In the over-parameterized regime ($d > n$), the effective regularization $\lambda_*$ will have an explicit formulation as $\lambda_* = \frac{d-n}{n}$, thus for the bias and variance of the test error, we have

$$\mathsf{B}^{\mathsf{LS}}_{\mathsf{R},0} = \frac{\lambda_n^2 \langle \boldsymbol{\beta}_*, \boldsymbol{\Sigma}(\boldsymbol{\Sigma} + \lambda_n \boldsymbol{I})^{-2} \boldsymbol{\beta}_* \rangle}{1 - n^{-1}\mathrm{Tr}(\boldsymbol{\Sigma}^2(\boldsymbol{\Sigma} + \lambda_n)^{-2})} = \frac{\lambda_n^2 \frac{1}{(1+\lambda_n)^2} \|\boldsymbol{\beta}_*\|_2^2}{1 - \frac{1}{n}\frac{d}{(1+\lambda_n)^2}} = \|\boldsymbol{\beta}_*\|_2^2 \frac{d-n}{d} \, ,$$

$$\mathsf{V}^{\mathsf{LS}}_{\mathsf{R},0} = \frac{\sigma^2 \mathrm{Tr}(\boldsymbol{\Sigma}^2(\boldsymbol{\Sigma} + \lambda_n \boldsymbol{I})^{-2})}{n - \mathrm{Tr}(\boldsymbol{\Sigma}^2(\boldsymbol{\Sigma} + \lambda_n \boldsymbol{I})^{-2})} = \frac{\sigma^2 \frac{d}{(1+\lambda_n)^2}}{n - \frac{d}{(1+\lambda_n)^2}} = \sigma^2 \frac{n}{d-n} \, ,$$

and combining the bias and variance, we have

$$\mathsf{R}^{\mathsf{LS}}_0 = \mathsf{B}^{\mathsf{LS}}_{\mathsf{R},0} + \mathsf{V}^{\mathsf{LS}}_{\mathsf{R},0} = \|\boldsymbol{\beta}_*\|_2^2 \frac{d-n}{d} + \sigma^2 \frac{n}{d-n} \, . \tag{56}$$

---

[6]Due to the complexity of the calculations, we use Mathematica Wolfram to eliminate $n$. The same approach is applied later whenever $n$ or $p$ elimination is required.

For the bias and variance of the norm, we have

$$\mathsf{B}_{\mathsf{N},0}^{\mathsf{LS}} = \langle \boldsymbol{\beta}_*, \boldsymbol{\Sigma}(\boldsymbol{\Sigma} + \lambda_n \boldsymbol{I})^{-1} \boldsymbol{\beta}_* \rangle = \frac{1}{1 + \lambda_n} \|\boldsymbol{\beta}_*\|_2^2 = \|\boldsymbol{\beta}_*\|_2^2 \frac{n}{d},$$

$$\mathsf{V}_{\mathsf{N},0}^{\mathsf{LS}} = \frac{\sigma \mathrm{Tr}(\boldsymbol{\Sigma}(\boldsymbol{\Sigma} + \lambda_n \boldsymbol{I})^{-2})}{n - \mathrm{Tr}(\boldsymbol{\Sigma}^2(\boldsymbol{\Sigma} + \lambda_n \boldsymbol{I})^{-2})} = \frac{\sigma^2 \frac{d}{(1+\lambda_n)^2}}{n - \frac{d}{(1+\lambda_n)^2}} = \sigma^2 \frac{n}{d - n},$$

and combining the bias and variance, we have

$$\mathsf{N}_0^{\mathsf{LS}} = \mathsf{B}_{\mathsf{N},0}^{\mathsf{LS}} + \mathsf{V}_{\mathsf{N},0}^{\mathsf{LS}} = \|\boldsymbol{\beta}_*\|_2^2 \frac{n}{d} + \sigma^2 \frac{n}{d - n}. \tag{57}$$

Finally, combining Eq. (56) and Eq. (57), we eliminate $n$ and thus obtain

$$\mathsf{R}_0^{\mathsf{LS}} = \sqrt{(\mathsf{N}_0^{\mathsf{LS}})^2 - 2(\|\boldsymbol{\beta}_*\|_2^2 - \sigma^2)\mathsf{N}_0^{\mathsf{LS}} + (\|\boldsymbol{\beta}_*\|_2^2 + \sigma^2)^2} - \sigma^2.$$

By taking the derivative of $\mathsf{R}_0^{\mathsf{LS}}$ with respect to $\mathsf{N}_0^{\mathsf{LS}}$, we get

$$\frac{\partial \mathsf{R}_0^{\mathsf{LS}}}{\partial \mathsf{N}_0^{\mathsf{LS}}} = \frac{\mathsf{N}_0^{\mathsf{LS}} - (\|\boldsymbol{\beta}_*\|_2^2 - \sigma^2)}{\sqrt{(\mathsf{N}_0^{\mathsf{LS}})^2 - 2(\|\boldsymbol{\beta}_*\|_2^2 - \sigma^2)\mathsf{N}_0^{\mathsf{LS}} + (\|\boldsymbol{\beta}_*\|_2^2 + \sigma^2)^2}}.$$

From the derivative function, we observe that $\mathsf{R}_0^{\mathsf{LS}}$ decreases monotonically with $\mathsf{N}_0^{\mathsf{LS}}$ when $\mathsf{N}_0^{\mathsf{LS}} < \|\boldsymbol{\beta}_*\|_2^2 - \sigma^2$, and increases monotonically with $\mathsf{N}_0^{\mathsf{LS}}$ when $\mathsf{N}_0^{\mathsf{LS}} > \|\boldsymbol{\beta}_*\|_2^2 - \sigma^2$. $\qquad\square$

**Relationship for min-$\ell_2$-norm interpolator in the under-parameterized regime.** Next, we consider the min-norm estimator, and we find that for the min-norm estimator, in the under-parameterized regime, the relationship between risk and norm is linear, and this linearity is independent of the data distribution.

**Proposition D.9** (Relationship for min-$\ell_2$-norm interpolator in the **under-parameterized** regime). *The deterministic equivalents $\mathsf{R}_0^{\mathsf{LS}}$ and $\mathsf{N}_0^{\mathsf{LS}}$, in under-parameterized regimes ($d < n$) admit the linear relationship*

$$\mathsf{R}_0^{\mathsf{LS}} = d \left( \mathsf{N}_0^{\mathsf{LS}} - \|\boldsymbol{\beta}_*\|_2^2 \right) / \mathrm{Tr}(\boldsymbol{\Sigma}^{-1}).$$

*Proof of Proposition D.9.* According to Proposition B.6 and Corollary D.5, for minimum $\ell_2$-norm estimator, in the under-parameterized regime ($d < n$), we have

$$\mathsf{B}_{\mathsf{R},0}^{\mathsf{LS}} = 0, \quad \mathsf{V}_{\mathsf{R},0}^{\mathsf{LS}} = \frac{\sigma^2 d}{n - d}; \qquad \mathsf{B}_{\mathsf{N},0}^{\mathsf{LS}} = \|\boldsymbol{\beta}_*\|_2^2, \quad \mathsf{V}_{\mathsf{N},0}^{\mathsf{LS}} = \frac{\sigma^2 \mathrm{Tr}(\boldsymbol{\Sigma}^{-1})}{n - d}.$$

From these expressions, we can conclude that

$$\mathsf{R}_0^{\mathsf{LS}} = \mathsf{B}_{\mathsf{R},0}^{\mathsf{LS}} + \mathsf{V}_{\mathsf{R},0}^{\mathsf{LS}} = \frac{\sigma^2 d}{n - d}; \qquad \mathsf{N}_0^{\mathsf{LS}} = \mathsf{B}_{\mathsf{N},0}^{\mathsf{LS}} + \mathsf{V}_{\mathsf{N},0}^{\mathsf{LS}} = \|\boldsymbol{\beta}_*\|_2^2 + \frac{\sigma^2 \mathrm{Tr}(\boldsymbol{\Sigma}^{-1})}{n - d}.$$

Finally, combing the above equation and eliminate $n$, in the under-parameterized regime, it follows that

$$\mathsf{R}_0^{\mathsf{LS}} = \frac{d}{\mathrm{Tr}(\boldsymbol{\Sigma}^{-1})} \left( \mathsf{N}_0^{\mathsf{LS}} - \|\boldsymbol{\beta}_*\|_2^2 \right). \tag{58}$$

$\qquad\square$

The relationship in the over-parameterized regime is more complicated. We present it in the special case of isotropic features in Corollary D.8 of Proposition D.7, and we also give an approximation in Proposition D.10 under the power-law assumption.

**Assumption 6** (Power-law assumption). For the covariance matrix $\boldsymbol{\Sigma}$ and the target function $\boldsymbol{\beta}_*$, we assume that $\sigma_k(\boldsymbol{\Sigma}) = k^{-\alpha}, \alpha > 0$ and $\boldsymbol{\beta}_{*,k} = k^{-\alpha\beta/2}, \beta \in \mathbb{R}$.

This assumption is close to classical source condition and capacity condition [9] and is similarly used in [45, Assumption 1].

**Relationship under power-law assumption.** Instead of assuming $\boldsymbol{\Sigma} = \boldsymbol{I}_d$, we next consider power-law features in Assumption 6 and characterize the relationship.

**Proposition D.10** (Power-law features for min-$\ell_2$ norm estimator). *Under Assumption 6, in the over-parameterized regime $(d > n)$, we consider some special cases for analytic formulation: if $\alpha = 1$ and $\beta = 0$, when $n \to d$, we have[7]*

$$\mathsf{V}_{\mathsf{R},0}^{\mathsf{LS}} \approx \frac{2(\mathsf{V}_{\mathsf{N},0}^{\mathsf{LS}})^2}{d\mathsf{V}_{\mathsf{N},0}^{\mathsf{LS}} - d^2\sigma^2}, \quad \mathsf{B}_{\mathsf{R},0}^{\mathsf{LS}} \approx \frac{2\mathsf{B}_{\mathsf{N},0}^{\mathsf{LS}}(d - \mathsf{B}_{\mathsf{N},0}^{\mathsf{LS}})}{d^2}.$$

**Remark:** The relationship between $\mathsf{R}_0^{\mathsf{LS}}$ and $\mathsf{N}_0^{\mathsf{LS}}$ is still linear in the under-parameterized regime, but is quite complex in the over-parameterized regime.

*Proof of Proposition D.10.* In the over-parameterized regime $(d > n)$, according to Proposition B.6 and Corollary D.5, under Assumption 6, we have

$$\mathsf{B}_{\mathsf{R},0}^{\mathsf{LS}} = \frac{\lambda_n^2 \langle \boldsymbol{\beta}_*, \boldsymbol{\Sigma}(\boldsymbol{\Sigma} + \lambda_n \boldsymbol{I})^{-2} \boldsymbol{\beta}_* \rangle}{1 - n^{-1}\mathrm{Tr}(\boldsymbol{\Sigma}^2(\boldsymbol{\Sigma} + \lambda_n \boldsymbol{I})^{-2})} = \frac{\lambda_n^2 \mathrm{Tr}(\boldsymbol{\Sigma}^{1+\beta}(\boldsymbol{\Sigma} + \lambda_n \boldsymbol{I})^{-2})}{1 - n^{-1}\mathrm{Tr}(\boldsymbol{\Sigma}^2(\boldsymbol{\Sigma} + \lambda_n \boldsymbol{I})^{-2})},$$

$$\mathsf{V}_{\mathsf{R},0}^{\mathsf{LS}} = \frac{\sigma^2 \mathrm{Tr}\left(\boldsymbol{\Sigma}^2(\boldsymbol{\Sigma} + \lambda_n \boldsymbol{I})^{-2}\right)}{n - \mathrm{Tr}\left(\boldsymbol{\Sigma}^2(\boldsymbol{\Sigma} + \lambda_n \boldsymbol{I})^{-2}\right)},$$

$$\mathsf{B}_{\mathsf{N},0}^{\mathsf{LS}} = \langle \boldsymbol{\beta}_*, \boldsymbol{\Sigma}(\boldsymbol{\Sigma} + \lambda_n \boldsymbol{I})^{-1} \boldsymbol{\beta}_* \rangle = \mathrm{Tr}(\boldsymbol{\Sigma}^{1+\beta}(\boldsymbol{\Sigma} + \lambda_n \boldsymbol{I})^{-1}),$$

$$\mathsf{V}_{\mathsf{N},0}^{\mathsf{LS}} = \frac{\sigma^2 \mathrm{Tr}\left(\boldsymbol{\Sigma}(\boldsymbol{\Sigma} + \lambda_n \boldsymbol{I})^{-2}\right)}{n - \mathrm{Tr}\left(\boldsymbol{\Sigma}^2(\boldsymbol{\Sigma} + \lambda_n \boldsymbol{I})^{-2}\right)}.$$

To compute these quantities, here we introduce the following continuum approximations to eigensums.

$$\int_1^{d+1} \frac{k^{-\alpha}}{k^{-\alpha} + \lambda_n} \, \mathrm{d}k \leq \mathrm{Tr}(\boldsymbol{\Sigma}(\boldsymbol{\Sigma} + \lambda_n)^{-1}) = \sum_{i=1}^d \frac{\sigma_i}{\sigma_i + \lambda_n} \leq \int_0^d \frac{k^{-\alpha}}{k^{-\alpha} + \lambda_n} \, \mathrm{d}k, \quad (59)$$

due to the fact that the integrand is non-increasing function of $k$. Similarly, we also have

$$\int_1^{d+1} \frac{k^{-2\alpha}}{(k^{-\alpha} + \lambda_n)^2} \, \mathrm{d}k \leq \mathrm{Tr}(\boldsymbol{\Sigma}^2(\boldsymbol{\Sigma} + \lambda_n)^{-2}) = \sum_{i=1}^d \frac{\sigma_i^2}{(\sigma_i + \lambda_n)^2} \leq \int_0^d \frac{k^{-2\alpha}}{(k^{-\alpha} + \lambda_n)^2} \, \mathrm{d}k. \quad (60)$$

We consider some special cases that are useful for discussion. When $\alpha = 1$, we have

$$\frac{\log(1 + d\lambda_n + \lambda_n) - \log(1 + \lambda_n)}{\lambda_n} \leq \mathrm{Tr}(\boldsymbol{\Sigma}(\boldsymbol{\Sigma} + \lambda_n)^{-1}) \leq \frac{\log(1 + d\lambda_n)}{\lambda_n}, \quad (61)$$

$$\frac{d+1}{\lambda_n d + \lambda_n + 1} - \frac{1}{\lambda_n + 1} \leq \mathrm{Tr}(\boldsymbol{\Sigma}^2(\boldsymbol{\Sigma} + \lambda_n)^{-2}) = \sum_{i=1}^d \frac{\sigma_i^2}{(\sigma_i + \lambda_n)^2} \leq \frac{d}{1 + d\lambda_n}. \quad (62)$$

Recall that $\lambda_n$ is defined by $\mathrm{Tr}(\boldsymbol{\Sigma}(\boldsymbol{\Sigma} + \lambda_n \boldsymbol{I})^{-1}) = n$. Using Eq. (59), we have

$$\frac{\log(1 + d\lambda_n)}{\lambda_n} \approx n.$$

Observe that as $n \to d$, $\lambda_n \to 0$, allowing us to apply a Taylor expansion:

$$\frac{\log(1 + d\lambda_n)}{\lambda_n} \approx \frac{d\lambda_n - \frac{1}{2}(d\lambda_n)^2}{\lambda_n} = d - \frac{1}{2}d^2\lambda_n.$$

Based on this approximation, $\lambda_n$ can be expressed as

$$\lambda_n \approx \frac{2(d - n)}{d^2}.$$

---

[7]The symbol $\approx$ here represents two types of approximations: i) approximation for self-consistent equations; ii) Taylor approximation of logarithmic function around zero (related to $n \to d$).

In the following discussion, we consider the case $n \to d$. Thus, we have the approximation

$$\mathrm{Tr}(\boldsymbol{\Sigma}(\boldsymbol{\Sigma} + \lambda_n)^{-1}) \approx n\,, \quad \mathrm{Tr}(\boldsymbol{\Sigma}^2(\boldsymbol{\Sigma} + \lambda_n)^{-2}) \approx \frac{d}{1 + d\lambda_n}\,.$$

Then we have

$$\mathsf{V}^{\mathsf{LS}}_{\mathsf{R},0} = \frac{\sigma^2 \mathrm{Tr}\left(\boldsymbol{\Sigma}^2(\boldsymbol{\Sigma} + \lambda_n \boldsymbol{I})^{-2}\right)}{n - \mathrm{Tr}\left(\boldsymbol{\Sigma}^2(\boldsymbol{\Sigma} + \lambda_n \boldsymbol{I})^{-2}\right)} \approx \frac{\sigma^2 \frac{d}{1 + d\lambda_n}}{n - \frac{d}{1 + d\lambda_n}} = \frac{\sigma^2 d}{n + d(n\lambda_n - 1)}\,,$$

$$\mathsf{V}^{\mathsf{LS}}_{\mathsf{N},0} = \frac{\sigma^2 \mathrm{Tr}\left(\boldsymbol{\Sigma}(\boldsymbol{\Sigma} + \lambda_n \boldsymbol{I})^{-2}\right)}{n - \mathrm{Tr}\left(\boldsymbol{\Sigma}^2(\boldsymbol{\Sigma} + \lambda_n \boldsymbol{I})^{-2}\right)} \approx \frac{\sigma^2 \frac{1}{\lambda_n}(d - \frac{1}{2}d^2\lambda_n - \frac{d}{1 + d\lambda_n})}{n - \frac{d}{1 + d\lambda_n}} = \frac{\sigma^2 d^2(d\lambda_n - 1)}{2(n + d(n\lambda_n - 1))}\,.$$

Use these two formulation to eliminate $n$, we obtain

$$\mathsf{V}^{\mathsf{LS}}_{\mathsf{R},0} \approx \frac{2(\mathsf{V}^{\mathsf{LS}}_{\mathsf{N},0})^2}{d\mathsf{V}^{\mathsf{LS}}_{\mathsf{N},0} - d^2\sigma^2}\,.$$

Next we discuss the situation under different $\beta$.

For $\beta = 0$, we have

$$\mathsf{B}^{\mathsf{LS}}_{\mathsf{R},0} = \frac{\lambda_n^2 \mathrm{Tr}(\boldsymbol{\Sigma}(\boldsymbol{\Sigma} + \lambda_n \boldsymbol{I})^{-2})}{1 - n^{-1}\mathrm{Tr}(\boldsymbol{\Sigma}^2(\boldsymbol{\Sigma} + \lambda_n \boldsymbol{I})^{-2})} \approx \frac{\lambda_n(d - \frac{1}{2}d^2\lambda_n - \frac{d}{1 + d\lambda_n})}{1 - \frac{d}{n(1 + d\lambda_n)}} = n\lambda_n\,,$$

$$\mathsf{B}^{\mathsf{LS}}_{\mathsf{N},0} = \mathrm{Tr}(\boldsymbol{\Sigma}(\boldsymbol{\Sigma} + \lambda_n \boldsymbol{I})^{-1}) \approx d - \frac{1}{2}d^2\lambda_n\,,$$

Use these two formulation to eliminate $n$, we obtain

$$\mathsf{B}^{\mathsf{LS}}_{\mathsf{R},0} \approx \frac{2\mathsf{B}^{\mathsf{LS}}_{\mathsf{N},0}(d - \mathsf{B}^{\mathsf{LS}}_{\mathsf{N},0})}{d^2}\,.$$

For $\beta = 1$, we have

$$\mathsf{B}^{\mathsf{LS}}_{\mathsf{R},0} = \frac{\lambda_n^2 \mathrm{Tr}(\boldsymbol{\Sigma}^2(\boldsymbol{\Sigma} + \lambda_n \boldsymbol{I})^{-2})}{1 - n^{-1}\mathrm{Tr}(\boldsymbol{\Sigma}^2(\boldsymbol{\Sigma} + \lambda_n \boldsymbol{I})^{-2})} \approx \frac{\lambda_n^2 \frac{d}{1 + d\lambda_n}}{1 - \frac{d}{n(1 + d\lambda_n)}} = \frac{nd\lambda_n^2}{n(1 + d\lambda_n) - d}\,,$$

$$\mathsf{B}^{\mathsf{LS}}_{\mathsf{N},0} = \mathrm{Tr}(\boldsymbol{\Sigma}^2(\boldsymbol{\Sigma} + \lambda_n \boldsymbol{I})^{-1}) = \mathrm{Tr}(\boldsymbol{\Sigma}) - \lambda_n \mathrm{Tr}(\boldsymbol{\Sigma}(\boldsymbol{\Sigma} + \lambda_n \boldsymbol{I})^{-1}) \approx \mathrm{Tr}(\boldsymbol{\Sigma}) - n\lambda_n\,.$$

Use these two formulation to eliminate $n$, we obtain

$$\mathsf{B}^{\mathsf{LS}}_{\mathsf{R},0} \approx \frac{2\sqrt{(\mathsf{B}^{\mathsf{LS}}_{\mathsf{N},0})^2 - 2\mathrm{Tr}(\boldsymbol{\Sigma})\mathsf{B}^{\mathsf{LS}}_{\mathsf{N},0} + \mathrm{Tr}(\boldsymbol{\Sigma})^2}}{\sqrt{d^2 + 2d^2\mathsf{B}^{\mathsf{LS}}_{\mathsf{N},0} - 2d^2\mathrm{Tr}(\boldsymbol{\Sigma})}} = \frac{2(\mathsf{B}^{\mathsf{LS}}_{\mathsf{N},0} - \mathrm{Tr}(\boldsymbol{\Sigma}))}{d\sqrt{1 + 2\mathsf{B}^{\mathsf{LS}}_{\mathsf{N},0} - 2\mathrm{Tr}(\boldsymbol{\Sigma})}}\,.$$

For $\beta = -1$, we need to use another two continuum approximations to eigensums

$$\mathrm{Tr}((\boldsymbol{\Sigma} + \lambda_n)^{-1}) = \sum_{i=1}^{d} \frac{1}{\sigma_i + \lambda_n} \approx \int_0^d \frac{1}{k^{-\alpha} + \lambda_n}\,\mathrm{d}k = \frac{d\lambda_n - \log(1 + d\lambda_n)}{\lambda_n^2}\,,$$

$$\mathrm{Tr}((\boldsymbol{\Sigma} + \lambda_n)^{-2}) = \sum_{i=1}^{d} \frac{1}{(\sigma_i + \lambda_n)^2} \approx \int_0^d \frac{1}{(k^{-\alpha} + \lambda_n)^2}\,\mathrm{d}k = \frac{\frac{d\lambda_n(2 + d\lambda_n)}{1 + d\lambda_n} - 2\log(1 + d\lambda_n)}{\lambda_n^3}\,.$$

Once again, we apply the Taylor expansion, but this time expanding to the third order

$$\log(1 + d\lambda_n) \approx d\lambda_n - \frac{1}{2}(d\lambda_n)^2 + \frac{1}{3}(d\lambda_n)^3\,.$$

Then we have

$$\mathrm{Tr}((\boldsymbol{\Sigma} + \lambda_n)^{-1}) \approx \frac{d\lambda_n - \log(1 + d\lambda_n)}{\lambda_n^2} \approx \frac{1}{2}d^2 - \frac{1}{3}d^3\lambda_n\,,$$

$$\mathrm{Tr}((\mathbf{\Sigma} + \lambda_n)^{-2}) \approx \frac{\frac{d\lambda_n(2+d\lambda_n)}{1+d\lambda_n} - 2\log(1+d\lambda_n)}{\lambda_n^3} = \frac{\frac{1}{3}d^3 - \frac{2}{3}d^4\lambda_n}{1+d\lambda_n}.$$

Using the approximation sated above, we have

$$\mathsf{B}_{\mathsf{R},0}^{\mathsf{LS}} = \frac{\lambda_n^2 \mathrm{Tr}((\mathbf{\Sigma}+\lambda_n \mathbf{I})^{-2})}{1 - n^{-1}\mathrm{Tr}(\mathbf{\Sigma}^2(\mathbf{\Sigma}+\lambda_n \mathbf{I})^{-2})} \approx \frac{\lambda_n^2\left((\frac{1}{3}d^3 - \frac{2}{3}d^4\lambda_n)/(1+d\lambda_n)\right)}{1 - \frac{d}{n(1+d\lambda_n)}},$$

$$\mathsf{B}_{\mathsf{N},0}^{\mathsf{LS}} = \mathrm{Tr}((\mathbf{\Sigma}+\lambda_n \mathbf{I})^{-1}) = \frac{1}{2}d^2 - \frac{1}{3}d^3\lambda_n.$$

Use these two formulation to eliminate $n$, we obtain

$$\mathsf{B}_{\mathsf{R},0}^{\mathsf{LS}} \approx \frac{216(\mathsf{B}_{\mathsf{N},0}^{\mathsf{LS}})^4 - 324d^2(\mathsf{B}_{\mathsf{N},0}^{\mathsf{LS}})^3 + 126d^4(\mathsf{B}_{\mathsf{N},0}^{\mathsf{LS}})^2 + d^6\mathsf{B}_{\mathsf{N},0}^{\mathsf{LS}} - 5d^8}{2d^5(6\mathsf{B}_{\mathsf{N},0}^{\mathsf{LS}} - d^2)}.$$

$\square$

Here we present some experimental results to check the relationship between $\mathsf{B}_{\mathsf{R},0}^{\mathsf{LS}}$ and $\mathsf{B}_{\mathsf{N},0}^{\mathsf{LS}}$, as well as $\mathsf{V}_{\mathsf{R},0}^{\mathsf{LS}}$ and $\mathsf{V}_{\mathsf{N},0}^{\mathsf{LS}}$, see Fig. 6. We can see that our approximate relationship on variance (see the red line in Fig. 6(d)) provides the precise estimation. For the bias (see the left three figures of Fig. 6), our approximate relationship is accurate if $\mathsf{B}_{\mathsf{N},0}^{\mathsf{LS}}$ is large.

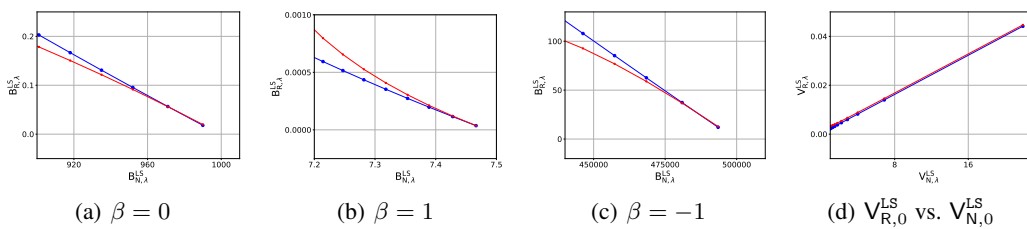

(a) $\beta = 0$      (b) $\beta = 1$      (c) $\beta = -1$      (d) $\mathsf{V}_{\mathsf{R},0}^{\mathsf{LS}}$ vs. $\mathsf{V}_{\mathsf{N},0}^{\mathsf{LS}}$

Figure 6: The left three figures (a) (b) (c) show the relationship between $\mathsf{B}_{\mathsf{R},0}^{\mathsf{LS}}$ and $\mathsf{B}_{\mathsf{N},0}^{\mathsf{LS}}$ when $\alpha = 1$ and $\beta$ takes on different values. The figure (d) shows the relationship between $\mathsf{V}_{\mathsf{R},0}^{\mathsf{LS}}$ and $\mathsf{V}_{\mathsf{N},0}^{\mathsf{LS}}$ when $\alpha = 1$. The blue line is the relationship obtained by deterministic equivalent experiments, and the red line is the approximate relationship we give.

## E   Proofs for random feature ridge regression

In this section, we provide the proof of deterministic equivalence for random feature ridge regression in both the asymptotic (Appendix E.1) and non-asymptotic (Appendix E.2) settings. Additionally, we provide the proof of the relationship between test risk and the $\ell_2$ norm given in the main text, as detailed in Appendix E.3.

Though the results [1] are for linear regression, we can still use the results for RFMs, which requires some knowledge from Eqs. (27) and (28).

We firstly confirm that Assumption 3 in Appendix B.1, used to derive all asymptotic results, can be replaced by the Hanson-Wright assumption employed in the non-asymptotic analysis. It is evident that Eqs. (8) and (10) are obtained directly by taking the limits of Eqs. (27) and (28) as $n \to \infty$.

Additionally, a key step in the proof of Eqs. (9) and (11) in [1] involves showing that $\Delta$ is almost surely negligible, where $\Delta$ is defined as

$$\Delta = \frac{1}{n}\sum_{i=1}^{n} \frac{\boldsymbol{x}_i\boldsymbol{x}_i^\top (\widehat{\mathbf{\Sigma}}_{-i} - z\boldsymbol{I})^{-1} - \mathbf{\Sigma}(\widehat{\mathbf{\Sigma}} - z\boldsymbol{I})^{-1}}{1 + \boldsymbol{x}_i^\top (n\widehat{\mathbf{\Sigma}}_{-i} - nz\boldsymbol{I})^{-1}\boldsymbol{x}_i},$$

with $\widehat{\mathbf{\Sigma}} = \frac{1}{n}\sum_{i=1}^{n}\boldsymbol{x}_i\boldsymbol{x}_i^\top$, $\widehat{\mathbf{\Sigma}}_{-i} = \frac{1}{n}\sum_{j\neq i}\boldsymbol{x}_j\boldsymbol{x}_j^\top$, and $z \in \mathbb{R}$.

In the analysis of [1], the negligibility of $\Delta$ arises from the assumption that the components of $\boldsymbol{x}_i$ follow a sub-Gaussian distribution, which leads to the Hanson-Wright inequality

$$\mathbb{P}\left[\left|\boldsymbol{x}_i^\top \boldsymbol{x}_i - \mathrm{tr}(\mathbf{\Sigma})\right| \leq c\left(t\|\mathbf{\Sigma}\|_{\mathrm{op}} + \sqrt{t}\|\mathbf{\Sigma}\|_F\right)\right] \geq 1 - 2e^{-t}.$$

In this way, Assumption 4 is also sufficient to establish the negligibility of $\Delta$.

After obtaining Eqs. (8) and (10) and the negligibility of $\Delta$, we can follow the argument of [1] and derive the rest asymptotic deterministic equivalence.

Finally, with these observations, we can eliminate the reliance on Assumption 3 and instead rely solely on Assumption 4 to derive all the asymptotic results.

### E.1 Asymptotic deterministic equivalence for random features ridge regression

In this section, we establish the asymptotic approximation guarantees for random feature regression in terms of its $\ell_2$-norm based capacity. Before presenting the proof of Theorem 3.1, we firstly give the proof of the bias-variance decomposition.

*Proof.* Here we give the bias-variance decomposition of $\mathbb{E}_\varepsilon \|\hat{a}\|_2^2$. The formulation of $\mathbb{E}_\varepsilon \|\hat{a}\|_2^2$ is given by
$$\mathbb{E}_\varepsilon \|\hat{a}\|_2^2 = \mathbb{E}_\varepsilon \|(\boldsymbol{Z}^\top \boldsymbol{Z} + \lambda \boldsymbol{I})^{-1} \boldsymbol{Z}^\top \boldsymbol{y}\|_2^2 \,,$$
which admits a similar bias-variance decomposition
$$\begin{aligned}
\mathbb{E}_\varepsilon \|\hat{a}\|_2^2 &= \mathbb{E}_\varepsilon \|(\boldsymbol{Z}^\top \boldsymbol{Z} + \lambda \boldsymbol{I})^{-1} \boldsymbol{Z}^\top (\boldsymbol{G}\boldsymbol{\theta}_* + \varepsilon)\|_2^2 \\
&= \|(\boldsymbol{Z}^\top \boldsymbol{Z} + \lambda \boldsymbol{I})^{-1} \boldsymbol{Z}^\top \boldsymbol{G}\boldsymbol{\theta}_*\|_2^2 + \mathbb{E}_\varepsilon \|(\boldsymbol{Z}^\top \boldsymbol{Z} + \lambda \boldsymbol{I})^{-1} \boldsymbol{Z}^\top \varepsilon\|_2^2 \\
&= \langle \boldsymbol{\theta}_*, \boldsymbol{G}^\top \boldsymbol{Z}(\boldsymbol{Z}^\top \boldsymbol{Z} + \lambda \boldsymbol{I})^{-2} \boldsymbol{Z}^\top \boldsymbol{G}\boldsymbol{\theta}_* \rangle + \sigma^2 \mathrm{Tr}\left(\boldsymbol{Z}^\top \boldsymbol{Z}(\boldsymbol{Z}^\top \boldsymbol{Z} + \lambda \boldsymbol{I})^{-2}\right) \\
&=: \mathcal{B}_{\mathcal{N},\lambda}^{\mathrm{RFM}} + \mathcal{V}_{\mathcal{N},\lambda}^{\mathrm{RFM}} \,.
\end{aligned}$$

Accordingly, we conclude the proof. $\qquad\square$

Now we are ready to present the proof of Theorem 3.1 as below.

*Proof of Theorem 3.1.* We give the asymptotic deterministic equivalents for the norm from the bias $\mathcal{B}_{\mathcal{N},\lambda}^{\mathrm{RFM}}$ and variance $\mathcal{V}_{\mathcal{N},\lambda}^{\mathrm{RFM}}$, respectively. We provide asymptotic expansions in two steps, by first considering the deterministic equivalent over $\boldsymbol{G}$, and then over $\boldsymbol{F}$.

Under Assumption 1, we can apply Propositions B.3 and B.4 and Corollaries D.2 and D.3 directly in the proof below.

**Deterministic equivalent over $\boldsymbol{G}$:** For the bias term, we use Eq. (13) in Proposition B.4 with $\boldsymbol{T} = \boldsymbol{G}$, $\boldsymbol{\Sigma} = \boldsymbol{F}^\top \boldsymbol{F}$, $\boldsymbol{A} = \boldsymbol{\theta}_* \boldsymbol{\theta}_*^\top$ and $\boldsymbol{B} = \boldsymbol{F}^\top \boldsymbol{F}$ and obtain

$$\begin{aligned}
\mathcal{B}_{\mathcal{N},\lambda}^{\mathrm{RFM}} &= \langle \boldsymbol{\theta}_*, \boldsymbol{G}^\top \boldsymbol{Z}(\boldsymbol{Z}^\top \boldsymbol{Z} + \lambda \boldsymbol{I})^{-2} \boldsymbol{Z}^\top \boldsymbol{G}\boldsymbol{\theta}_* \rangle \\
&= \mathrm{Tr}(\boldsymbol{\theta}_*^\top \boldsymbol{G}^\top \boldsymbol{Z}(\boldsymbol{Z}^\top \boldsymbol{Z} + \lambda \boldsymbol{I})^{-2} \boldsymbol{Z}^\top \boldsymbol{G}\boldsymbol{\theta}_*) \\
&= p\mathrm{Tr}(\boldsymbol{\theta}_* \boldsymbol{\theta}_*^\top \boldsymbol{G}^\top (\boldsymbol{G}\boldsymbol{F}^\top \boldsymbol{F}\boldsymbol{G}^\top + p\lambda \boldsymbol{I})^{-1} \boldsymbol{G}\boldsymbol{F}^\top \boldsymbol{F}\boldsymbol{G}^\top (\boldsymbol{G}\boldsymbol{F}^\top \boldsymbol{F}\boldsymbol{G}^\top + p\lambda \boldsymbol{I})^{-1} \boldsymbol{G}) \\
&\sim p\underbrace{\mathrm{Tr}(\boldsymbol{\theta}_* \boldsymbol{\theta}_*^\top (\boldsymbol{F}^\top \boldsymbol{F} + \nu_1 \boldsymbol{I})^{-1} \boldsymbol{F}^\top \boldsymbol{F}(\boldsymbol{F}^\top \boldsymbol{F} + \nu_1 \boldsymbol{I})^{-1})}_{\mathrm{I}_1} \qquad (63) \\
&\quad + p\nu_1^2 \underbrace{\mathrm{Tr}(\boldsymbol{\theta}_* \boldsymbol{\theta}_*^\top (\boldsymbol{F}^\top \boldsymbol{F} + \nu_1 \boldsymbol{I})^{-2})}_{:=I_2} \cdot \underbrace{\mathrm{Tr}(\boldsymbol{F}^\top \boldsymbol{F}(\boldsymbol{F}^\top \boldsymbol{F} + \nu_1 \boldsymbol{I})^{-2})}_{:=I_3} \cdot \frac{1}{n - \widehat{\mathrm{df}}_2(\nu_1)} \,,
\end{aligned}$$

where $\nu_1$ defined by $\nu_1(1 - \frac{1}{n}\widehat{\mathrm{df}}_1(\nu_1)) \sim \frac{p\lambda}{n}$, $\widehat{\mathrm{df}}_1(\nu_1)$ and $\widehat{\mathrm{df}}_2(\nu_1)$ are degrees of freedom associated to $\boldsymbol{F}^\top \boldsymbol{F}$ in Definition B.2.

For the variance term, we use Eq. (54) with $\boldsymbol{T} = \boldsymbol{G}$ in Proposition B.4, $\boldsymbol{A} = \boldsymbol{F}^\top \boldsymbol{F}$, $\boldsymbol{\Sigma} = \boldsymbol{F}^\top \boldsymbol{F}$ and obtain
$$\begin{aligned}
\mathcal{V}_{\mathcal{N},\lambda}^{\mathrm{RFM}} &= \sigma^2 \mathrm{Tr}\left(\boldsymbol{Z}^\top \boldsymbol{Z}(\boldsymbol{Z}^\top \boldsymbol{Z} + \lambda \boldsymbol{I})^{-2}\right) = \sigma^2 \mathrm{Tr}\left(\boldsymbol{Z}\boldsymbol{Z}^\top (\boldsymbol{Z}\boldsymbol{Z}^\top + \lambda \boldsymbol{I})^{-2}\right) \\
&= \sigma^2 p\mathrm{Tr}\left(\boldsymbol{G}\boldsymbol{F}^\top \boldsymbol{F}\boldsymbol{G}^\top (\boldsymbol{G}\boldsymbol{F}^\top \boldsymbol{F}\boldsymbol{G}^\top + p\lambda \boldsymbol{I})^{-2}\right) \\
&\sim \sigma^2 p\frac{\mathrm{Tr}(\boldsymbol{F}^\top \boldsymbol{F}(\boldsymbol{F}^\top \boldsymbol{F} + \nu_1 \boldsymbol{I})^{-2})}{n - \widehat{\mathrm{df}}_2(\nu_1)} \,.
\end{aligned}$$

**Deterministic equivalent over $F$:** In the next, we aim to eliminate the randomness over $F$ in Eq. (63) from the bias part. First our result depends on the asymptotic equivalents for $\widehat{\mathrm{df}}_1(\nu_1)$ and $\widehat{\mathrm{df}}_2(\nu_1)$. For $\widehat{\mathrm{df}}_1(\nu_1)$, we use Eq. (8) in Proposition B.3 with $X = F$ and obtain

$$\widehat{\mathrm{df}}_1(\nu_1) = \mathrm{Tr}(F^\top F(F^\top F + \nu_1 I)^{-1}) \sim \mathrm{Tr}(\Lambda(\Lambda + \nu_2 I)^{-1}) = \mathrm{df}_1(\nu_2)\,,$$

where $\nu_2$ defined by $\nu_2(1 - \frac{1}{p}\mathrm{df}_1(\nu_2)) \sim \frac{\nu_1}{p}$. Hence $\nu_1$ can be defined by $\nu_1(1 - \frac{1}{n}\mathrm{df}_1(\nu_2)) \sim \frac{p\lambda}{n}$ from Eq. (3).

For $\widehat{\mathrm{df}}_2(\nu_1)$, we use Eq. (9) in Proposition B.3 with $X = F$, $A = B = I$ and obtain

$$\begin{aligned}
\widehat{\mathrm{df}}_2(\nu_1) =\ & \mathrm{Tr}(F^\top F(F^\top F + \nu_1 I)^{-1} F^\top F(F^\top F + \nu_1 I)^{-1}) \\
\sim\ & \mathrm{Tr}(\Lambda^2(\Lambda + \nu_2 I)^{-2}) + \nu_2^2 \mathrm{Tr}(\Lambda(\Lambda + \nu_2 I)^{-2}) \cdot \mathrm{Tr}(\Lambda^2(\Lambda + \nu_2 I)^{-2}) \cdot \frac{1}{p - \mathrm{df}_2(\nu_2)} \\
=:\ & n\Upsilon(\nu_1, \nu_2)\,.
\end{aligned}$$

$$(64)$$

For $I_3 := \mathrm{Tr}(F^\top F(F^\top F + \nu_1 I)^{-2})$, we use Eq. (52) with $X = F$ and obtain

$$\mathrm{Tr}(F^\top F(F^\top F + \nu_1 I)^{-2}) \sim \mathrm{Tr}(\Lambda(\Lambda + \nu_2 I)^{-2}) \cdot \frac{1}{p - \mathrm{df}_2(\nu_2)}\,. \qquad (65)$$

Then we use Eq. (52) again with $X = F$, $A = \theta_* \theta_*^\top$ to obtain the deterministic equivalent of $I_1$

$$\begin{aligned}
\mathrm{Tr}(\theta_* \theta_*^\top (F^\top F + \nu_1 I)^{-1} F^\top F(F^\top F + \nu_1 I)^{-1}) &= \mathrm{Tr}(\theta_* \theta_*^\top F^\top F(F^\top F + \nu_1 I)^{-2}) \\
&\sim \mathrm{Tr}(\theta_* \theta_*^\top \Lambda(\Lambda + \nu_2 I)^{-2}) \cdot \frac{1}{p - \mathrm{df}_2(\nu_2)} \\
&= \theta_*^\top \Lambda(\Lambda + \nu_2 I)^{-2} \theta_* \cdot \frac{1}{p - \mathrm{df}_2(\nu_2)}\,.
\end{aligned}$$

Further, for $I_2$, use Eq. (11) with $A = \theta_* \theta_*^\top$ and $B = I$, we obtain

$$\begin{aligned}
\mathrm{Tr}(\theta_* \theta_*^\top (F^\top F + \nu_1 I)^{-2}) \sim\ & \frac{\nu_2^2}{\nu_1^2} \mathrm{Tr}(\theta_* \theta_*^\top (\Lambda + \nu_2 I)^{-2}) \\
& + \frac{\nu_2^2}{\nu_1^2} \mathrm{Tr}(\theta_* \theta_*^\top (\Lambda + \nu_2 I)^{-2} \Lambda) \cdot \mathrm{Tr}((\Lambda + \nu_2 I)^{-2} \Lambda) \cdot \frac{1}{p - \mathrm{df}_2(\nu_2)}\,.
\end{aligned}$$

Finally, combine the above equivalents, for the bias, we obtain

$$\begin{aligned}
\mathcal{B}_{\mathcal{N},\lambda}^{\mathrm{RFM}} \sim\ & p\theta_*^\top \Lambda(\Lambda + \nu_2 I)^{-2} \theta_* \cdot \frac{1}{p - \mathrm{df}_2(\nu_2)} \\
& + p\nu_1^2 \left( \frac{\nu_2^2}{\nu_1^2} \mathrm{Tr}(\theta_* \theta_*^\top (\Lambda + \nu_2 I)^{-2}) + \frac{\nu_2^2}{\nu_1^2} \mathrm{Tr}(\theta_* \theta_*^\top (\Lambda + \nu_2 I)^{-2} \Lambda) \cdot \frac{\mathrm{Tr}((\Lambda + \nu_2 I)^{-2} \Lambda)}{p - \mathrm{df}_2(\nu_2)} \right) \\
& \cdot \mathrm{Tr}(\Lambda(\Lambda + \nu_2 I)^{-2}) \cdot \frac{1}{p - \mathrm{df}_2(\nu_2)} \cdot \frac{1}{n - n\Upsilon(\nu_1, \nu_2)} \\
=\ & p\theta_*^\top \Lambda(\Lambda + \nu_2 I)^{-2} \theta_* \cdot \frac{1}{p - \mathrm{df}_2(\nu_2)} \\
& + \frac{p}{n} \left( \nu_2^2 \theta_*^\top (\Lambda + \nu_2 I)^{-2} \theta_* + \nu_2^2 \theta_*^\top \Lambda(\Lambda + \nu_2 I)^{-2} \theta_* \cdot \frac{\mathrm{Tr}(\Lambda(\Lambda + \nu_2 I)^{-2})}{p - \mathrm{df}_2(\nu_2)} \right) \\
& \cdot \mathrm{Tr}(\Lambda(\Lambda + \nu_2 I)^{-2}) \cdot \frac{1}{p - \mathrm{df}_2(\nu_2)} \cdot \frac{1}{1 - \Upsilon(\nu_1, \nu_2)} \\
=\ & \frac{p\langle \theta_*, \Lambda(\Lambda + \nu_2 I)^{-2} \theta_* \rangle}{p - \mathrm{Tr}(\Lambda^2(\Lambda + \nu_2 I)^{-2})} \\
& + \frac{p\chi(\nu_2)}{n} \cdot \frac{\nu_2^2 \left[ \langle \theta_*, (\Lambda + \nu_2 I)^{-2} \theta_* \rangle + \chi(\nu_2) \langle \theta_*, \Lambda(\Lambda + \nu_2 I)^{-2} \theta_* \rangle \right]}{1 - \Upsilon(\nu_1, \nu_2)}\,.
\end{aligned}$$

Similarly, for the variance, using Eq. (64) and Eq. (65) for $I_3$, we have

$$\mathcal{V}_{\mathcal{N},\lambda}^{\text{RFM}} \sim \sigma^2 p \text{Tr}(\mathbf{\Lambda}(\mathbf{\Lambda} + \nu_2 \mathbf{I})^{-2}) \cdot \frac{1}{p - \text{df}_2(\nu_2)} \cdot \frac{1}{n - n\Upsilon(\nu_1, \nu_2)}$$

$$\sim \sigma^2 \frac{\frac{p}{n}\chi(\nu_2)}{1 - \Upsilon(\nu_1, \nu_2)}.$$

Accordingly, we finish the proof. $\qquad\square$

In the next, we present the proof for min-$\ell_2$-norm interpolator under RFMs.

*Proof of Corollary 3.2.* Similar to linear regression, we separate the two regimes $p < n$ and $p > n$ as well. For both of them, we provide asymptotic expansions in two steps, first with respect to $\mathbf{G}$ and then $\mathbf{F}$ in the under-parameterized regime and vice-versa for the over-parameterized regime.

**Under-parameterized regime: Deterministic equivalent over $\mathbf{G}$**  For the variance term, we can use Eq. (54) with $\mathbf{T} = \mathbf{G}$, $\mathbf{\Sigma} = \mathbf{F}^\top \mathbf{F}$, $\mathbf{A} = \mathbf{F}^\top \mathbf{F}$ and obtain

$$
\begin{aligned}
\mathcal{V}_{\mathcal{N},0}^{\text{RFM}} &= \sigma^2 \cdot \text{Tr}(\mathbf{Z}^\top \mathbf{Z}(\mathbf{Z}^\top \mathbf{Z} + \lambda \mathbf{I})^{-2}) \\
&= \sigma^2 \cdot p\text{Tr}(\mathbf{F}\mathbf{G}^\top \mathbf{G}\mathbf{F}^\top (\mathbf{F}\mathbf{G}^\top \mathbf{G}\mathbf{F}^\top + p\lambda \mathbf{I})^{-2}) \\
&= \sigma^2 \cdot p\text{Tr}(\mathbf{F}^\top \mathbf{F}\mathbf{G}^\top (\mathbf{G}\mathbf{F}^\top \mathbf{F}\mathbf{G}^\top + p\lambda \mathbf{I})^{-2}\mathbf{G}) \\
&\sim \sigma^2 \cdot p\text{Tr}(\mathbf{F}^\top \mathbf{F}(\mathbf{F}^\top \mathbf{F} + \tilde{\lambda}\mathbf{I})^{-2}) \cdot \frac{1}{n - p} \\
&\sim \sigma^2 \cdot \text{Tr}((\mathbf{F}\mathbf{F}^\top)^{-1}) \cdot \frac{p}{n - p},
\end{aligned}
$$

where $\tilde{\lambda}$ is defined by

$$\tilde{\lambda}(1 - \frac{1}{n}\widetilde{\text{df}}_1(\tilde{\lambda})) \sim \frac{p\lambda}{n}, \tag{66}$$

where $\widetilde{\text{df}}_1(\tilde{\lambda})$ and $\widetilde{\text{df}}_2(\tilde{\lambda})$ are degrees of freedom associated to $\mathbf{F}^\top \mathbf{F}$. In the under-parameterized regime ($p < n$), when $\lambda$ goes to zero, we have $\tilde{\lambda} \to 0$ and $\widetilde{\text{df}}_2(\tilde{\lambda}) \to p$ [1].

For the bias term, we use Eq. (13) with $\mathbf{T} = \mathbf{G}$, $\mathbf{\Sigma} = \mathbf{F}^\top \mathbf{F}$, $\mathbf{A} = \boldsymbol{\theta}_* \boldsymbol{\theta}_*^\top$, $\mathbf{B} = \mathbf{F}^\top \mathbf{F}$ and then obtain

$$
\begin{aligned}
\mathcal{B}_{\mathcal{N},0}^{\text{RFM}} &= \text{Tr}(\boldsymbol{\theta}_*^\top \mathbf{G}^\top \mathbf{Z}(\mathbf{Z}^\top \mathbf{Z} + \lambda \mathbf{I})^{-2}\mathbf{Z}^\top \mathbf{G}\boldsymbol{\theta}_*) \\
&= p\text{Tr}(\boldsymbol{\theta}_*^\top \mathbf{G}^\top \mathbf{G}\mathbf{F}^\top (\mathbf{F}\mathbf{G}^\top \mathbf{G}\mathbf{F}^\top + p\lambda \mathbf{I})^{-2}\mathbf{F}\mathbf{G}^\top \mathbf{G}\boldsymbol{\theta}_*) \\
&= p\text{Tr}(\boldsymbol{\theta}_*\boldsymbol{\theta}_*^\top \mathbf{G}^\top (\mathbf{G}\mathbf{F}^\top \mathbf{F}\mathbf{G}^\top + p\lambda \mathbf{I})^{-1}\mathbf{G}\mathbf{F}^\top \mathbf{F}\mathbf{G}^\top (\mathbf{G}\mathbf{F}^\top \mathbf{F}\mathbf{G}^\top + p\lambda \mathbf{I})^{-1}\mathbf{G}) \\
&\sim p\text{Tr}(\boldsymbol{\theta}_*\boldsymbol{\theta}_*^\top (\mathbf{F}^\top \mathbf{F} + \tilde{\lambda}\mathbf{I})^{-1}\mathbf{F}^\top \mathbf{F}(\mathbf{F}^\top \mathbf{F} + \tilde{\lambda}\mathbf{I})^{-1}) \\
&\quad + p\tilde{\lambda}^2 \text{Tr}(\boldsymbol{\theta}_*\boldsymbol{\theta}_*^\top (\mathbf{F}^\top \mathbf{F} + \tilde{\lambda}\mathbf{I})^{-2}) \cdot \text{Tr}(\mathbf{F}^\top \mathbf{F}(\mathbf{F}^\top \mathbf{F} + \tilde{\lambda}\mathbf{I})^{-2}) \cdot \frac{1}{n - p} \\
&\sim p\text{Tr}(\boldsymbol{\theta}_*\boldsymbol{\theta}_*^\top \mathbf{F}^\top (\mathbf{F}\mathbf{F}^\top)^{-2}\mathbf{F}) + p\text{Tr}(\boldsymbol{\theta}_*\boldsymbol{\theta}_*^\top (\mathbf{I} - \mathbf{F}^\top (\mathbf{F}\mathbf{F}^\top)^{-1}\mathbf{F})) \cdot \text{Tr}((\mathbf{F}\mathbf{F}^\top)^{-1}) \cdot \frac{1}{n - p}.
\end{aligned}
$$

In the next, we are ready to eliminate the randomness over $\mathbf{F}$.

**Under-parameterized regime: deterministic equivalent over $\mathbf{F}$**  For the variance term, from [1, Sec 3.2] we know that $1/\lambda_p$ is almost surely the limit of $\text{Tr}((\mathbf{F}\mathbf{F}^\top)^{-1})$, thus we have

$$\text{Tr}((\mathbf{F}\mathbf{F}^\top)^{-1}) \sim \frac{1}{\lambda_p},$$

where $\lambda_p$ defined by $\text{df}_1(\lambda_p) = p$, where $\text{df}_1(\lambda_p)$ and $\text{df}_2(\lambda_p)$ are degrees of freedom associated to $\mathbf{\Lambda}$. Hence we can obtain

$$\mathcal{V}_{\mathcal{N},0}^{\text{RFM}} \sim \sigma^2 \cdot \frac{1}{\lambda_p} \cdot \frac{p}{n - p} = \frac{\sigma^2 p}{\lambda_p (n - p)}.$$

For the bias term, denote $D := F\Lambda^{-1/2}$, we first use Eq. (54) with $T = D$, $\Sigma = \Lambda$, $A = \Lambda^{1/2}\boldsymbol{\theta}_*\boldsymbol{\theta}_*^\top\Lambda^{1/2}$ and obtain the deterministic equivalent of the first term in $\mathcal{B}_{\mathcal{N},0}^{\mathrm{RFM}}$

$$\mathrm{Tr}(\boldsymbol{\theta}_*\boldsymbol{\theta}_*^\top F^\top (FF^\top)^{-2}F) = \mathrm{Tr}(\Lambda^{1/2}\boldsymbol{\theta}_*\boldsymbol{\theta}_*^\top\Lambda^{1/2}D^\top (D\Lambda D^\top)^{-2}D)$$
$$\sim \mathrm{Tr}(\boldsymbol{\theta}_*\boldsymbol{\theta}_*^\top\Lambda(\Lambda + \lambda_p)^{-2}) \cdot \frac{1}{n - \mathrm{df}_2(\lambda_p)}.$$

Then we use Eq. (13) with $T = D$, $\Sigma = \Lambda$, $A = \Lambda^{1/2}\boldsymbol{\theta}_*\boldsymbol{\theta}_*^\top\Lambda^{1/2}$ and obtain

$$\mathrm{Tr}(\boldsymbol{\theta}_*\boldsymbol{\theta}_*^\top F^\top (FF^\top)^{-1}F) = \mathrm{Tr}(\Lambda^{1/2}\boldsymbol{\theta}_*\boldsymbol{\theta}_*^\top\Lambda^{1/2}D^\top (D\Lambda D^\top)^{-1}D) \sim \mathrm{Tr}(\boldsymbol{\theta}_*\boldsymbol{\theta}_*^\top\Lambda(\Lambda + \lambda_p)^{-1}),$$

Then the deterministic equivalent of the second term in $\mathcal{B}_{\mathcal{N},0}^{\mathrm{RFM}}$ is given by

$$\mathrm{Tr}(\boldsymbol{\theta}_*\boldsymbol{\theta}_*^\top (I - F^\top (FF^\top)^{-1}F)) \sim \lambda_p\boldsymbol{\theta}_*^\top (\Lambda + \lambda_p)^{-1}\boldsymbol{\theta}_*.$$

Finally, combine the above equivalents and we have

$$\mathcal{B}_{\mathcal{N},0}^{\mathrm{RFM}} \sim \boldsymbol{\theta}_*^\top\Lambda(\Lambda + \lambda_p)^{-2}\boldsymbol{\theta}_* \cdot \frac{p}{n - \mathrm{df}_2(\lambda_p)} + \boldsymbol{\theta}_*^\top (\Lambda + \lambda_p)^{-1}\boldsymbol{\theta}_* \cdot \frac{p}{n - p}$$
$$= \frac{p\langle\boldsymbol{\theta}_*, \Lambda(\Lambda + \lambda_p)^{-2}\boldsymbol{\theta}_*\rangle}{n - \mathrm{Tr}(\Lambda^2(\Lambda + \lambda_n I)^{-2})} + \frac{p\langle\boldsymbol{\theta}_*, (\Lambda + \lambda_p)^{-1}\boldsymbol{\theta}_*\rangle}{n - p}.$$

**Over-parameterized regime: deterministic equivalent over $F$**   Denote $K := \Lambda^{1/2}G^\top G\Lambda^{1/2}$, for the variance term, we use Eq. (54) with $T = D$, $\Sigma = A = K$ and obtain

$$\mathcal{V}_{\mathcal{N},0}^{\mathrm{RFM}} = \sigma^2 \cdot p\mathrm{Tr}(FG^\top GF^\top (FG^\top GF^\top + p\lambda I)^{-2})$$
$$= \sigma^2 \cdot p\mathrm{Tr}(KD^\top (DKD^\top + p\lambda I)^{-2}D)$$
$$\sim \sigma^2 \cdot p\mathrm{Tr}(K(K + \hat{\lambda}I)^{-2}) \cdot \frac{1}{p - n}$$
$$\sim \sigma^2 \cdot \mathrm{Tr}((G\Lambda G^\top)^{-1}) \cdot \frac{p}{p - n},$$

where $\hat{\lambda}$ is defined by

$$\hat{\lambda}(1 - \frac{1}{n}\widehat{\mathrm{df}}_1(\hat{\lambda})) \sim \frac{p\lambda}{n}, \tag{67}$$

where $\widehat{\mathrm{df}}_1(\hat{\lambda})$ and $\widehat{\mathrm{df}}_2(\hat{\lambda})$ are degrees of freedom associated to $K$. In the over-parameterized regime $(p > n)$, when $\lambda$ goes to zero, we have $\hat{\lambda} \to 0$ and $\widehat{\mathrm{df}}_2(\hat{\lambda}) \to n$ [1].

For the bias term, we use Eq. (54) with $T = D$, $\Sigma = K$, $A = \Lambda^{1/2}G^\top G\boldsymbol{\theta}_*\boldsymbol{\theta}_*^\top G^\top G\Lambda^{1/2}$ and obtain

$$\mathcal{B}_{\mathcal{N},0}^{\mathrm{RFM}} = p\mathrm{Tr}(\boldsymbol{\theta}_*^\top G^\top GF^\top (FG^\top GF^\top + p\lambda I)^{-2}FG^\top G\boldsymbol{\theta}_*)$$
$$= p\mathrm{Tr}(\Lambda^{1/2}G^\top G\boldsymbol{\theta}_*\boldsymbol{\theta}_*^\top G^\top G\Lambda^{1/2}D(DKD^\top + p\lambda I)^{-2}D)$$
$$\sim p\mathrm{Tr}(\Lambda^{1/2}G^\top G\boldsymbol{\theta}_*\boldsymbol{\theta}_*^\top G^\top G\Lambda^{1/2}(K + \hat{\lambda}I)^{-2}) \cdot \frac{1}{p - n}$$
$$\sim \mathrm{Tr}(\boldsymbol{\theta}_*\boldsymbol{\theta}_*^\top G^\top (G\Lambda G^\top)^{-1}G) \cdot \frac{p}{p - n}.$$

**Over-parameterized regime: deterministic equivalent over $G$**   For the variance term, we have

$$\mathcal{V}_{\mathcal{N},0}^{\mathrm{RFM}} \sim \sigma^2 \cdot \frac{1}{\lambda_n} \cdot \frac{p}{p - n} = \frac{\sigma^2 p}{\lambda_n(p - n)}.$$

For the bias term, we have

$$\mathcal{B}_{\mathcal{N},0}^{\mathrm{RFM}} \sim \mathrm{Tr}(\boldsymbol{\theta}_*\boldsymbol{\theta}_*^\top (\Lambda + \lambda_n)^{-1}) \cdot \frac{p}{p - n}$$
$$= \boldsymbol{\theta}_*^\top (\Lambda + \lambda_n)^{-1}\boldsymbol{\theta}_* \cdot \frac{p}{p - n}$$
$$= \frac{p\langle\boldsymbol{\theta}_*, (\Lambda + \lambda_n)^{-1}\boldsymbol{\theta}_*\rangle}{p - n}.$$

Finally, we conclude the proof. □

To build the connection between the test risk and norm for the min-$\ell_2$-norm estimator for random features regression, we also need the deterministic equivalent of the test risk as below.

**Proposition E.1** (Asymptotic deterministic equivalence of the test risk of the min-$\ell_2$-norm interpolator). *Under Assumption 1, for the minimum $\ell_2$-norm estimator $\hat{a}_{\min}$, we have the following deterministic equivalence: for the under-parameterized regime ($p < n$), we have*

$$\mathcal{B}^{\text{RFM}}_{\mathcal{R},0} \sim \frac{n\lambda_p \langle \boldsymbol{\theta}_*, (\boldsymbol{\Lambda} + \lambda_p \boldsymbol{I})^{-1} \boldsymbol{\theta}_* \rangle}{n - p}, \quad \mathcal{V}^{\text{RFM}}_{\mathcal{R},0} \sim \frac{\sigma^2 p}{n - p},$$

*where $\lambda_p$ is defined by $\text{Tr}(\boldsymbol{\Lambda}(\boldsymbol{\Lambda} + \lambda_p \boldsymbol{I})^{-1}) \sim p$. In the over-parameterized regime ($p > n$), we have*

$$\mathcal{B}^{\text{RFM}}_{\mathcal{R},0} \sim \frac{n\lambda_n^2 \langle \boldsymbol{\theta}_*, (\boldsymbol{\Lambda} + \lambda_n \boldsymbol{I})^{-2} \boldsymbol{\theta}_* \rangle}{n - \text{Tr}(\boldsymbol{\Lambda}^2(\boldsymbol{\Lambda} + \lambda_n \boldsymbol{I})^{-2})} + \frac{n\lambda_n \langle \boldsymbol{\theta}_*, (\boldsymbol{\Lambda} + \lambda_n \boldsymbol{I})^{-1} \boldsymbol{\theta}_* \rangle}{p - n},$$

$$\mathcal{V}^{\text{RFM}}_{\mathcal{R},0} \sim \frac{\sigma^2 \text{Tr}(\boldsymbol{\Lambda}^2(\boldsymbol{\Lambda} + \lambda_n \boldsymbol{I})^{-2})}{n - \text{Tr}(\boldsymbol{\Lambda}^2(\boldsymbol{\Lambda} + \lambda_n \boldsymbol{I})^{-2})} + \frac{\sigma^2 n}{p - n},$$

*where $\lambda_n$ is defined by $\text{Tr}(\boldsymbol{\Lambda}(\boldsymbol{\Lambda} + \lambda_n \boldsymbol{I})^{-1}) \sim n$.*

*Proof of Proposition E.1.* For the proof, we separate the two regimes $p < n$ and $p > n$. For both of them, we provide asymptotic expansions in two steps, first with respect to $\boldsymbol{G}$ and then $\boldsymbol{F}$ in the under-parameterized regime and vice-versa for the over-parameterized regime.

**Under-parameterized regime: deterministic equivalent over $\boldsymbol{G}$** For the variance term, in the under-parameterized regime, when $\lambda \to 0$, the variance term will become $\mathcal{V}^{\text{RFM}}_{\mathcal{R},0} = \sigma^2 \cdot \text{Tr}(\hat{\boldsymbol{\Lambda}}_F (\boldsymbol{Z}^\top \boldsymbol{Z})^{-1})$. Accordingly, using [1, Eq. (12)], we have

$$\mathcal{V}^{\text{RFM}}_{\mathcal{R},0} = \sigma^2 \cdot \text{Tr}(\hat{\boldsymbol{\Lambda}}_F (\boldsymbol{Z}^\top \boldsymbol{Z})^{-1})$$
$$= \sigma^2 \cdot \text{Tr}(\boldsymbol{F}\boldsymbol{F}^\top (\boldsymbol{F}\boldsymbol{G}^\top \boldsymbol{G}\boldsymbol{F}^\top)^{-1})$$
$$\sim \frac{\sigma^2}{n - p} \cdot \text{Tr}(\boldsymbol{F}\boldsymbol{F}^\top (\boldsymbol{F}\boldsymbol{F}^\top)^{-1})$$
$$= \frac{\sigma^2 p}{n - p}.$$

For the bias term, it can be decomposed into

$$\mathcal{B}^{\text{RFM}}_{\mathcal{R},0} = \|\boldsymbol{\theta}_* - p^{-1/2}\boldsymbol{F}^\top (\boldsymbol{Z}^\top \boldsymbol{Z} + \lambda \boldsymbol{I})^{-1}\boldsymbol{Z}^\top \boldsymbol{G}\boldsymbol{\theta}_*\|_2^2$$
$$= \boldsymbol{\theta}_*^\top \boldsymbol{\theta}_* - 2p^{-1/2}\boldsymbol{\theta}_*^\top \boldsymbol{F}^\top (\boldsymbol{Z}^\top \boldsymbol{Z} + \lambda \boldsymbol{I})^{-1}\boldsymbol{Z}^\top \boldsymbol{G}\boldsymbol{\theta}_*$$
$$+ \boldsymbol{\theta}_*^\top \boldsymbol{G}^\top \boldsymbol{Z}(\boldsymbol{Z}^\top \boldsymbol{Z} + \lambda \boldsymbol{I})^{-1}\hat{\boldsymbol{\Lambda}}_F (\boldsymbol{Z}^\top \boldsymbol{Z} + \lambda \boldsymbol{I})^{-1}\boldsymbol{Z}^\top \boldsymbol{G}\boldsymbol{\theta}_*.$$

For the second term: $p^{-1/2}\boldsymbol{\theta}_*^\top \boldsymbol{F}^\top (\boldsymbol{Z}^\top \boldsymbol{Z} + \lambda \boldsymbol{I})^{-1}\boldsymbol{Z}^\top \boldsymbol{G}\boldsymbol{\theta}_*$, we can use Eq. (12) with $\boldsymbol{T} = \boldsymbol{G}$, $\boldsymbol{\Sigma} = \boldsymbol{F}^\top \boldsymbol{F}$, $\boldsymbol{A} = \boldsymbol{\theta}_*\boldsymbol{\theta}_*^\top \boldsymbol{F}^\top \boldsymbol{F}$ and obtain

$$p^{-1/2}\boldsymbol{\theta}_*^\top \boldsymbol{F}^\top (\boldsymbol{Z}^\top \boldsymbol{Z} + \lambda \boldsymbol{I})^{-1}\boldsymbol{Z}^\top \boldsymbol{G}\boldsymbol{\theta}_* = \text{Tr}(\boldsymbol{\theta}_*\boldsymbol{\theta}_*^\top \boldsymbol{F}^\top \boldsymbol{F}\boldsymbol{G}^\top (\boldsymbol{G}\boldsymbol{F}^\top \boldsymbol{F}\boldsymbol{G}^\top + p\lambda \boldsymbol{I})^{-1}\boldsymbol{G})$$
$$\sim \text{Tr}(\boldsymbol{\theta}_*\boldsymbol{\theta}_*^\top \boldsymbol{F}^\top \boldsymbol{F}(\boldsymbol{F}^\top \boldsymbol{F} + \tilde{\lambda}\boldsymbol{I})^{-1})$$
$$\sim \text{Tr}(\boldsymbol{\theta}_*\boldsymbol{\theta}_*^\top \boldsymbol{F}^\top (\boldsymbol{F}\boldsymbol{F}^\top)^{-1}\boldsymbol{F}),$$

where the implicit regularization parameter $\tilde{\lambda}$ is defined by Eq. (66).

For the third term: $\boldsymbol{\theta}_*^\top \boldsymbol{G}^\top \boldsymbol{Z}(\boldsymbol{Z}^\top \boldsymbol{Z} + \lambda \boldsymbol{I})^{-1}\hat{\boldsymbol{\Lambda}}_F (\boldsymbol{Z}^\top \boldsymbol{Z} + \lambda \boldsymbol{I})^{-1}\boldsymbol{Z}^\top \boldsymbol{G}\boldsymbol{\theta}_*$, we can use Eq. (13) with $\boldsymbol{T} = \boldsymbol{G}, \boldsymbol{\Sigma} = \boldsymbol{F}^\top \boldsymbol{F}, \boldsymbol{A} = \boldsymbol{\theta}_*\boldsymbol{\theta}_*^\top, \boldsymbol{B} = \boldsymbol{F}^\top \boldsymbol{F}\boldsymbol{F}^\top \boldsymbol{F}$ and obtain

$$\boldsymbol{\theta}_*^\top \boldsymbol{G}^\top \boldsymbol{Z}(\boldsymbol{Z}^\top \boldsymbol{Z} + \lambda \boldsymbol{I})^{-1}\hat{\boldsymbol{\Lambda}}_F (\boldsymbol{Z}^\top \boldsymbol{Z} + \lambda \boldsymbol{I})^{-1}\boldsymbol{Z}^\top \boldsymbol{G}\boldsymbol{\theta}_*$$
$$= \text{Tr}(\boldsymbol{\theta}_*\boldsymbol{\theta}_*^\top \boldsymbol{G}^\top \boldsymbol{G}\boldsymbol{F}^\top (\boldsymbol{F}\boldsymbol{G}^\top \boldsymbol{G}\boldsymbol{F}^\top + p\lambda \boldsymbol{I})^{-1}\boldsymbol{F}\boldsymbol{F}^\top (\boldsymbol{F}\boldsymbol{G}^\top \boldsymbol{G}\boldsymbol{F}^\top + p\lambda \boldsymbol{I})^{-1}\boldsymbol{F}\boldsymbol{G}^\top \boldsymbol{G})$$
$$= \text{Tr}(\boldsymbol{\theta}_*\boldsymbol{\theta}_*^\top \boldsymbol{G}^\top (\boldsymbol{G}\boldsymbol{F}^\top \boldsymbol{F}\boldsymbol{G}^\top + p\lambda \boldsymbol{I})^{-1}\boldsymbol{G}\boldsymbol{F}^\top \boldsymbol{F}\boldsymbol{F}^\top \boldsymbol{F}\boldsymbol{G}^\top (\boldsymbol{G}\boldsymbol{F}^\top \boldsymbol{F}\boldsymbol{G}^\top + p\lambda \boldsymbol{I})^{-1}\boldsymbol{G})$$
$$\sim \text{Tr}(\boldsymbol{\theta}_*\boldsymbol{\theta}_*^\top (\boldsymbol{F}^\top \boldsymbol{F} + \tilde{\lambda}\boldsymbol{I})^{-1}\boldsymbol{F}^\top \boldsymbol{F}\boldsymbol{F}^\top \boldsymbol{F}(\boldsymbol{F}^\top \boldsymbol{F} + \tilde{\lambda}\boldsymbol{I})^{-1})$$
$$+ \tilde{\lambda}^2 \text{Tr}(\boldsymbol{\theta}_*\boldsymbol{\theta}_*^\top (\boldsymbol{F}^\top \boldsymbol{F} + \tilde{\lambda}\boldsymbol{I})^{-2}) \cdot \text{Tr}(\boldsymbol{F}^\top \boldsymbol{F}\boldsymbol{F}^\top \boldsymbol{F}(\boldsymbol{F}^\top \boldsymbol{F} + \tilde{\lambda}\boldsymbol{I})^{-2}) \cdot \frac{1}{n - p}$$
$$\sim \text{Tr}(\boldsymbol{\theta}_*\boldsymbol{\theta}_*^\top \boldsymbol{F}^\top (\boldsymbol{F}\boldsymbol{F}^\top)^{-1}\boldsymbol{F}) + \text{Tr}(\boldsymbol{\theta}_*\boldsymbol{\theta}_*^\top (\boldsymbol{I} - \boldsymbol{F}^\top (\boldsymbol{F}\boldsymbol{F}^\top)^{-1}\boldsymbol{F})) \cdot \frac{p}{n - p}.$$

Combining the above equivalents, we have

$$\mathcal{B}^{\text{RFM}}_{\mathcal{R},0} = \boldsymbol{\theta}^\top_* \boldsymbol{\theta}_* - \text{Tr}(\boldsymbol{\theta}_* \boldsymbol{\theta}^\top_* \boldsymbol{F}^\top (\boldsymbol{F}\boldsymbol{F}^\top)^{-1}\boldsymbol{F}) + \text{Tr}(\boldsymbol{\theta}_*\boldsymbol{\theta}^\top_*(\boldsymbol{I} - \boldsymbol{F}^\top(\boldsymbol{F}\boldsymbol{F}^\top)^{-1}\boldsymbol{F})) \cdot \frac{p}{n-p}$$

$$= \boldsymbol{\theta}^\top_*\boldsymbol{\theta}_* \cdot \frac{n}{n-p} - \text{Tr}(\boldsymbol{\theta}_*\boldsymbol{\theta}^\top_*\boldsymbol{F}^\top(\boldsymbol{F}\boldsymbol{F}^\top)^{-1}\boldsymbol{F}) \cdot \frac{n}{n-p}\,.$$

**Under-parameterized regime: deterministic equivalent over $\boldsymbol{F}$**    For the bias term, we can use Eq. (12) with $\boldsymbol{T} = \boldsymbol{D} := \boldsymbol{F}\boldsymbol{\Lambda}^{-1/2}$, $\boldsymbol{A} = \boldsymbol{\Lambda}^{1/2}\boldsymbol{\theta}_*\boldsymbol{\theta}^\top_*\boldsymbol{\Lambda}^{1/2}$ and obtain

$$\text{Tr}(\boldsymbol{\theta}_*\boldsymbol{\theta}^\top_*\boldsymbol{F}^\top(\boldsymbol{F}\boldsymbol{F}^\top)^{-1}\boldsymbol{F}) = \text{Tr}(\boldsymbol{\Lambda}^{1/2}\boldsymbol{\theta}_*\boldsymbol{\theta}^\top_*\boldsymbol{\Lambda}^{1/2}\boldsymbol{D}^\top(\boldsymbol{D}\boldsymbol{\Lambda}\boldsymbol{D}^\top)^{-1}\boldsymbol{D})$$

$$\sim \text{Tr}(\boldsymbol{\Lambda}^{1/2}\boldsymbol{\theta}_*\boldsymbol{\theta}^\top_*\boldsymbol{\Lambda}^{1/2}(\boldsymbol{\Lambda} + \lambda_p)^{-1})$$

$$= \boldsymbol{\theta}^\top_*\boldsymbol{\Lambda}(\boldsymbol{\Lambda} + \lambda_p)^{-1}\boldsymbol{\theta}_*\,.$$

Thus, we finally obtain

$$\mathcal{B}^{\text{RFM}}_{\mathcal{R},0} \sim \boldsymbol{\theta}^\top_*\boldsymbol{\theta}_* \cdot \frac{n}{n-p} - \boldsymbol{\theta}^\top_*\boldsymbol{\Lambda}(\boldsymbol{\Lambda} + \lambda_p)^{-1}\boldsymbol{\theta}_* \cdot \frac{n}{n-p}$$

$$= \lambda_p\boldsymbol{\theta}^\top_*(\boldsymbol{\Lambda} + \lambda_p)^{-1}\boldsymbol{\theta}_* \cdot \frac{n}{n-p}$$

$$= \frac{n\lambda_p\langle\boldsymbol{\theta}_*, (\boldsymbol{\Lambda} + \lambda_p\boldsymbol{I})^{-1}\boldsymbol{\theta}_*\rangle}{n-p}\,.$$

**Over-parameterized regime: deterministic equivalent over $\boldsymbol{F}$**    For the variance term, with $\boldsymbol{D} := \boldsymbol{F}\boldsymbol{\Lambda}^{-1/2}$ and $\boldsymbol{K} := \boldsymbol{\Lambda}^{1/2}\boldsymbol{G}^\top\boldsymbol{G}\boldsymbol{\Lambda}^{1/2}$ we can obtain

$$\mathcal{V}^{\text{RFM}}_{\mathcal{R},0} = \sigma^2 \cdot \text{Tr}(\widehat{\boldsymbol{\Lambda}}_{\boldsymbol{F}}\boldsymbol{Z}^\top\boldsymbol{Z}(\boldsymbol{Z}^\top\boldsymbol{Z} + \lambda\boldsymbol{I})^{-2})$$

$$= \sigma^2 \cdot \text{Tr}(\boldsymbol{F}\boldsymbol{F}^\top\boldsymbol{F}\boldsymbol{G}^\top\boldsymbol{G}\boldsymbol{F}^\top(\boldsymbol{F}\boldsymbol{G}^\top\boldsymbol{G}\boldsymbol{F}^\top + p\lambda\boldsymbol{I})^{-2})$$

$$= \sigma^2 \cdot \text{Tr}(\boldsymbol{D}\boldsymbol{\Lambda}\boldsymbol{D}^\top\boldsymbol{D}\boldsymbol{\Lambda}^{1/2}\boldsymbol{G}^\top\boldsymbol{G}\boldsymbol{\Lambda}^{1/2}\boldsymbol{D}^\top(\boldsymbol{D}\boldsymbol{\Lambda}^{1/2}\boldsymbol{G}^\top\boldsymbol{G}\boldsymbol{\Lambda}^{1/2}\boldsymbol{D}^\top + p\lambda\boldsymbol{I})^{-2})$$

$$= \sigma^2 \cdot \text{Tr}(\boldsymbol{\Lambda}\boldsymbol{D}^\top(\boldsymbol{D}\boldsymbol{K}\boldsymbol{D}^\top + p\lambda\boldsymbol{I})^{-1}\boldsymbol{D}\boldsymbol{K}\boldsymbol{D}^\top(\boldsymbol{D}\boldsymbol{K}\boldsymbol{D}^\top + p\lambda\boldsymbol{I})^{-1}\boldsymbol{D})\,,$$

then we directly use Eq. (13) with $\boldsymbol{T} = \boldsymbol{D}$, $\boldsymbol{\Sigma} = \boldsymbol{K}$, $\boldsymbol{A} = \boldsymbol{\Lambda}$, $\boldsymbol{B} = \boldsymbol{K}$ and obtain

$$\text{Tr}(\boldsymbol{\Lambda}\boldsymbol{D}^\top(\boldsymbol{D}\boldsymbol{K}\boldsymbol{D}^\top + p\lambda\boldsymbol{I})^{-1}\boldsymbol{D}\boldsymbol{K}\boldsymbol{D}^\top(\boldsymbol{D}\boldsymbol{K}\boldsymbol{D}^\top + p\lambda\boldsymbol{I})^{-1}\boldsymbol{D})$$

$$\sim \text{Tr}(\boldsymbol{\Lambda}(\boldsymbol{K} + \hat{\lambda}\boldsymbol{I})^{-1}\boldsymbol{K}(\boldsymbol{K} + \hat{\lambda}\boldsymbol{I})^{-1}) + \hat{\lambda}^2\text{Tr}(\boldsymbol{\Lambda}(\boldsymbol{K} + \hat{\lambda}\boldsymbol{I})^{-2}) \cdot \text{Tr}(\boldsymbol{K}(\boldsymbol{K} + \hat{\lambda}\boldsymbol{I})^{-2}) \cdot \frac{1}{p-n}$$

$$\sim \text{Tr}(\boldsymbol{\Lambda}^2\boldsymbol{G}^\top(\boldsymbol{G}\boldsymbol{\Lambda}\boldsymbol{G}^\top)^{-2}\boldsymbol{G}) + \text{Tr}(\boldsymbol{\Lambda}(\boldsymbol{I} - \boldsymbol{\Lambda}^{1/2}\boldsymbol{G}^\top(\boldsymbol{G}\boldsymbol{\Lambda}\boldsymbol{G}^\top)^{-1}\boldsymbol{G}\boldsymbol{\Lambda}^{1/2})) \cdot \text{Tr}((\boldsymbol{G}\boldsymbol{\Lambda}\boldsymbol{G}^\top)^{-1}) \cdot \frac{1}{p-n}\,,$$

where the implicit regularization parameter $\hat{\lambda}$ is defined by Eq. (67).

For the bias term, first we have

$$p^{-1/2}\boldsymbol{\theta}^\top_*\boldsymbol{F}^\top(\boldsymbol{Z}^\top\boldsymbol{Z} + \lambda\boldsymbol{I})^{-1}\boldsymbol{Z}^\top\boldsymbol{G}\boldsymbol{\theta}_* = \text{Tr}(\boldsymbol{\theta}_*\boldsymbol{\theta}^\top_*\boldsymbol{F}^\top(\boldsymbol{F}\boldsymbol{G}^\top\boldsymbol{G}\boldsymbol{F}^\top + p\lambda\boldsymbol{I})^{-1}\boldsymbol{F}\boldsymbol{G}^\top\boldsymbol{G})$$

$$= \text{Tr}(\boldsymbol{\Lambda}^{1/2}\boldsymbol{G}^\top\boldsymbol{G}\boldsymbol{\theta}_*\boldsymbol{\theta}^\top_*\boldsymbol{\Lambda}^{1/2}\boldsymbol{D}^\top(\boldsymbol{D}\boldsymbol{K}\boldsymbol{D}^\top + p\lambda\boldsymbol{I})^{-1}\boldsymbol{D})\,,$$

then we use Eq. (12) with $\boldsymbol{T} = \boldsymbol{D}$, $\boldsymbol{\Sigma} = \boldsymbol{K}$, $\boldsymbol{A} = \boldsymbol{\Lambda}^{1/2}\boldsymbol{G}^\top\boldsymbol{G}\boldsymbol{\theta}_*\boldsymbol{\theta}^\top_*\boldsymbol{\Lambda}^{1/2}$ and obtain

$$\text{Tr}(\boldsymbol{\Lambda}^{1/2}\boldsymbol{G}^\top\boldsymbol{G}\boldsymbol{\theta}_*\boldsymbol{\theta}^\top_*\boldsymbol{\Lambda}^{1/2}\boldsymbol{D}^\top(\boldsymbol{D}\boldsymbol{K}\boldsymbol{D}^\top + p\lambda\boldsymbol{I})^{-1}\boldsymbol{D}) \sim \text{Tr}(\boldsymbol{\theta}_*\boldsymbol{\theta}^\top_*\boldsymbol{\Lambda}\boldsymbol{G}^\top(\boldsymbol{G}\boldsymbol{\Lambda}\boldsymbol{G}^\top)^{-1}\boldsymbol{G})\,.$$

Furthermore, we use Eq. (13) with $\boldsymbol{T} = \boldsymbol{D}$, $\boldsymbol{\Sigma} = \boldsymbol{K}$, $\boldsymbol{A} = \boldsymbol{\Lambda}^{1/2}\boldsymbol{G}^\top\boldsymbol{G}\boldsymbol{\theta}_*\boldsymbol{\theta}^\top_*\boldsymbol{G}^\top\boldsymbol{G}\boldsymbol{\Lambda}^{1/2}$, $\boldsymbol{B} = \boldsymbol{\Lambda}$ and obtain

$$\boldsymbol{\theta}^\top_*\boldsymbol{G}^\top\boldsymbol{Z}(\boldsymbol{Z}^\top\boldsymbol{Z} + \lambda\boldsymbol{I})^{-1}\widehat{\boldsymbol{\Lambda}}_{\boldsymbol{F}}(\boldsymbol{Z}^\top\boldsymbol{Z} + \lambda\boldsymbol{I})^{-1}\boldsymbol{Z}^\top\boldsymbol{G}\boldsymbol{\theta}_*$$

$$= \text{Tr}(\boldsymbol{\Lambda}^{1/2}\boldsymbol{G}^\top\boldsymbol{G}\boldsymbol{\theta}_*\boldsymbol{\theta}^\top_*\boldsymbol{G}^\top\boldsymbol{G}\boldsymbol{\Lambda}^{1/2}\boldsymbol{D}^\top(\boldsymbol{D}\boldsymbol{K}\boldsymbol{D}^\top + p\lambda\boldsymbol{I})^{-1}\boldsymbol{D}\boldsymbol{\Lambda}\boldsymbol{D}^\top(\boldsymbol{D}\boldsymbol{K}\boldsymbol{D}^\top + p\lambda\boldsymbol{I})^{-1}\boldsymbol{D})$$

$$\sim \text{Tr}(\boldsymbol{\Lambda}^{1/2}\boldsymbol{G}^\top\boldsymbol{G}\boldsymbol{\theta}_*\boldsymbol{\theta}^\top_*\boldsymbol{G}^\top\boldsymbol{G}\boldsymbol{\Lambda}^{1/2}(\boldsymbol{K} + \hat{\lambda}\boldsymbol{I})^{-1}\boldsymbol{\Lambda}(\boldsymbol{K} + \hat{\lambda}\boldsymbol{I})^{-1})$$

$$\quad + \hat{\lambda}^2\text{Tr}(\boldsymbol{\Lambda}^{1/2}\boldsymbol{G}^\top\boldsymbol{G}\boldsymbol{\theta}_*\boldsymbol{\theta}^\top_*\boldsymbol{G}^\top\boldsymbol{G}\boldsymbol{\Lambda}^{1/2}(\boldsymbol{K} + \hat{\lambda}\boldsymbol{I})^{-2}) \cdot \text{Tr}(\boldsymbol{\Lambda}(\boldsymbol{K} + \hat{\lambda}\boldsymbol{I})^{-2}) \cdot \frac{1}{p-n}$$

$$\sim \text{Tr}(\boldsymbol{\theta}_*\boldsymbol{\theta}^\top_*\boldsymbol{G}^\top(\boldsymbol{G}\boldsymbol{\Lambda}\boldsymbol{G}^\top)^{-1}\boldsymbol{G}\boldsymbol{\Lambda}^2\boldsymbol{G}^\top(\boldsymbol{G}\boldsymbol{\Lambda}\boldsymbol{G}^\top)^{-1}\boldsymbol{G})$$

$$\quad + \text{Tr}(\boldsymbol{\theta}_*\boldsymbol{\theta}^\top_*\boldsymbol{G}^\top(\boldsymbol{G}\boldsymbol{\Lambda}\boldsymbol{G}^\top)^{-1}\boldsymbol{G}) \cdot \text{Tr}(\boldsymbol{\Lambda}(\boldsymbol{I} - \boldsymbol{\Lambda}^{1/2}\boldsymbol{G}^\top(\boldsymbol{G}\boldsymbol{\Lambda}\boldsymbol{G}^\top)^{-1}\boldsymbol{G}\boldsymbol{\Lambda}^{1/2})) \cdot \frac{1}{p-n}\,.$$

In the next, we are ready to eliminate the randomness over $\boldsymbol{G}$.

**Over-parameterized regime: deterministic equivalent over $G$** For the variance term, we use Eq. (54) to obtain

$$\operatorname{Tr}(\mathbf{\Lambda}^2 \mathbf{G}^\top (\mathbf{G}\mathbf{\Lambda}\mathbf{G}^\top)^{-2}\mathbf{G}) \sim \frac{\mathrm{df}_2(\lambda_n)}{n - \mathrm{df}_2(\lambda_n)} \,.$$

Then we use Eq. (12) to obtain

$$\operatorname{Tr}(\mathbf{\Lambda}^2 \mathbf{G}^\top (\mathbf{G}\mathbf{\Lambda}\mathbf{G}^\top)^{-1}\mathbf{G}) \sim \operatorname{Tr}(\mathbf{\Lambda}^2(\mathbf{\Lambda} + \lambda_n)^{-1}),$$

where $\lambda_n$ is defined by $\mathrm{df}_1(\lambda_n) = n$. Hence we have

$$\operatorname{Tr}(\mathbf{\Lambda}(\mathbf{I} - \mathbf{\Lambda}^{1/2}\mathbf{G}^\top (\mathbf{G}\mathbf{\Lambda}\mathbf{G}^\top)^{-1}\mathbf{G}\mathbf{\Lambda}^{1/2})) \sim n\lambda_n.$$

Combine the above equivalents, we have

$$\begin{aligned}
\mathcal{V}_{\mathcal{R},0}^{\mathrm{RFM}} &\sim \sigma^2 \cdot \frac{\mathrm{df}_2(\lambda_n)}{n - \mathrm{df}_2(\lambda_n)} + \sigma^2 \cdot \frac{n}{p - n} \\
&= \frac{\sigma^2 \operatorname{Tr}(\mathbf{\Lambda}^2(\mathbf{\Lambda} + \lambda_n \mathbf{I})^{-2})}{n - \operatorname{Tr}(\mathbf{\Lambda}^2(\mathbf{\Lambda} + \lambda_n \mathbf{I})^{-2})} + \frac{\sigma^2 n}{p - n} \,.
\end{aligned}$$

For the bias term, we first use Eq. (12) to obtain

$$\operatorname{Tr}(\boldsymbol{\theta}_* \boldsymbol{\theta}_*^\top \mathbf{\Lambda}\mathbf{G}^\top (\mathbf{G}\mathbf{\Lambda}\mathbf{G}^\top)^{-1}\mathbf{G}) \sim \operatorname{Tr}(\boldsymbol{\theta}_* \boldsymbol{\theta}_*^\top \mathbf{\Lambda}(\mathbf{\Lambda} + \lambda_n)^{-1}) \,.$$

Moreover, we use Eq. (13) to obtain

$$\begin{aligned}
&\operatorname{Tr}(\boldsymbol{\theta}_* \boldsymbol{\theta}_*^\top \mathbf{G}^\top (\mathbf{G}\mathbf{\Lambda}\mathbf{G}^\top)^{-1}\mathbf{G}\mathbf{\Lambda}^2\mathbf{G}^\top (\mathbf{G}\mathbf{\Lambda}\mathbf{G}^\top)^{-1}\mathbf{G}) \\
&\sim \operatorname{Tr}(\boldsymbol{\theta}_* \boldsymbol{\theta}_*^\top \mathbf{\Lambda}^2(\mathbf{\Lambda} + \lambda_n)^{-2}) + \lambda_n^2 \cdot \operatorname{Tr}(\boldsymbol{\theta}_* \boldsymbol{\theta}_*^\top (\mathbf{\Lambda} + \lambda_n)^{-2}) \cdot \frac{\mathrm{df}_2(\lambda_n)}{n - \mathrm{df}_2(\lambda_n)}.
\end{aligned}$$

Accordingly, we finally conclude that

$$\begin{aligned}
\mathcal{B}_{\mathcal{R},0}^{\mathrm{RFM}} &\sim \lambda_n^2 \boldsymbol{\theta}_*^\top (\mathbf{\Lambda} + \lambda_n \mathbf{I})^{-2}\boldsymbol{\theta}_* \cdot \frac{n}{n - \mathrm{df}_2(\lambda_n)} + \lambda_n \boldsymbol{\theta}_*^\top (\mathbf{\Lambda} + \lambda_n \mathbf{I})^{-1}\boldsymbol{\theta}_* \cdot \frac{n}{p - n} \\
&= \frac{n\lambda_n^2 \langle \boldsymbol{\theta}_*, (\mathbf{\Lambda} + \lambda_n \mathbf{I})^{-2}\boldsymbol{\theta}_* \rangle}{n - \operatorname{Tr}(\mathbf{\Lambda}^2(\mathbf{\Lambda} + \lambda_n \mathbf{I})^{-2})} + \frac{n\lambda_n \langle \boldsymbol{\theta}_*, (\mathbf{\Lambda} + \lambda_n \mathbf{I})^{-1}\boldsymbol{\theta}_* \rangle}{p - n} \,.
\end{aligned}$$

$\square$

## E.2 Non-asymptotic deterministic equivalence for random features ridge regression

Here we present the proof for the non-asymptotic results on the variance and then discuss the related results on bias due to the insufficient deterministic equivalence.

### E.2.1 Proof on the variance term

**Theorem E.2** (Deterministic equivalence of variance part of the $\ell_2$ norm). *Assume the features $\{\boldsymbol{z}_i\}_{i \in [n]}$ and $\{\boldsymbol{f}_j\}_{j \in [p]}$ satisfy Assumption 1 with a constant $C_* > 0$. Then for any $D, K > 0$, there exist constant $\eta_* \in (0, 1/2)$ and $C_{*,D,K} > 0$ ensuring the following property holds. For any $n, p \geq C_{*,D,K}$, $\lambda > 0$, if the following condition is satisfied:*

$$\lambda \geq n^{-K}, \quad \gamma_\lambda \geq p^{-K}, \quad \widetilde{\rho}_\lambda(n,p)^{5/2} \log^{3/2}(n) \leq K\sqrt{n}, \quad \widetilde{\rho}_\lambda(n,p)^2 \cdot \rho_{\gamma_+}(p)^7 \log^4(p) \leq K\sqrt{p},$$

*then with probability at least $1 - n^{-D} - p^{-D}$, we have that*

$$\left| \mathcal{V}_{\mathcal{N},\lambda}^{\mathrm{RFM}} - \mathsf{V}_{\mathsf{N},\lambda}^{\mathrm{RFM}} \right| \leq C_{x,D,K} \cdot \mathcal{E}_V(n,p) \cdot \mathsf{V}_{\mathsf{N},\lambda}^{\mathrm{RFM}} \,.$$

*where the approximation rate is given by*

$$\mathcal{E}_V(n,p) := \frac{\widetilde{\rho}_\lambda(n,p)^6 \log^{5/2}(n)}{\sqrt{n}} + \frac{\widetilde{\rho}_\lambda(n,p)^2 \cdot \rho_{\gamma_+}(p)^7 \log^3(p)}{\sqrt{p}} \,.$$

*Proof of Theorem E.2.* First, note that $\mathcal{V}_{\mathcal{N},\lambda}^{\mathrm{RFM}}$ can be written in terms of the functional $\Phi_4$ defined in Eq. (37):

$$\mathcal{V}_{\mathcal{N},\lambda}^{\mathrm{RFM}} = \sigma^2 \cdot n\Phi_4(\boldsymbol{Z}; \widehat{\boldsymbol{\Lambda}}_{\boldsymbol{F}}^{-1}, \lambda).$$

Recall that $\mathcal{A}_{\mathcal{F}}$ is the event defined in [14, Eq. (79)]. Under the assumptions, we have

$$\mathbb{P}(\mathcal{A}_{\mathcal{F}}) \geq 1 - p^{-D}.$$

Hence, applying Proposition B.10 for $\boldsymbol{F} \in \mathcal{A}_{\mathcal{F}}$ and via union bound, we obtain that with probability at least $1 - p^{-D} - n^{-D}$,

$$\left| n\Phi_4(\boldsymbol{Z}; \widehat{\boldsymbol{\Lambda}}_{\boldsymbol{F}}^{-1}, \lambda) - n\widetilde{\Phi}_5(\boldsymbol{F}; \widehat{\boldsymbol{\Lambda}}_{\boldsymbol{F}}^{-1}, p\nu_1) \right| \leq C_{*,D,K} \cdot \mathcal{E}_1(p,n) \cdot n\widetilde{\Phi}_5(\boldsymbol{F}; \widehat{\boldsymbol{\Lambda}}_{\boldsymbol{F}}^{-1}, p\nu_1), \quad (68)$$

and we recall the expressions

$$n\widetilde{\Phi}_5(\boldsymbol{F}; \widehat{\boldsymbol{\Lambda}}_{\boldsymbol{F}}^{-1}, p\nu_1) = \frac{\widetilde{\Phi}_6(\boldsymbol{F}; \widehat{\boldsymbol{\Lambda}}_{\boldsymbol{F}}^{-1}, p\nu_1)}{n - \widetilde{\Phi}_6(\boldsymbol{F}; \boldsymbol{I}, p\nu_1)}, \qquad \widetilde{\Phi}_6(\boldsymbol{F}; \widehat{\boldsymbol{\Lambda}}_{\boldsymbol{F}}^{-1}, p\nu_1) = p\mathrm{Tr}\big(\boldsymbol{F}\boldsymbol{F}^\top (\boldsymbol{F}\boldsymbol{F}^\top + p\nu_1)^{-2}\big).$$

From [14, Lemma B.11], we have with probability at least $1 - p^{-D}$

$$\left| p\mathrm{Tr}(\boldsymbol{F}\boldsymbol{F}^\top (\boldsymbol{F}\boldsymbol{F}^\top + p\nu_1)^{-2}) - p^2\Psi_3(\nu_2; \boldsymbol{\Lambda}^{-1}) \right| \leq C_{*,D,K} \cdot \rho_{\gamma_+}(p) \cdot \mathcal{E}_3(p) \cdot p^2\Psi_3(\nu_2; \boldsymbol{\Lambda}^{-1}),$$

where the approximation rate $\mathcal{E}_3(p)$ is given by

$$\mathcal{E}_3(p) := \frac{\rho_{\gamma_+}(p)^6 \log^3(p)}{\sqrt{p}}.$$

Furthermore, from the proof of [14, Theorem B.12], we have with probability at least $1 - p^{-D}$,

$$\left| (1 - n^{-1}\widetilde{\Phi}_6(\boldsymbol{F}; \boldsymbol{I}, p\nu_1))^{-1} - (1 - \Upsilon(\nu_1, \nu_2))^{-1} \right| \leq C_{*,D,K} \cdot \widetilde{\rho}_\lambda(n,p)\rho_{\gamma_+}(p)\mathcal{E}_3(p) \cdot (1 - \Upsilon(\nu_1, \nu_2))^{-1}.$$

Combining those two bounds, we obtain

$$\left| \frac{\widetilde{\Phi}_6(\boldsymbol{F}; \widehat{\boldsymbol{\Lambda}}_{\boldsymbol{F}}^{-1}, p\nu_1)}{n - \widetilde{\Phi}_6(\boldsymbol{F}; \boldsymbol{I}, p\nu_1)} - \frac{p^2\Psi_3(\nu_2; \boldsymbol{\Lambda}^{-1})}{n - n\Upsilon(\nu_1, \nu_2)} \right| \leq C_{*,D,K} \cdot \widetilde{\rho}_\lambda(n,p)\rho_{\gamma_+}(p)\mathcal{E}_3(p) \cdot \frac{p^2\Psi_3(\nu_2; \boldsymbol{\Lambda}^{-1})}{n - n\Upsilon(\nu_1, \nu_2)}.$$

Finally, we can combine this bound with Eq. (68) to obtain via union bound that with probability at least $1 - n^{-D} - p^{-D}$,

$$\left| n\Phi_4(\boldsymbol{Z}; \widehat{\boldsymbol{\Lambda}}_{\boldsymbol{F}}^{-1}, \lambda) - \frac{p^2\Psi_3(\nu_2; \boldsymbol{\Lambda}^{-1})}{n - n\Upsilon(\nu_1, \nu_2)} \right| \leq C_{*,D,K} \left\{ \mathcal{E}_1(p,n) + \widetilde{\rho}_\lambda(n,p)\rho_{\gamma_+}(p)\mathcal{E}_3(p) \right\} \frac{p^2\Psi_3(\nu_2; \boldsymbol{\Lambda}^{-1})}{n - n\Upsilon(\nu_1, \nu_2)}.$$

Replacing the rate $\mathcal{E}_j$ by their expressions conclude the proof of this theorem. $\square$

### E.2.2 Discussion on the bias term

We present the deterministic equivalence of the bias term as an informal result, without a Existing deterministic equivalence results appear insufficient to directly establish this desired bias result. While we believe this is doable under additional assumptions, a complete proof is beyond the scope of this paper..

In the proof of the bias term, deterministic equivalences for functionals of the form

$$\mathrm{Tr}\left( \boldsymbol{A}\left( \boldsymbol{X}^\top \boldsymbol{X} \right)^2 \left( \boldsymbol{X}^\top \boldsymbol{X} + \lambda \right)^{-2} \right)$$

are required. However, such equivalences are currently unavailable, necessitating the introduction of technical assumptions to leverage the deterministic equivalences of $\Phi_2(\boldsymbol{X}; \boldsymbol{A}, \lambda)$ and $\Phi_4(\boldsymbol{X}; \boldsymbol{A}, \lambda)$.

Furthermore, the proof of the bias term in [14] suggests that deriving deterministic equivalences for the bias of the $\ell_2$ norm, analogous to [14, Proposition B.7], is also required but remains unresolved.

Addressing these gaps in deterministic equivalence is an important direction for future work, particularly to establish rigorous proofs for the currently missing results.

### E.3 Proofs on relationship in RFMs

To derive the relationship between test risk and norm for the random feature model, we first examine the linear relationship in the over-parameterized regime. Next, we analyze the case where $\boldsymbol{\Lambda} = \boldsymbol{I}_m$ with $n < m < \infty$ (finite rank), followed by the relationship under the power-law assumption.

#### E.3.1 Proof for min-norm interpolator in the over-parameterized regime

According to the formulation in Corollary 3.2 and Proposition E.1, we have for the under-parameterized regime ($p < n$), we have

$$\mathcal{B}_{\mathcal{N},0}^{\mathrm{RFM}} \sim \mathsf{B}_{\mathrm{N},0}^{\mathrm{RFM}} = \frac{p\langle \boldsymbol{\theta}_*, \boldsymbol{\Lambda}(\boldsymbol{\Lambda} + \lambda_p \boldsymbol{I})^{-2}\boldsymbol{\theta}_*\rangle}{n - \mathrm{Tr}(\boldsymbol{\Lambda}^2(\boldsymbol{\Lambda} + \lambda_p \boldsymbol{I})^{-2})} + \frac{p\langle \boldsymbol{\theta}_*, (\boldsymbol{\Lambda} + \lambda_p \boldsymbol{I})^{-1}\boldsymbol{\theta}_*\rangle}{n - p},$$

$$\mathcal{V}_{\mathcal{N},0}^{\mathrm{RFM}} \sim \mathsf{V}_{\mathrm{N},0}^{\mathrm{RFM}} = \frac{\sigma^2 p}{\lambda_p(n - p)},$$

$$\mathcal{B}_{\mathcal{R},0}^{\mathrm{RFM}} \sim \mathsf{B}_{\mathrm{R},0}^{\mathrm{RFM}} = \frac{n\lambda_p\langle \boldsymbol{\theta}_*, (\boldsymbol{\Lambda} + \lambda_p \boldsymbol{I})^{-1}\boldsymbol{\theta}_*\rangle}{n - p}, \quad \mathcal{V}_{\mathcal{R},0}^{\mathrm{RFM}} \sim \mathsf{V}_{\mathrm{R},0}^{\mathrm{RFM}} = \frac{\sigma^2 p}{n - p}.$$

In the over-parameterized regime ($p > n$), we have

$$\mathcal{B}_{\mathcal{N},0}^{\mathrm{RFM}} \sim \mathsf{B}_{\mathrm{N},0}^{\mathrm{RFM}} = \frac{p\langle \boldsymbol{\theta}_*, (\boldsymbol{\Lambda} + \lambda_n \boldsymbol{I})^{-1}\boldsymbol{\theta}_*\rangle}{p - n}, \quad \mathcal{V}_{\mathcal{N},0}^{\mathrm{RFM}} \sim \mathsf{V}_{\mathrm{N},0}^{\mathrm{RFM}} = \frac{\sigma^2 p}{\lambda_n(p - n)},$$

$$\mathcal{B}_{\mathcal{R},0}^{\mathrm{RFM}} \sim \mathsf{B}_{\mathrm{R},0}^{\mathrm{RFM}} = \frac{n\lambda_n^2\langle \boldsymbol{\theta}_*, (\boldsymbol{\Lambda} + \lambda_n \boldsymbol{I})^{-2}\boldsymbol{\theta}_*\rangle}{n - \mathrm{Tr}(\boldsymbol{\Lambda}^2(\boldsymbol{\Lambda} + \lambda_n \boldsymbol{I})^{-2})} + \frac{n\lambda_n\langle \boldsymbol{\theta}_*, (\boldsymbol{\Lambda} + \lambda_n \boldsymbol{I})^{-1}\boldsymbol{\theta}_*\rangle}{p - n},$$

$$\mathcal{V}_{\mathcal{R},0}^{\mathrm{RFM}} \sim \mathsf{V}_{\mathrm{R},0}^{\mathrm{RFM}} = \frac{\sigma^2 \mathrm{Tr}(\boldsymbol{\Lambda}^2(\boldsymbol{\Lambda} + \lambda_n \boldsymbol{I})^{-2})}{n - \mathrm{Tr}(\boldsymbol{\Lambda}^2(\boldsymbol{\Lambda} + \lambda_n \boldsymbol{I})^{-2})} + \frac{\sigma^2 n}{p - n}.$$

With these formulations we can introduce the relationship between test risk and norm in the over-parameterized regime as follows.

*Proof of Proposition 4.1.* In the over-parameterized regime ($p > n$), we have

$$\mathsf{N}_0^{\mathrm{RFM}} = \mathsf{B}_{\mathrm{N},0}^{\mathrm{RFM}} + \mathsf{V}_{\mathrm{N},0}^{\mathrm{RFM}} = \frac{p\langle \boldsymbol{\theta}_*, (\boldsymbol{\Lambda} + \lambda_n \boldsymbol{I})^{-1}\boldsymbol{\theta}_*\rangle}{p - n} + \frac{\sigma^2 p}{\lambda_n(p - n)} = \left[\langle \boldsymbol{\theta}_*, (\boldsymbol{\Lambda} + \lambda_n \boldsymbol{I})^{-1}\boldsymbol{\theta}_*\rangle + \frac{\sigma^2}{\lambda_n}\right]\frac{p}{p - n}.$$

$$\mathsf{R}_0^{\mathrm{RFM}} = \frac{n\lambda_n^2\langle \boldsymbol{\theta}_*, (\boldsymbol{\Lambda} + \lambda_n \boldsymbol{I})^{-2}\boldsymbol{\theta}_*\rangle}{n - \mathrm{Tr}(\boldsymbol{\Lambda}^2(\boldsymbol{\Lambda} + \lambda_n \boldsymbol{I})^{-2})} + \frac{n\lambda_n\langle \boldsymbol{\theta}_*, (\boldsymbol{\Lambda} + \lambda_n \boldsymbol{I})^{-1}\boldsymbol{\theta}_*\rangle}{p - n} + \frac{\sigma^2 \mathrm{Tr}(\boldsymbol{\Lambda}^2(\boldsymbol{\Lambda} + \lambda_n \boldsymbol{I})^{-2})}{n - \mathrm{Tr}(\boldsymbol{\Lambda}^2(\boldsymbol{\Lambda} + \lambda_n \boldsymbol{I})^{-2})} + \frac{\sigma^2 n}{p - n}$$

$$= \frac{n\lambda_n^2\langle \boldsymbol{\theta}_*, (\boldsymbol{\Lambda} + \lambda_n \boldsymbol{I})^{-2}\boldsymbol{\theta}_*\rangle + \sigma^2 \mathrm{Tr}(\boldsymbol{\Lambda}^2(\boldsymbol{\Lambda} + \lambda_n \boldsymbol{I})^{-2})}{n - \mathrm{Tr}(\boldsymbol{\Lambda}^2(\boldsymbol{\Lambda} + \lambda_n \boldsymbol{I})^{-2})} + \left[n\lambda_n\langle \boldsymbol{\theta}_*, (\boldsymbol{\Lambda} + \lambda_n \boldsymbol{I})^{-1}\boldsymbol{\theta}_*\rangle + \sigma^2 n\right]\frac{1}{p - n}.$$

Then we eliminate $p$ and obtain that the deterministic equivalents of the estimator's test risk and norm, $\mathsf{R}_0^{\mathrm{RFM}}$ and $\mathsf{N}_0^{\mathrm{RFM}}$, in over-parameterized regimes ($p > n$) admit

$$\mathsf{R}_0^{\mathrm{RFM}} = \lambda_n \mathsf{N}_0^{\mathrm{RFM}} - \left[\lambda_n\langle \boldsymbol{\theta}_*, (\boldsymbol{\Lambda} + \lambda_n \boldsymbol{I})^{-1}\boldsymbol{\theta}_*\rangle + \sigma^2\right] + \frac{n\lambda_n^2\langle \boldsymbol{\theta}_*, (\boldsymbol{\Lambda} + \lambda_n \boldsymbol{I})^{-2}\boldsymbol{\theta}_*\rangle + \sigma^2 \mathrm{Tr}(\boldsymbol{\Lambda}^2(\boldsymbol{\Lambda} + \lambda_n \boldsymbol{I})^{-2})}{n - \mathrm{Tr}(\boldsymbol{\Lambda}^2(\boldsymbol{\Lambda} + \lambda_n \boldsymbol{I})^{-2})}.$$

$\square$

#### E.3.2 Isotropic features with finite rank

**Corollary E.3** (Isotropic features for min-$\ell_2$-norm interpolator). *Consider covariance matrix* $\boldsymbol{\Lambda} = \boldsymbol{I}_m$ *($n < m < \infty$), in the over-parameterized regime ($p > n$), the deterministic equivalents* $\mathsf{R}_0^{\mathrm{RFM}}$ *and* $\mathsf{N}_0^{\mathrm{RFM}}$ *specifies the linear relationship in Eq.* (5) *as* $\mathsf{R}_0^{\mathrm{RFM}} = \frac{m-n}{n}\mathsf{N}_0^{\mathrm{RFM}} + \frac{2n-m}{m-n}\sigma^2$.
*While in the under-parameterized regime ($p < n$), we focus on bias and variance separately*

$$\text{Variance: } \left(\mathsf{V}_{\mathrm{R},0}^{\mathrm{RFM}}\right)^2 = \frac{m-n}{n}\mathsf{V}_{\mathrm{R},0}^{\mathrm{RFM}}\mathsf{V}_{\mathrm{N},0}^{\mathrm{RFM}} + \frac{m\sigma^2}{n}\mathsf{V}_{\mathrm{N},0}^{\mathrm{RFM}},$$

$$\text{Bias: } (m-n)\mathsf{B}_{\mathrm{N},0}^{\mathrm{RFM}}(m\mathsf{B}_{\mathrm{R},0}^{\mathrm{RFM}} - n\|\boldsymbol{\theta}_*\|_2^2)(m(\mathsf{B}_{\mathrm{R},0}^{\mathrm{RFM}})^2 - n\|\boldsymbol{\theta}_*\|_2^4)$$

$$= nm(\mathsf{B}_{\mathrm{R},0}^{\mathrm{RFM}} - \|\boldsymbol{\theta}_*\|_2^2)^2[m(\mathsf{B}_{\mathrm{R},0}^{\mathrm{RFM}})^2 + n\|\boldsymbol{\theta}_*\|_2^2\mathsf{B}_{\mathrm{R},0}^{\mathrm{RFM}} - 2n\|\boldsymbol{\theta}_*\|_2^4].$$

**Remark:** In the under-parameterized regime, $V_{R,0}^{RFM}$ and $V_{N,0}^{RFM}$ are related by a hyperbola, the asymptote of which is $V_{R,0}^{RFM} = \frac{m-n}{n}V_{N,0}^{RFM} + \frac{m}{m-n}\sigma^2$. Further, for $p \to n$, we have $B_{R,0}^{RFM} \approx \frac{m-n}{n}B_{R,0}^{RFM} + \frac{2(m-n)}{m}\|\boldsymbol{\theta}_*\|_2^2$, see discussion in Appendix E.3.

Next we present the proof of Corollary E.3 with $\boldsymbol{\Lambda} = \boldsymbol{I}_m$.

*Proof of Corollary E.3.* Here we consider the case where $\boldsymbol{\Lambda} = \boldsymbol{I}_m$. Under this condition, the definitions of $\lambda_p$ and $\lambda_n$ above are simplified to $\frac{m}{1+\lambda_p} = p$ and $\frac{m}{1+\lambda_n} = n$, respectively. Consequently, $\lambda_p$ and $\lambda_n$ have explicit expressions given by $\lambda_p = \frac{m-p}{p}$ and $\lambda_n = \frac{m-n}{n}$, respectively.

First, in the over-parameterized regime ($p > n$), we have

$$B_{N,0}^{RFM} = \frac{p\frac{1}{1+\lambda_n}\|\boldsymbol{\theta}_*\|_2^2}{p-n} = \frac{np}{m(p-n)}\|\boldsymbol{\theta}_*\|_2^2, \quad V_{N,0}^{RFM} = \frac{\sigma^2 p}{\lambda_n(p-n)} = \frac{\sigma^2 np}{(m-n)(p-n)}.$$

$$B_{R,0}^{RFM} = \frac{n\lambda_n^2 \frac{1}{(1+\lambda_n)^2}\|\boldsymbol{\theta}_*\|_2^2}{n - \frac{m}{(1+\lambda_n)^2}} + \frac{n\lambda_n \frac{1}{1+\lambda_n}\|\boldsymbol{\theta}_*\|_2^2}{p-n} = \frac{p(m-n)}{m(p-n)}\|\boldsymbol{\theta}_*\|_2^2,$$

$$V_{R,0}^{RFM} = \frac{\sigma^2 \frac{m}{(1+\lambda_n)^2}}{n - \frac{m}{(1+\lambda_n)^2}} + \frac{\sigma^2 n}{p-n} = \frac{\sigma^2 n}{m-n} + \frac{\sigma^2 n}{p-n}.$$

We eliminate $p$ and obtain that the relationship between $V_{R,0}^{RFM}$ and $V_{N,0}^{RFM}$ is

$$V_{R,0}^{RFM} = \frac{m-n}{n}V_{N,0}^{RFM} + \frac{2n-m}{m-n}\sigma^2.$$

similarly, the relationship between $B_{R,0}^{RFM}$ and $B_{N,0}^{RFM}$ is

$$B_{R,0}^{RFM} = \frac{m-n}{n}B_{N,0}^{RFM}.$$

Combining the above two relationship, we obtain the relationship between test risk $R_0^{RFM}$ and norm $N_0^{RFM}$ as

$$R_0^{RFM} = \frac{m-n}{n}N_0^{RFM} + \frac{2n-m}{m-n}\sigma^2.$$

Accordingly, in the under-parameterized regime ($p < n$), we have

$$B_{N,0}^{RFM} = \frac{p\frac{1}{(1+\lambda_p)^2}\|\boldsymbol{\theta}_*\|_2^2}{n - \frac{m}{(1+\lambda_p)^2}} + \frac{p\frac{1}{1+\lambda_p}\|\boldsymbol{\theta}_*\|_2^2}{n-p} = \frac{p}{m}\left(\frac{p^2}{nm-p^2} + \frac{p}{n-p}\right)\|\boldsymbol{\theta}_*\|_2^2,$$

$$V_{N,0}^{RFM} = \frac{\sigma^2 p}{\lambda_p(p-n)} = \frac{\sigma^2 p^2}{(m-p)(n-p)}.$$

$$B_{R,0}^{RFM} = \frac{n\lambda_p \frac{1}{1+\lambda_p}\|\boldsymbol{\theta}_*\|_2^2}{n-p} = \frac{n(m-p)}{m(n-p)}\|\boldsymbol{\theta}_*\|_2^2, \quad V_{R,0}^{RFM} = \frac{\sigma^2 p}{n-p}.$$

Then we eliminate $p$ and obtain that, in the under-parameterized regime ($p < n$), the relationship between $V_{R,0}^{RFM}$ and $V_{N,0}^{RFM}$ is

$$V_{R,0}^{RFM} = \frac{(m-n)V_{N,0}^{RFM} + \sqrt{(m-n)^2(V_{N,0}^{RFM})^2 + 4nm\sigma^2 V_{N,0}^{RFM}}}{2n},$$

which can be further simplified as a hyperbolic function

$$\left(V_{R,0}^{RFM}\right)^2 = \frac{m-n}{n}V_{R,0}^{RFM}V_{N,0}^{RFM} + \frac{m\sigma^2}{n}V_{N,0}^{RFM},$$

and the asymptote of this hyperbola is $V_{R,0}^{RFM} = \frac{m-n}{n}V_{N,0}^{RFM} + \frac{m}{m-n}\sigma^2$.

Besides, we eliminate $p$ and obtain the relationship between $\mathsf{B}_{\mathsf{R},0}^{\mathsf{RFM}}$ and $\mathsf{B}_{\mathsf{N},0}^{\mathsf{RFM}}$ as

$$\frac{\|\boldsymbol{\theta}_*\|_2^6 n^2 \left( 2\|\boldsymbol{\theta}_*\|_2^2 + \mathsf{B}_{\mathsf{N},0}^{\mathsf{RFM}} - \frac{\mathsf{B}_{\mathsf{N},0}^{\mathsf{RFM}} n}{m} \right)}{m} = (\mathsf{B}_{\mathsf{R},0}^{\mathsf{RFM}})^4 n + (\mathsf{B}_{\mathsf{R},0}^{\mathsf{RFM}})^2 \|\boldsymbol{\theta}_*\|_2^2 n \left( \|\boldsymbol{\theta}_*\|_2^2 + \mathsf{B}_{\mathsf{N},0}^{\mathsf{RFM}} - \frac{4\|\boldsymbol{\theta}_*\|_2^2 n}{m} - \frac{\mathsf{B}_{\mathsf{N},0}^{\mathsf{RFM}} n}{m} \right)$$

$$+ \mathsf{B}_{\mathsf{R},0}^{\mathsf{RFM}} \|\boldsymbol{\theta}_*\|_2^4 n \left( \mathsf{B}_{\mathsf{N},0}^{\mathsf{RFM}} + \frac{5\|\boldsymbol{\theta}_*\|_2^2 n}{m} - \frac{\mathsf{B}_{\mathsf{N},0}^{\mathsf{RFM}} n}{m} \right) + (\mathsf{B}_{\mathsf{R},0}^{\mathsf{RFM}})^3 \left( -\mathsf{B}_{\mathsf{N},0}^{\mathsf{RFM}} m - 2\|\boldsymbol{\theta}_*\|_2^2 n + \mathsf{B}_{\mathsf{N},0}^{\mathsf{RFM}} n + \frac{\|\boldsymbol{\theta}_*\|_2^2 n^2}{m} \right),$$

which can be simplified to

$$\mathsf{B}_{\mathsf{N},0}^{\mathsf{RFM}}(m-n)(m\mathsf{B}_{\mathsf{R},0}^{\mathsf{RFM}} - n\|\boldsymbol{\theta}_*\|_2^2)(m(\mathsf{B}_{\mathsf{R},0}^{\mathsf{RFM}})^2 - n\|\boldsymbol{\theta}_*\|_2^4)$$
$$= nm(\mathsf{B}_{\mathsf{R},0}^{\mathsf{RFM}} - \|\boldsymbol{\theta}_*\|_2^2)^2 (m(\mathsf{B}_{\mathsf{R},0}^{\mathsf{RFM}})^2 - 2n\|\boldsymbol{\theta}_*\|_2^4 + n\|\boldsymbol{\theta}_*\|_2^2 \mathsf{B}_{\mathsf{R},0}^{\mathsf{RFM}}).$$

We can find that in this case, the relationship can be easily written as

$$\mathsf{B}_{\mathsf{N},0}^{\mathsf{RFM}} = \frac{nm(\mathsf{B}_{\mathsf{R},0}^{\mathsf{RFM}} - \|\boldsymbol{\theta}_*\|_2^2)^2 (m(\mathsf{B}_{\mathsf{R},0}^{\mathsf{RFM}})^2 - 2n\|\boldsymbol{\theta}_*\|_2^4 + n\|\boldsymbol{\theta}_*\|_2^2 \mathsf{B}_{\mathsf{R},0}^{\mathsf{RFM}})}{(m-n)(m\mathsf{B}_{\mathsf{R},0}^{\mathsf{RFM}} - n\|\boldsymbol{\theta}_*\|_2^2)(m(\mathsf{B}_{\mathsf{R},0}^{\mathsf{RFM}})^2 - n\|\boldsymbol{\theta}_*\|_2^4)}.$$

Next we will show that when $p \to n$, which also implies that $\mathsf{B}_{\mathsf{N},0}^{\mathsf{RFM}} \to \infty$ and $\mathsf{B}_{\mathsf{R},0}^{\mathsf{RFM}} \to \infty$, this relationship is approximately linear.

Recall that the relationship between $\mathsf{B}_{\mathsf{R},0}^{\mathsf{RFM}}$ and $\mathsf{B}_{\mathsf{N},0}^{\mathsf{RFM}}$ is given by $\mathsf{B}_{\mathsf{R},0}^{\mathsf{RFM}} = \frac{(m-n)}{n}\mathsf{B}_{\mathsf{N},0}^{\mathsf{RFM}}$, and is equivalent to $\mathsf{B}_{\mathsf{N},0}^{\mathsf{RFM}} = \frac{n}{(m-n)}\mathsf{B}_{\mathsf{R},0}^{\mathsf{RFM}} := f(\mathsf{B}_{\mathsf{R},0}^{\mathsf{RFM}})$. We then do a difference and get

$$\mathsf{B}_{\mathsf{N},0}^{\mathsf{RFM}} - f(\mathsf{B}_{\mathsf{R},0}^{\mathsf{RFM}}) = \frac{nm(\mathsf{B}_{\mathsf{R},0}^{\mathsf{RFM}} - \|\boldsymbol{\theta}_*\|_2^2)^2 (m(\mathsf{B}_{\mathsf{R},0}^{\mathsf{RFM}})^2 - 2n\|\boldsymbol{\theta}_*\|_2^4 + n\|\boldsymbol{\theta}_*\|_2^2 \mathsf{B}_{\mathsf{R},0}^{\mathsf{RFM}})}{(m-n)(m\mathsf{B}_{\mathsf{R},0}^{\mathsf{RFM}} - n\|\boldsymbol{\theta}_*\|_2^2)(m(\mathsf{B}_{\mathsf{R},0}^{\mathsf{RFM}})^2 - n\|\boldsymbol{\theta}_*\|_2^4)} - \frac{n}{m-n}\mathsf{B}_{\mathsf{R},0}^{\mathsf{RFM}},$$

then take $\mathsf{B}_{\mathsf{R},0}^{\mathsf{RFM}} \to \infty$ and we get

$$\lim_{\mathsf{B}_{\mathsf{R},0}^{\mathsf{RFM}} \to \infty} \mathsf{B}_{\mathsf{N},0}^{\mathsf{RFM}} - f(\mathsf{B}_{\mathsf{R},0}^{\mathsf{RFM}}) = -\frac{2n}{m}\|\boldsymbol{\theta}_*\|_2^2.$$

Finally, organizing this equation and we get

$$\mathsf{B}_{\mathsf{R},0}^{\mathsf{RFM}} \approx \frac{m-n}{n}\mathsf{B}_{\mathsf{N},0}^{\mathsf{RFM}} + \frac{2(m-n)}{m}\|\boldsymbol{\theta}_*\|_2^2.$$

$\square$

### E.3.3 Proof on features under power law assumption

*Proof of Corollary 4.2.* First, we use integral approximation to give approximations to some quantities commonly used in deterministic equivalence to prepare for the subsequent derivations.

According to the integral approximation in [53, Lemma 1], we have

$$\mathrm{Tr}(\boldsymbol{\Lambda}(\boldsymbol{\Lambda}+\nu_2)^{-1}) \approx C_1 \nu_2^{-\frac{1}{\alpha}}, \ \mathrm{Tr}(\boldsymbol{\Lambda}^2(\boldsymbol{\Lambda}+\nu_2)^{-2}) \approx C_2 \nu_2^{-\frac{1}{\alpha}}, \ \mathrm{Tr}(\boldsymbol{\Lambda}(\boldsymbol{\Lambda}+\nu_2)^{-2}) \approx (C_1 - C_2)\nu_2^{-\frac{1}{\alpha}-1}, \quad (69)$$

where $C_1$ and $C_2$ are

$$C_1 = \frac{\pi}{\alpha \sin(\pi/\alpha)}, \quad C_2 = \frac{\pi(\alpha-1)}{\alpha^2 \sin(\pi/\alpha)}, \quad \text{with } C_1 > C_2. \tag{70}$$

Besides, according to definition of $T(\nu)$ Appendix B.5, we have

$$\langle \boldsymbol{\theta}_*, (\boldsymbol{\Lambda}+\nu_2)^{-1}\boldsymbol{\theta}_* \rangle = T_{2r,1}^1(\nu_2) \approx C_3 \nu_2^{(2r-1)\wedge 0},$$
$$\langle \boldsymbol{\theta}_*, \boldsymbol{\Lambda}(\boldsymbol{\Lambda}+\nu_2)^{-2}\boldsymbol{\theta}_* \rangle = T_{2r+1,2}^1(\nu_2) \approx C_4 \nu_2^{(2r-1)\wedge 0}.$$

When $r \in (0, \frac{1}{2})$, according to the integral approximation, we have

$$C_3 = \frac{\pi}{\alpha \sin(2\pi r)}, \quad C_4 = \frac{2\pi r}{\alpha \sin(2\pi r)}, \quad \text{with } C_3 > C_4. \tag{71}$$

Otherwise, if $r \in [\frac{1}{2}, \infty)$, we have

$$\frac{1}{\alpha(2r-1)} < C_3 < \frac{1}{\alpha(2r-1)} + 1\,, \quad \frac{1}{\alpha(2r-1)} < C_4 < \frac{1}{\alpha(2r-1)} + 1\,, \quad \text{with } C_3 > C_4.$$

For $\langle \boldsymbol{\theta}_*, (\boldsymbol{\Lambda} + \nu_2)^{-2} \boldsymbol{\theta}_* \rangle$, we have to discuss its approximation in the case $r \in (0, \frac{1}{2})$, $r \in [\frac{1}{2}, 1)$ and $r \in [\frac{1}{2}, \infty)$ separately.

$$\langle \boldsymbol{\theta}_*, (\boldsymbol{\Lambda} + \nu_2)^{-2} \boldsymbol{\theta}_* \rangle \approx \begin{cases} (C_3 - C_4)\nu_2^{2r-2}, & \text{if } r \in (0, \frac{1}{2})\,; \\ C_5 \nu_2^{2r-2}, & \text{if } r \in [\frac{1}{2}, 1)\,; \\ C_6, & \text{if } r \in [1, \infty)\,, \end{cases}$$

where $\frac{1}{2\alpha(r-1)} < C_6 < \frac{1}{2\alpha(r-1)} + 1$.

With the results of the integral approximation above, we next derive the relationship between $\mathsf{R}_0^{\mathrm{RFM}}$ and $\mathsf{N}_0^{\mathrm{RFM}}$ **separately in over-parameterized regime** ($p > n$) **and under-parameterized regime** ($p < n$).

**The relationship in over-parameterized regime** ($p > n$)    According to the self-consistent equation

$$1 + \frac{n}{p} - \sqrt{\left(1 - \frac{n}{p}\right)^2 + \frac{4\lambda}{p\nu_2}} = \frac{2}{p} \mathrm{Tr}\left(\boldsymbol{\Lambda}\left(\boldsymbol{\Lambda} + \nu_2\right)^{-1}\right),$$

$$\nu_1 = \frac{\nu_2}{2}\left[1 - \frac{n}{p} + \sqrt{\left(1 - \frac{n}{p}\right)^2 + \frac{4\lambda}{p\nu_2}}\right],$$

In the over-parameterized regime ($p > n$), as $\lambda \to 0$, for the first equation, $\frac{4\lambda}{p\nu_2}$ will approach 0, and $\mathrm{Tr}(\boldsymbol{\Lambda}(\boldsymbol{\Lambda} + \nu_2)^{-1})$ will converge to $n$. Consequently, by Eq. (69), $\nu_2$ will converge to the constant $(\frac{n}{C_1})^{-\alpha}$. Furthermore, from the second equation, $\nu_1$ will converge to $\nu_2(1 - \frac{n}{p})$. Thus, according to Eq. (69), we have

$$\mathrm{Tr}(\boldsymbol{\Lambda}(\boldsymbol{\Lambda} + \nu_2)^{-1}) \approx n, \quad \mathrm{Tr}(\boldsymbol{\Lambda}^2(\boldsymbol{\Lambda} + \nu_2)^{-2}) \approx \frac{C_2}{C_1} n, \quad \mathrm{Tr}(\boldsymbol{\Lambda}(\boldsymbol{\Lambda} + \nu_2)^{-2}) \approx (C_1 - C_2)(\frac{n}{C_1})^{\alpha+1}.$$

Thus, in the over-parameterized regime

$$\Upsilon(\nu_1, \nu_2) = \frac{p}{n}\left[\left(1 - \frac{\nu_1}{\nu_2}\right)^2 + \left(\frac{\nu_1}{\nu_2}\right)^2 \frac{\mathrm{Tr}\left(\boldsymbol{\Lambda}^2(\boldsymbol{\Lambda} + \nu_2)^{-2}\right)}{p - \mathrm{Tr}\left(\boldsymbol{\Lambda}^2(\boldsymbol{\Lambda} + \nu_2)^{-2}\right)}\right]$$

$$\approx \frac{p}{n}\left[\left(\frac{n}{p}\right)^2 + \left(1 - \frac{n}{p}\right)^2 \frac{\mathrm{Tr}\left(\boldsymbol{\Lambda}^2(\boldsymbol{\Lambda} + \nu_2)^{-2}\right)}{p - \mathrm{Tr}\left(\boldsymbol{\Lambda}^2(\boldsymbol{\Lambda} + \nu_2)^{-2}\right)}\right]$$

$$\approx \frac{\frac{C_2}{C_1}p - 2\frac{C_2}{C_1}n + n}{p - \frac{C_2}{C_1}n},$$

$$\chi(\nu_2) = \frac{\mathrm{Tr}(\boldsymbol{\Lambda}(\boldsymbol{\Lambda} + \nu_2)^{-2})}{p - \mathrm{Tr}(\boldsymbol{\Lambda}^2(\boldsymbol{\Lambda} + \nu_2)^{-2})} \approx \frac{(C_1 - C_2)(\frac{n}{C_1})^{\alpha+1}}{p - \frac{C_2}{C_1}n}\,.$$

According to the approximation, we have the deterministic equivalents of variance terms

$$\mathsf{V}_{\mathrm{R},0}^{\mathrm{RFM}} = \sigma^2 \frac{\Upsilon(\nu_1, \nu_2)}{1 - \Upsilon(\nu_1, \nu_2)} \approx \sigma^2 \frac{(C_1 - 2C_2)n + C_2 p}{(C_1 - C_2)(p - n)}\,,$$

$$\mathsf{V}_{\mathrm{N},0}^{\mathrm{RFM}} = \sigma^2 \frac{p}{n} \frac{\chi(\nu_2)}{1 - \Upsilon(\nu_1, \nu_2)} \approx \sigma^2 \frac{(\frac{n}{C_1})^\alpha p}{p - n}\,.$$

Then recall Eq. (70), we eliminate $p$ and obtain

$$\mathsf{V}_{\mathrm{R},0}^{\mathrm{RFM}} \approx \left(\frac{n}{C_1}\right)^{-\alpha} \mathsf{V}_{\mathrm{N},0}^{\mathrm{RFM}} + \sigma^2 \frac{2C_2 - C_1}{C_1 - C_2} = \left(\frac{n}{C_1}\right)^{-\alpha} \mathsf{V}_{\mathrm{N},0}^{\mathrm{RFM}} + \sigma^2(\alpha - 2)\,. \tag{72}$$

For the bias terms, due to the varying approximation behaviors of the quantities containing $\boldsymbol{\theta}_*$ for different values of $r$, we have to discuss their approximations in the conditions $r \in (0, \frac{1}{2})$, $r \in [\frac{1}{2}, 1)$ and $r \in [\frac{1}{2}, \infty)$ separately.

**Condition 1:** $r \in (0, \frac{1}{2})$

$$\mathsf{B}_{\mathsf{R},0}^{\mathsf{RFM}} = \frac{\nu_2^2}{1 - \Upsilon(\nu_1, \nu_2)} \left[ \langle \boldsymbol{\theta}_*, (\boldsymbol{\Lambda} + \nu_2)^{-2} \boldsymbol{\theta}_* \rangle + \chi(\nu_2) \langle \boldsymbol{\theta}_*, \boldsymbol{\Lambda}(\boldsymbol{\Lambda} + \nu_2)^{-2} \boldsymbol{\theta}_* \rangle \right]$$

$$\approx \frac{\left( \frac{n}{C_1} \right)^{-2\alpha r} ((C_1 C_4 - C_2 C_3) n + C_1 (C_3 - C_4) p)}{(C1 - C2)(p - n)},$$

$$\mathsf{B}_{\mathsf{N},0}^{\mathsf{RFM}} = \frac{\nu_2}{\nu_1} \langle \boldsymbol{\theta}_*, (\boldsymbol{\Lambda} + \nu_2)^{-1} \boldsymbol{\theta}_* \rangle - \frac{\lambda}{n} \frac{\nu_2^2}{\nu_1^2} \frac{\langle \boldsymbol{\theta}_*, (\boldsymbol{\Lambda} + \nu_2)^{-2} \boldsymbol{\theta}_* \rangle + \chi(\nu_2) \langle \boldsymbol{\theta}_*, \boldsymbol{\Lambda}(\boldsymbol{\Lambda} + \nu_2)^{-2} \boldsymbol{\theta}_* \rangle}{1 - \Upsilon(\nu_1, \nu_2)}$$

$$\approx \frac{\nu_2}{\nu_1} \langle \boldsymbol{\theta}_*, (\boldsymbol{\Lambda} + \nu_2)^{-1} \boldsymbol{\theta}_* \rangle$$

$$\approx \frac{\left( \frac{n}{C_1} \right)^{-\alpha(2r-1)} C_3 p}{p - n}.$$

Then we eliminate $p$ and obtain

$$\mathsf{B}_{\mathsf{R},0}^{\mathsf{RFM}} \approx \left( \frac{n}{C_1} \right)^{-\alpha} \mathsf{B}_{\mathsf{N},0}^{\mathsf{RFM}} + \left( \frac{n}{C_1} \right)^{-2\alpha r} \frac{C_2 C_3 - C_1 C_4}{C_1 - C_2}. \tag{73}$$

**Condition 2:** $r \in [\frac{1}{2}, 1)$

$$\mathsf{B}_{\mathsf{R},0}^{\mathsf{RFM}} = \frac{\nu_2^2}{1 - \Upsilon(\nu_1, \nu_2)} \left[ \langle \boldsymbol{\theta}_*, (\boldsymbol{\Lambda} + \nu_2)^{-2} \boldsymbol{\theta}_* \rangle + \chi(\nu_2) \langle \boldsymbol{\theta}_*, \boldsymbol{\Lambda}(\boldsymbol{\Lambda} + \nu_2)^{-2} \boldsymbol{\theta}_* \rangle \right]$$

$$\approx \frac{\left( \frac{n}{C_1} \right)^{-\alpha} \left( C_1 \left( C_4 n + C_5 \left( \frac{n}{C_1} \right)^{-\alpha(2r-1)} p \right) - C_2 n \left( C_4 + C_5 \left( \frac{n}{C_1} \right)^{-\alpha(2r-1)} \right) \right)}{(C1 - C2)(p - n)},$$

$$\mathsf{B}_{\mathsf{N},0}^{\mathsf{RFM}} = \langle \boldsymbol{\theta}_*, \boldsymbol{\Lambda}(\boldsymbol{\Lambda} + \nu_2)^{-2} \boldsymbol{\theta}_* \rangle \cdot \frac{p}{p - \mathrm{df}_2(\nu_2)}$$

$$+ \frac{p}{n} \nu_2^2 \left( \langle \boldsymbol{\theta}_*, (\boldsymbol{\Lambda} + \nu_2)^{-2} \boldsymbol{\theta}_* \rangle + \chi(\nu_2) \langle \boldsymbol{\theta}_*, \boldsymbol{\Lambda}(\boldsymbol{\Lambda} + \nu_2)^{-2} \boldsymbol{\theta}_* \rangle \right) \cdot \frac{\chi(\nu_2)}{1 - \Upsilon(\nu_1, \nu_2)}$$

$$\approx \frac{\left( C_4 + C_5 \left( \frac{n}{C_1} \right)^{-\alpha(2r-1)} \right) p}{p - n}.$$

Then we eliminate $p$ and obtain

$$\mathsf{B}_{\mathsf{R},0}^{\mathsf{RFM}} \approx \left( \frac{n}{C_1} \right)^{-\alpha} \mathsf{B}_{\mathsf{N},0}^{\mathsf{RFM}} + \left( \frac{n}{C_1} \right)^{-\alpha} \frac{-C_1 C_4 + C_2 C_4 + C_2 C_5 \left( \frac{n}{C_1} \right)^{-\alpha(2r-1)}}{C_1 - C_2}$$

$$\approx \left( \frac{n}{C_1} \right)^{-\alpha} \mathsf{B}_{\mathsf{N},0}^{\mathsf{RFM}} - \left( \frac{n}{C_1} \right)^{-\alpha} C_4.$$

The last "$\approx$" holds because $\left( \frac{n}{C_1} \right)^{-\alpha(2r-1)} = o(1)$.

**Condition 3:** $r \in [1, \infty)$

$$\mathsf{B}_{\mathsf{R},0}^{\mathsf{RFM}} = \frac{\nu_2^2}{1 - \Upsilon(\nu_1, \nu_2)} \left[ \langle \boldsymbol{\theta}_*, (\boldsymbol{\Lambda} + \nu_2)^{-2} \boldsymbol{\theta}_* \rangle + \chi(\nu_2) \langle \boldsymbol{\theta}_*, \boldsymbol{\Lambda}(\boldsymbol{\Lambda} + \nu_2)^{-2} \boldsymbol{\theta}_* \rangle \right]$$

$$\approx \frac{\left( \frac{n}{C_1} \right)^{-2\alpha} \left( C_1 \left( C_4 n \left( \frac{n}{C_1} \right)^{\alpha} + C_6 p \right) - C_2 n \left( C_6 + C_4 \left( \frac{n}{C_1} \right)^{\alpha} \right) \right)}{(C1 - C2)(p - n)},$$

$$\mathsf{B}_{\mathsf{N},0}^{\mathsf{RFM}} = \langle \boldsymbol{\theta}_*, \boldsymbol{\Lambda}(\boldsymbol{\Lambda} + \nu_2)^{-2} \boldsymbol{\theta}_* \rangle \cdot \frac{p}{p - \mathrm{df}_2(\nu_2)}$$

$$+ \frac{p}{n} \nu_2^2 \left( \langle \boldsymbol{\theta}_*, (\boldsymbol{\Lambda} + \nu_2)^{-2} \boldsymbol{\theta}_* \rangle + \chi(\nu_2) \langle \boldsymbol{\theta}_*, \boldsymbol{\Lambda}(\boldsymbol{\Lambda} + \nu_2)^{-2} \boldsymbol{\theta}_* \rangle \right) \cdot \frac{\chi(\nu_2)}{1 - \Upsilon(\nu_1, \nu_2)}$$

$$\approx \frac{\left( C_4 + C_6 \left( \frac{n}{C_1} \right)^{-\alpha} \right) p}{p - n} .$$

Then we eliminate $p$ and obtain

$$\mathsf{B}_{\mathsf{R},0}^{\mathsf{RFM}} \approx \left( \frac{n}{C_1} \right)^{-\alpha} \mathsf{B}_{\mathsf{N},0}^{\mathsf{RFM}} + \left( \frac{n}{C_1} \right)^{-\alpha} \frac{-C_1 C_4 + C_2 C_4 + C_2 C_6 \left( \frac{n}{C_1} \right)^{-\alpha}}{C_1 - C_2}$$

$$\approx \left( \frac{n}{C_1} \right)^{-\alpha} \mathsf{B}_{\mathsf{N},0}^{\mathsf{RFM}} - \left( \frac{n}{C_1} \right)^{-\alpha} C_4 .$$

The last "$\approx$" holds because $\left( \frac{n}{C_1} \right)^{-\alpha(2r-1)} = o(1)$.

Combining the above condition $r \in [\frac{1}{2}, 1)$ and $r \in [1, \infty)$, we have for $r \in [\frac{1}{2}, \infty)$

$$\mathsf{B}_{\mathsf{R},0}^{\mathsf{RFM}} \approx \left( \frac{n}{C_1} \right)^{-\alpha} \mathsf{B}_{\mathsf{N},0}^{\mathsf{RFM}} - \left( \frac{n}{C_1} \right)^{-\alpha} C_4 . \tag{74}$$

From Eqs. (72) to (74), we know that the relationship between $\mathsf{R}_0^{\mathsf{RFM}}$ and $\mathsf{N}_0^{\mathsf{RFM}}$ in the over-parameterized regime can be written as

$$\mathsf{R}_0^{\mathsf{RFM}} \approx (n/C_\alpha)^{-\alpha} \mathsf{N}_0^{\mathsf{RFM}} + C_{n,\alpha,r,1} .$$

**The relationship in under-parameterized regime** $(p < n)$  While in the under-parameterized regime $(p < n)$, When $\lambda \to 0$, $\mathrm{Tr}(\boldsymbol{\Lambda}(\boldsymbol{\Lambda} + \nu_2)^{-1})$ will converge to $p$, which means $\nu_2$ will converge to $(\frac{p}{C_1})^{-\alpha}$ and $\nu_1$ will converge to 0, with $\frac{\lambda}{\nu_1} \to n - p$.

Accordingly, in the under-parameterized regime

$$\Upsilon(\nu_1, \nu_2) = \frac{p}{n} \left[ \left( 1 - \frac{\nu_1}{\nu_2} \right)^2 + \left( \frac{\nu_1}{\nu_2} \right)^2 \frac{\mathrm{Tr}\left( \boldsymbol{\Lambda}^2(\boldsymbol{\Lambda} + \nu_2)^{-2} \right)}{p - \mathrm{Tr}\left( \boldsymbol{\Lambda}^2(\boldsymbol{\Lambda} + \nu_2)^{-2} \right)} \right] \to \frac{p}{n} ,$$

$$\chi(\nu_2) = \frac{\mathrm{Tr}\left( \boldsymbol{\Lambda}(\boldsymbol{\Lambda} + \nu_2)^{-2} \right)}{p - \mathrm{Tr}\left( \boldsymbol{\Lambda}^2(\boldsymbol{\Lambda} + \nu_2)^{-2} \right)} \to \frac{1}{\nu_2} \approx (\frac{p}{C_1})^{\alpha} .$$

Then we can further obtain that, for the variance

$$\mathsf{V}_{\mathsf{R},0}^{\mathsf{RFM}} = \sigma^2 \frac{\Upsilon(\nu_1, \nu_2)}{1 - \Upsilon(\nu_1, \nu_2)} \approx \sigma^2 \frac{p}{n - p},$$

$$\mathsf{V}_{\mathsf{N},0}^{\mathsf{RFM}} = \sigma^2 \frac{p}{n} \frac{\chi(\nu_2)}{1 - \Upsilon(\nu_1, \nu_2)} \approx \sigma^2 C_1^{-\alpha} \frac{p^{\alpha+1}}{n - p} .$$

For the relationship in the under-parameterized regime, we separately consider two cases, i.e. $p \ll n$ and $p \to n$.

First, we derive the relationship in the under-parameterized regime ($p < n$) as $p \to n$, based on the relationship in the over-parameterized regime. Recall the relationship between $\mathsf{V}_{\mathsf{R},0}^{\mathsf{RFM}}$ and $\mathsf{V}_{\mathsf{N},0}^{\mathsf{RFM}}$ in the over-parameterized regime, as presented in Eq. (72), given by

$$\mathsf{V}_{\mathsf{R},0}^{\mathsf{RFM}} \approx \left(\frac{n}{C_1}\right)^{-\alpha} \mathsf{V}_{\mathsf{N},0}^{\mathsf{RFM}} + \sigma^2(\alpha - 2) =: h(\mathsf{V}_{\mathsf{N},0}^{\mathsf{RFM}}).$$

Substituting the expression for $\mathsf{V}_{\mathsf{N},0}^{\mathsf{RFM}}$ in the under-parameterized regime into this relationship, we obtain

$$\mathsf{V}_{\mathsf{R},0}^{\mathsf{RFM}} \approx \left(\frac{n}{C_1}\right)^{-\alpha} \sigma^2 C_1^{-\alpha} \frac{p^{\alpha+1}}{n-p} + \sigma^2(\alpha - 2),$$

then we compute $\mathsf{V}_{\mathsf{R},0}^{\mathsf{RFM}} - h(\mathsf{V}_{\mathsf{N},0}^{\mathsf{RFM}})$ and obtain

$$\mathsf{V}_{\mathsf{R},0}^{\mathsf{RFM}} - h(\mathsf{V}_{\mathsf{N},0}^{\mathsf{RFM}}) = \sigma^2 \frac{p}{n-p} - \left(\frac{n}{C_1}\right)^{-\alpha} \sigma^2 C_1^{-\alpha} \frac{p^{\alpha+1}}{n-p} - \sigma^2(\alpha - 2)$$

$$= \sigma^2 \left(\frac{p - p^{\alpha+1} n^{-\alpha}}{n-p}\right) - \sigma^2(\alpha - 2).$$

Taking limits on the left and right sides of the equation, we get

$$\lim_{p \to n} \left(\mathsf{V}_{\mathsf{R},0}^{\mathsf{RFM}} - h(\mathsf{V}_{\mathsf{N},0}^{\mathsf{RFM}})\right) = 2\sigma^2.$$

Then when $p \to n$, we have

$$\mathsf{V}_{\mathsf{R},0}^{\mathsf{RFM}} \approx \left(\frac{n}{C_1}\right)^{-\alpha} \mathsf{V}_{\mathsf{N},0}^{\mathsf{RFM}} + \sigma^2\alpha. \tag{75}$$

For $p \ll n$, we have $\frac{1}{n-p} \approx \frac{1}{n}$, then

$$\mathsf{V}_{\mathsf{R},0}^{\mathsf{RFM}} = \sigma^2 \frac{\Upsilon(\nu_1, \nu_2)}{1 - \Upsilon(\nu_1, \nu_2)} \approx \sigma^2 \frac{p}{n},$$

$$\mathsf{V}_{\mathsf{N},0}^{\mathsf{RFM}} = \sigma^2 \frac{p}{n} \frac{\chi(\nu_2)}{1 - \Upsilon(\nu_1, \nu_2)} \approx \sigma^2 C_1^{-\alpha} \frac{p^{\alpha+1}}{n}.$$

Eliminate $p$ and we have

$$\mathsf{V}_{\mathsf{R},0}^{\mathsf{RFM}} \approx \left(\sigma^2\right)^{\frac{\alpha}{\alpha+1}} C_1^{\frac{\alpha}{\alpha+1}} \left(\mathsf{V}_{\mathsf{R},0}^{\mathsf{RFM}}\right)^{\frac{1}{\alpha+1}}.$$

Next, for the bias term we have

$$\mathsf{B}_{\mathsf{R},0}^{\mathsf{RFM}} = \frac{\nu_2^2}{1 - \Upsilon(\nu_1, \nu_2)} \left[\langle \boldsymbol{\theta}_*, (\boldsymbol{\Lambda} + \nu_2)^{-2} \boldsymbol{\theta}_* \rangle + \chi(\nu_2)\langle \boldsymbol{\theta}_*, \boldsymbol{\Lambda}(\boldsymbol{\Lambda} + \nu_2)^{-2} \boldsymbol{\theta}_* \rangle\right]$$

$$\approx \frac{\nu_2}{1 - \Upsilon(\nu_1, \nu_2)} \langle \boldsymbol{\theta}_*, (\boldsymbol{\Lambda} + \nu_2)^{-1} \boldsymbol{\theta}_* \rangle$$

$$\approx \frac{n}{n-p} C_3 \nu_2^{2r \wedge 1}.$$

$$\mathsf{B}_{\mathsf{N},0}^{\mathsf{RFM}} = p\langle \boldsymbol{\theta}_*, \boldsymbol{\Lambda}(\boldsymbol{\Lambda} + \nu_2)^{-2} \boldsymbol{\theta}_* \rangle \cdot \frac{1}{p - \mathrm{df}_2(\nu_2)}$$

$$+ \frac{p}{n} \chi(\nu_2) \frac{\nu_2^2}{1 - \Upsilon(\nu_1, \nu_2)} \left[\langle \boldsymbol{\theta}_*, (\boldsymbol{\Lambda} + \nu_2)^{-2} \boldsymbol{\theta}_* \rangle + \chi(\nu_2)\langle \boldsymbol{\theta}_*, \boldsymbol{\Lambda}(\boldsymbol{\Lambda} + \nu_2)^{-2} \boldsymbol{\theta}_* \rangle\right]$$

$$\approx p\langle \boldsymbol{\theta}_*, \boldsymbol{\Lambda}(\boldsymbol{\Lambda} + \nu_2)^{-2} \boldsymbol{\theta}_* \rangle \cdot \frac{1}{p - \mathrm{df}_2(\nu_2)} + \frac{p}{n} \chi(\nu_2) \frac{\nu_2}{1 - \Upsilon(\nu_1, \nu_2)} \langle \boldsymbol{\theta}_*, (\boldsymbol{\Lambda} + \nu_2)^{-1} \boldsymbol{\theta}_* \rangle$$

$$\approx \frac{p}{p - \frac{C_2}{C_1}p} C_4 \nu_2^{(2r-1) \wedge 0} + \frac{p}{n-p} C_3 \nu_2^{(2r-1) \wedge 0}$$

$$\approx \left(\frac{C_1 C_4}{C_1 - C_2} + \frac{p}{n-p} C_3\right) \nu_2^{(2r-1) \wedge 0}.$$

Then we use the approximation $\nu_2 \approx \left(\frac{p}{C_1}\right)^{-\alpha}$ and obtain

$$\mathsf{B}_{\mathsf{R},0}^{\mathsf{RFM}} \approx \frac{n}{n-p} C_3 \nu_2^{2r \wedge 1} \approx \frac{n}{n-p} C_3 \left(\frac{p}{C_1}\right)^{-\alpha(2r \wedge 1)},$$

$$\mathsf{B}_{\mathsf{N},0}^{\mathsf{RFM}} \approx \left( \frac{C_1 C_4}{C_1 - C_2} + \frac{p}{n-p} C_3 \right) \nu_2^{(2r-1)\wedge 0} \approx \left( \frac{C_1 C_4}{C_1 - C_2} + \frac{p}{n-p} C_3 \right) \left( \frac{p}{C_1} \right)^{-\alpha[(2r-1)\wedge 0]}.$$

Similarly to the bias term, we derive the relationship in the under-parameterized regime ($p < n$) as $p \to n$, based on the relationship in the over-parameterized regime. And we discuss the relationship when $r \in (0, \frac{1}{2})$ and $r \in [\frac{1}{2}, \infty)$ separately.

**Condition 1:** $r \in (0, \frac{1}{2})$. Recall the relationship between $\mathsf{B}_{\mathsf{R},0}^{\mathsf{RFM}}$ and $\mathsf{B}_{\mathsf{N},0}^{\mathsf{RFM}}$ in the over-parameterized regime, as presented in Eq. (73), given by:

$$\mathsf{B}_{\mathsf{R},0}^{\mathsf{RFM}} = \left( \frac{n}{C_1} \right)^{-\alpha} \mathsf{B}_{\mathsf{N},0}^{\mathsf{RFM}} + \left( \frac{n}{C_1} \right)^{-2\alpha r} \frac{C_2 C_3 - C_1 C_4}{C_1 - C_2} =: f(\mathsf{B}_{\mathsf{N},0}^{\mathsf{RFM}}).$$

Substituting the expression for $\mathsf{B}_{\mathsf{N},0}^{\mathsf{RFM}}$ in the under-parameterized regime into this relationship, we obtain:

$$f(\mathsf{B}_{\mathsf{N},0}^{\mathsf{RFM}}) = \left( \frac{n}{C_1} \right)^{-\alpha} \left( \frac{C_1 C_4}{C_1 - C_2} + \frac{p}{n-p} C_3 \right) \left( \frac{p}{C_1} \right)^{-\alpha(2r-1)} + \left( \frac{n}{C_1} \right)^{-2\alpha r} \frac{C_2 C_3 - C_1 C_4}{C_1 - C_2},$$

then we compute $\mathsf{B}_{\mathsf{R},0}^{\mathsf{RFM}} - f(\mathsf{B}_{\mathsf{N},0}^{\mathsf{RFM}})$ and obtain

$$\mathsf{B}_{\mathsf{R},0}^{\mathsf{RFM}} - f(\mathsf{B}_{\mathsf{N},0}^{\mathsf{RFM}}) = C_1^{2\alpha r} \Big( \frac{n}{n-p} C_3 p^{-2\alpha r} - \frac{C_1 C_4}{C_1 - C_2} p^{-\alpha(2r-1)} n^{-\alpha}$$
$$- \frac{p}{n-p} C_3 p^{-\alpha(2r-1)} n^{-\alpha} - \frac{C_2 C_3 - C_1 C_4}{C_1 - C_2} n^{-2\alpha r} \Big).$$

To simplify this equation, we begin by computing $\frac{n}{n-p} C_3 p^{-2\alpha r} - \frac{p}{n-p} C_3 p^{-\alpha(2r-1)} n^{-\alpha}$ and obtain

$$\frac{n}{n-p} C_3 p^{-2\alpha r} - \frac{p}{n-p} C_3 p^{-\alpha(2r-1)} n^{-\alpha} = C_3 p^{-\alpha(2r-1)} \left( \frac{n}{n-p} p^{-\alpha} - \frac{p}{n-p} n^{-\alpha} \right)$$
$$= C_3 p^{-\alpha(2r-1)} \frac{np^{-\alpha} - pn^{-\alpha}}{n-p},$$

where $\frac{np^{-\alpha} - pn^{-\alpha}}{n-p}$ is monotonically decreasing in $p$ (monotonicity can be obtained by simple derivatives), and by applying L'Hôpital's rule, we have:

$$\lim_{p \to n} \frac{np^{-\alpha} - pn^{-\alpha}}{n-p} = \lim_{p \to n} \frac{-\alpha np^{-\alpha-1} - n^{-\alpha}}{-1} = (\alpha+1) n^{-\alpha}.$$

Thus we have

$$\lim_{p \to n} C_3 p^{-\alpha(2r-1)} \frac{np^{-\alpha} - pn^{-\alpha}}{n-p} = (\alpha+1) C_3 n^{-2\alpha r}.$$

Thus we have

$$\lim_{p \to n} C_1^{2\alpha r} \Big( C_3 p^{-\alpha(2r-1)} \frac{np^{-\alpha} - pn^{-\alpha}}{n-p} - \frac{C_1 C_4}{C_1 - C_2} p^{-\alpha(2r-1)} n^{-\alpha} - \frac{C_2 C_3 - C_1 C_4}{C_1 - C_2} n^{-2\alpha r} \Big)$$
$$= C_1^{2\alpha r} \Big( (\alpha+1) C_3 n^{-2\alpha r} - \frac{C_1 C_4}{C_1 - C_2} n^{-2\alpha r} - \frac{C_2 C_3 - C_1 C_4}{C_1 - C_2} n^{-2\alpha r} \Big)$$
$$= C_1^{2\alpha r} C_3 n^{-2\alpha r} \Big( (\alpha+1) - \frac{C_2}{C_1 - C_2} \Big).$$

Recall that from Eq. (70) we have

$$C_1 = \frac{\pi}{\alpha \sin(\pi/\alpha)}, \quad C_2 = \frac{\pi(\alpha-1)}{\alpha^2 \sin(\pi/\alpha)},$$

thus

$$(\alpha+1) - \frac{C_2}{C_1 - C_2} = (\alpha+1) - \frac{\frac{\pi(\alpha-1)}{\alpha^2 \sin(\pi/\alpha)}}{\frac{\pi}{\alpha \sin(\pi/\alpha)} - \frac{\pi(\alpha-1)}{\alpha^2 \sin(\pi/\alpha)}} = 2.$$

Finally, we have

$$\lim_{p \to n} \left( B_{R,0}^{RFM} - f(B_{N,0}^{RFM}) \right) = 2C_1^{2\alpha r} C_3 n^{-2\alpha r} = 2C_3 \left( \frac{n}{C_1} \right)^{-2\alpha r},$$

and then the relationship between $B_{R,0}^{RFM}$ and $B_{N,0}^{RFM}$ is

$$
\begin{aligned}
B_{R,0}^{RFM} &\approx \left( \frac{n}{C_1} \right)^{-\alpha} B_{N,0}^{RFM} + \left( \frac{n}{C_1} \right)^{-2\alpha r} \frac{C_2 C_3 - C_1 C_4}{C_1 - C_2} + 2C_3 \left( \frac{n}{C_1} \right)^{-2\alpha r} \\
&\approx \left( \frac{n}{C_1} \right)^{-\alpha} B_{N,0}^{RFM} + \left( \frac{n}{C_1} \right)^{-2\alpha r} \frac{2C_1 C_3 - C_2 C_3 - C_1 C_4}{C_1 - C_2}.
\end{aligned}
\tag{76}
$$

**Condition 2:** $r \in [\frac{1}{2}, \infty)$. In this condition, the approximation of $B_{R,0}^{RFM}$ and $B_{N,0}^{RFM}$ can be simplified to

$$B_{R,0}^{RFM} \approx \frac{n}{n-p} C_3 \nu_2^{2r \wedge 1} \approx \frac{n}{n-p} C_3 \left( \frac{p}{C_1} \right)^{-\alpha(2r \wedge 1)} = \frac{n}{n-p} C_3 \left( \frac{p}{C_1} \right)^{-\alpha},$$

$$
\begin{aligned}
B_{N,0}^{RFM} &\approx \left( \frac{C_1 C_4}{C_1 - C_2} + \frac{p}{n-p} C_3 \right) \nu_2^{(2r-1) \wedge 0} \\
&\approx \left( \frac{C_1 C_4}{C_1 - C_2} + \frac{p}{n-p} C_3 \right) \left( \frac{p}{C_1} \right)^{-\alpha[(2r-1) \wedge 0]} \\
&= \frac{C_1 C_4}{C_1 - C_2} + \frac{p}{n-p} C_3.
\end{aligned}
$$

Recall the relationship between $B_{R,0}^{RFM}$ and $B_{N,0}^{RFM}$ in the over-parameterized regime is presented in Eq. (74), given by:

$$B_{R,0}^{RFM} \approx \left( \frac{n}{C_1} \right)^{-\alpha} B_{N,0}^{RFM} - \left( \frac{n}{C_1} \right)^{-\alpha} C_4 =: g(B_{N,0}^{RFM}).$$

Substituting the expression for $B_{N,0}^{RFM}$ in the under-parameterized regime into this relationship, we obtain:

$$g(B_{N,0}^{RFM}) = \left( \frac{n}{C_1} \right)^{-\alpha} \left( \frac{C_1 C_4}{C_1 - C_2} + \frac{p}{n-p} C_3 \right) - \left( \frac{n}{C_1} \right)^{-\alpha} C_4,$$

then we compute $B_{R,0}^{RFM} - g(B_{N,0}^{RFM})$ and obtain

$$B_{R,0}^{RFM} - g(B_{N,0}^{RFM}) = C_3 C_1^\alpha \frac{np^{-\alpha} - pn^{-\alpha}}{n-p} - \left( \frac{n}{C_1} \right)^{-\alpha} \left( \frac{C_2 C_4}{C_1 - C_2} \right).$$

Thus we have

$$
\begin{aligned}
\lim_{p \to n} \left( B_{R,0}^{RFM} - f(B_{N,0}^{RFM}) \right) &= \left( \frac{n}{C_1} \right)^{-\alpha} \left( (\alpha+1)C_3 - \frac{C_2 C_4}{C_1 - C_2} \right) \\
&\approx \left( \frac{n}{C_1} \right)^{-\alpha} \left( (\alpha+1)C_4 - \frac{C_2}{C_1 - C_2} C_4 \right) \\
&= \left( \frac{n}{C_1} \right)^{-\alpha} 2C_4,
\end{aligned}
$$

and the relationship between $B_{R,0}^{RFM}$ and $B_{N,0}^{RFM}$ is

$$
\begin{aligned}
B_{R,0}^{RFM} &\approx \left( \frac{n}{C_1} \right)^{-\alpha} B_{N,0}^{RFM} - \left( \frac{n}{C_1} \right)^{-\alpha} C_4 + \left( \frac{n}{C_1} \right)^{-\alpha} 2C_4 \\
&\approx \left( \frac{n}{C_1} \right)^{-\alpha} B_{N,0}^{RFM} + \left( \frac{n}{C_1} \right)^{-\alpha} C_4.
\end{aligned}
\tag{77}
$$

When $p \ll n$, we discuss cases $r \in (0, \frac{1}{2})$ and $r \in (\frac{1}{2}, \infty)$ separately.

If $r \in (0, \frac{1}{2})$, we have $\frac{n}{n-p} \approx 1$ and $\frac{p}{n-p} \approx 0$, then

$$B_{R,0}^{\text{RFM}} \approx C_3 \nu_2^{2r \wedge 1} \approx C_3 \left( \frac{p}{C_1} \right)^{-\alpha 2r},$$

$$B_{N,0}^{\text{RFM}} \approx \frac{C_1 C_4}{C_1 - C_2} \nu_2^{(2r-1) \wedge 0} \approx \frac{C_1 C_4}{C_1 - C_2} \left( \frac{p}{C_1} \right)^{-\alpha(2r-1)}.$$

Then we eliminate $p$ and obtain

$$B_{R,0}^{\text{RFM}} \approx C_3 \left( \frac{C_1 - C_2}{C_1 C_4} \right)^{2r/(2r-1)} \left( B_{N,0}^{\text{RFM}} \right)^{2r/(2r-1)}.$$

If $2r \geq 1$, we have

$$B_{R,0}^{\text{RFM}} \approx \frac{n}{n-p} C_3 \nu_2 \approx \frac{n}{n-p} C_3 \left( \frac{p}{C_1} \right)^{-\alpha},$$

$$B_{N,0}^{\text{RFM}} \approx \frac{C_1 C_4}{C_1 - C_2} + \frac{p}{n-p} C_3.$$

Then we eliminate $p$ and obtain

$$B_{R,0}^{\text{RFM}} \approx \left( \frac{C_1 C_3 - C_2 C_3 - C_1 C_4}{C_1 - C_2} + B_{N,0}^{\text{RFM}} \right) \left( \frac{n \left( B_{N,0}^{\text{RFM}} - \frac{C_1 C_4}{C_1 - C_2} \right)}{C_1 \left( C_3 + B_{N,0}^{\text{RFM}} - \frac{C_1 C_4}{C_1 - C_2} \right)} \right)^{-\alpha}.$$

From Eqs. (75) to (77), we know that the relationship between $R_0^{\text{RFM}}$ and $N_0^{\text{RFM}}$ in the under-parameterized regime when $p \to n$ can be written as

$$R_0^{\text{RFM}} \approx \left( n/C_\alpha \right)^{-\alpha} N_0^{\text{RFM}} + C_{n,\alpha,r,2}.$$

$\square$

# F   Scaling laws

To derive the scaling laws based on norm-based capacity, we first give the decay rate of the $\ell_2$ norm w.r.t. $n$.

The rate of the deterministic equivalent of the random feature ridge regression estimator's $\ell_2$ norm is given by

$$N_\lambda^{\text{RFM}} = \Theta \left( n^{-\gamma_{B_{N,\lambda}^{\text{RFM}}}} + \sigma^2 n^{-\gamma_{V_{N,\lambda}^{\text{RFM}}}} \right) = \Theta \left( n^{-\gamma_{N_\lambda^{\text{RFM}}}} \right),$$

where $\gamma_{N_\lambda^{\text{RFM}}} := \gamma_{B_{N,\lambda}^{\text{RFM}}} \wedge \gamma_{V_{N,\lambda}^{\text{RFM}}}$ for $\sigma^2 \neq 0$.

## F.1   Variance term

Using Eqs. (42) to (44), we have

$$V_{N,\lambda}^{\text{RFM}} = \sigma^2 \frac{p}{n} \frac{\chi(\nu_2)}{1 - \Upsilon(\nu_1, \nu_2)} = n^{q-1} n^{-q} O \left( \nu_2^{-1-1/\alpha} \right)$$

$$= O \left( n^{-(1-(\alpha+1)(1 \wedge q \wedge \ell/\alpha))} \right).$$

Hence, the variance term of the norm decays with $n$ with rate

$$\gamma_{V_{N,\lambda}^{\text{RFM}}}(\ell, q) = 1 - (\alpha + 1) \left( \frac{\ell}{\alpha} \wedge q \wedge 1 \right).$$

## F.2 Bias term

First, one could notice, using the integral approximation and Eqs. (41) and (42), that

$$\frac{p}{p - \mathrm{df}_2(\nu_2)} = \left(1 + n^{-q}O\left(\nu_2^{-1/\alpha}\right)\right) = \left(1 + O\left(n^{-q}n^{(1 \wedge q \wedge \ell/\alpha)}\right)\right) = O\left(1\right).$$

Thus for the bias term, using Eqs. (41) to (44) we have

$$\begin{aligned}
\mathsf{B}_{\mathsf{N},\lambda}^{\mathrm{RFM}} &= \langle \boldsymbol{\theta}_*, \boldsymbol{\Lambda}(\boldsymbol{\Lambda} + \nu_2)^{-2}\boldsymbol{\theta}_* \rangle \cdot \frac{p}{p - \mathrm{df}_2(\nu_2)} \\
&\quad + \frac{p}{n}\nu_2^2 \left(\langle \boldsymbol{\theta}_*, (\boldsymbol{\Lambda} + \nu_2)^{-2}\boldsymbol{\theta}_* \rangle + \chi(\nu_2)\langle \boldsymbol{\theta}_*, \boldsymbol{\Lambda}(\boldsymbol{\Lambda} + \nu_2)^{-2}\boldsymbol{\theta}_* \rangle\right) \cdot \frac{\chi(\nu_2)}{1 - \Upsilon(\nu_1, \nu_2)} \\
&= T_{2r+1,2}^1(\nu_2) + n^{q-1}\nu_2^2 \left(T_{2r,2}^1(\nu_2) + \chi(\nu_2)T_{2r+1,2}^1(\nu_2)\right)\chi(\nu_2) \\
&= \nu_2^{(2r-1)\wedge 0} + n^{q-1}\nu_2^2 O\left(\nu_2^{(2r-2)\wedge 0} + n^{-q}\nu_2^{-1-1/\alpha+(2r-1)\wedge 0}\right)n^{-q}O\left(\nu_2^{-1-1/\alpha}\right) \\
&= \nu_2^{(2r-1)\wedge 0} + n^{-1}O\left(\nu_2^{2r\wedge 2} + n^{-q}\nu_2^{-1/\alpha+2r\wedge 1}\right)O\left(\nu_2^{-1-1/\alpha}\right) \\
&= O\left(n^{-\alpha(1\wedge q\wedge \ell/\alpha)[(2r-1)\wedge 0]}\right) \\
&\quad + O\left(n^{-\alpha(1\wedge q\wedge \ell/\alpha)[(2r-1)\wedge 1]+(1\wedge q\wedge \ell/\alpha)-1} + n^{-\alpha(1\wedge q\wedge \ell/\alpha)[(2r-1)\wedge 0]+2(1\wedge q\wedge \ell/\alpha)-1-q}\right) \\
&= O\left(n^{-\alpha(1\wedge q\wedge \ell/\alpha)[(2r-1)\wedge 0]} + n^{-\alpha(1\wedge q\wedge \ell/\alpha)[(2r-1)\wedge 1]+(1\wedge q\wedge \ell/\alpha)-1}\right) \\
&= O\left(n^{-\alpha(1\wedge q\wedge \ell/\alpha)[(2r-1)\wedge 0]}\right).
\end{aligned}$$

Hence, the bias term of the norm decays with $n$ with rate

$$\gamma_{\mathsf{B}_{\mathsf{N},\lambda}^{\mathrm{RFM}}}(\ell, q) = \alpha\left(1 \wedge q \wedge \ell/\alpha\right)\left[(2r - 1) \wedge 0\right].$$

Recalling that we have

$$\gamma_{\mathsf{N}_\lambda^{\mathrm{RFM}}} := \gamma_{\mathsf{B}_{\mathsf{N},\lambda}^{\mathrm{RFM}}} \wedge \gamma_{\mathsf{V}_{\mathsf{N},\lambda}^{\mathrm{RFM}}},$$

according to which, we obtain the norm exponent $\gamma_{\mathsf{N}_\lambda^{\mathrm{RFM}}}$ as a function of $\ell$ and $q$, showing in Fig. 7. As observed in Fig. 7, $\gamma_{\mathsf{N}_\lambda^{\mathrm{RFM}}}$ is non-positive across all regions, indicating that the norm either increases or remains constant with $n$ in every case.

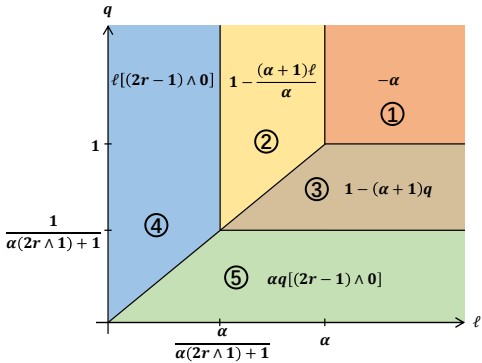

Figure 7: The norm rate $\gamma_{\mathsf{N}_\lambda^{\mathrm{RFM}}}$ as a function of $(\ell, q)$. Variance dominated region is colored by orange, yellow and brown, bias dominated region is colored by blue and green.

Next for the condition $r \in (0, \frac{1}{2})$, we derive the scaling law under norm-based capacity.

**Region 1: $\ell > \alpha$ and $q > 1$**    In this region, according to [14, Corollary 4.1], we have

$$\mathsf{R}_\lambda^{\mathrm{RFM}} = \Theta\left(n^{-0}\right) = \Theta\left(1\right),$$

and according to Fig. 7, we have
$$\mathsf{N}_\lambda^{\mathrm{RFM}} = \Theta\left(n^\alpha\right) ,$$
combing the above rate, we can obtain that
$$\mathsf{R}_\lambda^{\mathrm{RFM}} = \Theta\left(n^{-\alpha} \cdot \mathsf{N}_\lambda^{\mathrm{RFM}}\right) .$$

**Region 2:** $\frac{\alpha}{2\alpha r+1} < \ell < \alpha$ **and** $q > \frac{\ell}{\alpha}$    In this region, according to [14, Corollary 4.1], we have
$$\mathsf{R}_\lambda^{\mathrm{RFM}} = \Theta\left(n^{-\left(1-\frac{\ell}{\alpha}\right)}\right) ,$$
and according to Fig. 7, we have
$$\mathsf{N}_\lambda^{\mathrm{RFM}} = \Theta\left(n^{-\left(1-\frac{(\alpha+1)\ell}{\alpha}\right)}\right) ,$$
combing the above rate, we can obtain that
$$\mathsf{R}_\lambda^{\mathrm{RFM}} = \Theta\left(n^{-\ell} \cdot \mathsf{N}_\lambda^{\mathrm{RFM}}\right) .$$

**Region 3:** $\frac{1}{2\alpha r+1} < q < 1$ **and** $q < \frac{\ell}{\alpha}$    In this region, according to [14, Corollary 4.1], we have
$$\mathsf{R}_\lambda^{\mathrm{RFM}} = \Theta\left(n^{-(1-q)}\right) ,$$
and according to Fig. 7, we have
$$\mathsf{N}_\lambda^{\mathrm{RFM}} = \Theta\left(n^{-(1-(\alpha+1)q)}\right) ,$$
combing the above rate and eliminate $q$, we can obtain that
$$\mathsf{R}_\lambda^{\mathrm{RFM}} = \Theta\left(n^{-\frac{\alpha}{\alpha+1}} \cdot \left(\mathsf{N}_\lambda^{\mathrm{RFM}}\right)^{\frac{1}{\alpha+1}}\right) .$$

**Region 4:** $\ell < \frac{\alpha}{2\alpha r+1}$ **and** $q > \frac{\ell}{\alpha}$    In this region, according to [14, Corollary 4.1], we have
$$\mathsf{R}_\lambda^{\mathrm{RFM}} = \Theta\left(n^{-2\ell r}\right) ,$$
and according to Fig. 7, we have
$$\mathsf{N}_\lambda^{\mathrm{RFM}} = \Theta\left(n^{-\ell(2r-1)}\right) ,$$
combing the above rate, we can obtain that
$$\mathsf{R}_\lambda^{\mathrm{RFM}} = \Theta\left(n^{-1} \cdot \mathsf{N}_\lambda^{\mathrm{RFM}}\right) .$$

**Region 5:** $q < \frac{1}{2\alpha r+1}$ **and** $q < \frac{\ell}{\alpha}$    In this region, according to [14, Corollary 4.1], we have
$$\mathsf{R}_\lambda^{\mathrm{RFM}} = \Theta\left(n^{-2\alpha q r}\right) ,$$
and according to Fig. 7, we have
$$\mathsf{N}_\lambda^{\mathrm{RFM}} = \Theta\left(n^{-\alpha q(2r-1)}\right) ,$$
combing the above rate, we can obtain that
$$\mathsf{R}_\lambda^{\mathrm{RFM}} = \Theta\left(n^0 \cdot \left(\mathsf{N}_\lambda^{\mathrm{RFM}}\right)^{-\frac{2r}{1-2r}}\right) .$$

## G   Discussion

In this section, we discuss several issues related to the shape of generalization curves, norm control, and model complexity. In Appendix G.1, we examine the shape of generalization curves under various settings, emphasizing when theoretical predictions align with or diverge from empirical observations, particularly across synthetic and real-world datasets. In Appendix G.2, we analyze a practical approach to modifying norm by fixing the parameter count and imposing a norm constraint, and demonstrate its equivalence to adjusting the regularization strength. Finally, in Appendix G.3, we compare norm-based capacity with alternative complexity measures, including smoother-based metrics and degrees of freedom, and highlight their limitations in capturing test risk behavior.

Table 4: Generalization curves (test loss *vs.* $\ell_2$ norm) under different activation functions in RFMs. Training data $\{(\boldsymbol{x}_i, y_i)\}_{i \in [n]}$ are generated from a teacher-student model $y_i = \tanh(\langle \boldsymbol{\beta}, \boldsymbol{x}_i \rangle)$, where $\boldsymbol{x}_i \sim$ i.i.d. $\mathcal{N}(\boldsymbol{0}, \boldsymbol{I}_d)$ with $d = 100$. The number of training samples is fixed at $n = 300$. The random feature map is defined as $\varphi(\boldsymbol{x}, \boldsymbol{w}) = \varphi(\langle \boldsymbol{w}, \boldsymbol{x} \rangle)$ with random Gaussian initialization $\boldsymbol{w} \sim \mathcal{N}(\boldsymbol{0}, \boldsymbol{I}_d)$, where the activation function $\varphi(\cdot)$ is chosen from `ReLU`, `erf`, `tanh`, or `sigmoid`.

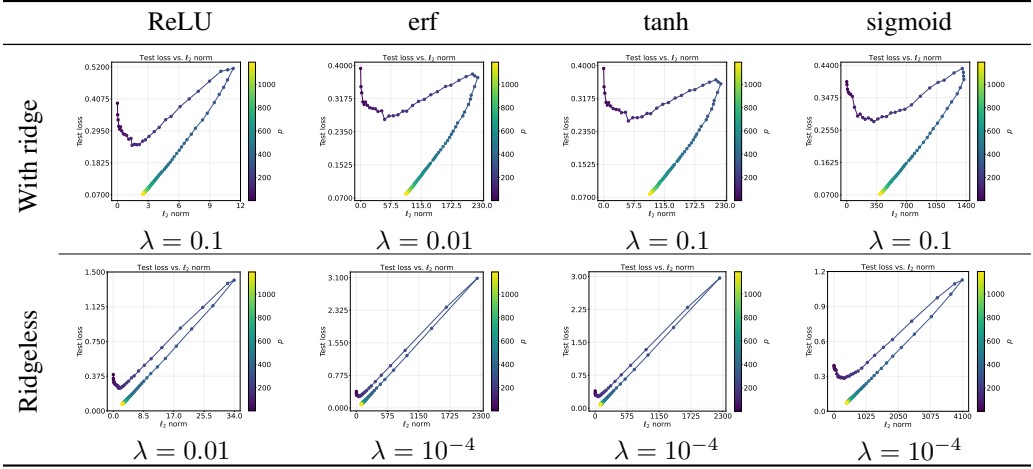

## G.1 Discussion on the shape of the generalization curve in Fig. 1

As illustrated in Fig. 1(a), and based on empirical observations from [43, Figure 8.12], the test risk in the over-parameterized regime initially exceeds that of the under-parameterized regime. However, as over-parameterization increases, the test risk begins to decrease. Eventually, in a sufficiently over-parameterized regime, the test risk becomes lower than in the under-parameterized case—indicating that sufficient over-parameterization can outperform under-parameterization.

In contrast, our experimental results in Fig. 1(b) reveal a slightly different behavior: the learning curve in the over-parameterized regime consistently remains below its under-parameterized counterpart throughout. This phenomenon presents an intriguing contrast, and the central question we address in this section is: What underlying factors cause this fundamental difference in behavior - where in some cases the over-parameterized curve initially above then crosses the under-parameterized curve, while in others it stays strictly lower?

We first conduct experiments on synthetic datasets to validate our theoretical findings. We generate training samples $\{(\boldsymbol{x}_i, y_i)\}_{i \in [n]}$ using a teacher-student model: $y_i = \tanh(\langle \boldsymbol{\beta}, \boldsymbol{x}_i \rangle)$, where input features $\boldsymbol{x}_i \overset{i.i.d}{\sim} \mathcal{N}(\boldsymbol{0}, \boldsymbol{I}_d)$ with dimension $d = 100$. As demonstrated in Table 4, our experimental results reveal that when the input features follow Gaussian distribution, the test loss curves in the over-parameterized regime consistently lie below those in the under-parameterized regime, regardless of the activation functions or ridge parameter values. This observation aligns perfectly with our theoretical predictions.

We further conducted experiments on the FashionMNIST data set [58]. In this practical setting, we observed a discrepancy between our experimental results and theoretical predictions.

As in Table 5, for cases with substantial ridge regularization, the test error curves in the over-parameterized regime remained below those in the under-parameterized regime, consistent with our synthetic data experiments. However, in the ridgeless case (corresponding to minimum-$\ell_2$-norm interpolators), we discovered a different phenomenon:

- Initially, the over-parameterized regime exhibited higher test error than the under-parameterized regime.

- As the number of parameters $p$ increased, the over-parameterized curve crossed under the under-parameterized curve. And this interaction formed a distinctive $\varphi$-shaped learning curve.

We attribute this behavior to the non-Gaussian distribution of input images $\boldsymbol{x}$ in FashionMNIST, which may violates our theoretical assumption like Assumption 1.

Table 5: Generalization curves (test error *vs.* $\ell_2$ norm) under different activation functions. Training data $\{(\boldsymbol{x}_i, y_i)\}_{i \in [n]}$ are sampled from the **FashionMNIST** data set [58], with input vectors normalized and flattened to $[-1, 1]^d$ for $d = 748$. The random feature map is defined as $\varphi(\boldsymbol{x}, \boldsymbol{w}) = \varphi(\langle \boldsymbol{w}, \boldsymbol{x} \rangle)$ with random Gaussian initialization $\boldsymbol{w} \sim \mathcal{N}(\boldsymbol{0}, \boldsymbol{I}_d)$, where the activation function $\varphi(\cdot)$ is chosen from ReLU, erf, tanh, or sigmoid. The number of training samples is fixed at $n = 300$.

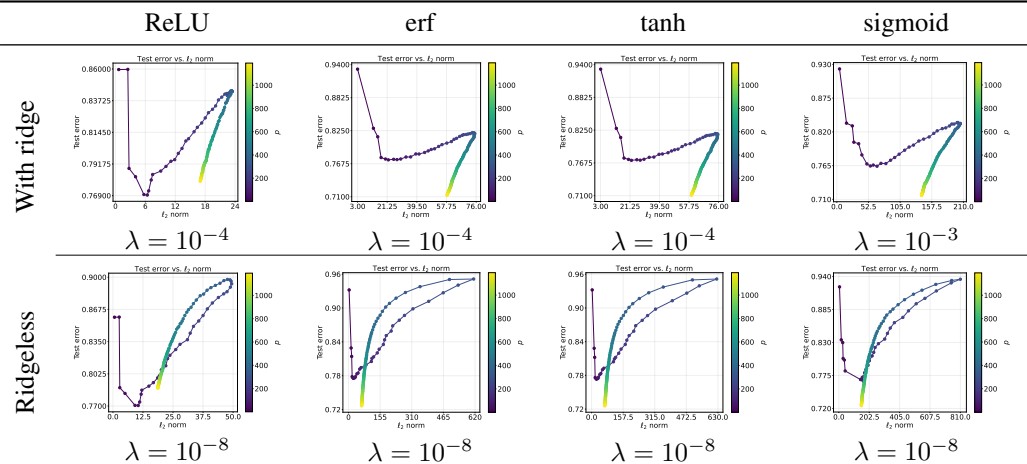

In summary, while our theoretical framework may not fully capture the generalization behavior when the dataset or activation functions deviate significantly from our assumptions, this does not undermine the core contributions of our work. When the data is well-behaved and aligns with our assumptions, our theory provides a highly accurate and effective characterization of the generalization curves under norm-based capacity control in the under-parameterized regime.

## G.2 Discussion on approaches to modifying the norm

Regarding approaches to controlling model norm, one method involves fixing the regularization strength while varying the model parameter count $p$, which serves as the primary focus of this paper. Alternatively, one can fix $p$ and constrain the weight norm to specific magnitudes. We later show that this approach is mathematically equivalent to fixing the parameter count while varying the regularization strength. In this section, we primarily focus on the latter approach.

We consider the problem of minimizing the squared loss under an $\ell_2$-norm constraint on the coefficients:

$$\min_{\boldsymbol{a}} \|\boldsymbol{y} - \boldsymbol{Z}\boldsymbol{a}\|^2 \quad \text{subject to} \quad \|\boldsymbol{a}\|_2^2 = B^2.$$

To incorporate the constraint, we introduce a Lagrange multiplier $\lambda$ and define the Lagrangian:

$$\mathcal{L}(\boldsymbol{a}, \lambda) = \|\boldsymbol{y} - \boldsymbol{Z}\boldsymbol{a}\|^2 + \lambda \left( \|\boldsymbol{a}\|_2^2 - B^2 \right).$$

Taking the gradient of $\mathcal{L}$ with respect to $\boldsymbol{a}$ and setting it to zero yields the first-order optimality condition:

$$\nabla_{\boldsymbol{a}} \mathcal{L} = -2\boldsymbol{Z}^\top \boldsymbol{y} + 2\boldsymbol{Z}^\top \boldsymbol{Z}\boldsymbol{a} + 2\lambda\boldsymbol{a} = \boldsymbol{0}.$$

Solving this equation gives the solution:

$$\hat{\boldsymbol{a}} = (\boldsymbol{Z}^\top \boldsymbol{Z} + \lambda I)^{-1} \boldsymbol{Z}^\top \boldsymbol{y}, \quad \text{subject to} \quad \|\hat{\boldsymbol{a}}\|_2^2 = B^2.$$

**Relation to Ridge Regression:** The solution resembles ridge regression, but $\lambda$ is chosen to strictly satisfy $\|\hat{\boldsymbol{a}}\|_2 = B$ rather than being a hyperparameter. $\lambda$ corresponds one-to-one with $B$, since $\lambda$ and $\|\hat{\boldsymbol{a}}\|_2^2$ are in one-to-one correspondence if $\lambda \geq 0$ ($\frac{\partial \|\hat{\boldsymbol{a}}\|_2^2}{\partial \lambda} = -2\boldsymbol{y}^\top \boldsymbol{Z}(\boldsymbol{Z}^\top \boldsymbol{Z} + \lambda\boldsymbol{I})^{-3}\boldsymbol{Z}^\top \boldsymbol{y} \leq 0$).

Therefore, we can say that changing the constraint $B$ (the restriction on $\|\boldsymbol{a}\|_2$) is equivalent to changing the regularization strength $\lambda$.

We conducted experiments on the random feature model by fixing the number of training samples and the aspect ratio $\gamma$, and varying the regularization parameter $\lambda$ to control the norm of the estimator. We then plotted the curves showing the relationships among test risk, norm, and $\lambda$ as in Fig. 8 (Figs. 8(a) to 8(c) for under-parameterized regimes and Figs. 8(d) to 8(f) for over-parameterized regimes). We can find that the norm is monotonically decreasing with the increasing $\lambda$, see Figs. 8(b) and 8(e). In fact, the relationship between the estimator's norm and the regularization parameter is called L-curve [21]. in both under- and over-parameterized ($\lambda < 1$ or $\lambda > 1$), the test risk is always a U-shaped curve of the regularization parameter $\lambda$ or norm, see Figs. 8(a) and 8(c) and Figs. 8(d) and 8(f), respectively.

To validate these observations on real data, we also conducted complementary experiments using the MNIST data set [30]. As shown in Fig. 9, all of the above phenomena persist.

Moreover, in modern ML practice, capacity can be steered by standard regularization—e.g., weight decay and early stopping—which explicitly or implicitly constrain model norms. Optimization itself also induces implicit regularization, notably with SGD. Recent work has begun to precisely characterize test risk under SGD for linear models [45, 44]. Extending our deterministic-equivalent framework to incorporate such optimization effects is both important and challenging, and we leave this for future work.

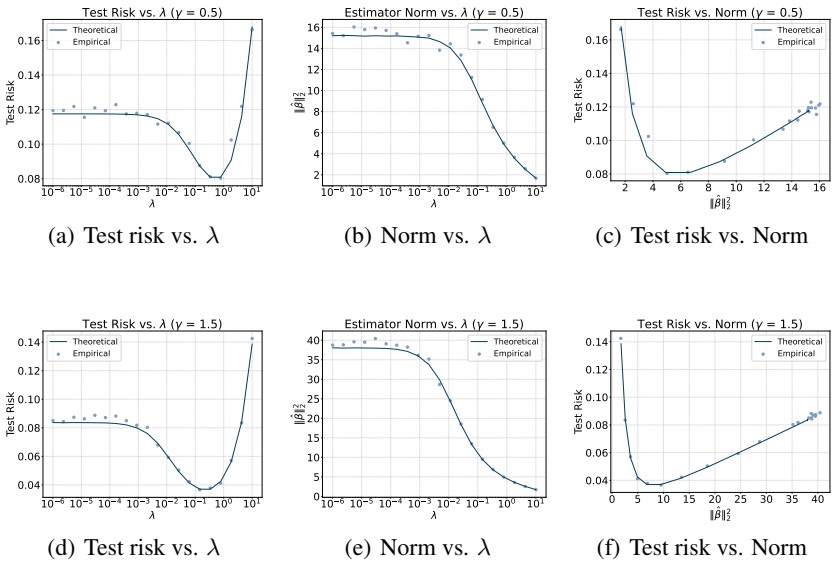

(a) Test risk vs. $\lambda$      (b) Norm vs. $\lambda$      (c) Test risk vs. Norm

(d) Test risk vs. $\lambda$      (e) Norm vs. $\lambda$      (f) Test risk vs. Norm

Figure 8: Relationship between test risk, $\ell_2$ norm, and $\lambda$ for different $\gamma = \frac{p}{n}$ for random feature ridge regression. Points in these figures are given by our experimental results, centering around the curves given by deterministic equivalents we derive. Training data $\{(\boldsymbol{x}_i, y_i)\}_{i \in [n]}$, $n = 100$, sampled from the model $y_i = \boldsymbol{g}_i^\top \boldsymbol{\theta}_* + \varepsilon_i$, $\sigma^2 = 0.01$, $\boldsymbol{g}_i \sim \mathcal{N}(0, \boldsymbol{I})$, $\boldsymbol{f}_i \sim \mathcal{N}(0, \boldsymbol{\Lambda})$ ($\boldsymbol{g}_i$ and $\boldsymbol{f}_i$ is defined in Section 2), with $\xi_k^2(\boldsymbol{\Lambda}) = k^{-3/2}$ and $\boldsymbol{\theta}_{*,k} = k^{-11/10}$, given by $\alpha = 1.5$, $r = 0.4$ in Assumption 2.

### G.3 Discussion with other model capacities

In this section, we discuss two other model capacities: *generalized effective number of parameters* and *degrees of freedom*, which are widely used to describe a model's generalization ability. From this discussion, we conclude that these two capacities are less suitable compared to norm-based model capacity.

**Generalized Effective Number of Parameters:** The authors [12] assess model complexity from the perspective of smoother by introducing a variance-based effective-parameter measure, termed the **generalized effective number of parameters**. In the context of ridge regression, this measure is

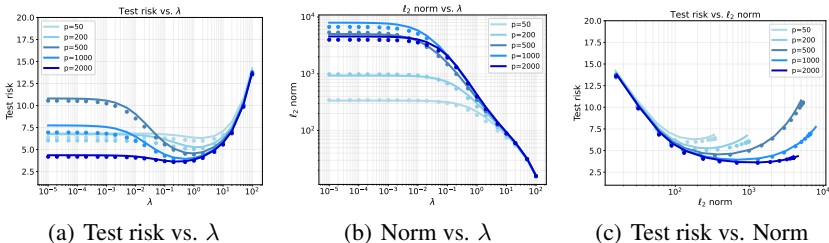

(a) Test risk vs. $\lambda$  (b) Norm vs. $\lambda$  (c) Test risk vs. Norm

Figure 9: Relationship between test risk, $\ell_2$ norm, and $\lambda$ for different $\gamma = \frac{p}{n}$ for random feature ridge regression. Points in these figures are given by our experimental results, centering around the curves given by deterministic equivalents we derive. Training data $\{(\boldsymbol{x}_i, y_i)\}_{i \in [n]}$, $n = 300$, sub-sampled from the **MNIST** data set [30], with feature map given by $\varphi(\boldsymbol{x}, \boldsymbol{w}) = \mathrm{erf}(\langle \boldsymbol{x}, \boldsymbol{w} \rangle)$ and $\boldsymbol{w} \sim \mathcal{N}(0, \boldsymbol{I})$.

given by

$$p_{\hat{\boldsymbol{s}}}^{\text{test}} = \frac{n}{|\mathcal{I}_{\text{test}}|} \sum_{j \in \mathcal{I}_{\text{test}}} \|\boldsymbol{x}_j^{\text{test}} (\boldsymbol{X}^\top \boldsymbol{X} + \lambda)^{-1} \boldsymbol{X}^\top\|_2^2 \,,$$

where $\{\boldsymbol{x}_j^{\text{test}}\}_{j \in \mathcal{I}_{\text{test}}}$ is the set of test inputs. Taking the expectation with respect to the test set yields

$$p_{\hat{\boldsymbol{s}}}^{\text{test}} = n \mathbb{E}_{\boldsymbol{x}_j^{\text{test}}} \|\boldsymbol{x}_j^{\text{test}} (\boldsymbol{X}^\top \boldsymbol{X} + \lambda)^{-1} \boldsymbol{X}^\top\|_2^2 = n \mathrm{Tr}(\boldsymbol{\Sigma} \boldsymbol{X}^\top \boldsymbol{X} (\boldsymbol{X}^\top \boldsymbol{X} + \lambda)^{-2}) \,,$$

which corresponds to the variance of the test risk $\mathcal{V}_{\mathcal{R}}^{\text{LS}}$ scaled by the factor $\frac{n}{\sigma^2}$.

For the random feature ridge regression, the generalized effective number of parameters can be similarly given by

$$p_{\hat{\boldsymbol{s}}}^{\text{test}} = n \mathbb{E}_{\boldsymbol{z}_j^{\text{test}}} \|\boldsymbol{z}_j^{\text{test}} (\boldsymbol{Z}^\top \boldsymbol{Z} + \lambda)^{-1} \boldsymbol{Z}^\top\|_2^2 = n \mathrm{Tr}(\widehat{\boldsymbol{\Lambda}}_{\boldsymbol{F}} \boldsymbol{Z}^\top \boldsymbol{Z} (\boldsymbol{Z}^\top \boldsymbol{Z} + \lambda)^{-2}) \,,$$

which corresponds to the variance of the test risk $\mathcal{V}_{\mathcal{R}}^{\text{RFM}}$ scaled by the factor $\frac{n}{\sigma^2}$.

The connection between variance and $p_{\hat{\boldsymbol{s}}}^{\text{test}}$ enables it to effectively capture the variance of test risk. However, due to the lack of information about the target function (without label information $y$), this model capacity cannot fully describe the behavior of test risk, as it neglects the bias component. This limitation becomes apparent when the test risk is dominated by bias.

**Degrees of freedom** For linear ridge regression, another measure of model capacity, known as the "degrees of freedom" [9, 23, 1], is defined as

$$\mathrm{df}_1(\lambda_*) := \mathrm{Tr}(\boldsymbol{\Sigma}(\boldsymbol{\Sigma} + \lambda_*)^{-1}) \,, \quad \mathrm{df}_2(\lambda_*) := \mathrm{Tr}(\boldsymbol{\Sigma}^2(\boldsymbol{\Sigma} + \lambda_*)^{-2}) \,.$$

$\mathrm{df}_1(\lambda_*)$ and $\mathrm{df}_2(\lambda_*)$ measures the number of "effective" parameters the model can fit. As the regularization strength $\lambda$ increases, model complexity decreases. From Definition B.8, we have $n - \frac{\lambda}{\lambda_*} = \mathrm{Tr}(\boldsymbol{\Sigma}(\boldsymbol{\Sigma} + \lambda_*)^{-1})$, implying that an increase in $\lambda$ raises $\lambda_*$, leading to a reduction in $\mathrm{df}_1(\lambda_*)$ and $\mathrm{df}_2(\lambda_*)$. This suggests that degrees of freedom can, to some extent, represent model complexity.

However, it is worth noting that since in linear ridge regression we only vary the number of training data $n$, according to the self-consistent equation $n - \frac{\lambda}{\lambda_*} = \mathrm{Tr}(\boldsymbol{\Sigma}(\boldsymbol{\Sigma} + \lambda_*)^{-1})$ we can tell that $\lambda_*$ decreases monotonically as $n$ increases, which leads to $\mathrm{df}_1$ and $\mathrm{df}_2$ increasing monotonically as $n$ increases. This monotonic relationship with $n$ suggests that when using degrees of freedom as a measure of model capacity, the double descent phenomenon still exists, as the effective capacity of the model continues to increase even beyond the interpolation threshold.

Similar to the generalized effective number of parameters mentioned above, these degrees of freedom also lack information about the target function, making them insufficient for accurately capturing the model's generalization ability.

Fig. 10 illustrates the relationship between test risk and different model capacity for linear ridge regression. It shows that double descent persists for degrees of freedom $\mathrm{df}_1$ and $\mathrm{df}_2$, indicating that degrees of freedom is not an appropriate measure of model capacity.

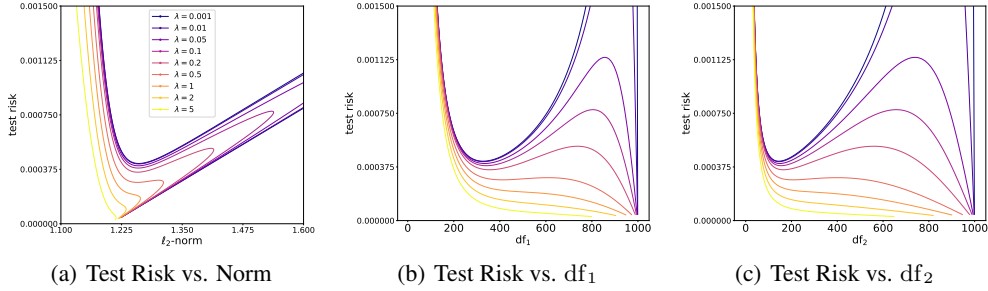

(a) Test Risk vs. Norm  (b) Test Risk vs. $\mathrm{df}_1$  (c) Test Risk vs. $\mathrm{df}_2$

Figure 10: Relationship between test risk and different model capacities. Training data $\{(\boldsymbol{x}_i, y_i)\}_{i \in [n]}$, $d = 1000$, sampled from a linear model $y_i = \boldsymbol{x}_i^\top \boldsymbol{\beta}_* + \varepsilon_i$, $\sigma^2 = 0.0004$, $\boldsymbol{x}_i \sim \mathcal{N}(0, \boldsymbol{\Sigma})$, with $\sigma_k(\boldsymbol{\Sigma}) = k^{-1}$, $\boldsymbol{\beta}_{*,k} = k^{-3/2}$.

# H  Experiment

To systematically validate our theoretical findings, we conduct a comprehensive empirical study across three distinct settings: (1) synthetic datasets (Appendix H.1), (2) real-world datasets (MNIST[30]/FashionMNIST[58]) with random features (Appendix H.2), and (3) two-layer neural networks with various norm-based capacity measures (Appendix H.3). All experiments can be conducted on a standard laptops with 16 GB memory.

## H.1  Experiment on synthetic dataset

To validate our theoretical framework, we conduct comprehensive experiments on synthetic datasets on linear regression in Fig. 11 and RFMs in Fig. 12, respectively. The strong agreement between theoretical predictions and empirical results confirms the accuracy of our theoretical analysis.

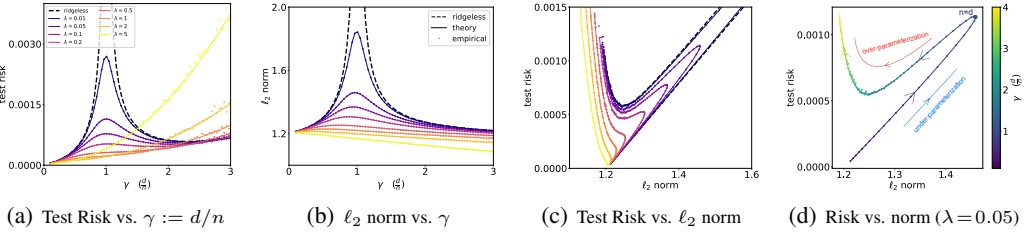

(a) Test Risk vs. $\gamma := d/n$  (b) $\ell_2$ norm vs. $\gamma$  (c) Test Risk vs. $\ell_2$ norm  (d) Risk vs. norm ($\lambda = 0.05$)

Figure 11: Results for the ridge regression estimator. Points in these four figures are given by our experimental results, and the curves are given by our theoretical results via deterministic equivalents. Training data $\{(\boldsymbol{x}_i, y_i)\}_{i \in [n]}$, $d = 1000$, sampled from a linear model $y_i = \boldsymbol{x}_i^\top \boldsymbol{\beta}_* + \varepsilon_i$, $\sigma^2 = 0.0004$, $\boldsymbol{x}_i \sim \mathcal{N}(0, \boldsymbol{\Sigma})$, with $\sigma_k(\boldsymbol{\Sigma}) = k^{-1}$, $\boldsymbol{\beta}_{*,k} = k^{-3/2}$.

## H.2  Experiment on real-world dataset

To complement the synthetic experiments presented in Appendix H.1, we additionally conducted experiments on the **MNIST** (Fig. 13) and **FashionMNIST** (Fig. 14) datasets [30, 58]. We applied the empirical diagonalization procedure introduced in [14, Algorithm 1] to estimate the key quantities $\boldsymbol{\Lambda}$ and $\boldsymbol{\theta}_*$ required for our analysis. The results on these real-world datasets are largely consistent with those observed on the synthetic data: in the under-parameterized regime, the curve of test risk versus norm exhibits a U-shape, while in the over-parameterized regime, the test risk increases monotonically with the norm and is approximately linear for ridge-less regression.

Notably, our random features model can be interpreted as a two-layer neural network with fixed first-layer weights $\boldsymbol{W}$, where the random features $\varphi(\boldsymbol{x}, \boldsymbol{w}_i)$ correspond to the hidden layer activations. This connection motivates our investigation of the Frobenius norm $\|\boldsymbol{W}\|_{\mathrm{F}}$ in Fig. 15, which captures the effective capacity of the frozen hidden layer. Furthermore, Fig. 16 examines the path norm—a

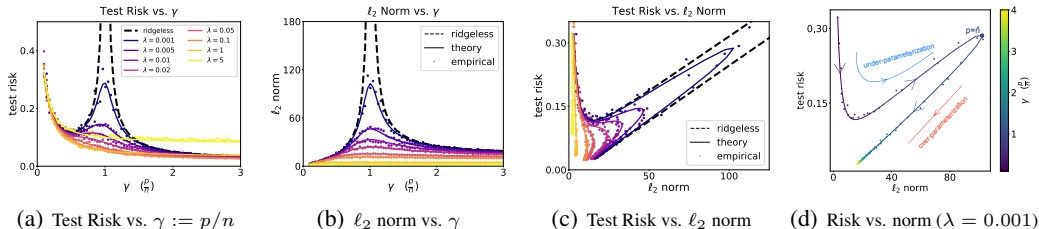

(a) Test Risk vs. $\gamma := p/n$     (b) $\ell_2$ norm vs. $\gamma$     (c) Test Risk vs. $\ell_2$ norm     (d) Risk vs. norm ($\lambda = 0.001$)

Figure 12: Relationship between test risk, ratio $\gamma := p/n$, and $\ell_2$ norm of the random feature ridge regression estimator (the regularization parameter is defined in Section 2). Points in these four figures are given by our experimental results, centering around the curves given by deterministic equivalents we derive. Training data $\{(\boldsymbol{x}_i, y_i)\}_{i \in [n]}$, $n = 100$, sampled from the model $y_i = \boldsymbol{g}_i^\top \boldsymbol{\theta}_* + \varepsilon_i$, $\sigma^2 = 0.01$, $\boldsymbol{g}_i \sim \mathcal{N}(0, \boldsymbol{I})$, $\boldsymbol{f}_i \sim \mathcal{N}(0, \boldsymbol{\Lambda})$ ($\boldsymbol{g}_i$ and $\boldsymbol{f}_i$ is defined in Section 2), with $\xi_k^2(\boldsymbol{\Lambda}) = k^{-3/2}$ and $\boldsymbol{\theta}_{*,k} = k^{-11/10}$, given by $\alpha = 1.5$, $r = 0.4$ in Assumption 2.

natural complexity measure for neural networks that sums over all input-output paths and is defined as

$$\mu_{\text{path-norm}} = \sum_{j=1}^{p} a_j^2 \|\boldsymbol{w}_j\|_2^2 \,.$$

This quantity can be interpreted as the product of the norms of the first-layer and second-layer weights. Prior empirical work by [26] demonstrates that among various norm-based complexity measures, the path norm shows the strongest correlation with generalization performance in neural networks. Motivated by this finding, we investigate the relationship between test risk and path norm in our setting.

Comparing Fig. 13, Fig. 15, and Fig. 16, we observe that the test risk curve aligns more closely with the norm-based capacity of the second-layer parameters in the random feature model, rather than with that of the first-layer weights. Therefore, it is meaningful to study the relationship between the test risk and the norm of the RFM estimator, i.e., the second-layer parameters, as this quantity plays a central role in determining the model's effective capacity and generalization behavior.

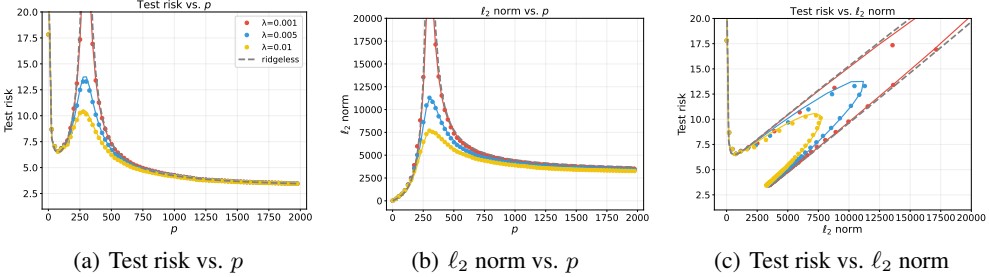

(a) Test risk vs. $p$     (b) $\ell_2$ norm vs. $p$     (c) Test risk vs. $\ell_2$ norm

Figure 13: The relationship between test risk, $\ell_2$ norm and the number of features $p$. Solid lines are obtained from the deterministic equivalent, and points are numerical simulations, with the different curves denoting different regularization strengths. Training data $\{(\boldsymbol{x}_i, y_i)\}_{i \in [n]}$, $n = 300$, sub-sampled from the **MNIST** data set [30], with feature map given by $\varphi(\boldsymbol{x}, \boldsymbol{w}) = \text{erf}(\langle \boldsymbol{x}, \boldsymbol{w} \rangle)$ and $\boldsymbol{w} \sim \mathcal{N}(0, \boldsymbol{I}/d)$, where $d = 748$.

### H.3   Norm-based capacity in two-layer neural networks

In this section, we investigate the relationship between test loss and different norm-based capacities for two-layer fully connected neural networks. Specifically, we evaluate four norm-based capacities: **Frobenius norm**, **Frobenius distance**, **spectral complexity**, and **path norm**. Our empirical results indicate that the path norm is the most suitable model capacity among these three norm-based capacities, which coincides with [26].

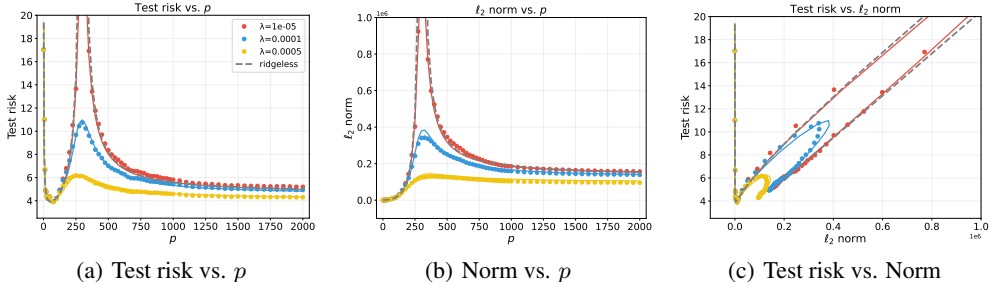

(a) Test risk vs. $p$          (b) Norm vs. $p$          (c) Test risk vs. Norm

Figure 14: The relationship between test risk, $\ell_2$ norm and the number of features $p$. Solid lines are obtained from the deterministic equivalent, and points are numerical simulations, with the different curves denoting different regularization strengths. Training data $\{(\boldsymbol{x}_i, y_i)\}_{i \in [n]}$, $n = 300$, sub-sampled from the **FashionMNIST** data set [58], with feature map given by $\varphi(\boldsymbol{x}, \boldsymbol{w}) = \mathrm{erf}(\langle \boldsymbol{x}, \boldsymbol{w} \rangle)$ and $\boldsymbol{w} \sim \mathcal{N}(0, \boldsymbol{I}/d)$, where $d = 748$.

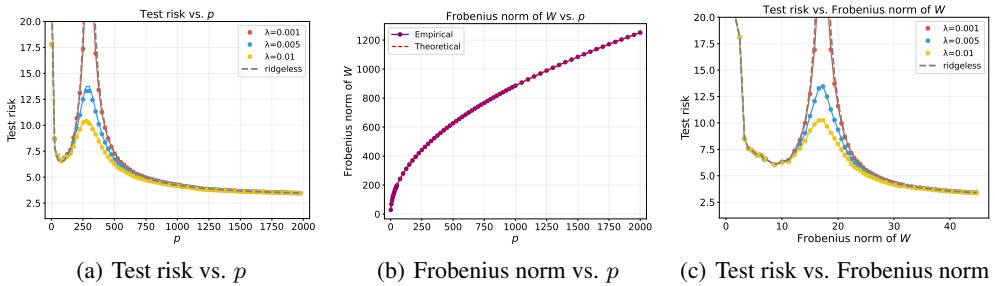

(a) Test risk vs. $p$          (b) Frobenius norm vs. $p$          (c) Test risk vs. Frobenius norm

Figure 15: The relationship between test risk, Frobenius norm of $\boldsymbol{W}$ (the weights in the **hidden layer**) and the number of features $p$. Training data $\{(\boldsymbol{x}_i, y_i)\}_{i \in [n]}$, $n = 300$, sub-sampled from the MNIST data set [30], with feature map given by $\varphi(\boldsymbol{x}, \boldsymbol{w}) = \mathrm{erf}(\langle \boldsymbol{x}, \boldsymbol{w} \rangle)$ and $\boldsymbol{w} \sim \mathcal{N}(0, \boldsymbol{I}/d)$, where $d = 748$.

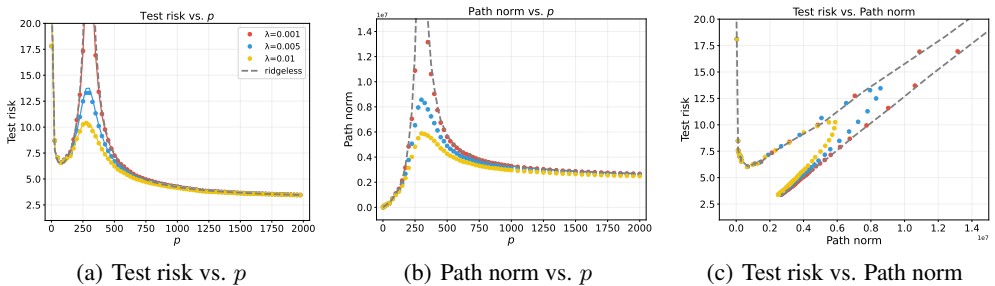

(a) Test risk vs. $p$          (b) Path norm vs. $p$          (c) Test risk vs. Path norm

Figure 16: The relationship between test risk, Path norm and the number of features $p$. Training data $\{(\boldsymbol{x}_i, y_i)\}_{i \in [n]}$, $n = 300$, sub-sampled from the MNIST data set [30], with feature map given by $\varphi(\boldsymbol{x}, \boldsymbol{w}) = \mathrm{erf}(\langle \boldsymbol{x}, \boldsymbol{w} \rangle)$ and $\boldsymbol{w} \sim \mathcal{N}(0, \boldsymbol{I}/d)$, where $d = 748$.

In our experiments, we use a balanced subset of the MNIST data set [30], consisting of 4,000 training samples from all the 10 classes. To simulate real-world noisy data, a noise level $\eta$ is introduced, meaning $\eta \cdot 100\%$ of the training labels are randomly corrupted.

The model is chosen as a two-layer fully connected neural network with parameters including a bias term. The network is initialized using the Xavier initialization scheme and trained using the Stochastic Gradient Descent (SGD) optimizer with a learning rate of 0.1 and momentum of 0.95 over 2,000 epochs. During training, a batch size of 128 is used.

To control model complexity, we vary the number of neurons in the hidden layer, thereby adjusting the number of model parameters. To ensure the robustness of the results, each experiment is repeated 10 times for each hidden layer dimension. The model's performance is evaluated using the Mean Squared Error (MSE) loss on both the training and test sets.

**Frobenius norm:** The parameter Frobenius norm is defined as for such two-layer neural networks

$$\mu_{\text{fro}}(f_{\boldsymbol{w}}) = \sum_{j=1}^{2} \|\boldsymbol{W}_j\|_{\text{F}}^2 \,,$$

where $\boldsymbol{W}_j$ is the parameter matrix of layer $j$.

Fig. 17 illustrates the relationship between test loss, Frobenius norm $\mu_{\text{fro}}$, and the number of parameters $p$. As the number of model parameters increases, the test loss exhibits the typical double descent phenomenon. However, the Frobenius norm consistently increases monotonically (with a slowdown in the growth rate in the over-parameterized regime). Consequently, when using the Frobenius norm as a measure of model capacity, the double descent phenomenon remains observable.

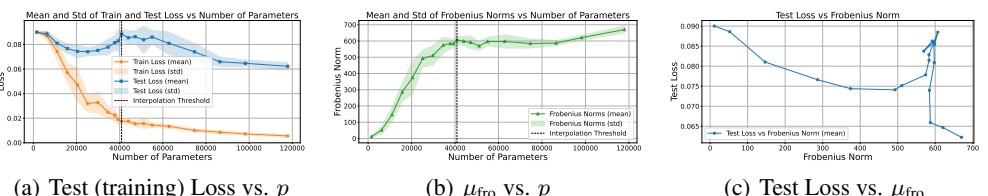

(a) Test (training) Loss vs. $p$      (b) $\mu_{\text{fro}}$ vs. $p$      (c) Test Loss vs. $\mu_{\text{fro}}$

Figure 17: Experiments on two-layer fully connected neural networks with noise level $\eta = 0.2$. The **left** figure shows the relationship between test (training) loss and the number of the parameters $p$. The **middle** figure shows the relationship between the Frobenius norm $\mu_{\text{fro}}$ and $p$. The **right** figure shows the relationship between the test loss and $\mu_{\text{fro}}$.

**Frobenius distance:** The Frobenius distance is defined as for such two-layer neural networks

$$\mu_{\text{fro-dis}}(f_{\boldsymbol{w}}) = \sum_{j=1}^{2} \|\boldsymbol{W}_j - \boldsymbol{W}_j^0\|_{\text{F}}^2 \,,$$

where $\boldsymbol{W}_j^0$ is the initialization of $\boldsymbol{W}_j^0$.

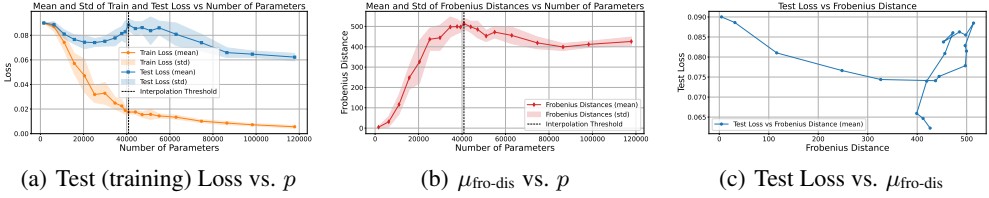

(a) Test (training) Loss vs. $p$      (b) $\mu_{\text{fro-dis}}$ vs. $p$      (c) Test Loss vs. $\mu_{\text{fro-dis}}$

Figure 18: Experiments on two-layer fully connected neural networks with noise level $\eta = 0.2$. The **left** figure is the same as Fig. 17(a). The **middle** figure shows the relationship between the Frobenius distance $\mu_{\text{fro-dis}}$ and $p$. The **right** figure shows the relationship between the test loss and $\mu_{\text{fro-dis}}$ .

Fig. 18 illustrates the relationship between test loss, Frobenius distance $\mu_{\text{fro}}$, and the number of parameters $p$. Different from Frobenius norm, Frobenius distance monotonically increases in the under-parameterized regime, but shows a decrease in the over-parameterized regime. However, since the change of Frobenius distance in the over-parameterized regime is gentle and even eventually appears to rise, using Frobenius distance as the model capacity does not reflect the generalization capacity of the model.

**Spectral complexity:** The spectral complexity is defined as for such two-layer neural networks

$$\mu_{\text{spec}}(f_{\boldsymbol{w}}) = \left(\prod_{i=1}^{2} \|\boldsymbol{W}_i\|\right) \left(\sum_{i=1}^{2} \frac{\|\boldsymbol{W}_i\|_{2,1}^{2/3}}{\|\boldsymbol{W}_i\|^{2/3}}\right)^{3/2} \,,$$

where $\| \cdot \|$ denote the spectral norm, and $\| \cdot \|_{p,q}$ denotes the $(p,q)$-norm of a matrix, defined as $\|\boldsymbol{M}\|_{p,q} := \| (\|\boldsymbol{M}_{:,1}\|_p, \cdots, \|\boldsymbol{M}_{:,m}\|_p) \|_q$ for $\boldsymbol{M} \in \mathbb{R}^{d \times m}$.

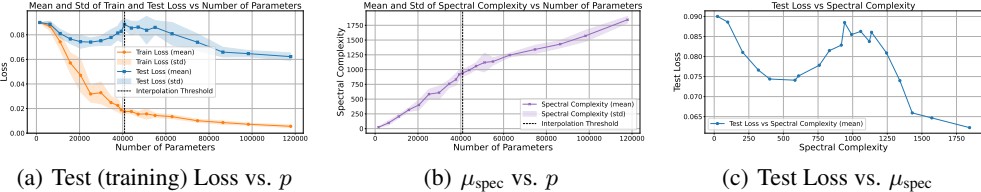

(a) Test (training) Loss vs. $p$      (b) $\mu_{\text{spec}}$ vs. $p$      (c) Test Loss vs. $\mu_{\text{spec}}$

Figure 19: Experiments on two-layer fully connected neural networks with noise level $\eta = 0.2$. The **left** figure is the same as Fig. 17(a). The **middle** figure shows the relationship between the path norm $\mu_{\text{spec}}$ and $p$. The **right** figure shows the relationship between the test loss and $\mu_{\text{spec}}$.

Fig. 19 illustrates the relationship between test loss, Spectral complexity $\mu_{\text{spec}}$, and the number of parameters $p$. We can see that $\mu_{\text{spec}}$ increases monotonically with $p$, so the same double descent phenomenon occurs with spectral complexity as model capacity.

**Path norm:** The path norm is defined as

$$\mu_{\text{path-norm}}(f_{\boldsymbol{w}}) = \sum_i f_{\boldsymbol{w}^2}(\mathbf{1})[i],$$

where $\boldsymbol{w}^2 = \boldsymbol{w} \circ \boldsymbol{w}$ is the element-wise square of the parameters, and $\mathbf{1}$ for all-one vector. The path norm represents the sum of the outputs of the neural network after squaring all the parameters and inputting the all-one vector.

Fig. 20 illustrates the relationship between test loss, Path norm $\mu_{\text{path}}$, and the number of parameters $p$. Path norm increases monotonically in the under-parameterized regime and decreases monotonically in the over-parameterized regime. This behavior resembles that of the $\ell_2$ norm of random feature estimators. Additionally, the relationship between test loss and path norm forms a U-shaped curve in the under-parameterized regime and increases monotonically in the over-parameterized regime. This pattern is strikingly similar to the relationship between test loss and the $\ell_2$ norm in random feature models.

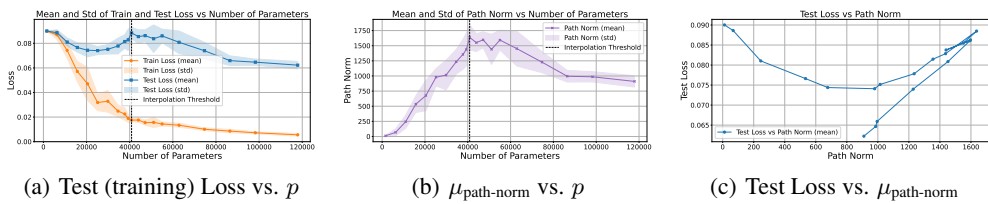

(a) Test (training) Loss vs. $p$      (b) $\mu_{\text{path-norm}}$ vs. $p$      (c) Test Loss vs. $\mu_{\text{path-norm}}$

Figure 20: Experiments on two-layer fully connected neural networks with noise level $\eta = 0.2$. The **left** figure is the same as Fig. 17(a). The **middle** figure shows the relationship between the path norm $\mu_{\text{path-norm}}$ and $p$. The **right** figure shows the relationship between the test loss and the path norm.

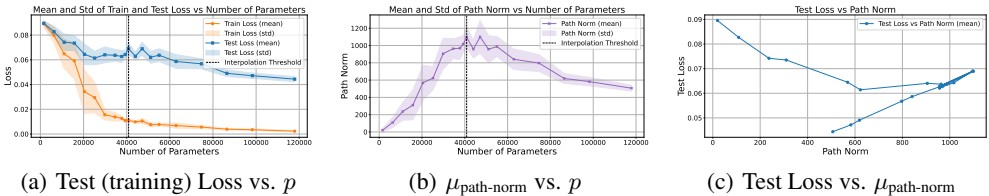

(a) Test (training) Loss vs. $p$      (b) $\mu_{\text{path-norm}}$ vs. $p$      (c) Test Loss vs. $\mu_{\text{path-norm}}$

Figure 21: Experiments on two-layer fully connected neural networks with noise level $\eta = 0.1$.

Besides, we also conduct experiments with the noise level $\eta = 0.1$ and $\eta = 0.3$ in Figs. 21 and 22, respectively. We can see that, when the noise level increases, we observe stronger peaks in the test

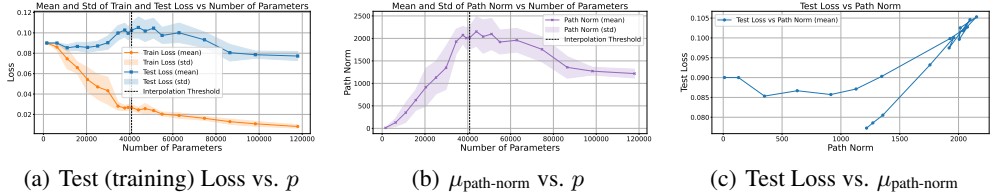

(a) Test (training) Loss vs. $p$  (b) $\mu_{\text{path-norm}}$ vs. $p$  (c) Test Loss vs. $\mu_{\text{path-norm}}$

Figure 22: Experiments on two-layer fully connected neural networks with noise level $\eta = 0.3$.

loss for double descent. However, the trend of test loss is similar at different noise levels with Path norm $\mu_{\text{path-norm}}$ as the model capacity, i.e., it shows a U-shape at the under-parameterized regime and an almost linear relationship at the over-parameterized regime.

These observations demonstrates the relationship between the test loss and norm, which is general, not limited to RFMs in the main text.

## H.4 Norm-based capacity in deep neural networks

To assess whether our norm-based capacity view extends beyond linear/RFM models and two-layer neural networks, we study the relationship between generalization and norm-based capacity on three deep families: *(i)* a 3-layer MLP trained on MNIST with 15% symmetric label noise (varying hidden width), *(ii)* a 3-layer CNN trained on MNIST with 15% symmetric label noise (varying channels), and *(iii)* ResNet18 [24] trained on CIFAR-10 with 15% noise (uniform width scaling across blocks). We train to (near) zero training error when feasible, then compute the path norm of the trained network and report test error on the clean test set. All runs are reproducible on a standard laptop with 16 GB memory. Code, scripts with pinned versions, and trained models are released at `github.com/yichenblue/norm-capacity` to facilitate verification and reuse.

**MLP.** We use **MNIST** dataset [30] with 16,000 samples and a 25% training split ($n_{\text{train}} = 4,000$, $n_{\text{test}} = 12,000$). The test set remains clean, while the training labels are corrupted with 15% symmetric noise: with probability 0.15, each label is replaced by a random class drawn uniformly from $\{0, \ldots, 9\} \setminus \{y\}$. The model is a three-layer MLP with ReLU activations, trained with SGD (momentum 0.9), learning rate 0.01, batch size 100, and CrossEntropyLoss for up to 500 epochs.

As shown in Fig. 23(a), plotting test error against width reproduces the familiar double-descent shape under label noise. When we instead index model capacity by the path norm of the trained network (also following [26] as in Appendix H.3),

$$\mu_{\text{path-norm}}(f_{\boldsymbol{w}}) = \sum_i f_{\boldsymbol{w}^2}(\mathbf{1})[i],$$

and plot test error against path norm (Fig. 23(c)), the curve exhibits a clear phase transition: a U-shaped trend in the under-parameterized regime, followed by a joint decrease of risk and norm once sufficiently over-parameterized. These observations are consistent with our findings in random feature models.

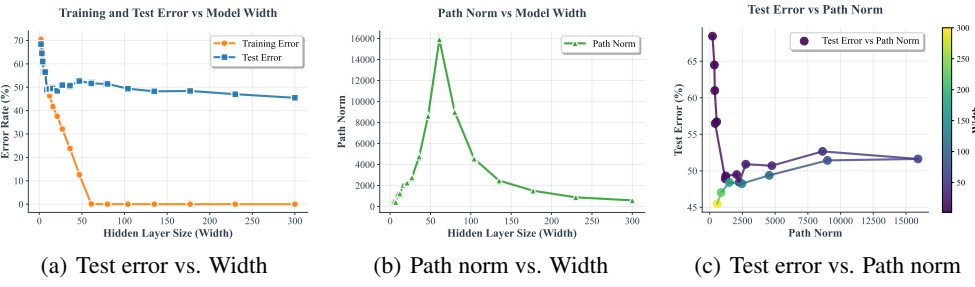

(a) Test error vs. Width  (b) Path norm vs. Width  (c) Test error vs. Path norm

Figure 23: Experiments on 3-layer MLP.

**CNN.** We next study a three-block CNN on **MNIST** with the same split and noise. Each block is Conv1d–ReLU with stride 2 and kernel size 3 (the first layer uses kernel size 5), followed by a linear classifier; we vary the number of channels to control capacity. We flatten each $28 \times 28$ image into a 1D signal before applying Conv1d. Results are qualitatively similar with Conv2d. Training uses the same optimizer and schedule as the MLP.

As shown in Fig. 24(a), test error as a function of channel count again shows double descent. In contrast, plotting against the path norm (Fig. 24(c)) produces the same pattern observed in the MLP: a U-shaped curve in the under-parameterized regime and a co-decrease of risk and norm when sufficiently over-parameterized, reinforcing the consistency of norm-based capacity across architectures.

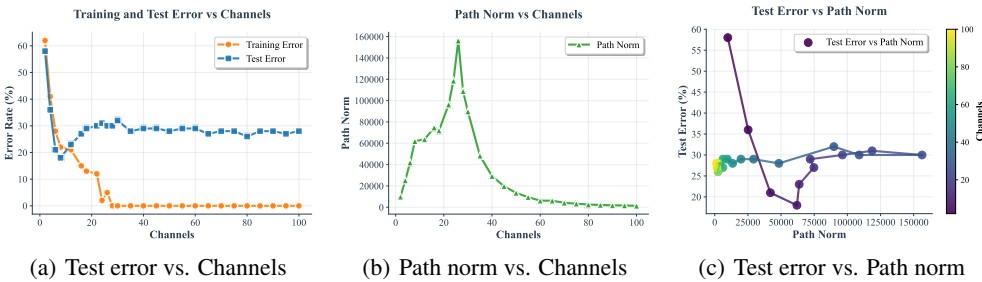

(a) Test error vs. Channels     (b) Path norm vs. Channels     (c) Test error vs. Path norm

Figure 24: Experiments on 3-layer CNN.

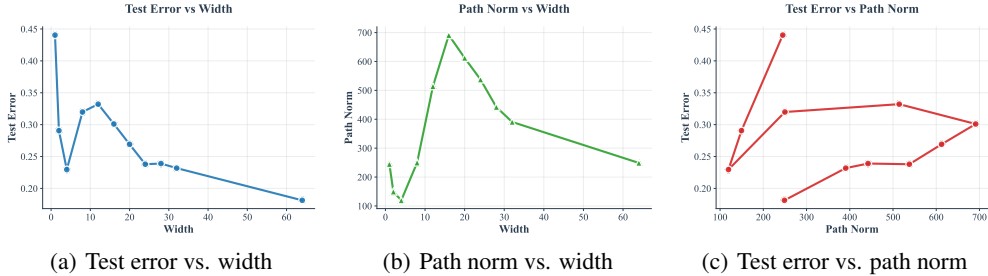

(a) Test error vs. width     (b) Path norm vs. width     (c) Test error vs. path norm

Figure 25: Experiments on ResNet18.

**ResNet18.** We further evaluate ResNet18 on **CIFAR-10** dataset [29] with 15% label noise, following the setup of OpenAI's deep double descent [40]. In addition to reproducing the reported deep double-descent behavior, we compute the path norm. Fig. 25 shows results across different widths. Based on Fig. 25 we can find that, in the sufficiently over-parameterized regime, the test risk and norm decrease together, ultimately aligning with the $\varphi$-curve. This suggests that double descent is a transient phenomenon, whereas the phase transition and the $\varphi$-shaped trend reflect more fundamental behavior if a suitable model capacity is used.

These results consistently demonstrate the existence of phase transitions, while double descent does not always occur—particularly under sufficient over-parameterization. Notably, the $\varphi$ curve exhibits a U-shaped trend, aligning with our theoretical predictions. All code and replication materials (including our reproduction of OpenAI's deep double-descent results) are available at `github.com/yichenblue/norm-capacity`.

