# OpenReview forum: "The $\varphi$ Curve: The Shape of Generalization through the Lens of Norm-based Capacity Control"
_NeurIPS.cc/2025/Conference — NeurIPS 2025 poster_

### Official Review · Reviewer_1YVx · 2025-06-06

**Clarity:** 2
**Significance:** 2
**Originality:** 3
**Rating:** 4
**Confidence:** 2

**Summary:**

The paper explains the double descent phenomenon for RFM from a theoretical perspective. Specifically, a simple neural network model was considered and the relationship between test error and network norm was calcualted.

**Questions:**

(1): The network and data used in the paper is simple, even close to a linear model, because the author only analyzed the linear function in the output layer of RFM, which can directly help derive the optimal formula (1) and simplify the calculation. And the label $y$  is also selected by linear with noise. However, commonly used neural networks and data are highly nonlinear, does your conclusion help us understand the  phenomenon of double decline in general situation?

(2): Is RFM being used in some field nowadays? Some examples?

**Ethical Concerns:**

["NO or VERY MINOR ethics concerns only"]

**Final Justification:**

I will keep my score.

**Limitations:**

Yes

**Quality:**

3

**Strengths And Weaknesses:**

Strengths:

The theory presented in this article is novel and undoubtedly has helped us further understand the double descent  phenomenon for RFM.

Weakness:

$The\ writing\ is\ not\ good\ enough,$ the author has placed too much basic settings in the appendix, which is not good, if only read the  main text, it will be hard to understand the paper.

$The setting is too simple.$ From the author's perspective, this is essentially a variant of a linear problem. The author needs to discuss how this differs from the classical linear problems?

$Need\ more\ References$, such as:《On the Double Descent of Random Features Models Trained with SGD》,nips2022.

---

> ### Author Rebuttal · Authors · 2025-07-29
>
> We thank the reviewer for acknowledging the novelty of our contribution. Below, we provide our point-by-point responses to the comments.
>
> >**Q1:** The writing is not good enough, too much basic settings is places in the appendix, which is not good, if only read the main text, it will be hard to understand the paper.
>
> **A1:** We thank the reviewer for checking the appendix. According to the reviewer's suggestions, we will include more preliminaries and proof sketches from the appendix into the main text for better readability, while keeping technical details in the appendix. We also plan to enhance the overview in the introduction to better guide the reader through the paper's structure.
>
> ---
>
> >**Q2:** The setting is too simple. From the author's perspective, this is essentially a variant of a linear problem. The author needs to discuss how this differs from the classical linear problems?
>
> **A2:** The results for RFMs differ from those of linear models in two key aspects: i) the structure of the deterministic equivalents; ii) the nature of the risk-norm relationship.
>
> Firstly, different from classical linear models only involving with data randomness, the random feature model (RFM) involves the random nonlinear feature mapping, leading to two sources of randomness—**data sampling and random feature initialization**. This makes deriving deterministic equivalents substantially more challenging with two self-consistency equations, leading to two effective regularization ($\nu_1$, $\nu_2$) for each source of randomness in the derivation of deterministic equivalence.
>
> For comparison, we have provided theoretical results for linear regression in Appendix D. To illustrate the differences in structure and complexity, we take the bias term of the risk and the norm as an example. To be specific,
>
> $B^{RFM}\sim\frac{\nu_2^2}{1-\Upsilon(\nu_1,\nu_2)}[\langle\theta_*,(\Lambda+\nu_2I)^{-2}\theta_*\rangle+\chi(\nu_2)\langle\theta_*,\Lambda(\Lambda+\nu_2I)^{-2}\theta_*\rangle],$
>
> $B^{LS}\sim\frac{\lambda_*^2\langle\beta_*,\Sigma(\Sigma+\lambda_* I)^{-2}\beta_*\rangle}{1-n^{-1}Tr(\Sigma^2(\Sigma+\lambda_*I)^{-2})},$
>
> Similarly, for the bias term of the estimator norm, we have:
>
> $B^{RFM}\sim\frac{p\langle\theta_*,\Lambda(\Lambda+\nu_2I)^{-2}\theta_*\rangle}{p-Tr(\Lambda^2(\Lambda+\nu_2I)^{-2})}+\frac{p\chi(\nu_2)}{n}\cdot\frac{\nu_2^2\left[\langle\theta_*,(\Lambda+\nu_2I)^{-2}\theta_*\rangle+\chi(\nu_2)\langle\theta_*,\Lambda(\Lambda+\nu_2I)^{-2}\theta_*\rangle\right]}{1-\Upsilon(\nu_1,\nu_2)},$
>
> $B^{LS}\sim\langle \beta_*,\Sigma^2(\Sigma+\lambda_* I)^{-2}\beta_*\rangle+\frac{\lambda_*^2\cdot\langle\beta_*,\Sigma(\Sigma+\lambda_* I)^{-2}\beta_*\rangle\cdot Tr(\Sigma(\Sigma+\lambda_* I)^{-2})}{ n-Tr(\Sigma^2(\Sigma+\lambda_* I)^{-2})}.$
>
> In the $p\to\infty$ limit, $\nu_1$ and $\nu_2$ in the RFMs converge to a single value $\nu_\kappa$. Consequently, $\Upsilon \to n^{-1} Tr\left( \Lambda^2(\Lambda + \nu_\kappa I)^{-2} \right)$, and $\chi \to 0$. Under this limit, the expressions for both the risk and the norm in RFMs reduce to those of linear regression, with the feature covariance matrix $\Sigma$ effectively replaced by $\Lambda$. This highlights that despite the structural differences, RFMs reduce to linear regression in the kernel limit, revealing a fundamental connection between the two models.
>
> Secondly, both RFMs and linear regression exhibit U-shaped trends in their risk–norm curves, but their behaviors in different regimes are exactly reversed: RFMs show a U-shape in the under-parameterized regime and monotonic increase in the over-parameterized regime, whereas linear regression shows the opposite.
>
> To illustrate this contrast, we refer to Proposition 4.1 for RFMs, which shows a linear relationship
>
> $R=\lambda_n N + C$
>
> for min-norm estimator within the over-parameterized regime (fully characterizing the curve in the under-parameterized regime remains analytically intractable). And for linear regression, Proposition D.8 (Appendix D.3.2) provides a closed-form expression for the risk–norm curve:
>
> $R=\begin{cases}N-{||\beta_*||}^2,&\text{if} d<n\\\\
> \sqrt{[N-({||\beta_*||}^2-\sigma^2)]^2+4{||\beta_*||}^2\sigma^2}-\sigma^2,&\text{otherwise}\end{cases},$
>
> which is linear in the under-parameterized regime and takes the form of a hyperbola in the over-parameterized regime.
>
> Despite these differences, **the core findings remain consistent across both models**: phase transitions are present, double descent does not occur, and the learning curves exhibit a U-shaped trend. This suggests that our conclusions are not limited to linear settings, but extend to more complex models as well.
>
> ---
>
> >**Q3:** Need more references, such as:《On the Double Descent of Random Features Models Trained with SGD》,nips2022.
>
> **A3:** We thank for the reviewer for pointing out this paper that provides a nice upper bound for RFF with SGD. Our current work does not tackle the optimization dynamics. Providing a **precise charecterization** of the test risk under (S)GD via deterministic equivalence, is highly nontrivial and often requires significantly more complex techniques. For instance, recent work [S1,S2] precisely charecterize the test risk under SGD in linear regression settings. We will cite these references and leave the SGD analysis as future work.
>
> ---
>
> >**Q4:** The network and data used in the paper is simple, even close to a linear model. However, commonly used neural networks and data are highly nonlinear, does your conclusion help us understand the phenomenon of double decline in general situation?
>
> **A4:** To validate our theory, apart from the existing experimental validation for two-layer neural networks in Appendix H.3, we also conducted comprehensive experiments on ResNet18, a three-layer MLP, and a CNN, respectively.
>
> Firstly, we reproduced OpenAI's results [S3] for deep double descent using ResNet18 on CIFAR-10 with 15% label noise, and computed the path norm following the definition in [S4]. The table below summarizes the results for ResNet:
>
> |ResNet18 Width|1|2|4|8|12|16|20|24|28|32|64|
> |-|-|-|-|-|-|-|-|-|-|-|-|
> |Test Error |0.4405|0.2905|0.2291|0.3203|0.3322|0.3011|0.2695|0.2381|0.2389|0.2317|0.1814|
> |Path Norm|245.01|149.48|119.41|249.93|514.35|691.31|612.41|537.77|442.29|390.63|248.85|
>
> Based on this, we can plot test error vs. path norm (we cannot put the figure here due to the NeurIPS rebuttal policy).
> We can find that, in the sufficiently over-parameterized regime, **the test risk and norm decrease together, ultimately aligning with the** $\varphi$ **curve**. This suggests that double descent is a transient phenomenon, whereas the phase transition and the $\varphi$-shaped trend reflect more fundamental behavior.
>
> Besides, we also conducted experiments on a 3-layer MLP and a 3-layer CNN. These results demonstrate similar $\varphi$-shaped trends in test error versus path norm and further confirm the generality of our findings. For detailed results, please refer to our answer to Reviewer miNz’s Q1.
>
> **Conclusion:** These results concide with our theory by demonstrating the existence of phase transitions, while double descent does not always occur—particularly under sufficient over-parameterization. Notably, the $\varphi$ curve exhibits a U-shaped trend, aligning with our theoretical predictions. These findings will be included in the updated version. Besides, we will **release the code, model parameters, and visualizations to foster further research in the community**, which would be helpful to understand the interplay between data, model complexity, and computational resources in modern ML. We will include these results in our updated version.
>
> ---
>
> >**Q5:** Is RFM being used in some field nowadays? Some examples?
>
>
> **A5:** Random Feature Models (RFMs) were originally proposed to accelerate large-scale kernel methods by approximating kernels via low-rank random projections [S5]. This idea was later adopted in linear attention mechanisms for Transformers, where the softmax function is closely related to an exponential kernel; see [S6, S7] for details. A survey [S8] provides an overview of this connection.
>
> More recently, the concept of low-rank random projection has also been applied to parameter-efficient fine-tuning techniques such as LoRA [S9] and its variants [S10]. For instance, [S10] leverages random features and learns their combination coefficients for fine-tuning. Notably, these random features can be generated on-the-fly using random number generators, significantly reducing memory overhead.
>
> We contend that RFMs retain both theoretical significance and practical relevance, even in the current era dominated by large language models.
>
> ---
>
> >Reference
>
>
> [S1] Courtney Paquette, Elliot Paquette, et al. "Homogenization of SGD in high-dimensions: Exact dynamics and generalization properties." Mathematical Programming, 2024.
>
> [S2] Elliot Paquette, Courtney Paquette, et al. "4+3 Phases of Compute-Optimal Neural Scaling Laws." NeruIPS'24.
>
> [S3] Preetum Nakkiran, Gal Kaplun, et al. "Deep Double Descent: Where Bigger Models and More Data Hurt." ICLR'20
>
> [S4] Yiding Jiang, Behnam Neyshabur, et al. "Fantastic generalization measures and where to find them." ICLR'20.
>
> [S5] Ali Rahimi, Benjamin Recht. "Random features for large-scale kernel machines." NeurIPS'07
>
> [S6] Krzysztof Choromanski, Valerii Likhosherstov, et al. "Rethinking attention with performers." ICLR'21
>
> [S7] Hao Peng, Nikolaos Pappas, et al. "Random feature attention." ICLR'21
>
> [S8] Fanghui Liu, Xiaolin Huang, et al. "Random Features for Kernel Approximation: A Survey on Algorithms, Theory, and Beyond." IEEE Transactions on Pattern Analysis and Machine Intelligence 44.10 (2021): 7128-7148.
>
> [S9] Edward J. Hu, Yelong Shen. "LoRA: Low-Rank Adaptation of Large Language Models." ICLR'22.
>
> [S10] Koohpayegani, Soroush Abbasi, et al. "NOLA: Compressing LoRA using Linear Combination of Random Basis." ICLR'24.

---

> > ### Author Response · Authors · 2025-08-05
> >
> > Dear Reviewer 1YVx,
> >
> > Thank you for your time and constructive feedback on our work. We have addressed your concerns with the following responses:
> >
> > - The setting for precise characterization we study lies at the frontier of current analytical understanding. In fact, our work highlights several open and mathematically challenging directions.
> >
> > - Our results differ from those in linear models, particularly in the structure of the deterministic equivalents and the nature of the risk–norm relationship.
> >
> > - We have included experiments on deep neural networks (e.g., ResNet18) that support and validate our theoretical findings.
> >
> > In addition, we have made efforts to improve the clarity and readability of the manuscript, and we also explicitly discuss potential future directions and associated technical challenges.
> >
> > As the author–reviewer response deadline approaches, we would greatly appreciate it if you could confirm whether our revisions adequately address your concerns. Please feel free to share any remaining questions—we would be happy to respond.
> >
> > Best regards,
> >
> > The Authors

---

> > ### Comment · Reviewer_1YVx · 2025-08-05
> >
> > I have read the author's reply, author has addressed most of my concerns, so I keep my score.

---

### Official Review · Reviewer_nNYU · 2025-06-24

**Clarity:** 3
**Significance:** 3
**Originality:** 4
**Rating:** 5
**Confidence:** 3

**Summary:**

This paper studies the relationship between norm-based capacity and test risk by applying the deterministic equivalent technique from random matrix theory. Using random feature models, the authors precisely characterize this relationship both theoretically and through empirical validation.

**Questions:**

1. In line 246, can you further explain as to why the bias follows a U-shaped curve, as opposed to the monotonic decrease shown in the classical bias-variance trade-off explanation? Specifically, what is the mechanism that causes a model's attempt to fit training data more closely to result in higher bias? The references shown in the sentence do not seem to discuss this in detail.
2. A natural question to ask is that, under what conditions does the test error curve in the overparameterized regime transition from being consistently below its underparameterized counterpart to above it ( $\varphi$ -shaped curve) and vice versa? It would be interesting to touch on this point.

**Ethical Concerns:**

["NO or VERY MINOR ethics concerns only"]

**Final Justification:**

Thank you for your response. The author has addressed all my questions, and I am keeping my score.

**Limitations:**

Yes

**Quality:**

4

**Strengths And Weaknesses:**

Strengths
1. The paper is clearly written and is methodically presented. The author's claim are sufficiently supported by formal mathematical proofs and are validated through empirical testing via synthetic and real-life datasets.
2. The proposed theory reveal that, under specific assumptions about the data, the test curve in the overparameterized regime consistently below the underparameterized test curve. This finding is novel in my opinion as it differs from the standard double descent phenomenon, where a " $\varphi$ " shaped test curve is typically observed.

Weaknesses
NA

---

> ### Author Rebuttal · Authors · 2025-07-29
>
> We thank the reviewer for the positive evaluation and are pleased that the clarity, theoretical rigor, and empirical validation of our work were well received. Below, we provide our point-by-point responses to the comments.
>
> >**Q1:** Further explain as to why the bias follows a U-shaped curve, as opposed to the monotonic decrease shown in the classical bias-variance trade-off explanation? Specifically, what is the mechanism that causes a model's attempt to fit training data more closely to result in higher bias?
>
> **A1:** We thank the reviewer for the insightful question. The U-shaped behavior of the bias in the under-parameterized regime arises due to the interaction between model capacity and the conditioning of the feature representation. To be specific,
>
> - **In the low-capacity regime** ($p \ll n$), increasing the number of random features improves the model’s ability to approximate the target function $f^\star$. Each additional feature offers a new direction in function space, leading to better alignment with $f^\star$ and reducing bias.
> - **Around $p = O(\sqrt{n} \log n)$**, this number of random features is theoretically proved to be sufficient for appromating $f^\star$ well [S1], yielding the lowest bias.
> - **As $p \to n$** (high-capacity regime), the kernel matrix becomes ill-conditioned. The model starts amplifying high-frequency components and fitting noise in the feature space, which degrades the alignment with $f^\star$ and increases bias.
>
> This intuition can be also formalized in our deterministic equivalence analysis. Under Assumption 2 (power-law spectrum) and the scaling $p = n^q$, $\lambda = n^{-(\ell - 1)}$ with $q, \ell > 1$, we derive the following asymptotic expressions (Appendix E.3.3):
>
> - **When $p \ll n$**, $\text{Bias} \approx C_3 \left( \frac{p}{C_1} \right)^{-2\alpha r}$, which decreases with $p$, as expected.
>
> - **When $p \to n$**, $\text{Bias} \approx \frac{n}{n - p} \cdot C_3 \left( \frac{p}{C_1} \right)^{-\alpha(2r \wedge 1)}$, with a positive derivative w.r.t. $p$ near $p = n$, indicating a **monotonic increase** in bias as the model becomes nearly interpolating.
>
> Thus, while classical bias-variance trade-offs assume monotonic bias decay, our setting reveals a **bias increase near interpolation threshold**, due to kernel ill-conditioning and projection distortion—resulting in a **U-shaped curve** in the under-parameterized regime.
>
>
> ---
>
> >**Q2:** Under what conditions does the test error curve in the overparameterized regime transition from being consistently below its underparameterized counterpart to above it ("$\varphi$"-shaped curve) and vice versa?
>
> **A2:** This is an interesting question! As shown in Appendix G.1, the transition to a “$\varphi$”-shaped curve depends on the data distribution, regularization, and to a lesser extent, the activation function.
>
> For Gaussian inputs (synthetic data), the over-parameterized test error consistently stays below the under-parameterized regime, aligning with our theory. On real data like FashionMNIST, we observe a $\varphi$-shaped curve in the ridgeless case, where test error initially rises before decreasing. With sufficient ridge regularization, the over-parameterized curve remains below throughout.
>
> These results suggest that deviations from our theoretical assumptions (e.g., independent feature inputs) play a key role in shaping the test error curve. Extension to the setting of correlated features requires additional efforts and new technical tools, leaving as a future work.
>
> ---
>
> >Reference
>
>
> [S1] Alessandro Rudi, Lorenzo Rosasco. "Generalization Properties of Learning with Random Features." NeurIPS'17.

---

> > ### Author Response · Authors · 2025-08-05
> >
> > Dear Reviewer nNYU,
> >
> > Many thanks for your support on our work.
> > According to your feedback, we have discussed the reason why the bias follows a U-shaped curve and when $\varphi$ curve exists.
> >
> > Feel free to let us know if you have any further concerns.
> >
> > Best,
> >
> > The Authors

---

### Official Review · Reviewer_arei · 2025-07-03

**Clarity:** 3
**Significance:** 3
**Originality:** 2
**Rating:** 4
**Confidence:** 4

**Summary:**

To tackle the challenge of characterizing the double descent learning curve based on the number of model parameters, this paper propose using the $l_2$ norm of the estimator's weights as a more suitable measure of capacity. When the learning curve is plotted against this norm-based capacity, the double descent disappears and is replaced by a more traditional U-shaped curve, aligning with classical statistical intuition. To prove this, the paper uses a mathematical tool from random matrix theory called deterministic equivalence to derive precise formulas for the test risk and the estimator's norm. This analysis is performed for linear regression and Random Feature Models (RFMs), which serve as theoretical stand-ins for two-layer neural networks. The theoretical results are shown to be in excellent agreement with experiments on both synthetic and real-world data.

**Questions:**

1.	Can you provide some intuition on how to generalize these results to deep, multi-layer neural network where all parameters are trained?
2.	How can the analyses be generalized to other norms? Are other measures like the "path norm" more suitable for complex models?
3.	How do these insights into norm-based capacity translate to tasks beyond regression, such as classification?

**Ethical Concerns:**

["NO or VERY MINOR ethics concerns only"]

**Final Justification:**

The authors have addressed my concerns. I maintain my original evaluation score for this submission.

**Limitations:**

1.	The theoretical analysis is restricted to linear models and Random Feature Models (RFMs), and does not extend to end-to-end trained deep neural networks.
2.	The study only considers models with closed-form solutions and overlooks the implicit regularization effects introduced by optimization methods such as Stochastic Gradient Descent (SGD). In practice, the choice of optimizer can significantly influence the model norm, which is not captured in the current analysis.
3.	The theoretical guarantees rely on strong assumptions about data structure, such as power-law eigenvalue decay. The paper also shows that when these assumptions are not satisfied—e.g., on real-world datasets—the empirical results can deviate from theoretical predictions.

**Paper Formatting Concerns:**

NA. The paper follows the formatting instructions.

**Quality:**

3

**Strengths And Weaknesses:**

Strengths:

1.	Strong Theoretical Foundation: The paper uses the deterministic equivalence in random matrix theory to provide precise, non-asymptotic characterizations of the relationship between estimator norm and test risk. A key strength is the development of new, more general deterministic quantities, which is a technical contribution of independent interest. The study is thorough, covering various model settings, including regularized $(\lambda>0)$ and min-norm interpolating $(\lambda=0)$ solutions.
2.	Clean Result: The key idea that a more appropriate capacity measure (norm) resolves the double descent puzzle is compelling and intuitive. It provides a clear explanation for the success of over-parameterized models without discarding classical statistical result.
3.	Convincing Validation: The paper's theoretical predictions are meticulously validated with experiments. The plots consistently show a near-perfect match between the derived theoretical curves and the empirical data points, lending strong support to the analysis.

Weaknesses

1.	Limited Scope of Models: The primary weakness, which the authors acknowledge, is that the theoretical analysis is confined to linear models and Random Feature Models (RFMs). The result also only holds for the second layer of RFMs, which is far from modern deep neural networks.
2.	Practicality of the Findings: The paper suggests that the model norm can be controlled via the regularization parameter, $\lambda$. However, it also points out that the behavior is different in the under- and over-parameterized regimes, meaning one must know which state the model is in to properly control the norm, which is a practical challenge.
3.	Dependence on Assumptions: The theoretical results rely on specific assumptions about the data, such as power-law eigenvalue decay (Assumption 2). The authors show that on real-world datasets like FashionMNIST, where these assumptions may not hold, the observed learning curve can deviate from the main theoretical prediction.
4.	Novelty Concern: The paper's contribution, while valuable, can be viewed as an incremental extension of the very recent work in reference [14]. The paper heavily leverages the framework, assumptions, and theorems from [14] for its analysis. The core methodological novelty is extending the tools from [14] to derive the deterministic equivalent for the estimator's norm—a different quantity than the test risk studied in [14]. This makes the contribution an insightful application and extension of an existing framework rather than the introduction of a fundamentally new one.

---

> ### Author Rebuttal · Authors · 2025-07-29
>
> We thank the reviewer for the constructive feedback and appreciate the recognition of the theoretical depth, clarity of insight, and empirical validation. Below, we provide our point-by-point responses to the comments.
>
> >**Q1:** The analysis is restricted to linear models and RFMs, focusing only on the second layer and lacking applicability to deep neural networks.
>
> **A1:** We acknowledge the theoretical focus on linear models and RFMs. Extension to nonlinear neural networks is non-trivial and remains an open question except for some extremely special cases [S1, S2].
>
> To validate our theory, we conducted experiments on ResNet18 by reproducing OpenAI's results [S3] on CIFAR-10 with 15% label noise and computing the path norm as defined in [S4] as below.
>
> | ResNet18 Width | 1      | 2      | 4      | 8      | 12     | 16     | 20     | 24     | 28     | 32     | 64     |
> |----------------|--------|--------|--------|--------|--------|--------|--------|--------|--------|--------|--------|
> | Test Error | 0.4405 | 0.2905 | 0.2291 | 0.3203 | 0.3322 | 0.3011 | 0.2695 | 0.2381 | 0.2389 | 0.2317 | 0.1814 |
> | Path Norm      | 245.01 | 149.48 | 119.41 | 249.93 | 514.35 | 691.31 | 612.41 | 537.77 | 442.29 | 390.63 | 248.85 |
>
> Based on this, we can plot test error vs. path norm (we cannot put the figure here due to the NeurIPS rebuttal policy).
> We can find that, in the sufficiently over-parameterized regime, **the test risk and norm decrease together, ultimately aligning with the** $\varphi$ **curve**. This suggests that double descent is a transient phenomenon, whereas the phase transition and the $\varphi$-shaped trend reflect more fundamental behavior.
>
> Besides, we also conducted experiments on a 3-layer MLP and a 3-layer CNN. These results demonstrate similar $\varphi$-shaped trends in test error versus path norm and further confirm the generality of our findings. For detailed results, please refer to our answer to Reviewer miNz’s Q1.
>
> **Conclusion:** These results concide with our theory by demonstrating the existence of phase transitions, while double descent does not always occur—particularly under sufficient over-parameterization. Notably, the $\varphi$ curve exhibits a U-shaped trend, aligning with our theoretical predictions. These findings will be included in the updated version. Besides, we will **release the code, model parameters, and visualizations to foster further research in the community**, which would be helpful to understand the interplay between data, model complexity, and computational resources in modern ML. We will include these results in our updated version.
>
> ---
>
> >**Q2:** The model norm can be controlled via $\lambda$. However, the behavior is different in the under- and over-parameterized regimes, meaning one must know which state the model is in to control the norm, which is a practical challenge.
>
> **A2:** Identifying whether a model is under- or over-parameterized is straightforward in practice, as it depends on knowing the number of model parameters and the training sample size. Thus, determining the regime is practically feasible.
>
> To guide model selection with the proper regularization parameter $\lambda$ in practice, for under/over-parameterized regimes, one can plot the empirical risk vs. the estimator norm as the regularization parameter (in weight decay) varies, i.e., **L-curve** [S5], respectively. The “corner” of L-curve reflects an optimal trade-off between data fitting and model complexity. **Selecting the regularization parameter at this corner** provides an effective strategy for balancing generalization and capacity. We will include more discussion on practical guidelines in our updated version.
>
> ---
>
> >**Q3:** Theoretical results rely on specific assumptions (e.g., power-law eigenvalue decay), which may not hold in real-world datasets.
>
> **A3:** We acknowledge that our theoretical results rely on assumptions such as power-law eigenvalue decay, which may not hold universally. However, these assumptions are standard in the theoretical literature [S6–S8], as they ensure the hypothesis class is well-behaved (e.g., trace-class kernels).
>
> While relaxing these assumptions is technically challenging, our empirical results on shallow and **deep** neural networks as mentioned before remain consistent with our theory: double descent is a transient phenomenon, whereas the phase transition and the $\varphi$-shaped trend reflect more fundamental behavior.
>
> ---
>
> >**Q4:** The work mainly extends prior results from [14] to estimator norm, making it an incremental contribution rather than a fundamentally new framework.
>
> **A4:** While we build on the analytical tools developed in [14] for deriving deterministic equivalents, the techniques in [14] are **insufficient to obtain the deterministic equivalent of the estimator norm**. For instance, we generalize their results by deriving expressions of the form $Tr(A X^\top X (X^\top X + \lambda)^{-1})$
> for arbitrary PSD matrices $A$, whereas [14] only considered the special case $A = I$.
>
> Moreover, establishing a quantitative relationship between test risk and estimator norm requires nontrivial analysis, including eliminating explicit model size dependence—a key challenge not addressed in [14].
>
> We do not claim to introduce a completely new proof framework, but our contribution aims to address the fudemental question: how test risk evolves with a suitable model capacity? We provide both theoretical and empirical evidence, and offer new insights beyond prior results.
>
> ---
>
> >**Q5:** How can the analyses be generalized to other norms? Are other measures like the path norm more suitable for complex models?
>
> **A5:** Generalizing our analysis to other norms is technically challenging, as it relies heavily on the resolvent structure specific to linear and random feature regression. While recent work has begun to explore deterministic equivalents for general loss functions [S9], the analysis remains considerably more complex and less mature.
>
> However, according to the reviewer's suggestions, we have empirically evaluated the path norm in deep networks, as given in **Q1**. This trend closely mirrors our theoretical findings for the $\ell_2$ norm, suggesting that the norm–risk relationship may generalize beyond $\ell_2$ and hold for more complex architectures.
>
> ---
>
> >**Q6:** How do these insights into norm-based capacity translate to tasks beyond regression, such as classification?
>
> **A6:** Extending the theory from regression to classification is analytically non-trivial due to differences in loss functions and optimization dynamics. A rigorous treatment would require new techniques beyond our current framework.
>
> Nonetheless, we present empirical results on classification tasks in Table 5, located in Appendix G (page 65), showing that test error (i.e., classification error rate) exhibits the same qualitative behavior as in regression: a U-shaped curve in the under-parameterized regime and a monotonic increase in the over-parameterized regime. These observations suggest that norm-based capacity control remains relevant and informative for understanding generalization in classification.
>
> ---
>
> >**Q7:** The study only considers models with closed-form solutions and overlooks the implicit regularization introduced by optimization methods such as SGD.
>
> A7: Thanks for the reviewer pointing out this. Our work requires closed-form solutions and does not address the effects of optimization dynamics. Some recent studies [S10–S13] precisely charecterize the test risk under SGD for linear models. Extending our deterministic equivalent framework to incorporate such optimization effects is an important and challenging direction, which we leave for future work. We will include these references and discuss the possiblity for study.
>
> ---
>
> We hope our responses provide a better understanding on this work in terms of problem settings, technical contributions, and practical guidelines. We're happy to discuss more if the reviewer has any concerns.
>
> ---
>
> >Reference
>
> [S1] Behrad Moniri, Donghwan Lee, et al. "A Theory of Non-Linear Feature Learning with One Gradient Step in Two-Layer Neural Networks." ArXiv:2310.07891.
>
> [S2] Yatin Dandi, Luca Pesce, et al. "A Random Matrix Theory Perspective on the Spectrum of Learned Features and Asymptotic Generalization Capabilities." ArXiv:2410.18938
>
> [S3] Preetum Nakkiran, Gal Kaplun, et al. "Deep Double Descent: Where Bigger Models and More Data Hurt." ICLR'20
>
> [S4] Yiding Jiang, Behnam Neyshabur, et al. "Fantastic generalization measures and where to find them." ICLR'20.
>
> [S5] Hansen, Per Christian. "Analysis of discrete ill-posed problems by means of the L-curve." SIAM review 34.4 (1992): 561-580.
>
> [S6] Leonardo Defilippis, Bruno Loureiro, and Theodor Misiakiewicz. "Dimension-free deterministic equivalents and scaling laws for random feature regression." NeurIPS'25.
>
> [S7] James B. Simon, Dhruva Karkada, et al. "More is Better in Modern Machine Learning: when Infinite Overparameterization is Optimal and Overfitting is Obligatory." ICLR'24.
>
> [S8] Dominic Richards, Jaouad Mourtada, and Lorenzo Rosasco. "Asymptotics of Ridge (less) Regression under General Source Condition." AISTATS'21
>
> [S9] Kiana Asgari, Andrea Montanari, and Basil Saeed. "Local minima of the empirical risk in high dimension: General theorems and convex examples." ArXiv:2502.01953.
>
> [S10] Courtney Paquette, Kiwon Lee, et al. "SGD in the Large: Average-case Analysis, Asymptotics, and Stepsize Criticality." COLT'21.
>
> [S11] Courtney Paquette, Elliot Paquette. "Dynamics of Stochastic Momentum Methods on Large-scale, Quadratic Models." NeurIPS'21.
>
> [S12] Courtney Paquette, Elliot Paquette, et al. "Homogenization of SGD in high-dimensions: Exact dynamics and generalization properties." Mathematical Programming, 2024.
>
> [S13] Elliot Paquette, Courtney Paquette, et al. "4+3 Phases of Compute-Optimal Neural Scaling Laws." NeurIPS'24.

---

> > ### Author Response · Authors · 2025-08-05
> >
> > Dear Reviewer arei,
> >
> > Thank you for your time and constructive feedback on our work.
> >
> > In response to your comments, we have conducted comprehensive experiments on deep neural networks to validate our theoretical findings, introduced a model selection procedure based on our theory, and discussed potential future directions along with the associated technical challenges.
> >
> > As the author-reviewer response deadline is approaching, we would greatly appreciate it if you could confirm whether our revisions address your concerns. Please feel free to let us know if there are any remaining issues—we would be happy to further clarify or revise.
> >
> > Best regards,
> >
> > The Authors

---

> > ### Comment · Reviewer_arei · 2025-08-06
> >
> > I thank the authors for addressing my concerns. I have no further comments and remain inclined to recommend acceptance.

---

### Official Review · Reviewer_miNz · 2025-07-03

**Clarity:** 3
**Significance:** 3
**Originality:** 3
**Rating:** 5
**Confidence:** 4

**Summary:**

For the problem of understanding how the test risk scales with model complexity in machine learning, this paper studies the shape of the generalization curve through a norm-based capacity measure. This paper uses deterministic equivalence as a key theoretical tool to characterize the relationship between risk and norm. In addition, this paper verifies that capacity control based on the $\ell_2$ norm can recover the classic U-shaped curve in RFM, which challenges the double descent behavior. Validation on synthetic data and MNIST shows that the theoretical predictions are consistent with the empirical results.

**Questions:**

Please refer to the Weaknesses for details.

**Ethical Concerns:**

["NO or VERY MINOR ethics concerns only"]

**Final Justification:**

The additional experiments provided by the authors in their feedback address my concerns about the limitations of their experimental validation. I recommend acceptance on the condition that the authors include more discussion of the relationship between norm capacity control and actual model selection in the final version, as the authors promised in their feedback, because this is very important.

**Limitations:**

This work does not seem to have any potential negative societal impact.

**Paper Formatting Concerns:**

NO.

**Quality:**

3

**Strengths And Weaknesses:**

Strengths:

1. This paper argues that when using a norm-based capacity metric, the learning curve of an over-parameterized deep learning model changes from a "double-descent" shape to a U-shaped curve that is more in line with classical statistical intuition. The authors verify this view through theoretical analysis and experiments using random feature models (RFMs).

2. Another key finding of this paper is that in the over-parameterized region, the test risk increases monotonically with the norm capacity. This seems to confirm the conclusion of classical learning theory to some extent: small-capacity models have better generalization performance. This finding is very interesting and seems to suggest that the current mainstream view that "classical learning theory is not applicable to deep learning" is a misunderstanding, that is, people's understanding of capacity is not in place.

3. This paper clarifies the essential connection between norm capacity and generalization, reveals the monotonic relationship between the regularization parameter $lambda$ and norm capacity, and provides a new perspective for model tuning.

Weaknesses:

I really appreciate this work, especially some of the insights it brings to the table through experimental phenomena, which, although not new in some ways, have been controversial in the learning theory community. The consistency between the theoretical analysis and experimental verification in this paper has answered some controversies very well. My only concern is the universality of the conclusions of this paper, because the analysis and verification involved are limited to RFM and linear models, lacking empirical evidence for deep models (more than 2 layers), so supplementary experimental verification in deep cases is necessary. In addition, it is necessary to discuss how to apply the control of norm capacity to actual model selection or training strategies (such as early stopping, weight decay).

---

> ### Author Rebuttal · Authors · 2025-07-29
>
> We thank the reviewer for the thoughtful feedback and appreciate the recognition of the theoretical-experimental consistency and the insights into generalization. Below, the concern regarding the universality of the conclusions and their applicability to model selection is addressed.
>
> >**Q1:** The conclusions may not be universally applicable, as the analysis is limited to RFM and linear models, lacking empirical evidence for **deep models** (more than 2 layers), so supplementary experimental verification in deep cases is necessary.
>
> **A1:** To validate our theory, according to the reviewer's suggestion, we conducted comprehensive experiments on ResNet18, a three-layer MLP, and a CNN, respectively.
>
> Firstly, we reproduced OpenAI's results [S1] for deep double descent using ResNet18 on CIFAR-10 with 15% label noise, and simultaneously computed the path norm following the definition in [S2]. The table below summarizes the results for various widths in each layer of ResNet:
>
> | ResNet18 Width | 1      | 2      | 4      | 8      | 12     | 16     | 20     | 24     | 28     | 32     | 64     |
> |----------------|--------|--------|--------|--------|--------|--------|--------|--------|--------|--------|--------|
> | Test Error | 0.4405 | 0.2905 | 0.2291 | 0.3203 | 0.3322 | 0.3011 | 0.2695 | 0.2381 | 0.2389 | 0.2317 | 0.1814 |
> | Path Norm      | 245.01 | 149.48 | 119.41 | 249.93 | 514.35 | 691.31 | 612.41 | 537.77 | 442.29 | 390.63 | 248.85 |
>
> Based on this, we can plot test error vs. path norm (we cannot put the figure here due to the NeurIPS rebuttal policy).
> We can find that, in the sufficiently over-parameterized regime, **the test risk and norm decrease together, ultimately aligning with the** $\varphi$ **curve**. This suggests that double descent is a transient phenomenon, whereas the phase transition and the $\varphi$-shaped trend reflect more fundamental behavior if a suitable model capacity is used.
>
> Secondly, we also conducted experiments on a 3-layer MLP and a 3-layer CNN trained on the MNIST dataset with 15% label noise, varying the model width (or number of channels) to study the relationship between test error and path norm. For the MLP, when plotting test risk vs. path norm, we can observe $\varphi$ curve, which aligns with our findings on RFMs.
>
> | MLP Width         | 2     | 3     | 4     | 5     | 7     | 9     | 12    | 27     | 47     | 61     | 80     | 300    |
> |-------------------|-------|-------|-------|-------|-------|-------|-------|--------|--------|--------|--------|--------|
> | Test Error (%)    | 68.43 | 64.48 | 60.98 | 56.72 | 56.48 | 48.90 | 49.30 | 50.90  | 52.66  | 51.62  | 51.42  | 45.48  |
> | Path Norm         | 216.28| 359.70| 389.95| 537.58| 417.88|1189.49|1230.08|2756.54 |8625.63 |15918.52|8996.63 | 581.03  |
>
>
> Similarly, in the CNN setting, we have the same observation as below.
>
> | CNN Channels      | 2     | 4     | 6     | 8     | 10    | 12    | 14    | 16    | 20     | 45     | 60    | 80    | 90    | 100   |
> |-------------------|-------|-------|-------|-------|-------|-------|-------|-------|--------|--------|-------|-------|-------|--------|
> | Test Error (%)    | 58.00 | 36.00 | 21.00 | 18.00 | 20.00 | 23.00 | 22.00 | 27.00 | 31.00  | 29.00  | 29.00 | 26.00 | 28.00 | 28.00  |
> | Path Norm         | 9846  |25203  |41980  |61969  |99322  |63675  |45474  |74785  |126608  |19764   | 6198  | 2523  | 1784  | 1338   |
>
> **Conclusion:** These results consistently demonstrate the existence of phase transitions, while double descent does not always occur—particularly under sufficient over-parameterization. Notably, the $\varphi$ curve exhibits a U-shaped trend, aligning with our theoretical predictions. These findings will be included in the updated version. Besides, we will **release the code, model parameters, and visualizations to foster further research in the community**, which would be helpful to understand the interplay between data, model complexity, and computational resources in modern ML. We will include these results in our updated version.
>
>
> ---
>
> >**Q2:** It is necessary to discuss how to apply the control of norm capacity to **actual model selection** or training strategies (such as early stopping, weight decay).
>
> **A2:** Norm capacity can be controlled in practice via standard regularization techniques such as weight decay and early stopping, which implicitly or explicitly constrain model norms. As shown in Figure 2 of our paper, increasing the $\ell_2$ regularization strength leads to a monotonic decrease in the estimator norm.
>
> To guide model selection, one can plot **the empirical risk vs. weight decay (i.e., estimator's norm)** as the regularization parameter (in weight decay) varies. This typically yields an **L-curve** [S3], where the “corner” reflects an optimal trade-off between data fitting and model complexity. Selecting the regularization parameter at this corner provides an effective strategy for balancing generalization and capacity. We will include more discussion in our updated version.
>
> ---
>
> >Reference
>
> [S1] Preetum Nakkiran, Gal Kaplun, et al. "Deep Double Descent: Where Bigger Models and More Data Hurt." ICLR'20
>
> [S2] Yiding Jiang, Behnam Neyshabur, et al. "Fantastic generalization measures and where to find them." ICLR'20.
>
> [S3] Hansen, Per Christian. "Analysis of discrete ill-posed problems by means of the L-curve." SIAM review 34.4 (1992): 561-580.

---

> ### Author Response · Authors · 2025-08-04
>
> Dear Reviewer miNz,
>
> Thank you for your support and valuable feedback on our work.
>
> According to your suggestions, we have conducted comprehensive experiments ranging from shallow neural networks to ResNet18 (to reproduce OpenAI's results), and we will release the code publicly to foster further research. Additionally, we have included a discussion on the use of the L-curve for practical model selection of the regularization parameter.
>
> Please feel free to let us know if you have any further questions or suggestions.
>
> Best regards,
>
> The Authors

---

> > ### Comment · Reviewer_miNz · 2025-08-06
> >
> > Thanks to the authors for their detailed responses, my concern has been addressed.

---

### Note · Authors · 2025-08-16

Dear AC and reviewers,

We would like to once again thank all the reviewers for their time, thoughtful feedback, and positive evaluations and thank for the AC on handling our submission.

All of the reviewers were convinced by the novelty and impact of our work.

According to the reviewers' suggestions, we have conducted comprehensive experiments ranging from shallow neural networks to ResNet18 (to reproduce OpenAI's results), and we will release the code publicly to foster further research.
Additionally, we have discussed the use of the L-curve for practical model selection of the regularization parameter.
We will update our submission according to the reviewers' suggestions.

Best,

Authors

---

### Decision · Program_Chairs · 2025-09-17

**Decision:**

Accept (poster)

**Comment:**

The paper revisits the problem of scaling of the test risk with the model complexity adopting a new norm-based capacity measure. By means of such a new measure, a more traditional U-shaped curve for the risk is found and the double-descent phenomenon disappears. The analysis, performed for linear regression on a random feature model, is verified by means of numerical experiments on both synthetic and real data. The paper has been appreciated for its clear and strong theoretical arguments, but suffer some limitations, such as the very simple set-up and strong theoretical assumptions on both the architecture and the data structure. Nevertheless, the Reviewers expressed unanimous appreciation for the work in light of the fact that it offers new insights on the problem, and the theoretical arguments are supported in more general settings by numerical experiments: I therefore recommend the authors to integrate in the paper such experiments. I recommend therefore its acceptance.